# Flux Vacua and Modularity for $\mathbb{Z}_2$ Symmetric Calabi-Yau Manifolds

PHILIP CANDELAS[*1], XENIA DE LA OSSA[*2], PYRY KUUSELA[*†3]

AND

JOSEPH MCGOVERN[*4]

[*]*Mathematical Institute*
*University of Oxford*
*Andrew Wiles Building*
*Radcliffe Observatory Quarter*
*Oxford, OX2 6GG, UK*

[†]*PRISMA+ Cluster of Excellence & Mainz Institute for Theoretical Physics*
*Johannes Gutenberg-Universität Mainz*
*55099 Mainz, Germany*

## Abstract

We find continuous families of supersymmetric flux vacua in IIB Calabi-Yau compactifications for multiparameter manifolds with an appropriate $\mathbb{Z}_2$ symmetry. We argue, supported by extensive computational evidence, that the numerators of the local zeta functions of these compactification manifolds have quadratic factors. These factors are associated with weight-two modular forms, these manifolds being said to be weight-two modular. Our evidence supports the flux modularity conjecture of Kachru, Nally, and Yang. The modular forms are related to a continuous family of elliptic curves. The flux vacua can be lifted to F-theory on elliptically fibred Calabi-Yau fourfolds. If conjectural expressions for Deligne's periods are true, then these imply that the F-theory fibre is complex-isomorphic to the modular curve. In three examples, we compute the local zeta function of the internal geometry using an extension of known methods, which we discuss here and in more detail in a companion paper. With these techniques, we are able to compare the zeta function coefficients to modular form Fourier coefficients for hundreds of manifolds in three distinct families, finding agreement in all cases. Our techniques enable us to study not only parameters valued in $\mathbb{Q}$ but also in algebraic extensions of $\mathbb{Q}$, so exhibiting relations to Hilbert and Bianchi modular forms. We present in appendices the zeta function numerators of these manifolds, together with the corresponding modular forms.

[1] candelas@maths.ox.ac.uk   [2] delaossa@maths.ox.ac.uk   [3] pyry.r.kuusela@gmail.com   [4] mcgovernjv@gmail.com

# 1. Introduction

## 1.1. Preamble

The study of the modularity of Calabi-Yau threefolds (for a review, see for example [1]) has in recent years been subject to a fresh mode of investigation: advances in computational techniques have permitted direct 'experimental' tests of conjectures and suggested a number of facts, some long-expected by theorists and others coming as a surprise. For instance, in the papers [2–4], examples of weight-four modular one-parameter manifolds were found by computing the zeta function. This is a generating function for point counts of the manifold considered as a variety over finite fields, and thus probes the arithmetic of the manifold. Such a function may at first glance seem to bear no relation to physics where the manifold is conventionally viewed as a real or complex variety. However, various quantisation conditions make it natural to study discrete or rational mathematical objects related to varieties $X$ over $\mathbb{Q}$ or $\mathbb{Z}_p$, an example being the integral cohomology groups $H^n(X, \mathbb{Z})$. In this paper, we fix attention to weight-two modular manifolds. This means that the local zeta functions of the manifolds possess quadratic factors, and there is a straightforward, though conceptually deep, way to read off a weight-two modular form from such a factor. According to the modularity theorems [5, 6], one can further associate this modular form with an elliptic curve $\mathcal{E}_R$ which we call the *modular curve.*

Some interesting connections between physics and modularity of Calabi-Yau manifolds have been investigated by Moore in [7, 8] where the weight-four case was studied. In this case there is a similar prescription that gives a weight-four modular form, and this is intimately connected with the attractor mechanism of $\mathcal{N} = 2$ extremal black holes. The first explicit examples of such one-parameter manifolds with full SU(3) holonomy were found in [2], with further consequences investigated in [9]. More examples of weight-four modularity were then announced in [3, 4]. The aforementioned manifolds are, strictly speaking, only conjecturally weight-four modular. These conjectures are supported by lengthy tables of zeta function numerators where factorisation into quadratics can be observed. It was proven in [10] that rigid Calabi-Yau manifolds are weight-four modular. It is also of interest that the emerging story of the computation of certain amplitudes via the geometry of Calabi-Yau manifolds is informed by modularity considerations. For reviews on the relation between Feynman integrals and motives, see for example [11–14]. Recent work on the relation between Calabi-Yau periods and these integrals includes [15–18].

To find manifolds that are weight-two modular, we concentrate on families of of Calabi-Yau manifolds whose complex structure moduli space possesses a $\mathbb{Z}_2$ symmetry which, in suitable coordinates, exchanges two of the moduli. By studying the action of this symmetry of the middle cohomology $H^3(X)$, we show that on the fixed locus of this action in the moduli space, the zeta function of every manifold has a quadratic factor.[1] It is natural to assume that such a factorisation arises from a reduction of a Galois representation, where one of the irreducible pieces is two-dimensional. Then *Serre's modularity conjecture* [20, 21] (which has since been proved [22, 23]) asserts that such a representation is associated to a modular form. In our case these modular forms are of weight two, which can be traced back to the fact that the two-dimensional representation acts on $H^{1,2} \oplus H^{2,1}$, so that the manifold is weight-two modular.

A priori, demanding a $\mathbb{Z}_2$ symmetry that exchanges moduli requires that we work with multiparameter manifolds so we require a practical means of computing the zeta function for multiparameter geometries. We introduce a new means of doing this, built on the work of [24] which solved this

---

[1]This result is related to, although distinct from, a conjecture made by Elmi in [19]. This states that when the moduli space of a one-parameter family possesses an involution symmetry compatible with the Hodge structure in the sense discussed in §5.3, the fixed-point locus of this operation is a rank-two attractor. In the one-parameter case such attractor points are always also isolated supersymmetric flux vacua.

problem for one-parameter geometries, and which is explained further in a companion paper devoted to our method [25]. These methods, without which the tests that we make would not be feasible, were also discussed and developed in [26].

One of our original motivations to study these symmetric manifolds in connection with modularity was that the fixed point loci solve the supersymmetric vacuum equations for type IIB flux compactifications. That supersymmetric flux vacua should be weight-two modular was first conjectured in [27], with further consequences investigated in [28, 29], and one of the major applications of our construction lies in testing this *flux modularity conjecture*: the Calabi-Yau manifolds whose moduli give supersymmetric vacua in IIB flux compactifications are weight-two modular. In future work, additional tests of the flux modularity conjecture could be made on other solution sets of the equations defining these supersymmetric vacua, such as those described in [30]. We verify the flux modularity conjecture in each of our examples for several hundred values of the moduli. We tabulate these results in appendices, giving the weight-two modular form associated to the moduli that support supersymmetric flux vacua. Moreover, we are able to do this for not only rational moduli, but also moduli in algebraic extensions of $\mathbb{Q}$. We pick out quadratic imaginary and totally real number fields, and demonstrate the relation to Bianchi and Hilbert modular forms respectively.

Varying the complex structure moduli $\boldsymbol{\varphi}$ of the compactification manifold causes the value of the axiodilaton field $\tau$ to vary. When $\tau(\boldsymbol{\varphi})$ is interpreted as a complex structure modulus, this gives rise to a family of elliptic curves. We find that the axiodilaton $\tau$ determined by the flux vacuum construction gives a function $j(\tau(\boldsymbol{\varphi}))$, which is always a rational function of $\boldsymbol{\varphi}$. One can write down a family of elliptic curves over $\mathbb{Q}$ that have $\tau(\boldsymbol{\varphi})$ as their complex structure parameter. Further, these curves $\mathcal{E}_S$ can be expressed in the form

$$y^2 \;=\; x^3 + \frac{(C(\boldsymbol{\varphi}) - 3)}{k(\boldsymbol{\varphi})^4}\, x + \frac{(C(\boldsymbol{\varphi}) - 2)}{k(\boldsymbol{\varphi})^6} \;, \tag{1}$$

where $(C(\boldsymbol{\varphi}) - 3)/k(\boldsymbol{\varphi})^4$ and $(C(\boldsymbol{\varphi}) - 2)/k(\boldsymbol{\varphi})^6$ are rational functions. We call this family of elliptic curves over $\mathbb{Q}$ a family of twisted Sen curves $\mathcal{E}_S$. This construction can also be viewed as defining a single elliptic curve over[2] $\mathbb{Q}(\boldsymbol{\varphi})$, which we call the twisted Sen curve. By slight abuse of notation we also denote this by $\mathcal{E}_S$. It should always be clear from context which is meant. To explain the terminology "twisted", consider a change of coordinates

$$x \mapsto k(\boldsymbol{\varphi})^{-2} x \;, \qquad y \mapsto k(\boldsymbol{\varphi})^{-3} y \;,$$

which, in general, is not defined over $\mathbb{Q}$, but instead[3] over $\mathbb{C}$, thus generically changing the *rational structure* of the elliptic curve. Such an isomorphism over $\mathbb{C}$ is known as a *twist*, and applying it gives a curve

$$y^2 \;=\; x^3 + (C(\boldsymbol{\varphi}) - 3)x + (C(\boldsymbol{\varphi}) - 2) \;. \tag{2}$$

As noted by Sen [31], elliptic curves of this form can be interpreted as the fibre of an F-theory lift of the IIB flux vacuum with orientifold planes. As the rational structure of the elliptic curve does not appear naturally in the F-theory description, it makes no difference whether we consider the original or the twisted curve, as they are complex isomorphic. Therefore, one can view the F-theory fibre $\mathcal{E}_F$ associated to the twisted Sen curve $\mathcal{E}_S$ as the curve (1) defined over the complex numbers.

Based on arguments using conjectural form of Deligne's periods and extensive numerical evidence, we find that it is possible to choose the twisted Sen curve in such a way that it is isomorphic with

---

[2]Here $\mathbb{Q}(\boldsymbol{\varphi})$ denotes the field of rational functions in variables $\varphi^i$ with rational coefficients, and should not be confused with the field extension of $\mathbb{Q}$ generated by the value the moduli take at a point.

[3]We could also consider these curves over $\overline{\mathbb{Q}}$, but opt instead to work with $\mathbb{C}$.

the modular curve $\mathcal{E}_R$ over $\mathbb{Q}$. This is a remarkable result, as it is not a priori clear that the modular $\mathcal{E}_R$ curve needs to vary with $\varphi$ in any nice way, let alone that the family of curves $\mathcal{E}_R$ varies with the moduli in a rational way. Combining these observations also reveals that, analogously to [28], as a complex curve, the modular curve can be given an interpretation as the F-theory fibre. To summarise this point, we find that the modular curves $\mathcal{E}_R$ form a family that can be expressed as an elliptic $\mathcal{E}_S$ curve over $\mathbb{Q}(\varphi)$. This family has the same complex strucutre as the F-theory fibre, so if we forget about the rational structure of the curves $\mathcal{E}_S$ we can identify these with the generic F-theory fibres $\mathcal{E}_F$ in the F-theory uplift of the IIB flux vacuum solution.

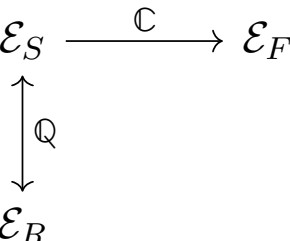

Figure 1: *The relations between the three families of elliptic curves appearing in this work: $\mathcal{E}_R$ denotes the modular elliptic curve associated to the zeta function, while $\mathcal{E}_S$ and $\mathcal{E}_F$ denote the twisted Sen curve and the F-theory fibre, respectively. It turns out that $\mathcal{E}_R$ and $\mathcal{E}_S$ are isomorphic over $\mathbb{Q}$, which is denoted by the vertical arrow. The F-theory fibre $\mathcal{E}_F$ is by construction the twisted Sen curve $\mathcal{E}_S$ considered to be defined over $\mathbb{C}$.*

Relating the F-theory curve to the modularity of the threefold may seem surprising. It is perhaps best understood, although not without invoking very deep concepts and conjectures, in terms of an elliptic motive. For a discussion of this concept with regard to Calabi-Yau modularity and supersymmetry, see [32, 33]. The numerator of the zeta function can be computed as the determinant a matrix that gives the action of the inverse Frobenius map on the middle étale cohomology of the manifold $X$. Hence, this numerator has a quadratic factor when this action is reducible, with a $2 \times 2$ block. The two dimensional subspace of $H^3_{\text{ét}}(X)$ left invariant by this action is a realisation of an elliptic motive that could, by comparison theorems, instead be realised in the de Rham cohomology. If this is done, one finds an integral lattice in $H^{1,2}(X, \mathbb{C}) \oplus H^{2,1}(X, \mathbb{C})$. In other words, the space $H^{1,2}(X, \mathbb{C}) \oplus H^{2,1}(X, \mathbb{C})$ intersects the lattice $H^3(X, \mathbb{Z})$ in a sublattice of rank at least two. This lattice can be generated by the flux vectors associated to the solution of the supersymmetric flux vacuum equations, and the lattice parameter is identified with the axiodilaton $\tau$. This is tautologically the modulus of the F-theory curve, but can be seen by the motivic considerations to also relate to the modularity of the threefold.

One of the surprising implications of the flux modularity conjecture is just how many families of Calabi-Yau manifolds should contain modular members. Our $\mathbb{Z}_2$ symmetric families already furnish a large collection, many of which are familiar and well-studied manifolds. For example, the mirror of any favourable CICY whose configuration matrix is symmetric under the exchange of two rows should, on the $h^{2,1} - 1$ dimensional locus that we describe, be weight-two modular. The computational results in this paper give a strong indication that modular manifolds were never that far away and need not be particularly exotic.

The wealth of data at our disposal allows us also to make many tests of *Deligne's conjecture* [34], which implies that there are relations between the periods of modular threefolds and the L-values of the associated modular forms. To a critical motive, roughly speaking a suitable piece of cohomology, can be associated *Deligne's periods* $c^{\pm}$. For Calabi-Yau manifolds, these can be formed as determinants of certain linear combinations of the periods and their derivatives, determined by

the action of the complex conjugation map on them. Deligne's conjecture states that a rational multiple of the period $c^+$ should give the special value of the L-function associated to the motive. We are able to verify this numerically for tens of manifolds belonging to the three families of threefolds we study. We also observe similar relation between this special value, the period $c^-$, and the modulus $\tau$ of the elliptic curve associated to the elliptic motive. Furthermore, we note an interesting numerical relation between derivatives of the periods which generate the elliptic motive, and the periods of the associated elliptic curve. This relation also holds in cases where the elliptic curve has a non-zero rank, and the special L-value vanishes by the Birch and Swinnerton-Dyer conjecture. Some examples of analogous relations on a discrete set of weight-four modular Calabi-Yau manifolds were provided already in [2, 4], and the relation to Deligne's conjecture was elaborated in [32, 33, 35]. Relations like this can see concrete application in physics due to the fact that the periods of Calabi-Yau threefolds may relate to other problems such as Feynman integrals in quantum field theories [36].

We illustrate the general theory with three worked examples. The first of these, which we present in greatest detail, is the five-parameter Hulek-Verrill manifold. This is mirror to the CICY

$$\begin{matrix}\mathbb{P}^1 \\ \mathbb{P}^1 \\ \mathbb{P}^1 \\ \mathbb{P}^1 \\ \mathbb{P}^1\end{matrix}\begin{bmatrix}1 & 1 \\ 1 & 1 \\ 1 & 1 \\ 1 & 1 \\ 1 & 1\end{bmatrix}_{\chi=-80}.$$

This was an early candidate for modularity, studied in [37]. The periods of this family of manifolds are related to banana diagrams [17], and it possesses a rank-two attractor [2]. This geometry was also studied in the context of mirror symmetry in [38].

In addition, we consider the mirror of the bicubic given by the CICY configuration

$$\begin{matrix}\mathbb{P}^2 \\ \mathbb{P}^2\end{matrix}\begin{bmatrix}3 \\ 3\end{bmatrix}_{\chi=-162}.$$

The third example is the mirror of the split quintic in $\mathbb{P}^4 \times \mathbb{P}^4$, so corresponding to the mirror of the configuration

$$\begin{matrix}\mathbb{P}^4 \\ \mathbb{P}^4\end{matrix}\begin{bmatrix}1\ 1\ 1\ 1\ 1 \\ 1\ 1\ 1\ 1\ 1\end{bmatrix}_{\chi=-100}.$$

Mirror symmetry for this family has previously been considered in [39, 40], and for a one-parameter quotient in [41]. We study this geometry in a forthcoming paper and find evidence that it possesses a rank-two attractor [42].

The Hodge conjecture suggests that the elliptic curves should be visible for any modular manifold for which the Hodge structure splits and thus the zeta function factors appropriately. We are able to construct an elliptic surface as a subvariety of the Hulek-Verrill threefold, which can be shown to give rise to non-trivial three-cycles of Hodge type $(1,2) + (2,1)$. Understanding how this occurs for other manifolds, and the relevance of arithmetic geometry to F-theory, are issues to which we hope to return.

## 1.2. Conventions

In this paper, we study $m$-parameter Calabi-Yau manifolds $X_{\boldsymbol{\varphi}}$, whose complex structure is parameterised by the coordinates $\boldsymbol{\varphi} = (\varphi^1, \ldots, \varphi^m)$, where $m = h^{2,1}$. We denote $m \times m$ matrices by symbols in blackboard bold, and $m$-component vectors by symbols in boldface.

Unless otherwise stated, we employ the Einstein summation convention, with the indices $a, b, c, \ldots$ from the beginning of the Latin alphabet taking values $0, \ldots, m$, and the indices $i, j, k, \ldots$ from the middle of the alphabet taking values $1, \ldots, m$.

The generator of quadratic number fields $\mathbb{Q}(\sqrt{n})$ is taken to be the smaller[4] root of the minimal polynomial given in the *L-Functions and Modular Forms Database LMFDB* [43], except in the case of the field $\mathbb{Q}(i)$, where, following their convention, we take the generator to be i.

Some symbols that appear in multiple sections are collected, together with their definitions, in table 1.

| Symbol | Definition/Description | Ref. |
|---|---|---|
| $\varphi$ | The coordinates $(\varphi^1, \ldots, \varphi^m)$ on the complex structure space of a Calabi-Yau manifold $X_{\boldsymbol{\varphi}}$. | |
| $\epsilon_i$ | Representation matrices of the homology algebra of the mirror of $X_{\boldsymbol{\varphi}}$. | (20) |
| $\varpi$ | The period vector of $X_{\boldsymbol{\varphi}}$ in the Frobenius basis. | (21) |
| $\Pi$ | The period vector of $X_{\boldsymbol{\varphi}}$ expressed in the integral symplectic basis. | (22) |
| $\mathbb{F}_{p^n}$ | The finite field with $p^n$ elements. | |
| $\zeta_p(X_{\boldsymbol{\varphi}}, T)$ | The local zeta function of a Calabi-Yau manifold $X_{\boldsymbol{\varphi}}$. | (32) |
| $R(X, T)$ | The numerator of the zeta function $\zeta_p(X, T)$. | (32) |
| $\mathcal{E}_R$ | The modular elliptic curve associated to the zeta function. | §2.3, §4.4 |
| $\mathcal{E}_F$ | The F-theory fibre of the uplift of the supersymmetric flux vacuum solution in IIB to F-theory. | (14) |
| $\mathcal{E}_S$ | The twisted Sen curve which is isomorphic to $\mathcal{E}_R$ over $\mathbb{Q}$ or the relevant number field $\mathbb{Q}(\sqrt{n})$. | §2.3, (4.4) |
| $c_p$ | The non-trivial coefficient in the quadratic factor $R_2(X, T)$ of the numerator of the zeta function $\zeta_p(X, T)$. | (36) |
| $\alpha_p$ | The coefficient of $q^p$ in the Fourier expansion of a modular form. | (30) |
| $k(\boldsymbol{\varphi})$ | The twist parameter relating the F-theory fibre $\mathcal{E}_F$ and the twisted Sen curve $\mathcal{E}_S$. | (37) |
| $U(\boldsymbol{\varphi})$ | The matrix representing the action of the inverse Frobenius map on the middle cohomology. | (50) |

Table 1: *Some symbols that are used throughout the paper with references to where they are defined.*

---

[4]We order the roots with respect to their real part in increasing order. Roots with the same real part are ordered with respect to the imaginary part.

## 2. Review of Supersymmetric Flux Vacua in IIB and F-Theory

### 2.1. Flux compactifications

Flux compactifications refer to a class of string theory compactifications with non-trivial background values for the $(p+1)$-form field strengths. Supersymmetric configurations were first studied in [44], with a more recent discussion given in [45].

The massless field content of type IIB string consists of the even-form Ramond-Ramond fields $C_0$, $C_2$, and $C_4$ with field strengths $F_i = dC_{i-1}$; the Kalb-Ramond two-form field $B_2$ with field strength $H_3 = dB_2$; the metric $g$; and the dilaton $\phi$. In addition to the field content, the theory contains odd-dimensional localised D-branes, that is D1, D3, D5, D7, and D9 branes, which couple to the gauge field potentials either electrically or magnetically.

The axion $C_0$ and dilaton $\phi$ are packaged into a single complex scalar field, the axiodilaton $\tau$:

$$\tau = C_0 + i\,e^{-\phi} \ . \tag{3}$$

The field strengths $H_3 = dB_2$ and $F_3 = dC_2$ are also for convenience combined into the complex field strength

$$G_3 = F_3 - \tau H_3 \ . \tag{4}$$

In the absence of fluxes, compactifying the ten-dimensional theory on an $m$-parameter Calabi-Yau threefold $X$ yields a four-dimensional $\mathcal{N}=2$ theory with $m=h^{1,2}$ vector multiplets, $h^{1,1}+1$ hypermultiplets, and one gravity multiplet (for a review, see for example [46, 47]). The scalar fields in the vector and hypermultiplets can be seen to parametrise the geometry of the internal Calabi-Yau manifold. Specifically, the scalars in the vector multiplets can be identified with the complex structure moduli space coordinates, while one of the scalars in each hypermultiplet, apart from the universal hypermultiplet, parametrise the complexified Kähler structure moduli space.

When compactifying in the presence of fluxes, the four-dimensional supersymmetry is broken to $\mathcal{N} = 1$. The moduli are now scalars in $\mathcal{N} = 1$ chiral multiplets [48]. The four-dimensional action includes a non-zero potential term which couples the scalars (including the axiodilaton $\tau$) to each other and the background fluxes $F_3$, $H_3$. This potential is constructed from a superpotential [49] which has the form

$$W \overset{\text{def}}{=} \int_X G_3 \wedge \Omega = (2\pi)^2\alpha' (F - \tau H)^T \Sigma \Pi \ , \qquad \Sigma = \begin{pmatrix} \mathbb{0} & \mathbb{1}_{m+1} \\ -\mathbb{1}_{m+1} & \mathbb{0} \end{pmatrix} \ , \tag{5}$$

where $\Omega$ is the holomorphic three-form on $X$. The form $\Omega$ can be expanded in an integral symplectic basis which gives the period vector in the integral basis, $\Pi$. This vector consists of functions of the $m$ complex structure moduli of $X$ that solve the system of Picard-Fuchs differential equations associated to $X$.

In order to obtain non-trivial supersymmetric solutions to the equations of motion the field strengths must satisfy

$$\int_X H_3 \wedge F_3 > 0 \ . \tag{6}$$

Decomposing the fluxes $H_3, F_3 \in H^3(X, \mathbb{Z})$ in an integral symplectic basis of $H^3(X, \mathbb{Z})$, we write

$$H_3 = (2\pi)^2\alpha' \left(h^a\alpha_a - h_b\,\beta^b\right), \qquad F_3 = (2\pi)^2\alpha' \left(f^a\alpha_a - f_b\,\beta^b\right), \quad \text{with} \quad h_b, h^a, f_b, f^a \in \mathbb{Z} \ .$$

The condition (6) can be satisfied by choosing suitable $h_b, h^a, f_b,$ and $f^a$, which are collected into vectors $F$ and $H$,

$$F = \begin{pmatrix} f^a \\ f_b \end{pmatrix}, \qquad H = \begin{pmatrix} h^a \\ h_b \end{pmatrix}.$$

There is also a tadpole cancellation condition which can be written in the F-theory language as

$$\frac{1}{2\kappa_{10}^2 T_3} \int_X H_3 \wedge F_3 + Q^{D3} = \frac{\chi(X_4)}{24},$$

where $X_4$ is the Calabi-Yau fourfold on which the F-theory is compactified. Since we are free to add localised D3-branes (but not anti-D3-branes) without breaking supersymmetry, this imposes, together with (6), the following condition on the fluxes:

$$0 < F^T \Sigma H \leqslant \frac{\chi(X_4)}{24}.$$

## 2.2. The scalar potential and supersymmetric vacuum equations

Compactification of type IIB supergravity on a Calabi-Yau threefold $X$ in the presence of fluxes $F_3$ and $H_3$ yields a four-dimensional $\mathcal{N}=1$ matter-coupled supergravity theory containing scalars that are coupled by an induced potential. This theory can be modified by quantities with string-theoretic origins. We do not incorporate all of these effects in the analysis performed in this paper. In this subsection, we explain the origin of the potential of the four-dimensional theory, highlighting where both perturbative and nonperturbative stringy corrections could affect our constructions.

Let us temporarily forget the fact that we have an internal Calabi-Yau manifold, and discuss some general features of these supergravity theories following [50]. In any four-dimensional $\mathcal{N}=1$ supergravity coupled to $N$ chiral multiplets, the scalars in those multiplets are coordinates on a projective Kähler manifold. The theory is specified by the choice of two functions of the moduli; the Kähler potential $\mathcal{K}$ for the scalar manifold and a superpotential $W$. From these two functions, one calculates the following potential:

$$V \stackrel{\text{def}}{=} e^{\mathcal{K}} \left( G^{\alpha\bar{\beta}} D_\alpha W \overline{D_\beta W} - 3|W|^2 \right). \tag{7}$$

The sum in the above expression runs over all moduli, so in the case of a Calabi-Yau compactification on $X$, $\alpha, \beta = 1, \ldots, h^{1,1}(X) + h^{1,2}(X) + 1$. The metric $G_{\alpha\bar{\beta}}$ is the metric on the scalar manifold, and, by virtue of the Kähler condition, $G_{\alpha\bar{\beta}} = \partial_\alpha \partial_{\bar{\beta}} \mathcal{K}$. The quantity $D_\alpha W$ is the Kähler covariant derivative of $W$ [51, 52], which has Kähler weight $(1,0)$, so

$$D_\alpha W = \partial_\alpha W + (\partial_\alpha \mathcal{K}) W.$$

In the case of a type IIB Calabi-Yau compactification, the total scalar manifold is the product of the upper half plane (in which the axiodilaton is valued) and the moduli space of the Calabi-Yau manifold $X$, which factorises, at least locally, into the the moduli spaces of complex structures and complexified Kähler structures of $X$. The dimensions of these spaces are, respectively, $h^{2,1}(X)$ and $h^{1,1}(X)$. The Kähler potential for the total moduli space is then a sum of the Kähler potentials for the axiodilaton, complex structure, and Kähler structure factors of the moduli space:

$$\mathcal{K} = K^{\text{AD}} + K^{\text{CS}} + K^{\text{CK}}.$$

These depend on the moduli as follows:

$$K^{\text{AD}} = -\log\left(-i(\tau - \bar{\tau})\right), \quad K^{\text{CS}} = -\log\left(-i\,\Pi^\dagger \Sigma\,\Pi\right), \quad K^{\text{CK}} = -\log\left(-i\,\amalg^\dagger \Sigma\amalg\right), \tag{8}$$

where $\Pi$ is the period vector for the mirror manifold, so a function of the Kähler moduli of $X$. The Kähler potentials for the axiodilaton and complex structure moduli are exact at tree level, while the potential for complexified Kähler structures, $K^{\mathbb{C}K}$, is corrected by fundamental string instantons and $\alpha'$ corrections (this corrected $K^{\mathbb{C}K}$ is the uncorrected $\widetilde{K}^{\mathbb{C}S}$ of the mirror manifold). Additionally, nonperturbative effects can modify $W$ in such a way as to give it a dependence on the Kähler moduli of $X$. If these latter corrections are neglected, then derivatives of $W$ with respect to Kähler moduli vanish and the potential (7) reduces to

$$V_{(W \text{ classical})} = e^{\mathcal{K}} \left( \text{Im}[\tau]^2 |D_\tau W|^2 + g^{i\bar{j}} D_i W \overline{D_j W} + \left( \widetilde{g}^{r\bar{s}} K^{\mathbb{C}K}_r K^{\mathbb{C}K}_{\bar{s}} - 3 \right) |W|^2 \right) . \tag{9}$$

In this expression $g_{i\bar{j}}$ is the metric on the space of complex structures and $\widetilde{g}_{r\bar{s}}$ is the metric on the space of complexified Kähler structures. The indices $i, j$ run from 1 to $h^{2,1}(X)$, and $r, s$ run from 1 to $h^{1,1}(X)$.

In supersymmetric configurations the superpotential $W$ vanishes, which greatly simplifies the above expression for the scalar potential $V$, which must vanish in a vacuum configuration.

$$V = e^{\mathcal{K}} \left( \text{Im}[\tau]^2 |\partial_\tau W|^2 + g^{i\bar{j}} \partial_{\varphi^i} W \, \overline{\partial_{\varphi^j} W} \right) . \tag{10}$$

This potential is manifestly positive semidefinite, and thus the equations defining a supersymmetric flux vacuum, requiring that both $W$ and $V$ vanish, read:

$$W = 0 , \qquad \partial_\tau W = 0 , \qquad \partial_{\varphi^i} W = 0 . \tag{11}$$

Depending on the vacuum expectation values of the field strengths $F_3$ and $H_3$, these equations might be solved for the complex structure moduli $\varphi^i$ as well as the value of the axiodilaton field. The Kähler moduli are unconstrained, although this ceases to be true when instanton corrections to the action are incorporated [53].

*Nonperturbative corrections to $W$*

In this paper, we only consider the case where the superpotential is given by the classical formula (5), that is, we work in the approximation where we can safely ignore the non-perturbative stringy corrections. Incorporating these corrections to $W$ could alter the space of supersymmetric flux vacua. In particular, the condition of vanishing $W$, which has a cohomological interpretation central to the modularity that we discuss, only holds when the classical expression (5) for $W$ is used. Additionally, these corrections give $W$ a dependence on the Kähler moduli, and so additional derivative terms will appear in the potential.

The nonperturbative corrections to $W$ that we are referring to are the same as those considered in [53], where it was argued that precisely these corrections allowed for a realisation of metastable de Sitter vacua. One source of these corrections is Euclidean D3-instantons, first discussed in [54]. Additionally, in some setups gluino condensation could occur on spacetime-filling D7-branes which gives a contribution to $W$. We will not discuss these possibilities further in this paper.

### 2.3. Lifting to F-theory

It is well known that the moduli space of elliptic curves can be taken to be the upper half plane, where two distinct points correspond to complex-isomorphic elliptic curves if and only if they are related by a Möbius transformation. The IIB theory possesses an $\text{SL}(2, \mathbb{Z})$ symmetry that acts on $\tau$ as a Möbius transformation and on $G_3$ by

$$\tau \mapsto \frac{a\tau + b}{c\tau + d} , \qquad G_3 \mapsto \frac{G_3}{c\tau + d} , \qquad \begin{pmatrix} a & b \\ c & d \end{pmatrix} \in \text{SL}(2, \mathbb{Z}) . \tag{12}$$

F-theory [55] (for a review, see for example [56]) geometrises the axiodilaton, giving it an interpretation as the complex structure parameter of an elliptic fibre $\mathcal{E}_F$ of an elliptically fibred Calabi-Yau fourfold.

The complex isomorphism classes of elliptic curves are characterised by the $j$-invariant

$$j(\tau) \;=\; 1728\,\frac{60^3 E_4^3(\tau)}{\Delta(\tau)} \;=\; \mathrm{e}^{-2\pi\mathrm{i}\tau} + 744 + 196884\,\mathrm{e}^{2\pi\mathrm{i}\tau} + 21493760\,\mathrm{e}^{4\pi\mathrm{i}\tau} + \dots$$

which is a modular function, taking a constant value on $\mathrm{SL}(2,\mathbb{Z})$ orbits of $\tau$. Elliptic curves with the same $j$-invariant are isomorphic as complex varieties. However, this isomorphism need not hold over subfields of $\mathbb{C}$. For instance, the curves

$$y^2 \;=\; x^3 + 2x \;, \qquad \eta^2 \;=\; \xi^3 - 2\xi \;,$$

both have $j$-invariant 1728. Indeed, one can obtain the latter from the former by the twist $(x,y) \mapsto (\mathrm{i}\xi, \mathrm{e}^{-\mathrm{i}\pi/4}\eta)$. Despite this, they are not isomorphic over $\mathbb{Q}$ as there is no *rational* morphism that will take one between the above pair of curves. We say that the two curves have different *rational structures*. This is reflected in various properties of the curves. For instance, the *Mordell–Weil group* of the first curve is $\mathbb{Z}_2$, and has rank 0, whereas the second curve has a rank-one Mordell–Weil group $\mathbb{Z} \oplus \mathbb{Z}_2$. As there is a morphism over $\mathbb{C}$ that relates these two curves, the two curves are said to be *twists* of one another.

Following [27, 31, 57], we can find an F-theory solution corresponding to a supersymmetric flux vacuum in type IIB theory compactified on a Calabi-Yau manifold $X$. The F-theory solutions are compactifications on an elliptically fibred Calabi-Yau manifold $X_4$ with the threefold $X$ being a double cover of the base of this fibration. In the weak coupling orbifold limit one retrieves the IIB theory on $X$ with a pair of D7-branes coincident with an O7-plane [31], and with the axiodilaton corresponding to a generic elliptic fibre of the fibration. In §4.7 we will argue that this curve has the same complex structure parameter $\tau$, and thus the same $j$-invariant, as the modular curve $\mathcal{E}_R$ related to the zeta function, by virtue of the construction of the F-theory lift. However, the F-theory fibre appears in the theory only as a complex curve, and thus this construction is agnostic regarding the rational structure of the fibre.

While F-theory does not distinguish curves with different rational structures, the fact that the modular curve and the F-theory fibre have the same complex structure makes it interesting to inquire whether there is a 'nice' family of elliptic curves over $\mathbb{Q}$ that, when considered as curves over $\mathbb{C}$, are isomorphic to the F-theory fibre. We find that it is possible to find an elliptic curve $\mathcal{E}_S$ over $\mathbb{Q}(\boldsymbol{\varphi})$, the field of rational functions of the complex structure moduli $\boldsymbol{\varphi}$, which can be interpreted as a family of rational elliptic curves whose members have the same rational structure as the modular curves $\mathcal{E}_R$. Therefore they are also complex isomorphic to the F-theory fibres and could be interpreted as a 'natural' rational model for the fibre $\mathcal{E}_F$, although we wish to emphasise that any other rational model would yield the same F-theory description, although other choices will not in general have a relation to the zeta function. This is analogous to the results of [27], where evidence was presented that for a class of flux compactifications on the mirror octic Calabi-Yau manifolds, whose zeta function was computed in [58], it is possible to find such a family of elliptic curves.

The generic elliptic fibre of the compactification manifold $X_4$ can be identified by starting with the most general elliptic fibration, finding the restrictions imposed by requiring that supersymmetry is preserved, and fixing the fibre by comparing to the IIB data. The most generic elliptic fibre over a base $B$ can be written in Weierstrass form as

$$y^2 \;=\; x^3 + f(u)x + g(u) \;, \tag{13}$$

where $u$ are the coordinates on the base $B$. In F-theory compactifications, the complex structure parameter $\tau$ of this elliptic curve is identified with the axiodilaton. For this reason, in the following we make no distinction between the two.

The points in the moduli space satisfying $\Delta = 4f^3 + 27g^2 = 0$, where the elliptic curve degenerates, can be identified with the locations of D7-branes. For this reason, it is convenient [31] to reparametrise the pair of functions $f$, $g$ in terms of functions[5] $\eta$, $h$ and $\chi$ such that

$$f(u) \;=\; C\eta(u) - 3h(u)^2 \;, \qquad g(u) \;=\; h(u)\left[C\eta(u) - 2h(u)^2\right] + C^2\chi(u) \;.$$

In this parametrisation, the limit $C \to 0$ corresponds to the weak coupling limit in which the locus where the elliptic fibre degenerates is given by $\eta(u) = \pm\sqrt{-12h(u)\chi(u)}$. This can be interpreted as the location of a pair of D7-branes which fill the spacetime and intersect the base on these four real-dimensional loci. Similarly, the locus $h(u) = 0$ gives the intersection of spacetime-filling O7-planes with the base.

Since we are looking for supersymmetric solutions, the solutions must reduce to a supersymmetric IIB configuration in the weak coupling limit $C \to 0$. To obtain a maximally supersymmetric background in this limit, we require that the O7-planes and D7-branes coincide. It turns out that, for the models we study, such solutions can be found among those studied in [57] where $\chi(u) = 0$, so that the O7 and D7 loci are, respectively, $h(u) = 0$ and $\eta(u) = 0$, and furthermore $\eta(u) = h(u)^2$ so that these loci coincide.

With these choices, the fibre with parameter $u$ in (13) becomes

$$y^2 \;=\; x^3 + h(u)^2(C - 3)\,x + h(u)^3(C - 2) \;.$$

Away from the singular locus $h(u) = 0$ this expression can be simplified by a fibrewise rescaling of the coordinates

$$y \to h^{3/2}(u)\,y \;, \qquad x \to h(u)\,x \;.$$

In these coordinates the equation defining the generic F-theory fibre $\mathcal{E}_F$ reads

$$y^2 \;=\; x^3 + (C - 3)\,x + (C - 2) \;. \tag{14}$$

The constant $C$ is fixed by matching the complex structure modulus of this fibre to the IIB axiodilaton profile. Once we have computed the value of $\tau$ as a function of the other moduli fields required by the supersymmetric flux vacuum equations, we will be able to evaluate $j(\tau)$. We will match this to the $j$-invariant of (14),

$$j_C \;=\; \frac{6912\,(C - 3)^3}{C^2\,(4C - 9)} \;, \tag{15}$$

This gives a cubic equation for $C$

$$j(\tau)C^2(4C - 9) - 6912(C - 3)^3 \;=\; 0 \;, \tag{16}$$

which we can solve for $C$, obtaining a curve of the form (14) with the $j$-invariant associated to the IIB construction. There are three solutions to the above equation, but these are all by construction complex isomorphic. As the F-theory construction does not specify the rational structure, only this isomorphism class enters the description and all three solutions are equivalent. The freedom of choosing a rational structure on the fibre can also be viewed as a freedom to choose any complex coordinates $x$ and $y$ on the fibre.

As $j(\tau)$ depends on the complex structure moduli $\boldsymbol{\varphi}$, we denote the solutions by $C(\boldsymbol{\varphi})$ to highlight the dependence of the solution on the complex structure of the compactification manifold.

---

[5]This function $\chi(u)$ is not to be confused with the Euler characteristic of the manifold.

### 3. Solutions via Permutation Symmetry

#### 3.1. The supersymmetric flux vacuum equations in Calabi-Yau compactifications

We wish to find solutions to the supersymmetric flux vacuum equations (11) in the case where the four-dimensional $\mathcal{N} = 1$ supergravity theory is obtained by compactifying the IIB theory on a Calabi-Yau threefold $X$. To perform this task one must specify a pair of integral flux vectors $F$ and $H$ satisfying

$$F^T \Sigma H \neq 0 .$$

In terms of these the complexified field strength $G_3$ has components $F - \tau H$, so that the superpotential (5) is

$$W = (2\pi)^2 \alpha' (F - \tau H)^T \Sigma \Pi .$$

Further, the flux vectors $F$ and $H$ must be such that the equations

$$W = 0 , \qquad \partial_\tau W = 0 , \qquad \partial_{\varphi^i} W = 0 \tag{17}$$

can be solved. The second of the equations (17) reads $H^T \Sigma \Pi = 0$, so that $H$ is symplectic orthogonal to the period vector $\Pi$, whence the second equation reduces to $F^T \Sigma \Pi = 0$. Thus we must find vectors $F$ and $H$ satisfying the following three conditions:

$$F^T \Sigma H \neq 0 , \qquad H^T \Sigma \Pi = 0 , \qquad F^T \Sigma \Pi = 0 . \tag{18}$$

Once a pair of fluxes satisfying these conditions has been specified, to obtain a flux vacuum one must still find a solution to the remaining $h^{2,1}$ equations :

$$\partial_{\varphi^i} \left[ (F - \tau H)^T \Sigma \Pi \right] = 0 , \tag{19}$$

which acts as a contraint on the moduli $\varphi^i$ and $\tau$. Note that these equations are overconstrained, and as such they have generically no solutions.

#### 3.2. Indicial algebras and the period vector

The period vector furnishes a basis of solutions to the Picard-Fuchs equation. To build a basis of solutions around a large complex structure point, one can begin with the unique holomorphic series solution $\varpi^0$, which has an expansion

$$\varpi^0 = \sum_{\boldsymbol{p} \in \mathbb{Z}_{\geqslant 0}^m} c(\boldsymbol{p}) \boldsymbol{\varphi}^{\boldsymbol{p}} ,$$

where the sum runs over over $m$-component multi-indices $\boldsymbol{p}$ whose components are all non-negative, and $\boldsymbol{\varphi} = (\varphi^1, \dots, \varphi^m)$ are coordinates in the complex structure moduli space. To obtain the other periods, we follow the process of [59] and first define a modified fundamental period

$$\varpi^{\boldsymbol{\epsilon}} = \sum_{\boldsymbol{p} \in \mathbb{Z}_{\geqslant 0}^m} \frac{c(\boldsymbol{p} + \boldsymbol{\epsilon})}{c(\boldsymbol{\epsilon})} \boldsymbol{\varphi}^{\boldsymbol{p}+\boldsymbol{\epsilon}} = \varpi^0 + \varpi^{\{i\}} \epsilon_i + \frac{1}{2} \varpi^{\{i,j\}} \epsilon_i \epsilon_j + \frac{1}{6} \varpi^{\{i,j,k\}} \epsilon_i \epsilon_j \epsilon_k + \mathcal{O}(\boldsymbol{\epsilon}^4) .$$

The functions $\varpi^{\{i\}}$, $\varpi^{\{i,j\}}$, $\varpi^{\{i,j,k\}}$, can be computed by differentiating the closed form for the coefficients $c(\boldsymbol{p})$, obtainable, for example, by Dwork-Griffiths-Katz reduction. We work in the ring where the product of any four $\epsilon_i$ vanishes, and introduce the following definitions and relations, requiring that the $\epsilon_i$ give a representation of the homology algebra of the mirror manifold $\widetilde{X}$:

$$\epsilon_i \epsilon_j = Y_{ijk} \mu^k , \qquad \epsilon_i \mu^j = \delta_i^j \eta , \qquad \epsilon_i \eta = 0 . \tag{20}$$

The numbers $Y_{ijk}$ appearing in (20) are the triple intersection numbers of two-cycles on the mirror Calabi-Yau manifold $\widetilde{X}$:

$$Y_{ijk} = \int_{\widetilde{X}} e_i \wedge e_j \wedge e_k ,$$

where the $e_i$ form a basis for the second integral cohomology of $\widetilde{X}$.

In terms of the algebra elements $\epsilon_i, \mu^i$ and $\eta$, we can rewrite

$$\varpi^{\boldsymbol{\epsilon}} = \varpi^0 + \varpi^i \epsilon_i + \varpi_i \mu^i + \varpi_0 \eta , \tag{21}$$

and one can read off a basis of $2m+2$ solutions to the Picard-Fuchs system $\varpi = (\varpi^0, \varpi^i, \varpi_i, \varpi_0)$ where the components besides $\varpi^0$ are

$$\varpi^i = \varpi^{\{i\}} , \qquad \varpi_i = \frac{1}{2} Y_{ijk} \varpi^{\{j,k\}} , \qquad \varpi_0 = \frac{1}{6} Y_{ijk} \varpi^{\{i,j,k\}} .$$

The basis of solutions $\varpi$ so described is termed the *Frobenius basis*. From this one can obtain the *complex Frobenius basis*, given by

$$\widehat{\varpi} = \nu^{-1} \varpi , \qquad \nu = \mathrm{diag}(1, 2\pi \mathrm{i} \mathbf{1}, (2\pi \mathrm{i})^2 \mathbf{1}, (2\pi \mathrm{i})^3) .$$

The period vector in the integral basis is given by

$$\Pi = \rho \nu^{-1} \varpi , \tag{22}$$

where $\rho$ is the matrix

$$\rho = \begin{pmatrix} -\frac{1}{3} Y_{000} & -\frac{1}{2} \boldsymbol{Y}_{00}^T & \mathbf{0}^T & 1 \\ -\frac{1}{2} \boldsymbol{Y}_{00} & -\mathbb{Y}_0 & -\mathbb{1} & \mathbf{0} \\ 1 & \mathbf{0}^T & \mathbf{0}^T & 0 \\ \mathbf{0} & \mathbb{1} & \mathbb{0} & \mathbf{0} \end{pmatrix} . \tag{23}$$

The vector $\boldsymbol{Y}_{00}$ has components defined in terms of the second Chern class $c_2$ of $\widetilde{X}$:

$$(\boldsymbol{Y}_{00})_i = Y_{00i} \stackrel{\mathrm{def}}{=} -\frac{1}{12} \int_{\widetilde{X}} c_2 \wedge e_i .$$

The number $Y_{000}$ is given by an integral involving the third Chern class, which evaluates to

$$Y_{000} = -\frac{3\chi(\widetilde{X})\zeta(3)}{(2\pi \mathrm{i})^3} = \frac{3\chi(X)\zeta(3)}{(2\pi \mathrm{i})^3} .$$

In order to explain the entries of the matrix $(\mathbb{Y}_0)_{ij} = Y_{0ij}$, we must recall some arguments from mirror symmetry following [59]. The complexified Kähler moduli space of $\widetilde{X}$ has the homogeneous coordinates $z^a$, $a = 0, \ldots, m = h^{1,1}(\widetilde{X})$, in terms of which can express the affine flat coordinates as $t^i = z^i/z^0$, $i = 1, \ldots, m$. The integral symplectic period vector $\Pi(\boldsymbol{t})$ is given in terms of the prepotential $\mathcal{G}$ by

$$\Pi = \begin{pmatrix} \frac{\partial}{\partial z^0} \mathcal{G} \\ \frac{\partial}{\partial z^i} \mathcal{G} \\ z^0 \\ z^0 t^i \end{pmatrix} , \qquad \mathcal{G} = -\frac{1}{6} Y_{abc} \frac{z^a z^b z^c}{z^0} + (z^0)^2 \sum_{\boldsymbol{p} \neq \boldsymbol{0}} n_{\mathfrak{p}} \mathrm{Li}_3(\boldsymbol{q}^{\boldsymbol{p}}) , \qquad q_i \stackrel{\mathrm{def}}{=} \exp(2\pi \mathrm{i} t^i) , \tag{24}$$

where $n_{\boldsymbol{p}}$ are the instanton numbers counting rational curves on $\widetilde{X}$, and the sum is taken over all positive $m$-component multi-indices $\boldsymbol{p}$.

The mirror map identifies $z^a = \widehat{\varpi}^a$. By comparing asymptotics as $t^i \to \infty$ the period vector in (24) is seen to be $z^0$ times the period vector computed via (22). The quantities $Y_{ijk}$, $Y_{00i}$, $Y_{000}$ are as we have given them, and the numbers $Y_{0ij}$ are fixed, up to shifting by integers following a symplectic transformation, by requiring that the monodromies $t^i \mapsto t^i + 1$ around the large complex structure point be given by an action of an integral symplectic matrix on the period vector $\Pi(\boldsymbol{t})$.

### 3.3. Solutions from $\mathbb{Z}_2$ symmetry

Having chosen coordinates $\boldsymbol{\varphi}$ on the parameter space[6] so as to obtain the periods in the standard Frobenius form, we concentrate on parameter spaces which are symmetric under $\varphi_i \leftrightarrow \varphi_j$ for a fixed pair $(i, j)$, meaning that the manifolds corresponding to the two points related by this operation are biholomorphic and the following equalities hold:

$$\widehat{\varpi}^i(\boldsymbol{\varphi}) \;=\; \widehat{\varpi}^j(\boldsymbol{\varphi}') \,, \qquad \widehat{\varpi}_i(\boldsymbol{\varphi}) \;=\; \widehat{\varpi}_j(\boldsymbol{\varphi}') \,. \tag{25}$$

Here $\boldsymbol{\varphi}'$ is obtained by acting on $\boldsymbol{\varphi}$ by a $\mathbb{Z}_2$ permutation $\mathfrak{A}_{i,j}$ defined by

$$\mathfrak{A}_{i,j} \,:\, \mathbb{C}^m \to \mathbb{C}^m \,:\quad \varphi^i \mapsto \varphi^j \,, \qquad \varphi^j \mapsto \varphi^i \,, \qquad \varphi_s \mapsto \varphi_s \text{ for } s \neq i, j$$

Additionally, this symmetry is reflected in the quantities $Y_{abc}$, as it in particular implies that two pairs of components of $\Pi$ are similarly exchanged.

When the vector $\widehat{\varpi}$ has the symmetry (25), it is possible to solve the supersymmetric flux vacuum equations (18) and (19) by choosing

$$H \;=\; (0, \boldsymbol{\delta}_i - \boldsymbol{\delta}_j, 0, \boldsymbol{0})^T \,, \qquad F \;=\; (0, \boldsymbol{0}, 0, \boldsymbol{\delta}_i - \boldsymbol{\delta}_j)^T \,,$$

imposing $\varphi^i = \varphi^j$, which is the fixed point of the involution $\mathfrak{A}$, but leaving the remaining $\varphi^s$ free, and giving $\tau$ the value

$$\tau(\boldsymbol{\varphi}) \;=\; -\frac{F^T \Sigma \left(\partial_{\varphi^i} - \partial_{\varphi^j}\right)\Pi}{H^T \Sigma \left(\partial_{\varphi^i} - \partial_{\varphi^j}\right)\Pi}\bigg|_{\varphi_i = \varphi_j} \;=\; \frac{\mathrm{i}}{2\pi} \frac{\partial_{\varphi^i} \left(\varpi_i - \varpi_j\right)}{\partial_{\varphi^i} \left(\varpi^i - \varpi^j\right)}\bigg|_{\varphi^i = \varphi^j} + Y_{0ij} - Y_{0ii} \,. \tag{26}$$

Note that the first term on the right-hand side above is purely imaginary for real moduli, and the remaining two terms give the real part of $\tau$. It is perhaps interesting that for real moduli the real part of the axiodilaton should only take the values $0$ or $\frac{1}{2}$, subject to the topological numbers $Y_{ijk}$ of the compactification manifold.

Later, in sections 6 and 7, we give several examples of manifolds with the property (25). We work with the mirrors of favourable CICY manifolds [60] whose configuration matrices are invariant under the exchange of two rows. "Favourable" means that the second cohomology is generated by the hyperplane classes of the factor spaces, which has the consequence that $h^{1,1}$ equals the number of rows in the configuration matrix. The periods near the large complex structure point in the moduli space can be written for example as integral expressions valid in certain patches, which allow us to give formulae for $\tau$ as a ratio of integrals of products of special functions. After some numerical work, these give rise to expressions for $j\big(\tau(\boldsymbol{\varphi})\big)$ as a rational functions of $\boldsymbol{\varphi}$.

---

[6]We differentiate between parameter space and moduli space in this paper. The latter involves a quotient by symmetries to obtain a space where different points correspond to varieties which are not biholomorphic.

## 4. Calabi-Yau Modularity

The modularity of Calabi-Yau manifolds can be seen as an aspect of the Langlands program, a highly-influential and far-reaching set of conjectures on relations between number theory, geometry, and representation theory. For an introductory exposition, see for example [61], and for a physicist-oriented review, see [62]. We give a very brief overview of the ideas relevant to Calabi-Yau modularity, focusing on the case of two-dimensional representations, which is the most relevant case for the rest of the paper. This overview is meant to provide some intuition behind the relation between modular forms and the zeta function, while making no attempt at rigour.

A special case of the correspondence can be stated [62] as a relation between two-dimensional representations of the *absolute Galois group* $\text{Gal}(\bar{\mathbb{Q}}/\mathbb{Q})$ and the space of Hecke eigenforms (see (76) for definition). Further, this correspondence should relate the eigenvalues of an element of the representation associated to the Frobenius map (33), and the Hecke eigenvalues of the Hecke eigenforms. Diagramatically,

$$\text{2-dimensional representations of } \text{Gal}(\bar{\mathbb{Q}}/\mathbb{Q}) \quad \longleftrightarrow \quad \text{Modular forms,}$$
$$\text{Frobenius eigenvalues} \quad \longleftrightarrow \quad \text{Hecke eigenvalues.}$$

The Galois representations arise naturally for manifolds $X(\mathbb{F}_q)$ defined over finite fields. The absolute Galois group acts on a certain cohomology theory[7] $H^3(X, \mathbb{Q}_\ell)$, which is a vector space over the field of $\ell$-adic numbers. This natural action of $\text{Gal}(\bar{\mathbb{Q}}/\mathbb{Q})$ on $H^3(X, \mathbb{Q}_\ell)$, defines a representation $\rho$ of the absolute Galois group.

If the middle cohomology group has a two-dimensional piece in a sense that will be made precise later, as is the case with elliptic curves, then the Langlands philosophy suggests that there should be a correspondence between the Frobenius eigenvalues and the Hecke eigenvalues of a modular form. In this paper we work with Calabi-Yau threefolds, where the middle cohomology is $H^3(X)$. For instance, it has been shown that *rigid* Calabi-Yau threefolds, for which the third cohomology group is two-dimensional, are modular of weight four [10]. It can also happen that the representation of $\text{Gal}(\bar{\mathbb{Q}}/\mathbb{Q})$ is reducible with one of the irreducible pieces being two-dimensional. This is the notion of modularity for the Calabi-Yau manifolds that we study: we look for Calabi-Yau threefolds whose Hodge structure splits, by which we mean that there exist two subspaces $V, V' \subset H^3(X, \mathbb{Q})$ with definite Hodge type, such that the third de Rham cohomology group can be written as a direct sum

$$H^3(X, \mathbb{Q}) \; = \; V \oplus V' \,. \tag{27}$$

It can then be shown that $H^3(X, \mathbb{Q}_\ell)$ has a similar split, and consequently the representation of the absolute Galois group reduces compatibly with this split. This process, which we briefly review, has been explained in detail in [27, 32].

Let $X(\mathbb{K})$ be a Calabi-Yau manifold defined over a number field $\mathbb{K}$. If there is a split of type (27), then there exists a number field $\mathbb{E} \supseteq \mathbb{K}$, such that the étale cohomology of $X(\mathbb{E})$ splits

$$H^3_{\text{ét}}(X(\mathbb{E}), \mathbb{Q}_\ell) \; = \; U \oplus U' \,,$$

where $U$ and $U'$ have the same dimensions and Hodge types as $V$ and $V'$, respectively. This in turn leads to reduction of the representation $\rho$ of the absolute Galois group:

$$\rho(X) \; = \; \rho(U) \oplus \rho(U') \,. \tag{28}$$

---

[7]The reader familiar with Langlands program will recognise this as the étale cohomology. However, comparison theorems show that for many purposes different cohomology theories are equivalent. Thus, in the present work, we need not use the language of étale cohomologies.

Modularity is most simply understood for varieties defined over $\mathbb{Q}$. In this case the two-dimensional subspaces with the Hodge type $(2,1)+(1,2)$ correspond to weight-two modular forms, whereas the spaces with Hodge type $(3,0)+(0,3)$ correspond to weight-four modular forms [27].

The modular forms associated to representations over field extensions of $\mathbb{Q}$ are more complicated. In [27], it was conjectured, in the spirit of the Langlands correspondence, that when the extension field $\mathbb{E}$ is a real quadratic field, the corresponding modular forms are Hilbert modular forms, instead of 'classical' elliptic modular forms. In §6 and §7, where we investigate supersymmetric flux vacua, we are indeed able to find significant evidence for this conjecture.

### 4.1. Elliptic curves over $\mathbb{Q}$

To illustrate some of the central ideas of modularity in their simplest setting, let us consider elliptic curves defined over $\mathbb{Q}$. Every such elliptic curve can be written in the Weierstrass form

$$ y^2 + a_1 xy + a_3 y \; = \; x^3 + a_2 x^2 + a_4 x + a_6 \; , $$

with $a_i \in \mathbb{Z}$. Since there is a natural map $\mathbb{Z} \to \mathbb{F}_{p^n}$, given by reduction mod $p$, this equation can also be seen as defining a variety $E(\mathbb{F}_{p^n})$ over finite fields $\mathbb{F}_{p^n}$. By $E(\mathbb{F}_{p^n})$ we mean the set of points $x \in \mathbb{F}_{p^n}$ satisfying the above equation, taking into account precisely one point 'at infinity', that does not lie on this affine patch. It is in principle straightforward to count the number of points on $E(\mathbb{F}_{p^n})$. Denoting this quantity $N_p^n(E)$, it is consequence of modularity theorem [5, 6, 63] that the quantities $c_p$, defined by

$$ c_p = p + 1 - N_p(E) \; , \tag{29} $$

are Fourier coefficients of a modular form, multiplying $q^p$ in the Fourier expansion.

To formulate this correspondence in terms that generalise better, we need to consider the *local zeta function* of the elliptic curve. It is defined as a generating function for the point counts $N_{p^n}(E)$:

$$ \zeta_p(T) \; = \; \exp\left( \sum_{n=1}^{\infty} \frac{N_{p^n}(E) T^n}{n} \right) \; . $$

The *Weil conjectures* [64] (which have been proved, see §4.2) can be used to show that this is, remarkably, a rational function of $T$,

$$ \zeta_p(T) \; = \; \frac{1 - c_p T + p T^2}{(1-T)(1-pT)} \; . $$

From the set of numerators of these zeta functions for each prime $p$, one can define a *Hasse-Weil L-function*

$$ L(E, s) \; = \; \prod_{p \text{ good}} \frac{1}{1 - c_p p^{-s} + p^{1-2s}} \prod_{p \text{ bad}} \frac{1}{1 \pm p^{-s}} \; . $$

Here the first product runs over the *primes of good reduction* $p$ for which the variety $E(\mathbb{F}_p)$ is smooth, and the second product is over *primes of bad reduction* $p$ for which $E(\mathbb{F}_p)$ has a singularity. The sign in the second product is chosen based on whether $E$ has a *split* or *non-split multiplicative reduction* at $p$ (for explanation of these terms, see for example [65]). The primes of bad reduction can also be used to define the *conductor*, an integer divisible precisely by the primes of bad reduction, although they may appear with multiplicity, according to whether the reduction is split or non-split multiplicative.

It is also possible to construct a *Hecke L-function* from a modular form. Consider a cusp form $f$ of a congruence subgroup $\Gamma_0(N)$ with Fourier expansion

$$f(\tau) = \sum_{n=1}^{\infty} \alpha_n q^n \ . \tag{30}$$

Then we define the associated L-function as

$$L(f,s) = \sum_{n=1}^{\infty} \frac{\alpha_n}{n^s} \ .$$

If $f$ belongs to a special class of cusp forms, known as Hecke eigenforms, which we briefly review in appendix A, then we can write the L-function in the suggestive form

$$L(f,s) = \prod_{p \nmid N} \frac{1}{1 - \alpha_p p^{-s} + p^{1-2s}} \prod_{p \mid N} \frac{1}{1 - \alpha_p p^{-s}} \ .$$

The modularity theorem states that for any elliptic curve $E$ defined over $\mathbb{Q}$ there exists a Hecke eigenform $f$ of level $N$, divisible only by the bad primes of $E$, and of weight 2, such that

$$L(E,s) = L(f,s) \ .$$

*An example of the correspondence between elliptic curves and modular forms*

To illustrate the previous discussion, take the elliptic curve with label **20.a3** in the L-functions and modular forms database (LMFDB) [43],

$$y^2 = x^3 + x^2 - x \ ,$$

Consider solutions over $\mathbb{F}_p \cong \mathbb{Z}/p\mathbb{Z}$. The primes of bad reduction are $p = 2, 5$ [43], and the conductor of this curve is given by $N = 2^2 \cdot 5 = 20$. We can count the number of solutions over $\mathbb{F}_p$, and use the relation (29) to tabulate the values of $c_p$ for the first ten primes in table 2.

| $p$ | 2 | 3 | 5 | 7 | 11 | 13 | 17 | 19 | 23 | 29 |
|-----|---|---|---|---|----|----|----|----|----|----|
| $c_p$ | 0 | $-2$ | $-1$ | 2 | 0 | 2 | $-6$ | $-4$ | 6 | 6 |

Table 2: *The point counts $c_p$ computed from the number of points on elliptic curves defined over finite fields $\mathbb{F}_p$ for the first ten primes.*

In accordance with the modularity theorem [5, 6], there is a modular form for $\Gamma_0(20)$ of weight 2 with LMFDB label **20.2.a.a**, whose Fourier expansion is given by

$$f(q) = q - \mathbf{2}q^3 - \mathbf{1}q^5 + \mathbf{2}q^7 + q^9 + \mathbf{2}q^{13} + 2q^{15} - \mathbf{6}q^{17} - \mathbf{4}q^{19} - 4q^{21} + \mathbf{6}q^{23} + q^{25} + 4q^{27} + \mathbf{6}q^{29} + \dots \ .$$

Note the agreement of the coefficients of $q^p$ with the point counts $c_p$, which persists to higher orders, as can be computationally verified. However, it is important to note that this modular form is not solely associated to the curve **20.a3**. There are three curves defined over $\mathbb{Q}$ related to **20.a3** by *isogenies* (see §4.5), **20.a1**, **20.a2**, and **20.a4**. Isogenies have the important property that they preserve point counts, thus isogenous curves are associated to the same modular form. For this reason, the point counts $c_p$ and the zeta function are naturally associated to an isogeny class consisting of all curves isogenous over $\mathbb{Q}$ (or an extension field if we are considering varieties over field extensions of $\mathbb{Q}$) rather than to a single elliptic curve. Specific curves within isogeny classes can be fixed by specifying additional data such as the $j$-invariant, which we use to pick a specific curve in the isogeny class in our setting. By abuse of language we refer to such a curve as *the* curve $\mathcal{E}_R$ associated to the zeta function.

### 4.2. The local zeta function

The reduction of the Galois representation (28) can be seen in the structure of the local zeta functions. We briefly review some of the important properties of the zeta function before discussing the factorisations related to the reduction of the Galois representation.

Let $X_{\boldsymbol{\varphi}}$ be an $m$-parameter family of Calabi-Yau manifolds defined over the rational numbers, parametrised by the moduli $\boldsymbol{\varphi} = (\varphi_1, \ldots, \varphi_m)$. Given the natural maps $\mathbb{Z} \to \mathbb{F}_{p^n}$, we can think of $X_{\boldsymbol{\varphi}}$ also as a family of manifolds over these finite fields. To avoid any confusion, where needed, we will denote the manifold $X_{\boldsymbol{\varphi}}$ defined over field $\mathbb{K}$ by $X_{\boldsymbol{\varphi}}(\mathbb{K})$.

Denoting the number of points on a manifold $X_{\boldsymbol{\varphi}}(\mathbb{F}_{p^n})$ by $N_{p^n}(\boldsymbol{\varphi})$, we define

$$\zeta_p(\boldsymbol{\varphi}; T) \;=\; \exp\left(\sum_{n=1}^{\infty} \frac{N_{p^n}(\boldsymbol{\varphi}) T^n}{n}\right).$$

The Weil conjectures, originally stated by Weil [64], and later proven by Dwork [66], Grothendiek [67], and Deligne [68, 69], can be stated as:

1. **Rationality**: $\zeta_p(X_{\boldsymbol{\varphi}}, T)$ is a rational function of $T$ of the form

$$\zeta_p(X_{\boldsymbol{\varphi}}, T) \;=\; \frac{P_1(X_{\boldsymbol{\varphi}}, T) P_3(X_{\boldsymbol{\varphi}}, T) \ldots P_{2d-1}(X_{\boldsymbol{\varphi}}, T)}{P_0(X_{\boldsymbol{\varphi}}, T) P_2(X_{\boldsymbol{\varphi}}, T) \ldots P_{2d}(X_{\boldsymbol{\varphi}}, T)} \;,$$

   where $P_i(X_{\boldsymbol{\varphi}}, T)$ is a polynomial in $T$ with integer coefficients whose degree is given by the Betti number $b_i$, and $d$ denotes the complex dimension of $X_{\boldsymbol{\varphi}}(\mathbb{C})$.

2. **Functional equation**: $\zeta_p(X, T)$ satisfies

$$\zeta_p\left(X_{\boldsymbol{\varphi}}, p^{-d} T^{-1}\right) \;=\; \pm p^{d\chi/2} T^{\chi} \zeta_p(X_{\boldsymbol{\varphi}}, T) \;, \tag{31}$$

   where $\chi$ is the Euler characteristic of $X_{\boldsymbol{\varphi}}$.

3. **Riemann hypothesis**: The polynomials $P_i(X_{\boldsymbol{\varphi}}, T)$ factorise over $\mathbb{C}$ as

$$P_i(X_{\boldsymbol{\varphi}}, T) \;=\; \prod_{j=1}^{b_j} (1 - \lambda_{ij}(X_{\boldsymbol{\varphi}}) T) \;,$$

   where the $\lambda_{ij}(X_{\boldsymbol{\varphi}})$ are algebraic integers of absolute value $p^{i/2}$.

The $m$-parameter Calabi-Yau threefolds have Hodge diamonds given by

$$
h^{p,q}(X_{\boldsymbol{\varphi}}) \;=\;
\begin{array}{ccccccc}
 & & & 1 & & & \\
 & & 0 & & 0 & & \\
 & 0 & & h^{1,1} & & 0 & \\
1 & & m & & m & & 1 \\
 & 0 & & h^{1,1} & & 0 & \\
 & & 0 & & 0 & & \\
 & & & 1 & & &
\end{array} \;.
$$

In this case, the form of the zeta function simplifies greatly, being determined by the degree-$(2m+2)$ polynomial $R(X_{\boldsymbol{\varphi}}, T) = P_3(X_{\boldsymbol{\varphi}}, T)$. This is due to the fact that, as $b_1 = b_5 = 0$, the polynomials

$P_1$ and $P_5$ are trivial, and when the Picard group is generated by divisors that are defined over $\mathbb{F}_p$, the zeta function takes the simple form:

$$\zeta_p(X_{\boldsymbol{\varphi}}, T) = \frac{R(X_{\boldsymbol{\varphi}}, T)}{(1-T)(1-pT)^{h^{11}}(1-p^2T)^{h^{11}}(1-p^3T)} \ . \tag{32}$$

As reviewed in detail in [24] (see also [70] for a physics-oriented exposition, or [71] for a physicist-friendly mathematical treatment), the polynomial $R(X_{\boldsymbol{\varphi}}, T)$ can be computed explicitly using the periods of the Calabi-Yau manifold. Roughly speaking, the relation to the periods is due to the fact that the polynomial $R(X_{\boldsymbol{\varphi}}, T)$ is the characteristic polynomial of the inverse *Frobenius map* acting on the third cohomology.

Denote by $\text{Frob}_p$ the Frobenius map

$$\mathbb{K}^k \to \mathbb{K}^k \ : \ \boldsymbol{x} = (x_1, \ldots, x_k) \mapsto (x_1^p, \ldots, x_k^p) \ = \ \boldsymbol{x}^p \ .$$

Recall Fermat's little theorem: for any prime $p$ and a non-zero integer $a$, $a^p \equiv a \mod p$. As a consequence of this fact, $\text{Frob}_p$ fixes the elements of $\mathbb{F}_p^k \subset \mathbb{K}^k$. Analogously, we define the Frobenius maps $\text{Frob}_{p^n}$ by

$$\text{Frob}_{p^n} = \text{Frob}_p^n \ : \ \boldsymbol{x} \mapsto \boldsymbol{x}^{p^n} \ , \tag{33}$$

which fix the elements of $\mathbb{F}_{p^n}^k \subset \mathbb{K}^k$. Importantly, this action maps the points on a manifold $X(\mathbb{F}_{p^n})$ to other points on the manifold. To see this, consider a manifold locally defined as a locus $P(\boldsymbol{x}) = 0$, with $P$ having coefficients in $\mathbb{F}_p$. Then, as the Frobenius map fixes the elements of $\mathbb{F}_{p^n}$, we have that $\text{Frob}_{p^n}(\boldsymbol{x}) = \boldsymbol{x}^{p^n}$ also satisfies the equation

$$0 \ = \ P(\boldsymbol{x})^{p^n} \ = \ P\left(\boldsymbol{x}^{p^n}\right) \ .$$

As explained in [24], it is possible to define corresponding Frobenius maps $\text{Fr}_{p^n}$, which act on certain $p$-adic cohomology groups $H^m(X_{\boldsymbol{\varphi}})$:

$$\text{Fr}_{p^n} : H^m(X_{\boldsymbol{\varphi}}) \to H^m(X_{\boldsymbol{\varphi}}) \ . \tag{34}$$

Using the Lefschetz fixed point theorem, which applies in this situation, it is then possible to count the number of points on manifolds $X_{\boldsymbol{\varphi}}(\mathbb{F}_{p^n})$ using traces of this map,

$$N_{p^n}(\boldsymbol{\varphi}) \ = \ \sum_{m=0}^{6} (-1)^m \, \text{Tr}\left(\text{Fr}_{p^n} \big| H^m(X_{\boldsymbol{\varphi}})\right) \ .$$

Furthermore, the characteristic polynomial of the inverse Frobenius map gives the polynomial $R(X_{\boldsymbol{\varphi}}, T)$:

$$R(X_{\boldsymbol{\varphi}}, T) \ = \ \det\left(\mathbb{I} - T\,\text{Fr}_p^{-1} \big| H^3(X_{\boldsymbol{\varphi}})\right) \ . \tag{35}$$

## 4.3. Factorisations of the local zeta function

The Frobenius map can be associated to an element of the Galois representation [62], so the reduction of the Galois representation (28) can be seen in the action of $\text{Fr}_p$, and hence in the structure of the zeta function by the relation (35). When the representation $\rho$ reduces, the matrices F and U which represent the actions of $\text{Fr}_p$ and $\text{Fr}_p^{-1}$ on $H^3(X_{\boldsymbol{\varphi}})$ are block diagonal in a suitable basis. This immediately implies that the characteristic polynomial $R(X_{\boldsymbol{\varphi}}, T)$ will factorise over

| smooth/sing. | $p$ | $R(X,T)$ | $c_p(3)$ |
|:---:|:---:|:---:|:---:|
| smooth | 7 | $\left(p^3T^2 - 2pT + 1\right)^4 \left(p^6T^4 + 2p^3T^3 - 54pT^2 + 2T + 1\right)$ | 2 |
| smooth | 11 | $\left(p^3T^2 + 1\right)^4 \left(p^3T^2 + 1\right) \left(p^3T^2 + 28T + 1\right)$ | 0 |
| singular | 13 | | — |
| smooth | 17 | $\left(p^3T^2 + 6pT + 1\right)^4 \left(p^6T^4 + 98p^3T^3 + 674pT^2 + 98T + 1\right)$ | −6 |
| smooth | 19 | $\left(p^3T^2 - 2pT + 1\right)^4 \left(p^6T^4 - 8p^3T^3 + 242pT^2 - 8T + 1\right)$ | 2 |
| smooth | 23 | $\left(p^3T^2 + 1\right)^4 \left(p^6T^4 - 68p^3T^3 + 98pT^2 - 68T + 1\right)$ | 0 |
| smooth | 29 | $\left(p^3T^2 + 6pT + 1\right)^4 \left(p^6T^4 - 6p^3T^3 + 722pT^2 - 6T + 1\right)$ | −6 |
| smooth* | 31 | | — |
| singular | 37 | | — |
| smooth | 41 | $\left(p^3T^2 + 12pT + 1\right)^4 \left(p^6T^4 - 230p^3T^3 + 2018pT^2 - 230T + 1\right)$ | −12 |
| smooth | 43 | $\left(p^3T^2 + 4pT + 1\right)^4 \left(p^6T^4 + 94p^3T^3 + 2186pT^2 + 94T + 1\right)$ | −4 |
| smooth | 47 | $\left(p^3T^2 + 1\right)^4 \left(p^3T^2 + 1\right) \left(p^3T^2 + 136T + 1\right)$ | 0 |

Table 3: *We tabulate here, for the first few primes p, the numerators of the zeta functions $\zeta_p$ of the family of $S_5$ symmetric Hulek-Verrill manifolds, defined in detail in §6, with $\varphi = 3$. As expected based on the symmetry, the numerators always factorise into four degree two pieces and into one degree four piece, which may exceptionally factorise further. One can read the coefficients $c_p(3)$ easily from the degree two piece, and these are also given explicitly for convenience.*

$\mathbb{Z}$ into a product of the characteristic polynomials of the blocks, corresponding to the irreducible representations.

The flux vacuum solutions found in §3 provide two integral vectors $F, H \in H^{2,1}(X_{\varphi}, \mathbb{Z}) \oplus H^{1,2}(X_{\varphi}, \mathbb{Z})$, which in turn correspond to two-dimensional piece of a split of the type (27). This implies that the characteristic polynomial $R(X_{\varphi}, T)$ appearing as the zeta function numerator factorises so that there is at least one quadratic factor $R_2(T)$ dividing $R(X_{\varphi}, T)$. The Weyl conjectures dictate that this factor will have the form

$$R_2(T) \ = \ 1 - c_p(\varphi)pT + p^3T^2 \ , \tag{36}$$

where we have written $c_p$ as a function of $\varphi$ to emphasise that it depends on the choice of the complex structure of the manifold as well as on the prime $p$.

The zeta function, and thus this quadratic polynomial, can be found using the techniques of [26] (see chapter 9), where a procedure for numerically computing zeta functions of multiparameter Calabi-Yau manifold is given. Adapted to the present case, this allows us to compute the polynomial $R(T)$ for the first few primes with the computational capacity being the bottleneck. In practice, we will compute the polynomials for the first 15 primes for the two-parameter models, and for the first 100 primes for the Hulek-Verrill manifold. This will be enough for our purposes, as it easily allows not only identifying the modular forms related to the zeta function, but also conducting highly non-trivial consistency checks on these results.

Based on the discussion in the beginning of this section, it is natural to assume that in the cases where the local zeta functions $\zeta_p(X, T)$ factorise for (almost) every prime, this factorisation arises from a reduction of a Galois representation, where one of the irreducible pieces is two-dimensional. Then Serre's modularity conjecture [20, 21] (which has been proved [22, 23]) asserts that such a representation is associated to a modular form. Namely, the coefficients $c_p(\varphi)$ appearing in the quadratic polynomial $R_2(T)$ should be Hecke eigenvalues of a Hecke eigenform, at least when $p$ is not a prime of bad reduction.

*An example: the $S_5$ symmetric Hulek-Verrill manifold with $\varphi = 3$*

Using the zeta functions computed for the models we study, we can fix the value of $\varphi$ and directly compare the coefficients $c_p(\varphi)$ to the modular form data collected in the L-functions and Modular Forms Database LMFDB [43]. For instance, in the case of the $S_5$ symmetric locus of the five-parameter Hulek-Verrill manifold we present in more detail in §6, taking $\boldsymbol{\varphi} = (\varphi, \ldots, \varphi)$ with $\varphi = 3$, we can find the relevant polynomials $R(X_{\boldsymbol{\varphi}}, T)$ among the zeta function data collected in appendix B. For convenience, the relevant polynomials are also gathered into table 3.

We can then directly read off the coefficients $c_p(3)$ from the quadratic piece, which can then be compared to the modular form data available in LMFDB. It turns out that there is exactly one modular form listed in LMFDB that has these coefficients. The modular form has label **156.2.a.b**, and has the Fourier expansion

$$\begin{aligned} f(q) &= q + q^3 + \mathbf{2}q^7 + q^9 + \mathbf{1}q^{13} - \mathbf{6}q^{17} + \mathbf{2}q^{19} + 2q^{21} - 5q^{25} + q^{27} - \mathbf{6}q^{29} + \mathbf{2}q^{31} + \mathbf{2}q^{37} + q^{39} \\ &\quad - \mathbf{12}q^{41} - \mathbf{4}q^{43} - 3q^{49} - 6q^{51} + \mathbf{6}q^{53} + 2q^{57} + \mathbf{12}q^{59} + \mathbf{2}q^{61} + 2q^{63} - \mathbf{10}q^{67} + \mathbf{12}q^{71} \\ &\quad + \mathbf{14}q^{73} - 5q^{75} + \mathbf{8}q^{79} + q^{81} + \mathbf{12}q^{83} - 6q^{87} + 2q^{91} + 2q^{93} - \mathbf{10}q^{97} + \ldots \end{aligned}$$

We can provide further evidence that the matching between the Fourier coefficients of this Hecke eigenform and the coefficients $c_p(3)$ arising from the zeta function is not accidental by computing the coefficients $c_p(3)$ for higher primes and comparing them to further Hecke eigenvalues of **156.2.a.b**. Doing this, we find continued agreement. For the first few primes, this is illustrated in table 4.

| $p$ | 7 | 11 | 13 | 17 | 19 | 23 | 29 | 31 | 37 | 41 | 43 | 47 | 53 | 59 | 61 | 67 | 71 | 73 | 79 | 83 | 89 | 97 |
|---|---|---|---|---|---|---|---|---|---|---|---|---|---|---|---|---|---|---|---|---|---|---|
| $c_p(3)$ | 2 | 0 | 1 | -6 | 2 | 0 | -6 | 2 | 2 | -12 | -4 | 0 | 6 | 12 | 2 | -10 | 12 | 14 | 8 | 12 | 0 | -10 |
| $\alpha_p$ | 2 | 0 | 1 | -6 | 2 | 0 | -6 | 2 | 2 | -12 | -4 | 0 | 6 | 12 | 2 | -10 | 12 | 14 | 8 | 12 | 0 | -10 |

Table 4: *The zeta functions coefficients $c_p(3)$ and the Hecke eigenvalues $\alpha_p$ for primes $7 \leqslant p < 100$. Here we have also included the singularities and apparent singularities by treating these cases more carefully.*

We can repeat the search for various values of $\varphi$, and should always be able to find a corresponding modular form. When $\varphi \in \mathbb{Q}$, these are elliptic modular forms, but also other types of modular forms will appear. For instance, we will have Hilbert or Bianchi modular forms appearing when $\varphi$ belongs to a real or imaginary quadratic field, respectively.

## 4.4. Relating flux vacua and modular elliptic curves

The search for modular forms corresponding to a quadratic factor of the zeta function numerator is made easier by using the intuition provided by the physical flux vacuum construction that allowed us to find the families of modular Calabi-Yau manifolds in the first place. Namely, by modularity, weight-two modular forms are associated to an elliptic curve $\mathcal{E}_R$, and in IIB theory there is a complex

scalar field, the axiodilaton $\tau$, which in F-theory constructions is identified with the elliptic fibre of an elliptically fibred fourfold. It is natural to speculate that these two curves could be identified. Indeed, in the case of flux vacua on the locus $\psi = 0$ of the two-parameter family of mirror octic threefolds, in [27, 28] evidence was provided in support of the conjecture that the elliptic curve $\mathcal{E}_R$ corresponding to the quadratic factor $R_2(T)$ is complex-isomorphic to the F-theory fibre $\mathcal{E}_F$. In fact, at least in the cases we have studied and in agreement with the analysis of [28], it is possible to find an elliptic curve $\mathcal{E}_S$, isomorphic to $\mathcal{E}_R$ over $\mathbb{Q}$, with coefficients that are rational functions of $\boldsymbol{\varphi}$ such that $\mathcal{E}_S$ is complex-isomorphic to $\mathcal{E}_F$ at every value of $\boldsymbol{\varphi}$. We will in this subsection explain the relations between the curves $\mathcal{E}_R$, $\mathcal{E}_F$, and $\mathcal{E}_S$. In the cases we have studied, a similar correspondence can be readily anticipated from the relation of the periods of the threefold to periods of an elliptic curve. This correspondence, which we explain in more detail in §4.7, essentially means that the two-dimensional piece of cohomology generated by the flux vectors $F$ and $H$,

$$\langle F, H \rangle \subset H^{1,2}(X, \mathbb{Z}) \oplus H^{2,1}(X, \mathbb{Z}) \ ,$$

can be expressed in terms of the two periods $\omega_1$ and $\omega_2$ of an elliptic curve. By construction, the axiodilaton is then a function of these periods. Thus, at least in the simplest cases, the axiodilaton can be expressed as the ratio $\tau = \omega_1/\omega_2$, explaining why $\mathcal{E}_F$ can be identified, as a complex curve, with the modular curve $\mathcal{E}_R$.

However, more is actually true: As mentioned in the introduction, we provide later extensive numerical evidence that there exists an elliptic curve $\mathcal{E}_S$ over[8] $\mathbb{Q}(\boldsymbol{\varphi})$, which can be also viewed as a moduli-dependent family of elliptic curves such that for every value of the moduli $\boldsymbol{\varphi}$ the curve $\mathcal{E}_S$ is complex-isomorphic to $\mathcal{E}_F$ and isomorphic to the modular curve $\mathcal{E}_R$ over $\mathbb{Q}$. This is equivalent to requiring that we can find a moduli-dependent twist so that applying it the F-theory fibre (14) gives the curve $\mathcal{E}_S$.

Consider now a twist of the form

$$x \mapsto k(\varphi)^{-2} x \ , \qquad y \mapsto k(\varphi)^{-3} y \ , \tag{37}$$

where $k : \mathcal{M}_{\mathbb{C}S} \to \mathbb{C}$ is a moduli-dependent complex twist parameter. This gives an isomorphism between the twisted Sen curve $\mathcal{E}_S$ and the F-theory fibre (14) if and only if $\mathcal{E}_S$ is of the form

$$y^2 \ = \ x^3 + \frac{C(\varphi) - 3}{k(\varphi)^4} \, x + \frac{C(\varphi) - 2}{k(\varphi)^6} \ . \tag{38}$$

The claim is that the twist parameter $k(\boldsymbol{\varphi})$ can be chosen so that this curve is isomorphic to the modular curve $\mathcal{E}_R$ over $\mathbb{Q}$, and that the coefficients $(C(\varphi) - 3)/k(\varphi)^4$ and $(C(\varphi) - 2)/k(\varphi)^6$ are rational functions of $\varphi$.

The existence of such a choice, either everywhere or in any open patch in the moduli space, is not at all obvious. While at any point $\boldsymbol{\varphi}'$ in the moduli space, one can always find a constant $k$ so that the rational structures of $\mathcal{E}_S$ and $\mathcal{E}_R$ agree, what is not a priori clear that this pointwise choice can be extended to a simple algebraic function $k(\boldsymbol{\varphi})$ on the moduli space, that makes $\mathcal{E}_S$ an elliptic curve over $\mathbb{Q}(\boldsymbol{\varphi})$. However, in the cases we have studied we always find such a choice, essentially indicating that the modular curve $\mathcal{E}_R$ has rational dependence on the moduli.

To find the family of curves $\mathcal{E}_S$ with the desired properties, we first find the F-theory curve $\mathcal{E}_F$. Its complex structure modulus $\tau$ is given by the axiodilaton $\tau$, which can be directly computed from the equation (26). In practice, it is most convenient to instead work with the $j$-invariant $j(\tau(\varphi))$,

---

[8]Note that the $\mathbb{Q}(\boldsymbol{\varphi})$ here denotes the field of rational functions in variables $\varphi^i$ with rational coefficients, and should not be confused with the field extension of $\mathbb{Q}$ generated by the value the moduli take at a point.

which is a rational function of $\varphi$. Given the $j$-invariant, the F-theory fibre (14) can be found as a function of the moduli.

As we will explain in §4.7, the conjectural identities for Deligne's periods imply that the axiodilaton can also be identified with the complex structure modulus of the modular curve $\mathcal{E}_R$. Therefore there should exist a twist given by a twist parameter $k$ that (38) has the same rational structure as $\mathcal{E}_R$.

To find the twist required to make the rational structures of $\mathcal{E}_F$ and $\mathcal{E}_R$ agree, we fix $\varphi$ and compute the coefficients $c_p(\varphi)$ for the first few primes (say $p < 50$). This is enough to find the corresponding elliptic curve $\mathcal{E}_R$ as this search is simplified by the F-theory curve being complex-isomorphic to $\mathcal{E}_R$: we need to only look for elliptic curves whose $j$-invariant is given by $j(\tau(\varphi))$. The curve $\mathcal{E}_R$ can be written, when $\varphi \in \mathbb{Q}$, as

$$y^2 \;=\; x^3 + r_1(\varphi)x - r_2(\varphi) \;,$$

where $r_1(\varphi), r_2(\varphi) \in \mathbb{Q}$. Note, however, that we make no a priori assumptions on the functional form of $r_1$ and $r_2$. By comparing this to (38), it is clear that $k(\varphi)$ has to satisfy

$$k(\varphi)^4 \;=\; r_1(\varphi)(C(\varphi) - 3) \;, \qquad k(\varphi)^6 \;=\; r_2(\varphi)(C(\varphi) - 2) \;.$$

It follows that $k(\varphi)^2$ satisfies a cubic equation

$$\frac{1}{r_2(\varphi)}k(\varphi)^6 + \frac{1}{r_1(\varphi)}k(\varphi)^4 - 1 \;=\; 0 \;. \tag{39}$$

We wish to emphasise that it is not a priori clear what kind of dependence the rational constants $r_1$ and $r_2$ have on $\varphi$. However, as the periods of $\mathcal{E}_R$ can be expressed in terms of the periods of the threefold, they vary continuously, and so should $r_1$ and $r_2$. Further, for every rational value of $\varphi$, the curve $\mathcal{E}_R$ should be defined over $\mathbb{Q}$. Then it is natural to expect that $r_1$ and $r_2$ could be expressed as rational functions of $\varphi$. In the present examples, this expectation is indeed satisfied, and we can fix both the twist $k(\varphi)$, up to an overall rational factor, and the coefficient functions $(C(\varphi) - 3)/k(\varphi)^4$ and $(C(\varphi) - 2)/k(\varphi)^6$ in the equation for $\mathcal{E}_S$, by computing their values at a small number of different values of $\boldsymbol{\varphi}$.

The last part of our claim, that the twisted Sen curves $\mathcal{E}_S$ should be isomorphic to $\mathcal{E}_R$ over $\mathbb{Q}$, or a number field $\mathbb{Q}(\alpha)$ in case the modulus takes values in this field instead of $\mathbb{Q}$, can then be tested by comparing the resulting formula (38) to the curve $\mathcal{E}_R$. In every case we study, we are able to verify this expectation for hundreds of different values of the moduli $\varphi$.

### 4.5. Scaling of fluxes and isogenies

The relation between the flux vectors $F$ and $H$ and the axiodilaton $\tau$ seemingly produces an antinomy whose solution nicely illustrates some subtleties regarding the relation between the zeta function and elliptic curves. Recall that the axiodilaton is identified with the complex structure of the F-theory fibre $\mathcal{E}_F$, which is in the same complex isomorphism class with the modular elliptic curve $\mathcal{E}_R$ associated to the zeta function. However, given a point $\boldsymbol{\varphi}$ in the moduli space, the zeta function and thus the coefficients $c_p(\mathcal{E}_R)$, giving the number of points on $\mathcal{E}_R$ over finite fields $\mathbb{F}_p$, are uniquely determined, and in particular do not depend on the flux vectors $F$ and $H$. The seeming paradox is that under a scaling $F \to mF$, $H \mapsto nH$ with $m, n \in \mathbb{Z}_+$, which preserves the quantisation conditions, the parameter $\tau$ scales as $\mathrm{Im}\tau \mapsto \frac{m}{n}\mathrm{Im}\tau$, thus associating a different elliptic curve to the flux vacuum. In particular, when $Y_{ij0} = 0$, which is the case for two of the manifolds we study here, $\tau = \mathrm{i}\,\mathrm{Im}\tau$, and the scaling is simply $\tau \mapsto \frac{m}{n}\tau$.

The resolution to this apparent tension is that a rational scaling of $\tau$ gives an *isogeny* of elliptic curves. As remarked earlier in §4.1, the zeta function is really associated to an *isogeny class* of elliptic curves, rather than to a single elliptic curve. Indeed, it is known that isogenous elliptic curves $\mathcal{E}$ and $\mathcal{E}'$ have in particular the same number of points over all finite fields $\mathbb{F}_p$. Thus the modular form coefficients associated to these elliptic curves satisfy $c_p(\mathcal{E}) = c_p(\mathcal{E}')$ for every good $p$. More is actually true: by the isogeny theorem (see for example [65] and refrences therein), if $c_p(\mathcal{E}) = c_p(\mathcal{E}')$ for all but finitely many primes, then $\mathcal{E}$ and $\mathcal{E}'$ are isogenous. Thus having multiple non-isomorphic F-theory fibres $\mathcal{E}_F$ associated to flux vacua supported by the same compactification manifold but having different values of the axiodilaton $\tau$ is not contradictory with identifying the $\mathcal{E}_F$ with the modular curves $\mathcal{E}_R$ as long as the different fibres $\mathcal{E}_F$ are isogenous.

If the conjectural relations (41) and (43) regarding Deligne's periods hold, it is always possible to choose $\mathcal{E}_R$ in the isogeny class so that its $j$-invariant agrees with the $j$-invariant of the F-theory curve $\mathcal{E}_F$. This $j$-invariant is uniquely determined by the coordinates in the moduli space and the flux vectors.

Let us briefly review some aspects of isogenies corresponding to rational scalings $\tau \mapsto r\tau$ with $r \in \mathbb{Q}$. To see this, recall (for a nice textbook treatment, see [65, 72, 73]) that the Weierstass elliptic functions $\wp(z; \Lambda)$ provide a one-to-one correspondence $\phi_\Lambda$ between a torus $\mathbb{C}/\Lambda$ and the elliptic curve

$$ y^2 = 4x^3 - g_2(\Lambda)\, x - g_3(\Lambda) \ . $$

This correspondence is defined by

$$ \phi_\Lambda(z) = (\wp(z; \Lambda), \wp'(z; \Lambda)) \ , $$

where $g_2(\Lambda)$ and $g_3(\Lambda)$ are defined by Eisenstein series

$$ g_2(\Lambda) = 60 \sum_{\substack{l \in \Lambda \\ l \neq 0}} \frac{1}{l^4} \ , \qquad g_3(\Lambda) = 140 \sum_{\substack{l \in \Lambda \\ l \neq 0}} \frac{1}{l^6} \ , $$

and the lattice $\Lambda \cong \mathbb{Z}^2$ is given in terms of the periods $\omega_1$ and $\omega_2$ by

$$ \Lambda = \omega_1 \mathbb{Z} + \omega_2 \mathbb{Z} \ , $$

and the complex structure parameter $\tau$ can be written as

$$ \tau = \frac{\omega_1}{\omega_2} \ . $$

An isogeny between two elliptic curves $\mathcal{E}_1$ and $\mathcal{E}_2$ is defined as a morphism $\Phi : \mathcal{E}_1 \to \mathcal{E}_2$ that maps the basepoint $O_1$ of $\mathcal{E}_1$ to the basepoint $O_2$ of $\mathcal{E}_2$, $\phi(O_1) = O_2$. The elliptic curves are said to be *isogenous* if there exists an isogeny $\phi : \mathcal{E}_1 \to \mathcal{E}_2$ that is not the constant isogeny mapping every point of $\mathcal{E}_1$ to the basepoint $O_2$.

*Isogenies from integer scalings of $\tau$*

As an example, consider the lattices $\Lambda = \omega_1 \mathbb{Z} + \omega_2 \mathbb{Z}$ and $\Lambda' = n\omega_1 \mathbb{Z} + \omega_2 \mathbb{Z}$ with $n \in \mathbb{Z}_+$, whose associated lattice parameters satisfy $\tau' = n\tau$. Then there exists a continuous $n$-to-one map of the associated tori $\psi : \mathbb{C}/\Lambda' \to \mathbb{C}/\Lambda$ defined by

$$ \psi(z) = z \mod \Lambda \ . $$

Combining this map with the bijections $\phi_\Lambda$ and $\phi_{\Lambda'}$ gives an $n$-to-one map $\Phi : \mathcal{E}_\Lambda \to \mathcal{E}_{\Lambda'}$ between the elliptic curves $\mathcal{E}_\Lambda$ and $\mathcal{E}_{\Lambda'}$ associated to the lattices $\Lambda$ and $\Lambda$.

$$\Phi \;=\; \phi_\Lambda \circ \psi \circ \phi_{\Lambda'}^{-1} \;,$$

which maps

$$(\wp(z, \Lambda), \wp'(z, \Lambda)) \mapsto (\wp(z, \Lambda'), \wp'(z, \Lambda'))$$

To see that this map is a morphism, note that the functions $\wp(z, \Lambda)$ and $\wp'(z, \Lambda)$ are periodic in $\Lambda'$ as $\Lambda' \subseteq \Lambda$, and thus are elliptic functions relative to $\Lambda'$. It is well known (see for example [72] for a proof) that the field of elliptic functions relative to $\Lambda'$ is given by $\mathbb{C}(\wp(z, \Lambda'), \wp(z, \Lambda'))$, thus there exist two rational functions $A(x, y)$ and $B(x, y)$ such that

$$\wp(z, L) \;=\; A(\wp(z, \Lambda'), \wp(z, \Lambda')) \;, \qquad \wp'(z, \Lambda) \;=\; B(\wp(z, \Lambda'), \wp(z, \Lambda')) \;.$$

Then the action of the map $\Phi$ can be written as

$$\Phi \;:\; (x, y) \mapsto (A(x, y), B(x, y)) \;, \tag{40}$$

making the fact that $\Phi$ is a morphism, and thus an isogeny, manifest.

Once one is able to compute the maps $\Phi$ for these integer scalings, it is possible to obtain similar maps for rational scalings $\tau \mapsto \frac{m}{n}\tau$ by composition.

*An example: isogeneous curves **17.a1**, **17.a2**, and **17.1-a1***

Consider for example the elliptic curve with LMFDB label **17.a1**, whose Weierstrass equation is

$$y^2 \;=\; x^3 - 1451x - 21274 \;.$$

The periods of this curve are

$$\omega_1 \;=\; -2.74573911808975\ldots i \;, \qquad \omega_2 \;=\; -0.773539876775560\ldots \;.$$

The rational rescaling of $\tau$, $\tau \mapsto \tau/2$ corresponds to $\omega_2 \mapsto 2\omega_2$, and the lattice $\Lambda' = \omega_1 \mathbb{Z} + 2\omega_2 \mathbb{Z}$ is the period lattice of the isogenous elliptic curve **17.a2** with the Weierstrass equation

$$y^2 \;=\; x^3 - 91x - 330 \;.$$

Note that in the above equation (40) the rational functions $A$ and $B$ are not necessarily defined over the rational numbers, but could be defined over any subfield of $\mathbb{C}$. Consequently, not all scalings $\tau \mapsto \frac{\alpha}{\beta}\tau$ correspond to isogenies between curves both defined over $\mathbb{Q}$.

For another example, take the elliptic curve **17.a1** from the previous example and make the scaling $\tau \mapsto 2\tau$. This gives a curve with the $j$-invariant

$$j(2\tau) \;=\; 11770747169144273640 + \frac{48532033870822191393}{17}\sqrt{17} \;,$$

implying that the corresponding elliptic curve is not defined over $\mathbb{Q}$. Indeed, this is the $j$-invariant of the elliptic curve **17.1-a1**, defined over the number field $\mathbb{Q}(\sqrt{17})$, with Weierstrass equation

$$y^2 + xy + y \;=\; x^3 - x^2 - \left(771 - 165\sqrt{17}\right)x - 10612 + 2498\sqrt{17} \;.$$

*Modular polynomials*

To find scalings of $\tau$ that give isogenies over $\mathbb{Q}$, we can use *modular polynomials* $\Psi_n(x, y)$, which are the minimal polynomials of $j(n\tau)$ over $\mathbb{C}(j(\tau))$ [74]. Thus, once we know the $j$-function $j(\tau)$, we can find the $j$-function $j(n\tau)$ as a root of the polynomial $\Psi_n(x, j(\tau))$. Looking for roots that are given by rational functions of $\varphi$, gives an easy way of finding isogenies that are defined over $\mathbb{Q}$. The modular polynomials $\Psi_n(x, y)$ can be computed using the algorithms in [75] and are given, for all $n$ with $1 \leqslant n \leqslant 400$ and for primes $400 < p < 1000$, on [76].

As a simple example, consider the $j$-functions of the isogenous curves **17.a1** and **17.a2** from the previous subsections:

$$j(\tau) \;=\; \frac{82483294977}{17} \;, \qquad j(\tau/2) \;=\; \frac{20346417}{289} \;.$$

Using these values it is easy to verify that $\Psi_2(j(\tau), j(\tau/2)) = 0$, where

$$\begin{aligned}
\Psi_2(X, Y) \;=\; & -X^2 Y^2 + (X^3 + Y^3) + 1488(X^2 Y + XY^2) - 162000(X^2 + Y^2) + 40773375 XY \\
& + 2^8 3^7 5^6 (X+Y) - 2^{12} 3^9 5^9 \;.
\end{aligned}$$

## 4.6. Field extensions

The discussion in sections 4.3 and 4.5 can be generalised to include the cases where the complex structure moduli are not rational, but rather lie in some *number field*, an algebraic field extension of $\mathbb{Q}$ of finite degree. In these cases, the identification of the curve $\mathcal{E}_S$ with the modular elliptic curve $\mathcal{E}_R$ predicts that $\mathcal{E}_R$ associated to the quadratic piece of the zeta function numerator is generically no longer defined over $\mathbb{Q}$, but rather over the field extension in which the complex structure moduli are valued. The conjecture that the elliptic curve $\mathcal{E}_R$ associated to the quadratic factor $R_2(T)$ of the zeta function is of the form (38) opens up new ways of investigating the cases where this elliptic curve is defined over a number field.

For simplicity, we mainly consider quadratic field extensions, although most of this section applies also to more general field extensions, and we expect similar modularity properties to hold for general number fields. However, the elliptic curves defined over these extensions and the corresponding modular forms become complicated very quickly as the degree of the field extension is increased, and LMFDB contains fewer examples to compare to.

*Real quadratic points and Hilbert modular forms*

A particularly interesting case is when elliptic curves are defined over a real quadratic field $\mathbb{Q}(\sqrt{n})$, where $n$ is a positive square-free integer. These elliptic curves have been proven to be modular [77] in the sense that for an elliptic curve $E(\mathbb{Q}(\sqrt{n}))$ defined over $\mathbb{Q}(\sqrt{n})$ there exists a Hilbert cusp form $f$ over $\mathbb{Q}(\sqrt{n})$ of weight $(2, 2)$ so that $L(E(\mathbb{Q}(\sqrt{n}))) = L(f)$.

At points in the moduli space where $\varphi \in \mathbb{Q}(\sqrt{n})$, the curve $\mathcal{E}_S$ given by the formula (38) in general has coefficients in $\mathbb{Q}(\sqrt{n})$. So when we identify this with the modular curve $\mathcal{E}_R$, as in the case of elliptic curves defined over $\mathbb{Q}$, we would expect to again have a correspondence between the coefficients $c_p(\varphi)$ appearing in the quadratic factor $R_2(\varphi)$ of the zeta function numerator and the Fourier series coefficients of Hilbert modular forms.

This correspondence is slightly more subtle than in the case of elliptic curves defined over $\mathbb{Q}$, as we have to take into account that the notion of prime differs in field extensions of $\mathbb{Q}$ slightly from the usual definition, and we need to make sense of square roots in the finite fields $\mathbb{F}_p$.

As discussed in more detail in appendix A, the Hecke eigenvalues of Hilbert cusp forms are, like their elliptic counterparts, each associated to a prime $\mathfrak{p}$ of the number field $\mathbb{Q}(\sqrt{n})$. However, in

the case of number fields, the primes are prime ideals of the ring of integers $\mathcal{O}_{\mathbb{Q}(\sqrt{n})}$ rather than elements of the field $\mathbb{Q}(\sqrt{n})$.

It turns out that the correct prescription to use in this case to find a correspondence between the Hecke eigenvalues of Hilbert cusp forms and the zeta function coefficients $c_p(\varphi)$ is to identify the coefficient $c_p(\varphi)$ with the Hecke eigenvalue $\alpha_{\mathfrak{p}}$ such that $N(\mathfrak{p}) = p$, where $N(\mathfrak{p})$ is the *ideal norm* of the prime ideal $\mathfrak{p}$. As rational primes $q \in \mathbb{Q}$ can factor in field extensions $(q) = \mathfrak{p}_1 \ldots \mathfrak{p}_k$, there are multiple primes with the same norm. Specifically, in quadratic field extensions, if $(q) = \mathfrak{p}\mathfrak{p}'$, then $\mathfrak{p}$ and $\mathfrak{p}'$ have the same norm. This property is in fact crucial for the identification of eigenvalues and the zeta function coefficients, as in general there are two coefficients $c_p(\varphi)$ for each $p$ if $\varphi \in \mathbb{Q}(\sqrt{n})\backslash\mathbb{Q}$, as we will see imminently.

To find the zeta function coefficients $c_p(\varphi)$ with $\varphi \in \mathbb{Q}(\sqrt{n})$ that should be compared with the Fourier coefficients of Hilbert modular forms, we must find a representative of $\sqrt{n}$ in $\mathbb{F}_p$. By this we mean an element $x \in \mathbb{F}_p$ that satisfies the equation $x^2 - n = 0 \mod p$. This equation can only be satisfied if $n$ is a quadratic residue mod $p$, in which case there are exactly two solutions we can think of corresponding to $\pm\sqrt{n}$, with the sign depending on the choice of embedding of $\mathbb{F}_p$ into $\mathbb{C}$. We need to keep in mind that both choices of embedding are equally valid, and thus we should think of both solutions representing $\sqrt{n}$ in $\mathbb{F}_p$.

When $n$ is a quadratic residue mod $p$, we obtain then two values of $\varphi \in \mathbb{F}_p$ corresponding to $a_1 + a_2\sqrt{n}$ with $a_i \in \mathbb{Q}$ and $a_2 \neq 0$, and thus two zeta function coefficients $c_p(\varphi)$ for each such prime $p$. Analogously to the rational case we expect that these coefficients match the Hecke eigenvalues $\alpha_{\mathfrak{p}}$ of Hilbert modular forms. Specifically, the Fourier coefficients $\alpha_{\mathfrak{p}}$ corresponding to the two primes $\mathfrak{p} \in \mathbb{Q}(n)$ with the field norm $N(\mathfrak{p}) = p$ should match the two coefficicents $c_p(\varphi)$.

We are able to test this in various cases, and find again that our results agree with this expectation. An analogous prediction in the case of a family of octic Calabi-Yau manifolds was made in [27]. However, in that case the lack of zeta function data prevented the authors from explicitly testing their predictions.

## Imaginary quadratic fields and Bianchi modular forms

The situation is analogous in the case of imaginary quadratic field extensions $\mathbb{Q}(\sqrt{-n})$ with $n > 0$ a square-free integer. For such field extensions the corresponding elliptic curves are conjectured to be modular, like those defined over real quadratic field extensions, with the difference being that now the corresponding modular forms should be Bianchi cusp forms (see for example [78]). The identification of the zeta function with an elliptic curve $\mathcal{E}_R$ defined over $\mathbb{Q}(\sqrt{-n})$ and the relevant Bianchi modular form works completely analogously to the case of real quadratic fields: Hecke eigenvalues of Bianchi cusp forms correspond again to prime ideals of the ring of integers $\mathcal{O}_{\mathbb{Q}(\sqrt{-n})}$, and the zeta function coefficients $c_p(\varphi)$ should correspond to Hecke eigenvalues $c_{\mathfrak{p}}$ with $N(\mathfrak{p}) = p$. Generically there are two such prime ideals $\mathfrak{p}$, and i has two representatives corresponding to $\pm i$, giving us two coefficients $c_p\left(a_1 + a_2\sqrt{-n}\right)$ when $a_i \in \mathbb{Q}$, $a_2 \neq 0$, and $-n$ is a quadratic residue mod $p$.

## Base changes, from Hilbert and Bianchi to elliptic modular forms

In some cases, even if the value of the complex structure modulus takes on a value in a totally real or quadratic imaginary field extension of $\mathbb{Q}$, the $j$-invariant may still lie in the base field $\mathbb{Q}$, and further the twisted Sen curve $\mathcal{E}_S$ itself may be defined over $\mathbb{Q}$. In this case $\mathcal{E}_S$ is a *base change* of an elliptic curve in $\mathbb{Q}$. To such a curve we thus can associate two modular forms: the Hilbert or Bianchi modular form of the base-change curve, and the elliptic modular form of the curve defined over $\mathbb{Q}$.

In this case, one can again find multiple representatives of the complex structure modulus $\varphi$, and thus multiple coefficients $c_p$ associated to these representatives. As expected, these turn out to be coefficients of a Hilbert or a Bianchi modular form. However, these modular forms are related to an elliptic modular form by the base change. Explicitly, the Hecke eigenvalues $\alpha_{\mathfrak{p}}$ are equal for all $\mathfrak{p}$ with same norm, and are coefficients of an elliptic modular form.

## 4.7. Deligne's conjecture: periods and L-function values

Deligne's conjectures provide a link between the periods of the modular threefold and L-functions related to the arithmetic properties of the modular curve associated to the zeta function of the threefold. This provides a concrete reason to expect that the F-theory curve $\mathcal{E}_F$, which is constructed out of the period data, can be identified, as a complex elliptic curve, with the modular curve $\mathcal{E}_R$.

The appropriate language to discuss Deligne's conjectures is that of *motives*, which can be thought of as generalisations of cohomology theories, keeping track of some properties that are independent of the choice of a 'good' cohomology theory. We will not attempt to review motives in any detail here, and instead refer an interested reader to an accessible summary [79] and references therein. For the purposes of the following discussion, it will be enough to think of two concrete *realisations* of the motive in question: the Betti and de Rham realisations (see for example [32]).

We are interested in *critical* 'elliptic' motives $M_{\mathrm{ell}}$, which correspond to two-dimensional sublattices $\Lambda_2 \subset H^3_{\mathrm{dR}}(X, \mathbb{Z})$ such that $\mathbb{C} \otimes \Lambda_2 \subset H^{1,2}(X, \mathbb{C}) \oplus H^{2,1}(X, \mathbb{C})$. There is a standard *comparison isomorphism* $I_\infty$ relating the de Rham cohomology to Betti cohomology, giving us a realisation of $\Lambda_2$ in the singular cohomology $H^3_{\mathrm{B}}(X, \mathbb{Z})$. Criticality means, in the cases we are interested in, that these realisations allow a Hodge decomposition such that $h^{p,q} \neq 0$ if and only if $p \geqslant 0$ and $q < 0$ or $p < 0$ and $q \geqslant 0$. Note that negative values of $p$ and $q$ are possible in the setting of abstract Hodge theory. We will see shortly how these arise in the cases we are interested in.

For these motives, one can define Deligne's period $c^+(M)$ as follows: Define $M_{\mathrm{B}}^{\pm}$ as subspaces of the Betti realisation $H^3_{\mathrm{B}}(X, \mathbb{Q})$ of the motive $M$ by the action of complex conjugation[9] $F_\infty$ on these spaces:

$$F_\infty\big|_{M_{\mathrm{B}}^{\pm}} = \pm \mathbb{I} \ .$$

We define also the subspaces $F^{\pm} \subset H^3(X, \mathbb{Q})$ as some spaces $F^p H^3(X, \mathbb{Q})$ in the Hodge filtration such that

$$\dim_{\mathbb{Q}} F^{\pm} \ = \ \dim_{\mathbb{Q}} M_{\mathrm{B}}^{\pm} \ .$$

Recall that a *Hodge filtration* of is a finite decreasing filtration of complex subspaces of $H()F$ From the definition of a Hodge filtration it then follows that $F^+ = F^-$ and under the comparison isomorphism $F^+ \otimes \mathbb{C}$ corresponds to $\bigoplus_{p > q} H^{p,q}_{\mathrm{B}}(X)$, where we have recalled that $H^3_{\mathrm{B}}(X, \mathbb{Z}) \otimes \mathbb{C}$ admits a natural Hodge decomposition

$$H^3_{\mathrm{B}}(X, \mathbb{Z}) \otimes \mathbb{C} \ = \ \bigoplus_{p+q=3} H^{p,q}_{\mathrm{B}}(X) \ .$$

Finally, using this, we define the subspaces $M_{\mathrm{dR}}^{\pm}$ of the de Rham realisation by

$$M_{\mathrm{dR}}^{\pm} \ = \ M_{\mathrm{dR}}/F^{\pm} \ .$$

The standard comparison isomorphism induces a map $I_\infty^+ : M_{\mathrm{B}}^+ \otimes \mathbb{C} \to M_{\mathrm{dR}}^+ \otimes \mathbb{C}$, which can be shown to be an isomorphism [34]. If one chooses rational bases for $M_{\mathrm{B}}^+$ and $M_{\mathrm{dR}}^+$, then Deligne's

---

[9]This map can in fact be indentified as the Frobenius map $\mathrm{Fr}_p$, at the infinite prime.

period is defined, up to an overal rational factor, which depends on the choices of bases, as the determinant

$$c^+(M) \stackrel{\text{def}}{=} \det(I_\infty^+) \; .$$

In other words, if we choose bases $\omega_i^+$ for $M_{\text{B}}^+$ and $A_i^+$ for the Poincaré dual of $M_{\text{dR}}^+$, $F^+ M_{\text{dR}}^\vee$, where $M_{\text{dR}}^\vee$ is the Poincaré dual of $M_{\text{dR}}$, Deligne's period is the determinant of the matrix

$$[I_\infty^+]_{ij} = \int_{A_i^+} \omega_j^+ \; .$$

In the cases we consider $M_{\text{dR}} = H_{\text{dR}}^3(X, \mathbb{Q})$ with its Poincaré dual given by $H_{\text{dR}}^3(X, \mathbb{Q}) \otimes \mathbb{Q}(3)$, and the dual pairing is given by

$$\langle *, * \rangle \; : \; H_{\text{dR}}^3(X, \mathbb{Q}) \times (H_{\text{dR}}^3(X, \mathbb{Q}) \otimes \mathbb{Q}(3)) \to \mathbb{Q} \; , \qquad \langle \alpha, \beta \rangle \; = \; \frac{1}{(2\pi \text{i})^3} \int_X \alpha \wedge \beta \; .$$

Thus, if $\alpha_i^+ \in H_{\text{dR}}^3(X, \mathbb{Q})$ are the images of the basis vectors $A_i^+$ of $M_B^+$ under the map $I_\infty^+$, and $\beta_j^+$ is a basis of $M_{\text{dR}}^+ \otimes \mathbb{Q}(-3) = F^+ M_{\text{dR}}^\vee \otimes \mathbb{Q}(-3) \subset H_{\text{dR}}^3(X, \mathbb{Q})$, Deligne's period is given by the determinant of

$$[I_\infty^+]_{ij} = \int_X \alpha_i^+ \wedge \beta_j^+ \; .$$

Deligne's conjecture predicts that for critical motives $M$, the $L$-value $L(M, 0)$ is a rational, possibly zero, multiple of $c^+(M)$,

$$\frac{L(M, 0)}{c^+(M)} \in \mathbb{Q} \; , \tag{41}$$

with $L(M, 0)$ the motivic L-function defined analogously to the L-function of an elliptic curve

$$L(M, s) \; = \; \prod_{p \text{ good}} \frac{1}{1 - c_p \, p^{-s} + p^{1-2s}} \prod_{p \text{ bad}} \frac{1}{1 \pm p^{-s}} \; ,$$

where now the coefficients $c_p$ are eigenvalues of the inverse Frobenius map acting on the two-dimensional piece of the cohomology that gives a realisation of the motive $M^{\text{ell}}$. In the cases we are considering, these are essentially L-functions of the elliptic curve $\mathcal{E}_R$ associated to the zeta function, so

$$L(M, s) \; = \; L(\mathcal{E}_R, s + 1) \; , \tag{42}$$

We are not aware of a corresponding conjecture for the period $c^-(M)$, but we conjecture, supported by strong numerical evidence in many examples, that for critical elliptic motives $M$, it is possible to find an elliptic curve in the isogeny class of $\mathcal{E}_R$ determined by the zeta function associated to the motive, such that its $\tau$-parameter, denoted $\tau(M)$, satisfies

$$\frac{L(M, 0)}{c^-(M) \, \text{Im} \, [\tau(M)]} \in \text{i} \, \mathbb{Q} \; . \tag{43}$$

In the examples we have studied, it is always possible to identify the aforementioned elliptic curve with a family of twisted Sen curves $\mathcal{E}_S$. The choice of the family does not affect the statement (43) as their complex structure parameters $\tau$ only differ by a rational scaling.

To make the motive $M^{\text{ell}}$ critical, we can use a *Tate twist* on it. Roughly speaking, this means that instead of considering $H_{\text{B}}^3(X, \mathbb{Q})$, we instead consider $H_{\text{B}}^3(X, \mathbb{Q}) \otimes (2\pi \text{i})^2 \mathbb{Q}$. The $(2\pi \text{i})\mathbb{Q}$, corresponding to the *Tate motive* $\mathbb{Q}(1)$, has a Hodge type of $(-1, -1)$, and the Hodge type is additive under tensor product, so to get a motive of Hodge type $(0, -1) + (-1, 0)$, $M^{\text{ell}}(2)$, we define

$$M^{\text{ell}}(2) \stackrel{\text{def}}{=} M^{\text{ell}} \otimes \mathbb{Q}(2)$$

The Tate twist has the effect of introducing multiples of $2\pi i$. In particular the form in $H^3_B(X, \mathbb{Z})$, dual to the three-cycle over which we integrate the relevant element of $H^3_{dR}(X, \mathbb{C})$ to obtain Deligne's period, gets multiplied by $(2\pi i)^2$, so that in the case we are interested in

$$c^{\pm}(M^{ell}(2)) = (2\pi i)^2 c^{\pm}(M^{ell}) .$$

In addition the twist shifts the argument of the motivic L-function:

$$L(M^{ell}(2), s) = L(M^{ell}, s + 2) = L(\mathcal{E}_R \otimes \mathbb{Q}(1), s) = L(\mathcal{E}_S, s + 1) .$$

where in the penultimate equation, which also explains the shift of the argument in (42), we have noted that a Tate twist $\mathbb{Q}(1)$ is needed to turn the motive of Hodge type $(1, 0) + (0, 1)$ corresponding to the middle cohomology $H^{1,0}(\mathcal{E}_R) \oplus H^{0,1}(\mathcal{E}_R)$ into a critical motive $M^{ell}(2)$. Thus Deligne's conjecture can be stated as

$$\frac{L(M^{ell}(2), 0)}{c^+(M^{ell}(2))} = \frac{L(\mathcal{E}_R, 1)}{(2\pi i)^2 c^+(M^{ell})} \in \mathbb{Q} .$$

Following the technique in [32, 33, 35], we can compute Deligne's period $c^+(M_{ell})$ of the rank-two motive $M_{ell}$ corresponding to the flux vacua. To begin, we need to find the action of complex conjugation on the integral Betti cohomology $H^3_B(X, \mathbb{Z})$, which can be done by generalising the technique of [33] to multiparameter manifolds. A convenient basis of the de Rham cohmology $H^3_{dR}(X)$ is given by the row vectors of a matrix $\nu^{-1}E$, where the matrix E consists of real logarithmic series in $\varphi$, which we will later define in (52). As we have briefly reviewed in §3.2, this is related to periods in the integral basis of $H^3_B(X, \mathbb{Z})$ by the matrix $\rho$ defined in (23), at least up to an overall scaling. As the overall scaling affects the action of complex conjugation, we have to fix this. It can be shown [33] that the correct scaling corresponds to multiplying the change-of-basis matrix $\rho$ by $(2\pi i)^3$. Noting that complex conjugation acts on $\nu^{-1}E$ as

$$V = \text{diag}(1, -\mathbf{1}, \quad \mathbf{1}, -1) ,$$

we can deduce that the action of the complex conjugation on the basis of $H^3_B(X, \mathbb{Z})$ is given by

$$F_\infty = (2\pi i)^3 \rho V \left((-2\pi i)^3 \rho^*\right)^{-1} = \begin{pmatrix} 1 & \mathbf{0}^T & 0 & \mathbf{0}^T \\ \mathbf{0} & -\mathbb{1} & \mathbf{0} & -2\mathbb{Y}_0 \\ 0 & \mathbf{0}^T & -1 & \mathbf{0}^T \\ \mathbf{0} & \mathbb{0} & \mathbf{0} & \mathbb{1} \end{pmatrix} .$$

The eigenvalues of this matrix are $\pm 1$, and the corresponding eigenvectors are

$$\begin{aligned} 1 : & \quad (1, \mathbf{0}, \mathbf{0}, 0)^T , & (0, -\mathbf{Y}_{i0}, 0, \boldsymbol{\delta}_i)^T , \\ -1 : & \quad (0, \boldsymbol{\delta}_i, 0, \mathbf{0})^T , & (0, \mathbf{0}, 1, \mathbf{0})^T . \end{aligned} \tag{44}$$

For fixed $i, j$, let us define the following useful eigenvectors:

$$V^- \overset{\text{def}}{=} 2(0, \boldsymbol{\delta}_i - \boldsymbol{\delta}_j, 0, \mathbf{0})^T = 2H ,$$

$$V^+ \overset{\text{def}}{=} 2(0, -\mathbf{Y}_{i0} + \mathbf{Y}_{j0}, 0, \boldsymbol{\delta}_i - \boldsymbol{\delta}_j)^T = 2(Y_{ij0} - Y_{ii0}) H + 2F .$$

The last equalities hold when we have a $\mathbb{Z}_2$ symmetry of the type described in §3, as then $Y_{ik0} = Y_{jk0}$ for every $k = 1, \ldots, m$, and $Y_{ii0} = Y_{jj0}$.

The motives $M_{\mathrm{ell}}$ of the type discussed above arise in the context of flux compactifications from the two-dimensional lattices $\Lambda_2$ spanned by the flux vectors $F$ and $H$ as in §3. By the construction of these solutions, the space $\mathbb{C} \otimes \Lambda_2$ can be alternatively spanned by

$$(D_i - D_j)\Pi = (\partial_i - \partial_j)\Pi \overset{\mathrm{def}}{=} \varpi_F(\boldsymbol{\varphi})F + \varpi_H(\boldsymbol{\varphi})H \;,$$

and its complex conjugate. Here the last equation acts as a definition of two new moduli-dependent quantities $\varpi_F$ and $\varpi_H$. By Griffiths transversality, $(\partial_i - \partial_j)\Pi \in H^{2,1}(X, \mathbb{C})$, and so $\overline{(\partial_i - \partial_j)\Pi} \in H^{1,2}(X, \mathbb{C})$. Thus the motive generated by $F$ and $H$ is indeed of the form $M_{\mathrm{ell}}$, and the Hodge filtration on $M_{\mathrm{ell}}$ is

$$F^3(M_{\mathrm{dR}}^{\mathrm{ell}}) = 0 \;,$$
$$F^2(M_{\mathrm{dR}}^{\mathrm{ell}}) = \big\langle (\partial_i - \partial_j)\Pi \big\rangle \;,$$
$$F^1(M_{\mathrm{dR}}^{\mathrm{ell}}) = \big\langle (\partial_i - \partial_j)\Pi \;, \; \overline{(\partial_i - \partial_j)\Pi} \big\rangle = M_{\mathrm{dR}}^{\mathrm{ell}} = F^0(M_{\mathrm{dR}}^{\mathrm{ell}}) \;.$$

Therefore
$$\mathbb{C} \otimes M_{\mathrm{dR}}^+ = \big\langle (\partial_i - \partial_j)\Pi \;, \; \overline{(\partial_i - \partial_j)\Pi} \big\rangle \;,$$

and the Poincaré dual is

$$\mathbb{C} \otimes F^+ M_{\mathrm{dR}}^{\vee} = \mathbb{C} \otimes F^+\big(H_{\mathrm{dR}}^3(X) \otimes \mathbb{Q}(3)\big) = \big\langle (\partial_i - \partial_j)\Pi \big\rangle \;.$$

To compute Deligne's period, we now need simply to identify the space $M_{\mathrm{B}}^+ \subset H^3(X, \mathbb{Z})$ on which $F_\infty$ acts as $+1$. From the expressions (44) this is easy:

$$M_{\mathrm{B}}^+ = \big\langle V^+ \big\rangle = \big\langle 2\,(Y_{ij0} - Y_{ii0})\,H + 2F \big\rangle \;.$$

Then Deligne's period $c^+(M^{\mathrm{ell}}(2))$ is given by

$$c^+(M^{\mathrm{ell}}(2)) = (2\pi\mathrm{i})^2 \int_X V^+ \wedge (\partial_i - \partial_j)\Pi = (2\pi\mathrm{i})^2\big(2\,(Y_{ij0} - Y_{ii0})\,H + 2F\big)^T \Sigma\,(\partial_i - \partial_j)\Pi \;. \quad (45)$$

Similarly, the period $c^-(M^{\mathrm{ell}}(2))$ is given by

$$c^-(M^{\mathrm{ell}}(2)) = (2\pi\mathrm{i})^2 \int_X V^- \wedge (\partial_i - \partial_j)\Pi = (2\pi\mathrm{i})^2\,(2H)^T \Sigma\,(\partial_i - \partial_j)\Pi \;. \quad (46)$$

To compute these numerically, use the expressions for the periods as integrals of special functions which we give in sections §6 and §7. By using also the identification of modular elliptic curves $\mathcal{E}_R$ related to the zeta function, gathered in the appendices B, C, and D, we can evaluate the special value $L(\mathcal{E}_R, 1)$ to 100 digits. This is easily done with Sage [80], for example. When $L(\mathcal{E}_R, 1) \neq 0$, we are able to test Deligne's conjecture numerically. Using the integral or series expressions for periods, we compute Deligne's period to the same accuracy, which allows us to compute the ratio in (41), which we expect to be a rational number of a low height[10]. We have tested the values of $\varphi$ that appear in the tables of appendices B, C, and D and lie near the large complex structure point. To at least 100 digits the following ratio is a rational number of height at most 18:

$$\frac{\mathrm{denominator}(\varphi)}{\varphi}\,\frac{L(\mathcal{E}_R, 1)}{c^+(M^{\mathrm{ell}}(2))} \in \mathbb{Q} \;.$$

We have similarly computed the periods $c^-$ numerically in these cases and found that the conjectured relation (43) holds to the same accuracy. Further, we observe that it is possible to choose the

---

[10]The *height* of an irreducible rational number $p/q$ is defined as $|p| + |q|$.

isogeny class of the elliptic curve corresponding to the parameter $\tau$ such that the rational constants appearing in these two statements agree, that is

$$-\mathrm{i}\operatorname{Im}\tau(M^{\mathrm{ell}}) \;=\; \frac{c^+(M^{\mathrm{ell}}(2))}{c^-(M^{\mathrm{ell}}(2))} \;. \tag{47}$$

Remarkably, if we accept the relation the above relation as true, then we are able to explain why the F-theory curve $\mathcal{E}_F$ and the modular elliptic curve $\mathcal{E}_R$ agree as complex curves. Namely, from the above relation it follows that for $\varphi > 0$ near the large complex structure point

$$\mathrm{i}\operatorname{Im}\tau(M^{\mathrm{ell}}) \;=\; -\frac{F^T\Sigma(\partial_i-\partial_j)\Pi}{H^T\Sigma(\partial_i-\partial_j)\Pi} + Y_{ii0}-Y_{ij0} \;=\; \mathrm{i}\operatorname{Im}\tau(\varphi)\;,$$

where, to arrive at the last equality, we have recognised $\tau(\varphi)$ from equation (26) and used the fact that $\varphi > 0$ .

*The Birch-Swinnerton-Dyer conjecture: L-function values and periods*

We can also invert the relations (45) and (46) to express $(\partial_i - \partial_j)\Pi$ in terms of Deligne's periods

$$(\partial_i - \partial_j)\Pi \;=\; \frac{1}{(2\pi\mathrm{i})^2 V^+\Sigma V^-}\left(c^+(M^{\mathrm{ell}}(2))V^- - c^-(M^{\mathrm{ell}}(2))V^+\right)\;.$$

When $L(M^{\mathrm{ell}},1){=}L(\mathcal{E}_R,1){\neq}0$, it is possible to express this in terms of L-function values using Deligne's conjecture and the conjectural relations (43) and (47). However, the Birch-Swinnerton-Dyer (BSD) conjecture [81, 82] provides a link between the special value of the L-function associated to an elliptic curve, and the real period. Specifically, the conjecture states that the leading coefficient in the Taylor expansion at $s = 1$ of an L-function $L(\mathcal{E}, s)$ associated to an elliptic curve $\mathcal{E}$, is, up to a constant of proportionality that is rational when $L(\mathcal{E}, 1) \neq 0$, the real period $\omega_1(\varphi)$ of $\mathcal{E}$. Therefore, when $L(\mathcal{E}_R, 1) \neq 0$, the Deligne and BSD conjectures together imply that

$$(\partial_i - \partial_j)\Pi \;=\; \frac{2C_Q(\varphi)}{(2\pi\mathrm{i})^2}\left(\omega_1(\varphi)H + \eta(\varphi)\big((Y_{ij0}-Y_{ii0})\,H + F\big)\right) \;=\; \frac{C_Q(\varphi)}{(2\pi\mathrm{i})^2}\left(\omega_1(\varphi)V^- + \eta(\varphi)V^+\right)\;,$$

where $C_Q(\varphi)$ is a moduli-dependent rational constant, and $\eta$ is an as-yet undetermined function. If, in addition to these conjectures, we assume the relations (43) and (47), we can express this in terms of elliptic curve periods, up to a rational constant:

$$(\partial_i - \partial_j)\Pi \;=\; \frac{C_Q(\varphi)}{(2\pi\mathrm{i})^2}\left(\omega_1(\varphi)V^- + \mathrm{i}\frac{|\omega_2(\varphi)|^2}{\operatorname{Im}\omega_2(\varphi)}V^+\right)\;.$$

In particular, if the combination of 'anomalous' terms $Y_{ij0} - Y_{ii0}$ vanishes, the motive is essentially that corresponding to the middle cohomology of an elliptic curve

$$(\partial_i - \partial_j)\Pi \;=\; \frac{C_Q(\varphi)}{(2\pi\mathrm{i})^2}\left(\omega_1(\varphi)V^- + \omega_2(\varphi)V^+\right)\;.$$

This can be viewed as a reflection of the 'elliptic' nature of the motive $M^{\mathrm{ell}}$. Note that this formula does not make reference to $L(\mathcal{E}_R,1)$. Indeed, in all cases we have studied this relation holds even when $L(\mathcal{E}_R,1) = 0$, so it is tempting to conjecture that this relation, or an appropriate generalisation outside of the large complex structure region, holds generally for any elliptic motive.

# 5.   Practical Evaluation of Zeta Functions of Multiparameter Manifolds

Some of the central developments that makes the analysis of the modular properties of the manifolds studied in this paper possible are the techniques that allow effective computation of the zeta functions of multiparameter Calabi-Yau manifolds. In this section, we give a brief overview of these methods that generalise the earlier work [24] developing the procedure for one-parameter manifolds. The material in this section is adapted from one the of authors' doctoral thesis [26]. The procedure for computing zeta functions of multiparameter manifolds, together with extensive examples, will be presented in detail in future work [25].

Following [24], which builds on the work of Dwork and Lauder [66, 83, 84], we compute the zeta function of a Calabi-Yau manifold $X$ by finding the matrix $\mathrm{F}(\varphi)$ representing the action of the Frobenius map defined in (33), and then using the relation (35) between the Frobenius map and the zeta function.

To better understand how this works, we first define a vector bundle $\mathcal{H}$, whose base is the complex structure moduli space $\mathcal{M}_{\mathbb{C}S}$ of the family $X_\varphi$ of Calabi-Yau manifolds we are studying. The fibre over a point $\varphi \in \mathcal{M}_{\mathbb{C}S}$ is the middle cohomology group $H^3(X_\varphi)$ of the manifold $X_\varphi$ corresponding to that point.

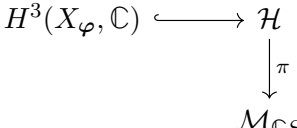

Figure 2: *The vector bundle $\mathcal{H}$.*

On this bundle, we can define the action of the Frobenius maps $\mathrm{Fr}_{p^n} : \Gamma(\mathcal{M}_{\mathbb{C}S}, H^3) \to \Gamma(\mathcal{M}_{\mathbb{C}S}, H^3)$. There is also a canonical Gauss-Manin connection $\nabla : \Gamma(\mathcal{M}_{\mathbb{C}S}, H^3) \to \Gamma(\mathcal{M}_{\mathbb{C}S}, H^3 \otimes T^*\mathcal{M}_{\mathbb{C}S})$. For our purposes, it will be enough to study the covariant derivatives along the vector fields $\theta_i = \varphi^i \partial_i$, where no sum over $i$ is implied. Define the covariant derivatives

$$\nabla_i \stackrel{\text{def}}{=} \nabla_{\theta_i} : \Gamma(\mathcal{M}_{\mathbb{C}S}, H^3) \to \Gamma(\mathcal{M}_{\mathbb{C}S}, H^3) \ . \tag{48}$$

The Frobenius map and this derivative satisfy a compatibility relation, as well as Leibniz rule and a linearity relation: for any section $v \in \Gamma(\mathcal{M}_{\mathbb{C}S}, H^3)$ and any function $f : \mathcal{M}_{\mathbb{C}S} \to \mathbb{C}$ on the moduli space, the following relations hold:

$$\begin{aligned}
p\,\mathrm{Fr}(\nabla_i v) &= \nabla_i(\mathrm{Fr}\,v) \ , \\
\nabla_i\left(f(\varphi)v\right) &= (\theta_i f)(\varphi)\,v + f(\varphi)\nabla_i v \ , \\
\mathrm{Fr}\left(f(\varphi)v\right) &= f(\varphi^p)\,\mathrm{Fr}(v) \ .
\end{aligned} \tag{49}$$

Over a generic point in the moduli space, the action of the Frobenius map $\mathrm{Fr}_p$ cannot be reduced, in a natural way, to an action on the middle cohomology as the fibre is not kept fixed under $\varphi \mapsto \varphi^p$, and the Frobenius map can be viewed as a map between distinct fibres $\mathrm{Fr}_p : \mathcal{H}_\varphi \to \mathcal{H}_{\varphi^p}$. This is an automorphism only at fixed points of $\mathrm{Fr}_p$, such that $\varphi^p = \varphi$. Examples of such points are provided by the Teichmüller representatives $\mathrm{Teich}(\varphi)$ of integral vectors $\varphi \in \mathbb{Z}^m$. At these points, it is possible to identify the action of $\mathrm{Fr}_p$ on the middle cohomology $H^3(X_{\mathrm{Teich}(\varphi)})$. Recalling the discussion in §4.2, this action determines the characteristic polynomial $R(X_\varphi, T)$, which appears as the numerator of the zeta function $\zeta(X_\varphi, T)$ and in turn determines this function. We denote the matrix describing this action, in the basis defined by (51) below, by $\mathrm{F}(\varphi)$. The Teichmüller

representative Teich($\varphi$) provides a natural embedding of $\mathbb{Z}_p$ to $\mathbb{Q}_p$, and hence we can identify $X_{\text{Teich}(\varphi)}$ with the manifold $X_\varphi(\mathbb{F}_p)$ defined over the finite field $\mathbb{F}_p$.

We have thus far purposefully left unspecified the type of the cohomology theory we study. We will be concentrating on two types of cohomologies: The first one is the usual Dolbeault cohomology $H^{p,q}(X_\varphi, \mathbb{C})$ of a complex manifold, which we can use for performing various computations in a familiar manner. However, for the purposes of finding the zeta function, we need a $p$-adic cohomology theory, such as the Dwork cohomology [85], which is a more natural theory to study on a manifold defined over a finite field. These cohomology groups $H^3(X, \mathbb{Q}_p)$ are finite-dimensional vector spaces over the field $\mathbb{Q}_p$ of $p$-adic numbers. Importantly, many of the properties we need to find the action of the Frobenius map are essentially independent of the choice of the particular cohomology theory [2].

Restating the previous paragraphs in more elementary terms: we wish find the matrix $\text{F}(\varphi)$, which can be evaluated in terms of the periods and their derivatives. This should not be a surprise, since the Picard-Fuchs equation follows from the Gauss-Manin connection, and this appears in the compatibility conditions satisfied by the Frobenius map. Having found this map, the zeta function numerator can be computed as the characteristic polynomial of its inverse $\text{F}^{-1}(\varphi) = \text{U}(\varphi)$:

$$R(X_\varphi, T) = \det(1 - T\,\text{U}(\varphi)) \ . \tag{50}$$

As the matrix $\text{U}(\varphi)$ is defined in such a way that it is only well-defined when $\varphi$ is a fixed point of the Frobenius maps $\text{Fr}_{p^n}$, we have to evaluate the matrix U at Teich($\varphi$).

### 5.1. Basis of the vector bundle $\mathcal{H}$

In order to study the Frobienus map, we need to find a basis of sections of the vector bundle $\mathcal{H}$. A natural choice of sections is given by the holomorphic three-form $\Omega$ together with a suitable set of its logarithmic derivatives. These derivatives must be chosen so that the Frobenius map in the given basis is regular in the large complex structure limit. In particular, this implies that the basis must have a certain asymptotic form, and that the sections must be linearly independent in the large complex structure limit.

We will shortly show that such set of sections is given by

$$\Omega \ , \qquad \theta_i \Omega \ , \qquad \widehat{Y}^{ijk} \theta_j \theta_k \Omega \ , \qquad \widehat{Y}^{ijk} \theta_i \theta_j \theta_k \Omega \ , \tag{51}$$

where the constants $\widehat{Y}^{ijk}$ are 'inverse triple intersection numbers' that are defined as a set of constants that satisfy the relation

$$Y_{ijk} \widehat{Y}^{ijs} = \delta_k^s \ .$$

Note that this does not define the quantities $\widehat{Y}^{ijk}$ uniquely. Rather, one can shift $\widehat{Y}^{ijk}$ by any $A^{ijk}$ which is 'orthogonal' to the triple intersection numbers, that is

$$Y_{ijk} A^{ijk} = 0 \ .$$

In this paper, we always choose these constants so that the basis (51) is symmetric under the symmetry group we are using the construct flux vacuum solutions.

It is useful to gather the combinations of derivatives appearing here into a vector

$$\Theta \overset{\text{def}}{=} (\Theta_0, \Theta_i, \Theta^i, \Theta^0) \overset{\text{def}}{=} \left( 1, \theta_i, \widehat{Y}^{ijk} \theta_j \theta_k, \frac{\widehat{Y}^{ijk}}{m} \theta_i \theta_j \theta_k \right) \ .$$

Let us denote by $\mathrm{E}(\varphi)$ the change-of-basis matrix from the constant basis to this basis. Explicitly, the components of this matrix are given by

$$\mathrm{E}(\varphi)_a^{\ b} = \Theta^b \, \varpi_a(\varphi) \, , \tag{52}$$

where the indices $a$ and $b$ are understood to stand for both upper and lower indices. With this definition, the asymptotic form of $\mathrm{E}(\varphi)$ in the large complex structure limit, $\varphi_i \to 0$ for all $i$, is given by

$$\mathrm{E}(\varphi) \;=\; \begin{pmatrix} 1 & \mathbf{0}^T & \mathbf{0}^T & 0 \\ \boldsymbol{\ell} & \mathbb{I} & 0 & \mathbf{0} \\ \frac{1}{2!}\boldsymbol{\ell}^T\mathbb{Y}_i\boldsymbol{\ell} & \ell^i\mathbb{Y}_i & \mathbb{I} & 0 \\ \frac{1}{6!}Y_{ijk}\ell^i\ell^j\ell^k & \frac{1}{2!}\boldsymbol{\ell}^T\mathbb{Y}_i\boldsymbol{\ell} & \boldsymbol{\ell}^T & 1 \end{pmatrix} + \mathcal{O}(\varphi\log^3\varphi) \;=\; \varphi^{\boldsymbol{\epsilon}} + \mathcal{O}(\varphi\log^3\varphi) \, , \tag{53}$$

where $\ell^i = \log\varphi^i$, and a free index $i$ corresponds to a row or a column vector.

We also note that the logarithmic derivatives of $\mathrm{E}$ can be written in the form

$$(\theta_i\mathrm{E})(\varphi) = \mathrm{E}(\varphi)\mathrm{B}_i(\varphi) \, ,$$

which follows from the fact that the columns of $\mathrm{E}$ give a basis of the third cohomology, and can be seen as the statement that $\mathrm{B}_i(\varphi)$ are the connection matrices of the Gauss-Manin connections (48). We can work out the asymptotics of the matrices $B_i$ in the large complex structure limit by studying the asymptotic form of $\theta_i\mathrm{E}$ as $\varphi \to 0$.

$$\theta_i\mathrm{E} \;\sim\; \begin{pmatrix} 0 & \mathbf{0} & \mathbf{0} & 0 \\ \boldsymbol{\delta}_i & 0 & 0 & \mathbf{0} \\ \boldsymbol{\ell}^T\mathbb{Y}_j\boldsymbol{\ell} & \mathbb{Y}_i & 0 & \mathbf{0} \\ \frac{1}{2!}\boldsymbol{\ell}^T\mathbb{Y}_i\boldsymbol{\ell} & \mathbf{Y}_{ij}^T\boldsymbol{\ell} & \boldsymbol{\delta}_i & 1 \end{pmatrix} \;=\; \varphi^{\boldsymbol{\epsilon}}\epsilon_i \;=\; \varphi^{\boldsymbol{\epsilon}}\mathrm{B}_i(\mathbf{0}) \, .$$

Comparing this to the equation (53), we deduce that in the large complex structure limit

$$\mathrm{B}_i(\mathbf{0}) \;=\; \epsilon_i \, .$$

## 5.2. The differential equation satisfied by the Frobenius map

The basic idea is to derive a differential equation satisfied by $\mathrm{F}(\varphi)$, and find the solution given the value of the matrix $\mathrm{F}$ at some point $\varphi_0$ as an initial condition.

Using the compatibility conditions (49), one can derive the following differential equation for the matrix of the Frobenius action

$$\theta_i(\mathrm{F})(\varphi) \;=\; p\,\mathrm{F}(\varphi)\,(\mathrm{B}_i(\varphi^p)) - \mathrm{B}_i(\varphi)\mathrm{F}(\varphi) \, .$$

The solution to these equations is given by

$$\mathrm{F}(\varphi) \;=\; \mathrm{E}^{-1}(\varphi)\mathrm{F}(\varphi_0)\mathrm{E}(\varphi^p) \, , \tag{54}$$

where $\varphi_0$ is a fixed initial value. In the following, we will take $\mathrm{F}(\varphi_0)$ to be the limiting value $\mathrm{F}(\mathbf{0}) \stackrel{\text{def}}{=} \lim_{\varphi_i \to 0}\mathrm{F}(\varphi)$, as it turns out that it is possible to highly constrain the form that the matrix $\mathrm{F}(\mathbf{0})$ takes in this limit. For computing the zeta function, it is moreover more convenient to work with the inverse Frobenius map

$$\mathrm{U}(\varphi) \stackrel{\text{def}}{=} \mathrm{F}^{-1}(\varphi) \;=\; \mathrm{E}^{-1}(\varphi^p)\mathrm{U}(\mathbf{0})\mathrm{E}(\varphi) \, .$$

We already know how to compute $E(\varphi)$ so the problem of finding the matrix $U(\varphi)$ representing the action of the inverse Frobenius map[11] is reduced to finding the large complex structure limit $U(\mathbf{0})$ of $U(\varphi)$.

*Large complex structure limit*

Taking the limit $\varphi \to \mathbf{0}$ of (54), one obtains the following relations, the one-parameter analogue of which was used in [24] to constrain the form of the matrix $U(\mathbf{0})$,

$$p\, \epsilon_i U(\mathbf{0}) \;=\; U(\mathbf{0})\, \epsilon_i \,, \qquad i = 1, \ldots, m \,. \tag{55}$$

The most general solution to these conditions can be written in terms of the homology algebra elements of the mirror manifold $\widetilde{X}$, (20), as

$$U(\mathbf{0}) = u\Lambda \left( I + \alpha^i\, \epsilon_i + \beta_i\, \mu^i + \gamma\, \eta \right) \,, \qquad \Lambda \;=\; \mathrm{diag}(1, p\,\mathbf{1}, p^2\,\mathbf{1}, p^3) \,. \tag{56}$$

In addition, we can impose a consistency relation with inner product $H^3 \times H^3 \to \mathbb{C}$

$$(\xi, \omega) \;=\; \int_X \xi \wedge \omega \,.$$

In the Frobenius basis where the period vector is $\varpi$, the matrix corresponding to the inner product is given by

$$\sigma \;=\; \nu^{-1} \rho^T \Sigma \rho \nu^{-1} \;=\; \frac{1}{(2\pi\mathrm{i})^3} \begin{pmatrix} 0 & \mathbf{0}^T & \mathbf{0}^T & -1 \\ \mathbf{0} & \mathbb{0} & \mathbb{1} & \mathbf{0} \\ \mathbf{0} & -\mathbb{1} & \mathbb{0} & \mathbf{0} \\ 1 & \mathbf{0}^T & \mathbf{0}^T & 0 \end{pmatrix} \,.$$

The matrix $U(\mathbf{0})$ should satisfy

$$U(\mathbf{0})\, \sigma\, U(\mathbf{0})^T \;=\; p^3 \sigma \,.$$

This imposes the conditions

$$u^2 \;=\; 1 \,, \qquad \beta_i \;=\; \frac{1}{2} Y_{ijk} \alpha^j \alpha^k \,.$$

In the one-parameter case of [24] it was conjectured, supported with extensive numerical evidence, that the correct solution is obtained by imposing (in the basis we are using)

$$u \;=\; 1 \,, \quad \alpha^i \;=\; 0 \,, \quad \gamma \;=\; \chi(X)\zeta_p(3) \,,$$

where $\zeta_p(3)$ is the $p$-adic zeta function.

We conjecture that this choice of constants applies also in the multi-parameter cases we study in this paper. In fact, it turns out that this choice of constants makes $U(\varphi)$ into a matrix of rational functions of $\varphi$, at least up to the $p$-adic order determined by the Riemann hypothesis necessary to

---

[11]The radius of convergence of the series in the matrix $F(\varphi)$ is $1 + \delta$ for some $\delta > 0$, while the matrix $E(\varphi)$ only converges in the open disk with unit radius centred at the origin. Therefore, when evaluating $F(\varphi)$ at Teichmüller values $\varphi^i = \mathrm{Teich}(n^i)$ with $n^i \in \mathbb{Z}$, we cannot simply evaluate the individual matrices at the Teichmüller point and multiply them together, as the series for $E(\varphi)$ and $E(\varphi^p)$ do not converge for such values of $\varphi$. Rather, we compute the series for $F(\varphi)$ to high enough $p$-adic order and evaluate these convergent series at the Teichmüller representative. For further discussion with references, see for example [70].

compute the coefficients of the polynomial $R(X_\varphi, T)$, which is the form that $\mathrm{U}(\varphi)$ is expected to take [24, 83, 84].

However, in general this choice of constants only works on the symmetric lines in the moduli spaces we are studying. On a generic point in the complex structure moduli space of a multiparameter manifold, these constants do not lead to a consistent solution. Rather, $p$-adic expansions for the coefficients $\alpha_i$ and $\gamma$ can be found order-by-order by requiring that the matrix $\mathrm{U}(\varphi)$ takes the form of a rational matrix, to a given $p$-adic accuracy.

### 5.3. Zeta function factorisations from $\mathbb{Z}_2$ symmetry

The explicit construction described above for the polynomial $R(X_\varphi, T)$, as the characteristic polynomial of the matrix $\mathrm{U}(\varphi)$, gives a way of seeing that the existence of $\mathbb{Z}_2$ symmetries discussed in this paper imply the existence of a quadratic factor in the $R(X_\varphi, T)$ when factored over the integers. This uses elementary linear algebra techniques, without needing to resort to the high-brow language of étale cohomologies and Galois group actions. Although the arguments used here are elementary, we feel that giving some details on these illustrates the role of the symmetries and underlines the hands-on control of the number theoretic objects that the methods outlined in this section allow.

Consider a matrix $\varsigma$ giving a $\mathbb{Z}_2$ action on the period vector $\varpi$. If this $\mathbb{Z}_2$ corresponds to a symmetry of the manifold at a point $\varphi$ in the moduli space, it will leave the period vector invariant

$$\varsigma \varpi(\varphi) \;=\; \varpi(\varphi) \;.$$

Equivalently, the matrix $\varsigma$ commutes with the period matrix $\mathrm{E}(\varphi)$. This symmetry can be used to split the middle cohomology into two eigenspaces $W^\pm$ correspond to the two possible eigenvalues of the $\mathbb{Z}_2$ action.

In the cases we study, the eigenspace $W^-$ of $\varsigma$ is always of Hodge type $(2,1)+(1,2)$, and $\varsigma$ commutes with $\mathrm{U}(0)$, and thus also with $\mathrm{U}(\varphi)$. Explicitly, taking the $\mathbb{Z}_2$ symmetry to exchange the moduli $\varphi^1 \leftrightarrow \varphi^2$, the matrix $\mathbb{Z}_2$ is given by

$$\varsigma \;=\; \mathrm{diag}\big(1,\, \sigma_1,\, \overbrace{1,\dots,1}^{m-2 \text{ times}},\, \sigma_1,\, \overbrace{1,\dots,1}^{m-2 \text{ times}},\, 1\big) \;, \qquad \sigma_1 \;=\; \begin{pmatrix} 0 & 1 \\ 1 & 0 \end{pmatrix}.$$

The matrix $\varsigma$ has a two-dimensional eigenspace $W^-$ with eigenvalue $-1$ which is of Hodge type $(1,2)+(2,1)$. The complementary space $W^+$ corresponds to the eigenvalue 1. The eigenspace $W^-$ corresponds exactly to the two-dimensional subspace $\langle F, H \rangle$ generated by the flux vectors. The fact that the matrices $\varsigma$ and $\mathrm{U}(\varphi)$ commute implies, by elementary linear algebra arguments, that $\mathrm{U}(\varphi)$ has a block-diagonal form. A $2\times 2$ block corresponds to the eigenspace $W^-$, and a $2m\times 2m$ block corresponds to $W^+$. Therefore the characteristic polynomial $R(X_\varphi, T)$ factorises over the integers.

As an aside, if we allow a more general $\mathbb{Z}_2$ action represented by a matrix $\varsigma$, the split into the corresponding eigenspaces $W^\pm$ may be trivial or it might not be compatible with the Hodge structure of the middle cohomology in the sense that the eigenspaces need not be of the Hodge type $(3,0)+(0,3)$ or $(1,2)+(2,1)$ that we expect to find in connection with modularity. For instance, a case where the fixed point of a $\mathbb{Z}_2$ action on a one-parameter moduli space gives a split into two subspaces of Hodge types $(3,0)+(1,2)$ and $(2,1)+(0,3)$ has been studied in [19]. Such a split clearly does not give a split of $H^3(X, \mathbb{Q})$, as the decomposition is not invariant under complex conjugation. We do not consider these more general cases in this paper.

## 6. The Main Example: The Hulek-Verrill Manifold

In this section and the next, we give a few examples of manifolds that have a $\mathbb{Z}_2$ or a larger symmetry group, and thus admit supersymmetric flux vacua given by the construction in §3. In particular, we will give a very brief overview of these geometries and list their periods before restricting to a $\mathbb{Z}_2$ symmetric locus. On this locus, we will find the complex F-theory curve $\mathcal{E}_F$, and by choosing a suitable rational structure we can find a family $\mathcal{E}_S$ of curves that will have the same rational structure as the modular curves $\mathcal{E}_R$. In all of the cases, we find a striking agreement between the modular forms associated to the curves $\mathcal{E}_S$ and the coefficients $c_p(\varphi)$ appearing in the quadratic part of the zeta function numerator, giving strong evidence that these manifolds are indeed modular and that the associated modular curves vary as rational functions of the moduli. The zeta function data, as well as a list of modular forms associated to the zeta functions of the manifolds is included appendix B.

We discuss the case of Hulek-Verrill manifold in detail to illustrate in practice the general theory discussed in the previous sections 3 and 4. The analysis of the remaining cases proceeds largely along the same lines, so we opt to leave out the details and content ourselves with just listing the results of this analysis in the following section 7.

### 6.1. The Hulek-Verrill manifold

Our first example is mirror to the complete intersection Calabi-Yau described by the CICY matrix

$$
\begin{array}{c}
\mathbb{P}^1 \\
\mathbb{P}^1 \\
\mathbb{P}^1 \\
\mathbb{P}^1 \\
\mathbb{P}^1
\end{array}
\left[
\begin{array}{cc}
1 & 1 \\
1 & 1 \\
1 & 1 \\
1 & 1 \\
1 & 1
\end{array}
\right]_{\chi=-80} .
$$

These five-parameter manifolds, called Hulek-Verrill manifolds $\mathrm{HV}_{\boldsymbol{\varphi}}$ [37], can be realised as the toric compactification of the hypersurface in $\mathbb{T}^5 = \mathbb{P}^5 \setminus \{X_\mu = 0\}$ given by the vanishing of

$$
P^1(\mathbf{X}) \;=\; \sum_{\mu=0}^{5} X_\mu \,, \qquad P^2(\mathbf{X}; \boldsymbol{\varphi}) \;=\; \sum_{\mu=0}^{5} \frac{\varphi^\mu}{X_\mu} \,. \tag{57}
$$

The manifolds obtained in this way are smooth outside the discriminant locus $\Delta = 0$ with

$$
\Delta \;=\; \prod_{\eta_i \in \{\pm 1\}} \left( \sqrt{\varphi^0} + \eta_1 \sqrt{\varphi^1} + \eta_2 \sqrt{\varphi^2} + \eta_3 \sqrt{\varphi^3} + \eta_4 \sqrt{\varphi^4} + \eta_5 \sqrt{\varphi^5} \right) . \tag{58}
$$

The triple intersection numbers $Y_{ijk}$ and the other $Y_{abc}$ of the mirror Hulek-Verrill manifolds can be computed from their description as complete intersection varieties, and are given by

$$
Y_{ijk} \;=\; \begin{cases} 2 & i,j,k \text{ distinct.} \\ 0 & \text{otherwise,} \end{cases} \qquad Y_{ij0} \;=\; 0 \,, \qquad Y_{i00} \;=\; -2 \,, \qquad Y_{000} \;=\; 240 \frac{\zeta(3)}{(2\pi\mathrm{i})^3} \,.
$$

The six $\varphi^\mu$ furnish projective coordinates for the complex structure moduli space of this manifold. From the defining equations (57), it is clear that interchanging the complex structure parameters $\varphi^\mu$ gives a biholomorphic manifold. Thus the complex structure moduli space has an $S_6$ symmetry group that we use later to find supersymmetric flux vacua. Due to the symmetry, we are also able to, without loss of generality, work exclusively in the patch $\varphi^0 = 1$ in which the five remaining $\varphi^i$ are the affine complex structure coordinates.

As detailed in [38], in the large complex structure region of the patch $\varphi^0 = 1$, given by the condition

$$\mathrm{Re}\left[\sum_{i=1}^{5} \sqrt{a_i}\right] < 1 , \tag{59}$$

the 12 periods of the the Hulek-Verrill manifold are given in the Frobenius basis by linear combinations of Bessel function moments. The periods relevant to us are

$$\varpi^0(\boldsymbol{\varphi}) = \int_0^{\infty} \mathrm{d}z \, z \, \mathrm{K}_0(z) \prod_{i=1}^{5} \mathrm{I}_0\left(\sqrt{\varphi^i}\, z\right) ,$$

$$\varpi^j(\boldsymbol{\varphi}) = -2 \int_0^{\infty} \mathrm{d}z \, z \, \mathrm{K}_0(z) \mathrm{K}_0\left(\sqrt{\varphi^j}\, z\right) \prod_{i\neq j} \mathrm{I}_0\left(\sqrt{\varphi^i}\, z\right) , \tag{60}$$

$$\varpi_j(\boldsymbol{\varphi}) = 8 \sum_{\substack{m<n \\ m,n\neq j}} \int_0^{\infty} \mathrm{d}z \, z \, \mathrm{K}_0(z) \mathrm{K}_0\left(\sqrt{\varphi^m}\, z\right) \mathrm{K}_0\left(\sqrt{\varphi^n}\, z\right) \prod_{i\neq m,n} \mathrm{I}_0\left(\sqrt{\varphi^i}\, z\right) - 4\pi^2 \varpi_0(\boldsymbol{\varphi}) .$$

The five-parameter Hulek-Verrill manifolds support supersymmetric flux vacua on the loci where any two complex structure moduli $\varphi^i$ are equal. For definiteness, let us set two of the $\varphi^1$ and $\varphi^2$ equal to $\varphi$, and relabel the three remaining $\varphi^i$ as $\psi^1$, $\psi^2$, $\psi^3$. Use of Bessel function identities reveals that in the case of Hulek-Verrill manifolds, the axiodilaton given in (26) is

$$\tau\left(\psi^1,\, \psi^2,\, \psi^3\right) = \frac{2\mathrm{i}}{\pi} \frac{\int_0^{\infty} \mathrm{d}z \, z \, \mathrm{K}_0(z) \left[\mathrm{K}_0\left(\sqrt{\psi^1}\, z\right) \mathrm{I}_0\left(\sqrt{\psi^2}\, z\right) \mathrm{I}_0\left(\sqrt{\psi^3}\, z\right) \, + \mathrm{cyclic}\right]}{\int_0^{\infty} \mathrm{d}z \, z \, \mathrm{K}_0(z) \, \mathrm{I}_0\left(\sqrt{\psi^1}\, z\right) \mathrm{I}_0\left(\sqrt{\psi^2}\, z\right) \mathrm{I}_0\left(\sqrt{\psi^3}\, z\right)} . \tag{61}$$

The dependence of the $j$-invariant can be found numerically by computing the value of $j(\tau)$ on numerous points on the moduli space, and fitting the points with a rational function. It turns out that the $j$-function takes the remarkably simple form

$$j\left(\tau\left(\psi^1,\, \psi^2,\, \psi^3\right)\right) = \frac{\left(\Delta_F + 16\psi^1\psi^2\psi^3\right)^3}{\Delta_F \left(\psi^1\psi^2\psi^3\right)^2} , \tag{62}$$

where the polynomial $\Delta_F$, related to the discriminant (58), is given by

$$\Delta_F = \prod_{\eta_i=\pm 1} \left(1 + \eta_1\sqrt{\psi^1} + \eta_2\sqrt{\psi^2} + \eta_3\sqrt{\psi^3}\right)$$

$$= \left(\left(1 - \psi^1 - \psi^2 - \psi^3\right)^2 - 4\left(\psi^1\psi^2 + \psi^2\psi^3 + \psi^3\psi^1\right)\right)^2 - 64\,\psi^1\psi^2\psi^3 . \tag{63}$$

We also note in passing that there is an isogeny over $\mathbb{Q}$ corresponding to the rescaling $\tau = \widetilde{\tau}/2$, or equivalently to a rescaling of fluxes, as discussed in §4.5. Thus there is another valid solution with

$$j\left(\widetilde{\tau}\left(\psi^1,\, \psi^2,\, \psi^3\right)\right) = \frac{\left(\Delta_F + 256\,\psi^1\psi^2\psi^3\right)^3}{\Delta_F^2\,\psi^1\psi^2\psi^3} . \tag{64}$$

Note that to derive (61) and then subsequently arrive at (62), we have used the expressions (60) for the periods, which are valid only in the region (59). However, since the expression (62) for the

$j$-function is well-defined everywhere outside of the discriminant locus, this is the unique analytic continuation of the left-hand side into this region and the expression (62) is correct throughout moduli space.

To fix the F-theory fibre $\mathcal{E}_F$, which was shown in §2 to take on the general form (14), we must choose $C = C(\psi^1, \psi^2, \psi^3)$ such that the $j$-invariant of this curve agrees with (62). The values of $C(\psi^1, \psi^2, \psi^3)$ satisfying this are given by solving the cubic equation (16). In this case, among the three solutions, there is one that is a rational function of the moduli $\psi^i$. As discussed in sections 3 and 4.4, we can can choose any of the three roots without loss of generality. Thus we take

$$C(\psi^1, \psi^2, \psi^3) \;=\; \frac{144\psi^1\psi^2\psi^3}{\Delta_F + 64\psi^1\psi^2\psi^3} \;. \tag{65}$$

We denote the value of $C$ corresponding to the isogenous curve with the $j$-invariant given by $j(\widetilde{\tau})$ in (64) as $\widetilde{C}(\psi^1, \psi^2, \psi^3)$, and choose the root that makes it a rational function in the complex structure moduli:

$$\widetilde{C}(\psi^1, \psi^2, \psi^3) \;=\; \frac{9\Delta_F}{4\left(\Delta_F + 64\,\psi^1\psi^2\psi^3\right)} \;. \tag{66}$$

There are two more isogenies over $\mathbb{Q}$ corresponding to the scalings $\tau \mapsto \tau/3$ and $\tau \mapsto \tau/6$, which can be treated similarly.

## 6.2. The $S_5$ symmetric family $\varphi^i = \varphi$

Let us now specialise to the locus in the complex structure moduli space where all complex structure moduli are equal, $\varphi^i = \varphi$ for $i = 1, \ldots, 5$. The manifolds parametrised by $\varphi$ are $S_5$ symmetric, with the action of $S_5$ corresponding to permutations of the complex structure parameters $\varphi^i$. The group $S_5$ contains various $\mathbb{Z}_2$ subgroups, and thus the manifold supports numerous flux vacua. In fact, it is easy to see that there are four independent choices for the pair $F, H$ of flux vectors, indicating the factorisation of $H^3(X, \mathbb{C})$ into $\mathbb{C} \otimes (\Lambda_2^4 \oplus \Lambda_4)$, where $\Lambda_2$ is a two-dimensional sublattice of $H^3(X, \mathbb{Z})$ of Hodge type $(1, 2) + (2, 1)$, and $\Lambda_4$ is a four-dimensional sublattice of mixed Hodge type $(3, 0) + (2, 1) + (1, 2) + (0, 3)$. Correspondingly, the polynomial $R(X, T)$ determining the zeta function factorises as

$$R(X, T) \;=\; R_2(T)^4 R_4(T) \;,$$

where the polynomials $R_2(T)$ and $R_4(T)$ are determined by three constants $a_p, b_p$, and $c_p$:

$$R_2(T) \;=\; 1 - c_p(\varphi)pT + p^3T \;, \qquad R_4(T) \;=\; 1 + a_p(\varphi)T + b_p(\varphi)T^2 + a_p(\varphi)p^3T^3 + p^6T^4 \;.$$

These have been tabulated for primes up to $p = 47$ in appendix B, and more can be easily computed using the methods developed in [26].

Specialising the expressions in the previous subsection to the case $\psi^1 = \psi^2 = \psi^3 = \varphi$, the $j$-invariant simplifies to

$$j(\tau(\varphi)) \overset{\text{def}}{=} j\left(\tau(\varphi, \varphi, \varphi)\right) \;=\; \frac{(3\varphi - 1)^3 \left(3\varphi^3 - 3\varphi^2 + 9\varphi - 1\right)^3}{\varphi^6 (\varphi - 1)^3 (9\varphi - 1)},$$

and the corresponding F-theory fibre $\mathcal{E}_F$ of the form (14) is

$$y^2 \;=\; x^3 + \left(\frac{144\varphi^3}{(3\varphi^2 + 6\varphi - 1)^2} - 3\right) x + \frac{144\varphi^3}{(3\varphi^2 + 6\varphi - 1)^2} - 2 \;. \tag{67}$$

The j-invariant associated to the isogenous curves obtained by scaling $\tau \mapsto \tau/2 = \widetilde{\tau}$, which takes elliptic curves defined over $\mathbb{Q}$ to other elliptic curves defined over $\mathbb{Q}$, is given by

$$j(\widetilde{\tau}(\varphi)) \stackrel{\text{def}}{=} j\left(\widetilde{\tau}(\varphi, \varphi, \varphi)\right) = \frac{(3\varphi + 1)^3 \left(3\varphi^3 + 75\varphi^2 - 15\varphi + 1\right)^3}{\varphi^3 (9\varphi - 1)^2 (\varphi - 1)^6} \,.$$

The elliptic curve $\widetilde{\mathcal{E}}_F$ of the form (14) with this $j$-invariant is

$$y^2 = x^3 - \left(\frac{144\varphi^3}{(3\varphi^2 + 6\varphi - 1)^2} + \frac{3}{4}\right) x + \frac{1}{4} - \frac{144\varphi^3}{(3\varphi^2 + 6\varphi - 1)^2} \,. \tag{68}$$

There are two further isogenies over $\mathbb{Q}$, associated to scalings $\tau \mapsto \tau/3$ and $\tau \mapsto \tau/6$. The corresponding $j$-invariants are

$$j(\tau/3) = \frac{(3\varphi - 1)^3 \left(243\varphi^3 - 243\varphi^2 + 9\varphi - 1\right)^3}{\varphi^2 (9\varphi - 1)^3 (\varphi - 1)} \,,$$

$$j(\tau/6) = \frac{(3\varphi + 1)^3 \left(243\varphi^3 - 405\varphi^2 + 225\varphi + 1\right)^3}{\varphi (9\varphi - 1)^6 (\varphi - 1)^2} \,.$$

One can work out the corresponding F-theory fibres and the twisted Sen curves in a manner that is completely analogous to the way we treat the curves associated to $\tau$ and $\widetilde{\tau}$. We refrain from giving further details on these curves, although the corresponding data is included in appendix B.

### 6.3. Rational points and elliptic modular forms

Based on the discussion in §4.4, we expect to be able to find an elliptic curve $\mathcal{E}_S$ over $\mathbb{Q}(\varphi)$ of the form (38), which can also be thought of of as a family of elliptic curves parametrised by $\varphi$, such that every member of the family is isomorphic over $\mathbb{Q}$ to the corresponding modular curve $\mathcal{E}_R$. In practice, to find the family, we use the values of $C(\boldsymbol{\varphi})$ found above, fix the value of $\varphi$ and compute the coefficients $c_p(\varphi)$ for the first few primes to find the corresponding elliptic curve $\mathcal{E}_R$. Then we look for a coefficient $k(\varphi)$ such that a curve given by (38) is rational isomorphic to $\mathcal{E}_R$.

For example, consider the case $\varphi = \frac{15}{2}$. The $j$-invariant corresponding to $\mathcal{E}_F$ at this point is

$$j\left(\tau\left(15/2\right)\right) = \frac{64096528489320601}{13313408062500} \,.$$

The LMFDB [43] lists 5 rational elliptic curves and corresponding elliptic modular forms with this $j$-invariant. We expect to find the modular form associated to the zeta function at $\varphi = \frac{15}{2}$ among these. To find this modular form, we compute the coefficients $c_p(\varphi)$ for the primes $p < 100$.

With the zeta function numerators in hand, finding the coefficients is straightforward, possibly apart from the slight subtlety associated with the representing the rational number $r/s$ in the finite fields $\mathbb{F}_p$. When $p \nmid s$, there is a unique element $x \in \mathbb{F}_p$ such that $xs = 1 \bmod p$. This $x$ gives the representative of $1/s$ in $\mathbb{F}_p$, and $r/s$ is represented by $rx \in \mathbb{F}_p$.

We can compute the coefficients $c_p(15/2)$ and compare them to the modular form coefficients (see table 5). We find perfect agreement with the modular form with LMFDB label **51870.2.a.bs**, whereas the coefficient of the modular forms associated to the other four curves differ at various primes by a sign, as is expected for modular forms given by twists.

The modular form **51870.2.a.bs** corresponds to the elliptic curve **51870.bs4** with the equation

$$y^2 + xy + y = x^3 - 8338x - 235312 \,, \tag{69}$$

defined up to rational isomorphisms. The elliptic curve (69) can be obtained from the expression (67) for $\mathcal{E}_F$ by effecting a twist (37) on the fibre coordinates $x$ and $y$ with

$$k\left(15/2\right) \;=\; 2\sqrt{\frac{3}{851}}\,,$$

which is unique up to multiplication by a rational number (so up to to rational isomorphisms). This fixes the curve $\mathcal{E}_S$ at the point $\varphi = 15/2$.

| $p$ | $\varphi \in \mathbb{F}_p$ | $c_p$ |
|-----|-----|-----|
| 5 | 0 | — |
| 7 | 4 | 1 |
| 11 | 2 | 6 |
| 13 | 1 | — |
| 17 | 16 | −6 |
| 19 | 17 | 1 |
| 23 | 19 | 0 |
| 29 | 22 | −6 |
| 31 | 23 | 2 |
| 37 | 26 | 2 |
| 41 | 28 | 0 |
| 43 | 29 | 8 |
| 47 | 31 | 0 |
| 53 | 34 | 0 |
| 59 | 37 | −6 |
| 61 | 38 | 8 |
| 67 | 41 | 8 |
| 71 | 43 | 0 |
| 73 | 44 | 2 |
| 79 | 47 | −10 |
| 83 | 49 | −12 |
| 89 | 52 | 0 |
| 97 | 56 | 8 |

| $p$ | $\alpha_p$ |
|-----|-----|
| 5 | 1 |
| 7 | 1 |
| 11 | 6 |
| 13 | 1 |
| 17 | −6 |
| 19 | 1 |
| 23 | 0 |
| 29 | −6 |
| 31 | 2 |
| 37 | 2 |
| 41 | 0 |
| 43 | 8 |
| 47 | 0 |
| 53 | 0 |
| 59 | −6 |
| 61 | 8 |
| 67 | 8 |
| 71 | 0 |
| 73 | 2 |
| 79 | −10 |
| 83 | −12 |
| 89 | 0 |
| 97 | 8 |

Table 5: *On the left, we list the coefficients $c_p$ appearing in the quadratic factor $R_2(T)$ of the zeta function numerator at $\varphi = 15/2$. The second column displays the image of $\varphi = 15/2$ in the finite field $\mathbb{F}_p$. Where the value for a coefficient $c_p$ is not displayed, this is due to $\varphi$ corresponding to an apparent singularity (see [24]) for that prime. On the right we display the Hecke eigenvalues $\alpha_p$ corresponding to a prime $p$ of the weight-two modular form with LMFDB label **51870.2.a.bs**. Note the perfect agreement when $c_p$ is listed.*

This procedure can be repeated for various values of $\varphi$. We tabulate these twists along with the corresponding curves for a few rational values of $\varphi$ in table 6. Many more points can be studied, and we include more extensive data in appendix B.

| $\varphi$ | $k(\varphi)$ | Modular Form | Elliptic Curve | Elliptic Curve Equation |
|---|---|---|---|---|
| $2$ | $2\sqrt{\frac{3}{23}}$ | **34.2.a.a** | **34.a4** | $xy + y^2 = x^3 - 3x + 1$ |
| $\frac{5}{2}$ | $2\sqrt{\frac{3}{131}}$ | **1290.2.a.h** | **1290.h4** | $xy + y^2 + y = x^3 - 108x + 118$ |
| $3$ | $\sqrt{\frac{3}{11}}$ | **156.2.a.b** | **156.b4** | $y^2 = x^3 + x^2 - 13x - 4$ |
| $\frac{7}{2}$ | $2\sqrt{\frac{3}{227}}$ | **4270.2.a.c** | **4270.c4** | $xy + y^2 + y = x^3 - 388x - 594$ |
| $4$ | $2\sqrt{\frac{3}{71}}$ | **210.2.a.d** | **210.d7** | $xy + y^2 = x^3 - 41x - 39$ |
| $\frac{9}{2}$ | $2\sqrt{\frac{3}{347}}$ | **3318.2.a.e** | **3318.e4** | $xy + y^2 + y = x^3 - 1051x - 6286$ |
| $5$ | $\sqrt{\frac{3}{26}}$ | **220.2.a.a** | **220.a3** | $y^2 = x^3 + x^2 - 100x - 252$ |
| $\frac{11}{2}$ | $2\sqrt{\frac{3}{491}}$ | **6402.2.a.f** | **6402.f4** | $xy + y^2 + y = x^3 - 2361x - 28280$ |
| $6$ | $2\sqrt{\frac{3}{143}}$ | **1590.2.a.u** | **1590.u4** | $xy + y^2 = x^3 - 210x - 828$ |
| $\frac{13}{2}$ | $2\sqrt{\frac{3}{659}}$ | **32890.2.a.b** | **32890.b4** | $xy + y^2 + y = x^3 - 4654x - 90324$ |
| $7$ | $\sqrt{\frac{3}{47}}$ | **2604.2.a.d** | **2604.d4** | $y^2 = x^3 + x^2 - 393x - 2448$ |
| $\frac{15}{2}$ | $2\sqrt{\frac{3}{851}}$ | **51870.2.a.bs** | **51870.bs4** | $xy + y^2 + y = x^3 - 8338x - 235312$ |
| $8$ | $2\sqrt{\frac{3}{239}}$ | **994.2.a.e** | **994.e4** | $xy + y^2 = x^3 - 678x - 5660$ |

Table 6: *Some modular forms corresponding to the quadratic factors of the numerator of the zeta function, the corresponding elliptic curves and twists relating these curves to the F-theory fibre (67).*

Based on the data in table 6 and the more complete data collected in the appendices, we find a root of a simple rational function

$$k(\varphi) = \sqrt{\frac{3}{3\varphi^2 + 6\varphi - 1}}$$

that gives the parameters $k(\varphi)$ up to an overall multiplicative rational constant, which does not affect the rational structure of the curves $\mathcal{E}_S$. Substituting this parameter to the equation (38) gives the curve over $\mathbb{Q}(\varphi)$

$$y^2 = x^3 - \frac{9\varphi^4 - 12\varphi^3 + 30\varphi^2 - 12\varphi + 1}{3}x - \frac{2\left(27\varphi^6 - 54\varphi^5 - 135\varphi^4 + 180\varphi^3 - 99\varphi^2 + 18\varphi - 1\right)}{27} \ . \quad (70)$$

This family $\mathcal{E}_S$ of twisted Sen curves coincides with the modular curve $\mathcal{E}_R$, up to a rational isomorphism. It should be noted that since this result relies on data obtained by numerical computations,

this should strictly speaking be viewed as a conjecture. However, we have tested this for hundreds of rational values of $\varphi$, finding that the results agree perfectly, giving a strong indication of this conjecture's veracity. We tabulate the values of $\varphi$ with the corresponding modular forms and elliptic curves in appendix B.

One interesting observation that can be made from the data gathered in appendix B is that among the curves with equal $j$-invariant $j(\varphi)$, the one corresponding to the correct modular form is, for all examples we have studied, given by the curve with the smallest conductor. This phenomenon does not appear in the examples studied in [27], nor is it true for the other manifolds we study.

The family of fibres with the complex structure parameter $\widetilde{\tau} = \tau/2$ can be treated similarly, which reveals that the twist can be taken to be given by

$$\widetilde{k}(\varphi) = \mathrm{i}\sqrt{\frac{3}{2(3\varphi^2 + 6\varphi - 1)}} = \sqrt{\frac{3}{2(1 - 6\varphi - 3\varphi^2)}} \,,$$

where the second expression is to remind us that the branch of the square root is chosen such that $\widetilde{k}(0) = \sqrt{3/2}$. The family of Sen curves $\mathcal{E}_S$ with the same rational structures as the curves $\mathcal{E}_R$ is given by

$$y^2 = x^3 - \frac{(3\varphi+1)(3\varphi^3+75\varphi^2-15\varphi+1)}{3}x + \frac{2(3\varphi^2+6\varphi-1)(9\varphi^4+30\varphi^2+30\varphi^2-12\varphi+1)}{27} \,. \quad (71)$$

Note that when $\varphi \in \mathbb{Q}$, the curves $\mathcal{E}_S$ and $\widetilde{\mathcal{E}}_S$ given by (70) and (71) can be verified to be isogenous, as demanded by the fact that both are identified with the same modular form associated to the zeta function. For instance, when $\varphi = 3$, these curves have LMFDB labels **156.b4** and **156.b3**, and both belong to the isongeny class with label **156.b**.

### 6.4. Field extensions

Following the discussion in §4.6, we wish to consider also values of $\varphi$ in number fields $\mathbb{Q}(\sqrt{n})$, and in this subsection give explicit examples of how the zeta function coefficients $c_p(\varphi)$ can be identified as Fourier coefficients of Hilbert or Bianchi modular forms.

*Real quadratic points and Hilbert modular forms*
To see explicitly how the correspondences work in the real quadratic case, let us take $\varphi = 2 + \sqrt{3}$. In this case the $j$-invariant is given by

$$j\left(\tau\left(2 + \sqrt{3}\,\right)\right) = \frac{1}{23}\left(22288 + 11904\sqrt{3}\right) \,,$$

and the twisted Sen curve $\mathcal{E}_S$, which we now expect to be isomorphic to $\mathcal{E}_R$ over $\mathbb{Q}(\sqrt{3})$, is given by

$$y^2 = x^3 - \left(\frac{187}{12} + 9\sqrt{3}\right)x - \frac{21\sqrt{3}}{4} + \frac{491}{54} \,.$$

This is isomorphic to the elliptic curve with LMFDB label **92.2-b2**, whose corresponding Hilbert cusp forms have labels **2.2.12.1-92.1-b** and **2.2.12.1-92.2-b**. These two modular forms differ only by swapping eigenvalues corresponding to primes of $\mathbb{Q}(\sqrt{3})$ with the same norm. The Hecke eigenvalues corresponding to the first few primes of $\mathbb{Q}(\sqrt{3})$ are given in the first subtable of table 7.

To find the corresponding coefficients $c_p$ in the quadratic factor of the zeta function, we must find a representative of $\sqrt{3}$ in $\mathbb{F}_p$, which as discussed in §4.6 means solving the equation $x^2 - 3 = 0$

| $\mathfrak{p}$ | $N(\mathfrak{p})$ | $\alpha_{\mathfrak{p}}$ |
|---|---|---|
| $(11, 1 - 2\sqrt{3})$ | 11 | $-6$ |
| $(11, 1 + 2\sqrt{3})$ | 11 | $0$ |
| $(13, 4 + \sqrt{3})$ | 13 | $-4$ |
| $(13, 4 - \sqrt{3})$ | 13 | $-4$ |
| $(23, 2 - 3\sqrt{3})$ | 23 | $-1$ |
| $(23, 2 + 3\sqrt{3})$ | 23 | $0$ |
| $(37, -7 + 2\sqrt{3})$ | 37 | $-4$ |
| $(37, -7 - 2\sqrt{3})$ | 37 | $2$ |
| $(47, -1 + 4\sqrt{3})$ | 47 | $0$ |
| $(47, -1 - 4\sqrt{3})$ | 47 | $0$ |
| $(59, -4 + 5\sqrt{3})$ | 59 | $12$ |
| $(59, -4 - 5\sqrt{3})$ | 59 | $-12$ |
| $(61, -8 + \sqrt{3})$ | 61 | $8$ |
| $(61, -8 - \sqrt{3})$ | 61 | $-10$ |
| $(71, -2 + 5\sqrt{3})$ | 71 | $0$ |
| $(71, -2 - 5\sqrt{3})$ | 71 | $12$ |
| $(73, -10 + 3\sqrt{3})$ | 73 | $2$ |
| $(73, -10 - 3\sqrt{3})$ | 73 | $-10$ |
| $(83, -5 + 6\sqrt{3})$ | 83 | $-12$ |
| $(83, -5 - 6\sqrt{3})$ | 83 | $-6$ |
| $(97, -10 - \sqrt{3})$ | 97 | $2$ |
| $(97, -10 + \sqrt{3})$ | 97 | $14$ |

| $p$ | $\varphi \in \mathbb{F}_p$ | $c_p(\varphi)$ |
|---|---|---|
| 11 | 8 | $-6$ |
| 11 | 7 | $0$ |
| 13 | 6 | $-4$ |
| 13 | 11 | $-4$ |
| 23 | 18 | $-1$ |
| 23 | 9 | $0$ |
| 37 | 24 | $-4$ |
| 37 | 17 | $2$ |
| 47 | 14 | $0$ |
| 47 | 37 | $0$ |
| 59 | 50 | $12$ |
| 59 | 13 | $-12$ |
| 61 | 10 | $8$ |
| 61 | 55 | $-10$ |
| 71 | 45 | $0$ |
| 71 | 30 | $12$ |
| 73 | 54 | $2$ |
| 73 | 23 | $-10$ |
| 83 | 72 | $-12$ |
| 83 | 15 | $-6$ |
| 97 | 89 | $2$ |
| 97 | 12 | $14$ |

Table 7: *On the left, we list the Hecke eigenvalues $\alpha_{\mathfrak{p}}$ of the Hilbert cusp form with the LMFDB label* **2.2.12.1-92.1-b**, *corresponding to primes $\mathfrak{p}$ of $\mathbb{Q}(\sqrt{3})$ with norms $N(\mathfrak{p})$. On the right, we display the representatives of $\varphi = 2 + \sqrt{3}$ in the finite fields $\mathbb{F}_p$ and the coefficients $c_p$ in the quadratic factor of the local zeta function numerator corresponding to these representatives. If a prime is not listed, it means that the representative of $2 + \sqrt{3}$ does not exist for that prime. Note again the perfect agreement of $\alpha_{\mathfrak{p}}$ with $c_p$.*

mod $p$. We list the representatives of $\varphi = 2 + \sqrt{3}$ alongside the zeta function coefficients $c_p(\varphi)$ in the second subtable of table 7

These considerations can be repeated for multiple values of the complex structure modulus $\varphi$ in totally real number fields. As expected, in every case we have studied where the curve $\mathcal{E}_S$ can be found on LMFDB, the Hilbert modular form associated to the this curve agrees with the modular form related to the zeta function. We list more values of $\varphi$ in totally real field extensions together with the corresponding Hilbert modular forms in appendix B.

The considerations regarding isogenies continue to hold for members of the family $\mathcal{E}_S$ defined over field extensions. In particular, the rescaling $\tau \mapsto \widetilde{\tau} = \tau/2$ of the complex structure parameter that we saw to give an isogeny over $\mathbb{Q}$ for the family of rational elliptic curves, also gives an isogeny over $\mathbb{Q}(\sqrt{3})$. For $\varphi = 2 + \sqrt{3}$, the corresponding value of the $j$-function is

$$j\left(\widetilde{\tau}(2 + \sqrt{3}\,)\right) \;=\; \frac{1}{529}\left(35168888 + 15980748\sqrt{3}\right) ,$$

and the curve $\widetilde{\mathcal{E}}_S$ is given, up to an isomorphism over $\mathbb{Q}(\sqrt{3})$, by

$$y^2 \;=\; x^3 - \left(\frac{1747}{12} + 84\sqrt{3}\right)x - \frac{25616}{27} + \frac{2191}{4}\sqrt{3} .$$

This is isomorphic over $\mathbb{Q}(\sqrt{3})$ to the elliptic curve with LMFDB label **92.2-b3**, and belongs to the same isogeny class as the curve **92.2-b2** corresponding to the unscaled lattice parameter.

*Imaginary quadratic fields and Bianchi modular forms*

As a simple example of a modular curve defined over quadratic imaginary number field, consider the curve $\mathcal{E}_S$ with $\varphi = i$. This is isomorphic over $\mathbb{Q}(i)$ to the elliptic curve with the LMFDB label **164.1-a2**, given by

$$y^2 \;=\; x^3 + \frac{5}{12}x + \left(\frac{2}{27} + \frac{i}{4}\right) .$$

Its $j$-invariant is

$$j(\tau(i)) \;=\; -\frac{1}{41}\left(10000 + 8000\,i\right) .$$

This elliptic curve has two associated Bianchi modular forms with LMFDB labels **2.0.4.1-164.1-a** and **2.0.4.1-164.2-a**. These forms differ, as was the case with the Hilbert modular forms in the previous section, only by permutation of the Hecke eigenvalues corresponding to the two primes with the same norm. The Hecke eigenvalues $\alpha_{\mathfrak{p}}$ of **2.0.4.1-164.1-a** are listed in table 8. As with Hilbert modular forms, we can find representatives of $\varphi = i$ in finite fields $\mathbb{F}_p$ by finding solutions to the equation $x^2 + 1 = 0 \mod p$. When such solutions exist, this allows us to associate two coefficients $c_p$ appearing in the quadratic factor of the zeta function numerator to the point $\varphi = i$. As expected, we find agreement with these coefficients and the Hecke eigenvalues of the Bianchi modular form **2.0.4.1-164.1-a**. We display the coefficients $\alpha_{\mathfrak{p}}$ and $c_p$ for primes $5 \leqslant p \leqslant 100$ in table 8, although we have checked the agreement for the first 100 primes.

One can check that the scaling $\tau \mapsto \tau/2$ gives again an isogeny over $\mathbb{Q}(i)$. The curve associated to $\widetilde{\tau}$ at $\varphi = i$ is **164.1-a1**, and has the Weierstrass equation

$$y^2 \;=\; x^3 + \left(\frac{5}{12} + 5i\right)x + \left(\frac{193}{54} + \frac{31}{12}i\right) .$$

The $j$-invariant is given by

$$j(\widetilde{\tau}(i)) \;=\; -\frac{1}{1681}\left(16234000 + 31900500\,i\right) .$$

| $\mathfrak{p}$ | $N(\mathfrak{p})$ | $\alpha_{\mathfrak{p}}$ |
|---|---|---|
| $(-2-i)$ | 5 | 0 |
| $(1+2i)$ | 5 | 0 |
| $(-2-3i)$ | 13 | 2 |
| $(3+2i)$ | 13 | $-4$ |
| $(4+i)$ | 17 | 6 |
| $(-4+i)$ | 17 | $-6$ |
| $(5-2i)$ | 29 | 6 |
| $(5+2i)$ | 29 | 0 |
| $(6+i)$ | 37 | 8 |
| $(-6+i)$ | 37 | $-4$ |
| $(5+4i)$ | 41 | $-6$ |
| — | 41 | — |
| $(7-2i)$ | 53 | 6 |
| $(7+2i)$ | 53 | 12 |
| $(-5-6i)$ | 61 | 2 |
| $(6+5i)$ | 61 | $-10$ |
| $(-8-3i)$ | 73 | 14 |
| $(-8+3i)$ | 73 | $-10$ |
| $(8-5i)$ | 89 | 6 |
| $(5+8i)$ | 89 | 6 |
| $(9-4i)$ | 97 | 2 |
| $(9+4i)$ | 97 | 2 |

| $p$ | $\varphi \in \mathbb{F}_p$ | $c_p$ |
|---|---|---|
| 5 | 3 | 0 |
| 5 | 2 | 0 |
| 13 | 8 | 2 |
| 13 | 5 | $-4$ |
| 17 | 13 | 6 |
| 17 | 4 | $-6$ |
| 29 | 17 | 6 |
| 29 | 12 | 0 |
| 37 | 31 | 8 |
| 37 | 6 | $-4$ |
| 41 | 9 | $-6$ |
| 41 | 32 | $-1$ |
| 53 | 30 | 6 |
| 53 | 23 | 12 |
| 61 | 50 | 2 |
| 61 | 11 | $-10$ |
| 73 | 46 | 14 |
| 73 | 27 | $-10$ |
| 89 | 55 | 6 |
| 89 | 34 | 6 |
| 97 | 75 | 2 |
| 97 | 22 | 2 |

Table 8: *On the left, we tabulate the Hecke eigenvalues $\alpha_{\mathfrak{p}}$ corresponding to primes $\mathfrak{p}$ of $\mathbb{Q}(i)$ with norm $N(\mathfrak{p})$ of the Bianchi cusp form with LMFDB label **2.0.4.1-164.1-a**. On the right, the coefficients $c_p$ appearing in the quadratic part of the local zeta function numerator are listed together with the representatives of $\varphi = i$ in finite fields $\mathbb{F}_p$. The Hecke eigenvalues and the zeta function coefficients agree when both exist.*

This procedure can be repeated for various values of $\varphi$ in quadratic imaginary number fields. In all cases we have studied, when the curve $\mathcal{E}_S$ is found in LMFDB, agreement between the elliptic curves $\mathcal{E}_S$ and $\mathcal{E}_R$ is found. We compile the points studied, together with their associated elliptic curves and modular forms in appendix B.

*Base changes*

To illustrate the base changes discussed in §4.6, consider $\varphi = -1 + i\sqrt{3} \in \mathbb{Q}\left(\sqrt{-3}\right)$. The $j$-invariant of the fibre corresponding to this elliptic curve is a rational number

$$j\left(-1 + i\sqrt{3}\right) = \frac{9938375}{21952} \; ,$$

and indeed $\mathcal{E}_S$, which is isomorphic over $\mathbb{Q}(\sqrt{-3})$ to the elliptic curve with the LMFDB label **196.2-a3**, is defined over $\mathbb{Q}$ and given by

$$y^2 = x^3 + \frac{215}{48}x - \frac{5291}{864} \; .$$

Thus we can think of this as a base-change curve.

As in the previous subsection, we can find the coefficients $c_p$ in the zeta function corresponding to $\varphi = -1 + i\sqrt{3}$. We list them in table 9 and identify them with the coefficients of the Bianchi modular form **2.0.3.1-196.2-a**. The Hecke eigenvalues corresponding to primes with the same norm are equal, which is a reflection of the base change, as these are exactly the Hecke eigenvalues $c_{N(\mathfrak{p})}$ of the elliptic modular forms **14.2.a.a** and **126.2.a.b**.

| $p$ | 13 | 13 | 19 | 19 | 31 | 31 | 37 | 37 | 43 | 43 | 61 | 61 | 67 | 67 | 73 | 73 | 79 | 79 | 97 | 97 |
|---|---|---|---|---|---|---|---|---|---|---|---|---|---|---|---|---|---|---|---|---|
| $\varphi \in \mathbb{F}_p$ | 5 | 6 | 3 | 14 | 10 | 19 | 15 | 20 | 12 | 29 | 26 | 33 | 7 | 58 | 16 | 55 | 31 | 46 | 25 | 70 |
| $c_p$ | -4 | -4 | 2 | 2 | -4 | -4 | 2 | 2 | 8 | 8 | 8 | 8 | -4 | -4 | 2 | 2 | 8 | 8 | -10 | -10 |

Table 9: *The coefficients $c_p$ appearing in the quadratic part of the zeta function numerator are listed together with the representatives of $\varphi = -1 + i\sqrt{3}$ in finite fields $\mathbb{F}_p$. These numbers can be identified with the Hecke eigenvalues of the Bianchi cusp form **2.0.3.1-196.2-a3**, and of the newforms **14.2.a.a** and **126.2.a.b**.*

## 6.5. Modularity and geometry

The Hulek-Verrill manifold, which is birational to the simultaneous vanishing locus of (57), stands out among our examples as being particularly well-studied with respect to modularity. In particular, modularity of varieties associated to certain conifold points in the moduli space of the family (57) was demonstrated rigorously in [37]. Above, we have provided arguments that similar modularity properties hold for all manifolds on the $\mathbb{Z}_2$ locus in the moduli space, and also provided extensive evidence in the case of the $S_5$ symmetric family.

In this section, we review some of the observations made by Hulek and Verrill in [37], and adapt them to our case in order to give an intuitive account explaining the splitting of the cohomology as a consequence of $\mathcal{E}_R$ defining a non-trivial cycle in the threefold. We anticipate that there is a similar relation between the geometry of the threefold and the modular curve for all of our examples. This is curious from the F-theory perspective, as the elliptic curve then makes two appearances in the fourfold: once as the fibre over a generic point of the base, and again in the base.

*The elliptic surface $\mathcal{G}_{abc}$*

To proceed with the analysis of the threefold, consider the elliptic surface $\mathcal{G}_{abc}$, which is a resolution of the following singular surface in $\mathbb{P}^2 \times \mathbb{P}^1$:

$$\mathcal{G}'_{abc}: \quad (x + y + z)(c\,xy + a\,yz + b\,xz)t_0 - t_1 xyz = 0 \;, \qquad (x : y : z) \in \mathbb{P}^2 \;, \quad (t_0 : t_1) \in \mathbb{P}^1 \;.$$

This surface is a fibration over $\mathbb{P}^1$ with coordinate $t = t_1/t_0$, and it will be convenient to denote by $\mathcal{E}'_{abc}(t)$ the fibre over the point $t \in \mathbb{P}^1$.

The singularities of the surface $\mathcal{G}'_{abc}$ are the singularities of the fibre at $t = \infty$. The fibre $\mathcal{E}'_{abc}(\infty)$ is given by $xyz = 0$, which has the three singular points $(1 : 0 : 0)$, $(0 : 1 : 0)$, and $(0 : 0 : 1)$. The resolution of these singularities results gives a smooth surface $\mathcal{G}_{abc}$ whose fibre $\mathcal{E}_{abc}(\infty)$ is an $I_6$ fibre.

Generically, in addition to this $I_6$ fibre, the smooth surface $\mathcal{G}_{abc}$ has five singular fibres. One of these is the singular fibre at $t = 0$ which is an $I_2$ Kodaira fibre, consisting of two rational curves meeting at two points. For the nongeneric case $a^2 + b^2 + c^2 = 2(ab + ac + bc)$, this fibre is of type III, and has two components that meet at a triple point.

Additionally, there are generically four singular fibres at the following values of $t$:

$$t_{++} = \sqrt{a} + \sqrt{b} + \sqrt{c} \,, \quad t_{+-} = \sqrt{a} + \sqrt{b} - \sqrt{c} \,, \quad t_{-+} = \sqrt{a} - \sqrt{b} + \sqrt{c} \,, \quad t_{--} = \sqrt{a} - \sqrt{b} - \sqrt{c} \,.$$

For each such value of $t$, $\mathcal{E}_{abc}(t)$ is generically an $I_1$ fibre. For nongeneric values of the parameters $a, b, c$ these $I_1$ fibres can collide so that we get more complicated singular fibres. Of particular interest is the case $b = c$: then $t_{-+} = t_{+-} = \sqrt{a}$ and the fibre over this point is of type $I_2$. This is exactly the case where the manifold is $\mathbb{Z}_2$ symmetric.

A Weierstrass model can be given for the elliptic curve $\mathcal{E}_{abc}(t)$. This is

$$y^2 = \left( x + \frac{2\left(t^2 + a^2 + b^2 + c^2\right) - (t + a + b + c)^2}{8a^2} \right)^2 x - \frac{A(a, b, c, t)}{64a^4} x \,,$$

where $\quad A(a, b, c, t) = \prod_{\alpha, \beta = \pm 1} \left( t - \left( \sqrt{a} + \alpha\sqrt{b} + \beta\sqrt{c} \right)^2 \right) \,.$

For $t = 1$ this quantity $A$ becomes $\Delta_F$ as in (63). The $j$-invariant of this elliptic curve is

$$j\left(\mathcal{E}_{abc}(t)\right) = \frac{\left(A(a, b, c, t) + 16abct\right)^3}{(abct)^2 A(a, b, c, t)} \,, \tag{72}$$

which was also computed in [86].

*Fibred product and modularity*

The Hulek-Verrill manifold is birational to the fibred product $\mathcal{G}_{\varphi^0, \varphi^1, \varphi^2} \times_{\mathbb{P}^1} \mathcal{G}_{\varphi^3, \varphi^4, \varphi^5}$ over $\mathbb{P}^1$. To see this, consider again the equations (57):

$$X_0 + X_1 + X_2 = -(X_3 + X_4 + X_5) \,,$$

$$\frac{\varphi^0}{X_0} + \frac{\varphi^1}{X_1} + \frac{\varphi^2}{X_2} = -\left( \frac{\varphi^3}{X_3} + \frac{\varphi^4}{X_4} + \frac{\varphi^5}{X_5} \right) \,,$$

defining a variety on $\mathbb{T}^5$ birational to the Hulek-Verrill manifold. As a consequence of these relations, we can write

$$(X_0 + X_1 + X_2)\left( \frac{\varphi^0}{X_0} + \frac{\varphi^1}{X_1} + \frac{\varphi^2}{X_2} \right) t_0 = t_1 \,,$$

$$(X_3 + X_4 + X_5)\left( \frac{\varphi^3}{X_3} + \frac{\varphi^4}{X_4} + \frac{\varphi^5}{X_5} \right) t_0 = t_1 \,, \qquad (t_0 : t_1) \in \mathbb{P}^1 \,,$$

which defines a fibred product $\mathcal{E}_{\varphi^0,\varphi^1,\varphi^2}(t) \times_{\mathbb{P}^1} \mathcal{E}_{\varphi^3,\varphi^4,\varphi^5}(t) \subset \mathbb{P}^2 \times \mathbb{P}^2 \times \mathbb{P}^1$ birational to the Hulek-Verrill manifold. This fibration structure can be used to show that $\mathbb{Z}_2$ symmetric Hulek-Verrill manifolds contain an elliptic surface birational to $\mathcal{E} \times \mathbb{P}^1$, with $\mathcal{E}$ an elliptic curve. To be concrete, let us take $\varphi^1 = \varphi^2 = \varphi$. We also make a choice of patch, setting $\varphi^0 = 1$, and denote the remaining three $\varphi^\mu$ by $\psi^1, \psi^2, \psi^3$. Then consider the intersection of $\mathrm{HV}_{\boldsymbol{\varphi}} \cap \mathbb{T}^5$ with a hypersurface $X_1 + X_2 = 0$. From the above equations it follows that the intersection is given by

$$(X_3 + X_4 + X_5)\left(\frac{\psi^1}{X_3} + \frac{\psi^2}{X_4} + \frac{\psi^3}{X_5}\right) = 1 ,$$

and so is birational to $\mathcal{E}_{\psi^1,\psi^2,\psi^3}(1) \times \mathbb{P}^1$. An alternative way of seeing this is to note that this elliptic surface is a subvariety of the fibred product $\mathcal{E}_{1,\varphi,\varphi}(t) \times_{\mathbb{P}^1} \mathcal{E}_{\psi^1,\psi^2,\psi^3}(t)$ at $t = 1$. At this point, the fibre $\mathcal{E}_{1,\varphi,\varphi}(t)$ is a singular fibre of type $I_2$ consisting of two copies of $\mathbb{P}^1$ intersecting at two points. The intersection picks out one of these projective lines, thus giving $\mathcal{E}_{\psi_1,\psi_2,\psi_3}(1) \times \mathbb{P}^1$. The fibred product $\mathcal{E}_{1,\varphi,\varphi}(t) \times_{\mathbb{P}^1} \mathcal{E}_{\psi^1,\psi^2,\psi^3}(t)$ is sketched in figure 3 with the elliptic surface corresponding to the product of curves highlighted in magenta. The embedding map,

$$\mathcal{E}_{\psi^1,\psi^2,\psi^3}(1) \times \mathbb{P}^1 \dashrightarrow \mathrm{HV}_{(1,\varphi,\varphi,\psi^1,\psi^2,\psi^3)} \cap \mathbb{T}^5 ,$$

induces a corresponding map of homologies

$$H_1\big(\mathcal{E}_{\psi^1,\psi^2,\psi^3}(1), \mathbb{Z}\big) \times H_2\big(\mathbb{P}^1, \mathbb{Z}\big) \rightarrow H_3\big(\mathrm{HV}_{(1,\varphi,\varphi,\psi^1,\psi^2,\psi^3)}, \mathbb{Z}\big) .$$

It can be shown that the cycles corresponding to the two three-cycles on the elliptic surface

$$\{X_1 + X_2 = 0\} \cap \mathrm{HV}_{\boldsymbol{\varphi}} \cap \mathbb{T}^5 \stackrel{\mathrm{bir.}}{\sim} \mathcal{E}_{\psi^1,\psi^2,\psi^3}(1) \times \mathbb{P}^1$$

are non-trivial in the homology. The corresponding non-trivial cohomology elements correspond to the product of one of the one-forms on the elliptic curve $\mathcal{E}_{\psi^1,\psi^2,\psi^3}(1)$ with the volume form on $\mathbb{P}^1$, and thus are of the Hodge types $(1,0) + (1,1) = (2,1)$ and $(0,1) + (1,1) = (1,2)$. This can be used to explain the splitting of the middle cohomology, and therefore the modularity, of the $\mathbb{Z}_2$ symmetric Hulek-Verrill manifolds. A rigorous account that solidifies our explanation can be found in the original work of Hulek and Verrill [37].

*Asides: other examples and supersymmetric F-theory configurations*

We have seen that, for the Hulek-Verrill example, the modular curve $\mathcal{E}_R$ appears in a subvariety $\mathcal{E}_R \times \mathbb{P}^1 \subset \mathrm{HV}_{(1,\varphi,\varphi,\psi^1,\psi^2,\psi^3)}$ of the threefold which defines non-trivial elements in the third homology, thus explaining the modularity of the manifold geometrically. We expect, in the spirit of the Hodge conjecture, that there exists a similar geometric reason for modularity in other cases, including those which we study in the next section.

One question for future work concerns the Calabi-Yau fourfold $Y$, where F theory on $Y$ reduces in the orientifold limit to a supersymmetric vacuum of an orientifolded IIB flux compactification [31] on a threefold $X$. The fourfold $Y$ is a fibration $\mathcal{E} \rightarrow Y \rightarrow \mathcal{B}$, and $X$ is a double cover of $\mathcal{B}$, which can be explicity constructed via the methods of [87, 88]. Do $\mathcal{B}$ and $X$ contain surfaces $\mathcal{E} \times \mathbb{P}^1$ that give rise to three-cycles inducing an appropriate splitting of cohomology? If this were true for $X$ then it would establish the flux modularity conjecture. It is also an interesting question whether the fourfolds themselves are modular, and if the F-theory fibre is related to its modularity, especially in the light of the fact that, on the level of the Picard-Fuchs equations, these fourfolds[12] can be viewed as a product space of the elliptic fibre and the base [89–91].

---

[12]The fourfolds appearing in the F-theory lift of the IIB flux vacuum solutions are in what is referred to in [89] as a global Sen limit, which means that the elliptic fibre has a constant complex structure along the base.

While our analysis for the case of $\mathbb{Z}_2$ symmetric Hulek-Verrill manifolds, thus far, shows that the modular curve is a subvariety of the base, it would be interesting to further investigate this and we speculate that the base does in fact contain a surface $\mathcal{E} \times \mathbb{P}^1$ which is non-trivial in homology. Proving this might be accomplished by some adaptation of the analysis of Hulek and Verrill to the variety $\mathcal{B}$.

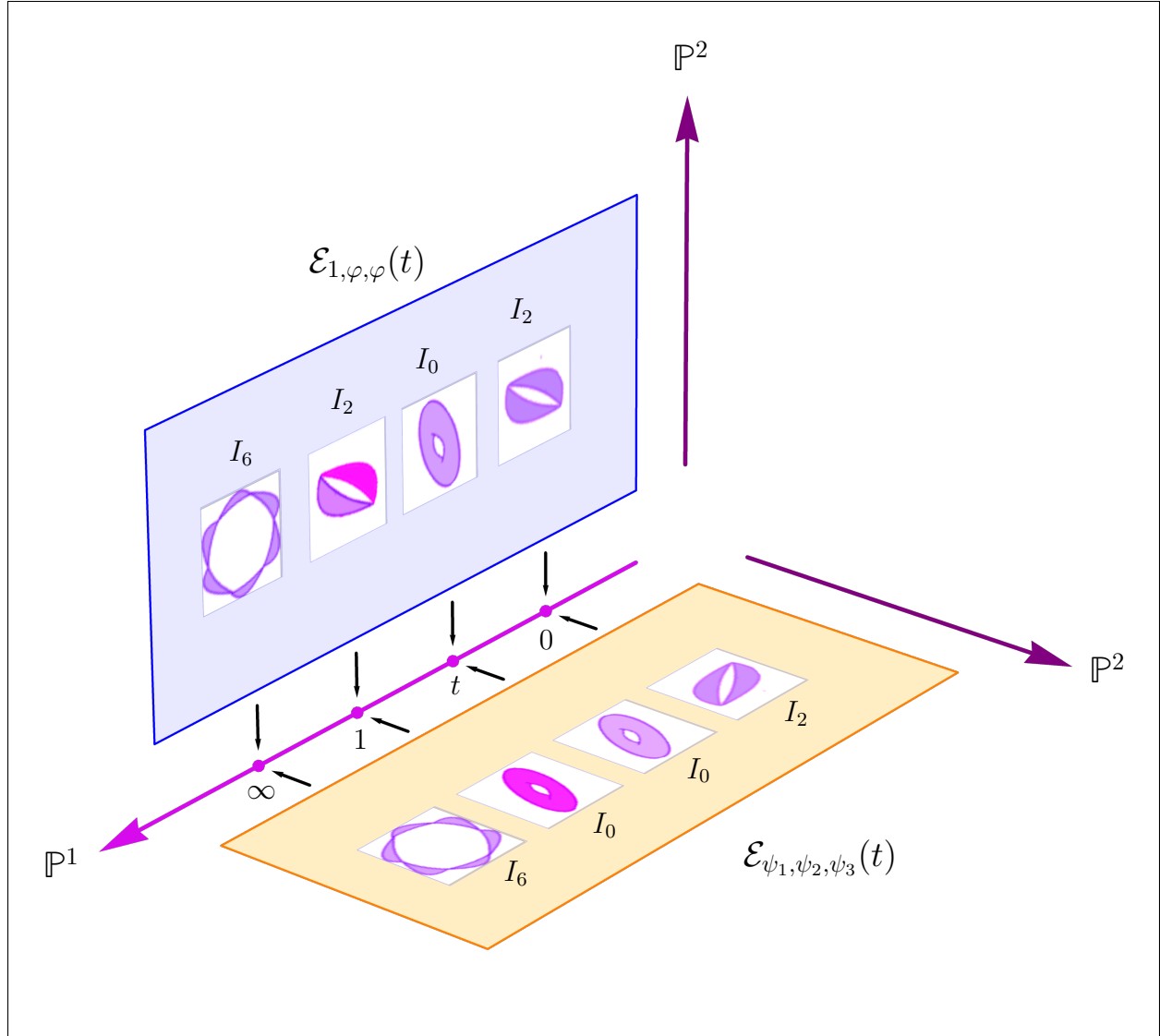

Figure 3: *The $\mathbb{Z}_2$ symmetric Hulek-Verrill manifolds are birational to fibred products $\mathcal{E}_{1,\varphi,\varphi}(t) \times_{\mathbb{P}^1} \mathcal{E}_{\psi_1,\psi_2,\psi_3}(t)$ over $\mathbb{P}^1$ with generic fibers elliptic curves $\mathcal{E}_{\psi_1,\psi_2,\psi_3}(t)$ and $\mathcal{E}_{1,\varphi,\varphi}(t)$. In this sketch, this situation is depicted for generic values of $\varphi, \psi_1, \psi_2$ and $\psi_3$. For generic $t \in \mathbb{P}^1$ the two elliptic curves are both smooth, however the fibers are singular for $t = 0, 1, \infty$, and certain other values. When $t = 0$, both elliptic curves are singular curves of type $I_2$; when $t = 1$ one elliptic curve is smooth, while the other is of type $I_2$; and when $t = \infty$, both elliptic curves are singular of type $I_6$. The remaining singular fibers over the points $t_{\pm\pm}$, which we do not display, contain $I_1$ singularities. The elliptic surface birational to the intersection $\{X_1 + X_2 = 0\} \cap HV_{\boldsymbol{\varphi}} \cap \mathbb{T}^5$ consists of one of the distinct components of the singular fibre $\mathcal{E}_{1,\varphi,\varphi}(t)$, which is of type $I_2$, together with the smooth elliptic curve $\mathcal{E}_{\psi^1,\psi^2,\psi^3}(1)$. This elliptic surface $\mathcal{E}_{\psi^1,\psi^2,\psi^3}(1) \times \mathbb{P}^1$ is highlighted in magenta.*

## 7. Additional Examples

The methods described in §3 and §4, and illustrated in detail with the example of the $S_5$ symmetric family of Hulek-Verrill manifolds in the previous section, can be applied to a wide variety of manifolds with at least a $\mathbb{Z}_2$ symmetry. In this section, we give two additional examples of families of such manifolds, supersymmetric flux vacua supported by them, and aspects of their F-theory lifts. We also briefly explore some concepts related to modularity, such as isogenies and twists.

### 7.1. The mirror bicubic

This manifold is mirror to the CICY[13]

$$
\begin{matrix} \mathbb{P}^2 \\ \mathbb{P}^2 \end{matrix} \begin{bmatrix} 3 \\ 3 \end{bmatrix}_{\chi=-162} ,
$$

and birational [92] to the singular hypersurface in $\mathbb{P}^2 \times \mathbb{P}^2$ given by the vanishing locus of

$$
X_0 X_1 X_2 Y_0 Y_1 Y_2 + \left( X_1^2 X_2 + X_1 X_2^2 - \varphi^1 X_0^3 \right) Y_0 Y_1 Y_2 + \left( Y_1^2 Y_2 + Y_1 Y_2^2 - \varphi^2 Y_0^3 \right) X_0 X_1 X_2 .
$$

The constants $Y_{abc}$ are given by

$$
Y_{ijk} = \begin{cases} 0 & i = j = k \\ 3 & \text{otherwise} \end{cases} , \quad Y_{ij0} = \begin{cases} 0 & i = j \\ \frac{1}{2} & i \neq j \end{cases} , \quad Y_{i00} = -3 , \quad Y_{000} = 486 \frac{\zeta(3)}{(2\pi i)^3} .
$$

The periods can be written in terms of integrals of hypergeometric and Meijer G-functions (see [93] for the definitions used here):

$$
\varpi^0 = \int_0^\infty du\, e^{-u}\, {}_0F_2 \left( 1, 1;\ u^3 \varphi^1 \right)\, {}_0F_2 \left( 1, 1;\ u^3 \varphi^2 \right) ,
$$

$$
\varpi^i = -\int_0^\infty du\, e^{-u}\, G_{0,3}^{2,0} \left( 0,0,0 | u^3 \varphi^i \right)\, {}_0F_2 \left( 1, 1;\ u^3 \varphi^j \right) , \qquad j \neq i ,
$$

$$
\varpi_i = 3 \int_0^\infty du\, e^{-u} \Big[ G_{0,3}^{2,0} \left( 0,0,0 | u^3 \varphi^i \right)\, G_{0,3}^{2,0} \left( 0,0,0 | u^3 \varphi^j \right)
$$

$$
+ {}_0F_2 \left( 1, 1;\ u^3 \varphi^i \right) \left( G_{0,3}^{3,0} \left( 0,0,0 | -u^3 \varphi^j \right) + i\pi G_{0,3}^{2,0} \left( 0,0,0 | u^3 \varphi^j \right) \right) \Big]
$$

$$
- 2\pi^2 \varpi_0 , \qquad\qquad\qquad j \neq i .
$$

We have not displayed $\varpi_0$, which does not enter into the following analysis. From the above expressions it follows that the axiodilaton profile on the $\mathbb{Z}_2$ symmetric locus $\varphi^1 = \varphi^2 = \varphi$ is given by

$$
\tau(\varphi) = \frac{3i}{2\pi} \frac{\int_0^\infty du\, e^{-u} \left[ {}_0F_2 \left( 1, 1;\ u^3 \varphi \right) G_{0,3}^{3,0}(0,0,1 | -u^3 \varphi) - {}_0F_2 \left( 2, 2;\ u^3 \varphi \right) G_{0,3}^{3,0}(1,1,1 | -u^3 \varphi) \right]}{\int_0^\infty du\, e^{-u} \left[ {}_0F_2 \left( 1, 1;\ u^3 \varphi \right) G_{0,3}^{2,0}(0,1,0 | u^3 \varphi) + {}_0F_2 \left( 2, 2;\ u^3 \varphi \right) G_{0,3}^{2,0}(1,1,1 | u^3 \varphi) \right]} - 1 .
$$

Even though it is not immediately obvious, this expression has, for real $\varphi$, a real part equal to $\frac{1}{2}$. When comparing to (26), one should bear in mind that we have further simplified the ratio of integrals to arrive at the above expression.

---

[13]This configuration should not be confused with its transpose, $\mathbb{P}^5[3, 3]$ .

Numerical methods strongly suggest that the $j$-invariant $j(\tau(\varphi))$ is given by the following rational function of the moduli:

$$j(\tau(\varphi)) \;=\; -\frac{(24\varphi+1)^3}{\varphi^3(27\varphi+1)} \; . \tag{73}$$

The modulus-dependent constant $C(\varphi)$ determining the F-theory fibre given by (14) can be found by equating $j_C$ with the $j$-invariant of the elliptic fibre of the F-theory lift, given by (73). There are three choices that are related by a twist, as discussed in §4.4, and are thus ultimately equivalent. In this section, we choose the real root for $C(\varphi)$ but the resulting expression is somewhat complicated, so we will not display it here.

We can use the zeta function data to find the isogeny class of $\mathcal{E}_R$, and then use the $j$-invariant (73) to fix the curve within this equivalence class. This can, in turn, be used to find a twist that turns $\mathcal{E}_F$ into the twisted Sen curve $\mathcal{E}_S$ that is isomorphic to $\mathcal{E}_R$ over $\mathbb{Q}$. This twist parameter $k(\varphi)$ satisfies

$$\frac{1}{216}\, k(\varphi)^6 \left(216\varphi^2 + 36\varphi + 1\right) + \frac{1}{36}\, k(\varphi)^4 \left(72\varphi + 3\right) - 4 \;=\; 0 \; .$$

Effecting this twist on the F-theory fibre gives the curve $\mathcal{E}_S$ with the Weierstrass equation

$$y^2 \;=\; x^3 - \frac{24\varphi + 1}{48}\, x + \frac{216\varphi^2 + 36\varphi + 1}{864} \; .$$

The modularity conjectures can now be tested by comparing this curve with $\mathcal{E}_R$. Doing this, we find that in all cases we have tested, the curve $\mathcal{E}_S$ is isomorphic to $\mathcal{E}_R$ over $\mathbb{Q}$.

There is also an isogeny over $\mathbb{Q}$, corresponding to the rescaling of the lattice parameter

$$\mathrm{Im}\,\tau \mapsto \mathrm{Im}\,\widetilde{\tau} = \mathrm{Im}\,\tau/3 \; .$$

The $j$-invariant associated to the family of $F$-theory fibres $\widetilde{\mathcal{E}}_F$ so obtained is given by

$$j(\widetilde{\tau}(\varphi)) \;=\; \frac{(216\varphi - 1)^3}{\varphi(27\varphi + 1)^3} \; . \tag{74}$$

The three expressions for $C(\varphi)$ were obtained by equating (74) with $j_C$ in (15), but these equations and their solutions are again too complicated to display here. The twist parameter $k(\varphi)$ relating the F-theory fibre to the twisted Sen curve $\mathcal{E}_S$ satisfies now the polynomial equation

$$\frac{1}{216}\, k(\varphi)^6 \left(-5832\varphi^2 + 540\varphi + 1\right) + \frac{1}{36}\, k(\varphi)^4 \left(-648\varphi + 3\right) - 4 \;=\; 0 \; ,$$

and the family of elliptic curves $\mathcal{E}_S$ is given by the equation

$$y^2 \;=\; x^3 + \frac{216\varphi - 1}{48}\, x + \frac{5832\varphi^2 - 540\varphi - 1}{864} \; .$$

## 7.2. The mirror split quintic

This manifold is the mirror of the split of the quintic hypersurface in $\mathbb{P}^5$, given by the configuration described by the CICY matrix

$$\begin{array}{c} \mathbb{P}^4 \\ \mathbb{P}^4 \end{array} \left[ \begin{array}{ccccc} 1 & 1 & 1 & 1 & 1 \\ 1 & 1 & 1 & 1 & 1 \end{array} \right]_{\chi=-100} .$$

This has two Kähler parameters, so the mirror manifolds form a two-parameter family. This is birational to [39] the redundantly parametrised family of varieties in $(\mathbb{C}^*)^4 \times (\mathbb{C}^*)^4$ given by

$$a_i + b_i U_i + c_i V_i = 0 , \qquad i = 1, \dots 4 ,$$

$$a_5 + \frac{b_5}{U_1 U_2 U_3 U_4} + \frac{c_5}{V_1 V_2 V_3 V_4} = 0 .$$

We opt to work in the patch where $b_1 = -\varphi^1$, $c_1 = -\varphi^2$, and all other $a_i, b_i, c_i$ equal 1. The quantities $Y_{abc}$ of the split quintic are

$$Y_{ijk} = \begin{cases} 5 & i = j = k \\ 10 & \text{otherwise} \end{cases} , \quad Y_{0ij} = \begin{cases} \frac{1}{2} & i = j \\ 0 & i \neq j \end{cases} , \quad Y_{00i} = -\frac{25}{6} , \quad Y_{000} = 300 \frac{\zeta(3)}{(2\pi i)^3} .$$

Denoting the relevant Meijer G-functions as

$$G_{0,d}^{a,0}(0,0,0,0,0|x) \overset{\text{def}}{=} G_{0,d}^{a,0}(x) \qquad \text{and} \qquad G_{0,d}^{a,0}(1,1,1,1,1|x) \overset{\text{def}}{=} G_{0,d}^{a,0}(1|x) ,$$

and the hypergeometric functions as

$$_0F_b(1,1,1,1;x) \overset{\text{def}}{=} {}_0F_b(x) \qquad \text{and} \qquad {}_0F_b(2,2,2,2;x) \overset{\text{def}}{=} {}_0F_b(2;x) ,$$

the periods relevant to the analysis in this paper can be written as

$$\varpi^0 = \int_0^\infty \mathrm{d}u \, G_{0,5}^{5,0}(u) \, {}_0F_4(u\varphi^1) \, {}_0F_4(u\varphi^2) ,$$

$$\varpi^i = -\int_0^\infty \mathrm{d}u \, G_{0,5}^{5,0}(u) \, G_{0,5}^{2,0}(u\varphi^i) \, {}_0F_4(u\varphi^j) , \qquad\qquad\qquad i \neq j ,$$

$$\varpi_i = 5 \int_0^\infty \mathrm{d}u \, G_{0,5}^{5,0}(u) \left[ 2 G_{0,5}^{2,0}(u\varphi^i) \, G_{0,5}^{2,0}(u\varphi^j) + \left( G_{0,5}^{3,0}(-u\varphi^i) + i\pi G_{0,5}^{2,0}(u\varphi^i) \right) {}_0F_4(u\varphi^j) \right.$$

$$\left. + 2 \left( G_{0,5}^{3,0}(-u\varphi^j) + i\pi G_{0,5}^{2,0}(u\varphi^j) \right) {}_0F_4(u\varphi^i) - \frac{5\pi^2}{3} \, {}_0F_4(u\varphi^i) \, {}_0F_4(u\varphi^j) \right] , \qquad i \neq j .$$

The axiodilaton is given on the $\mathbb{Z}_2$ symmetric locus $\varphi^1 = \varphi^2 = \varphi$ by

$$\tau(\varphi) = \frac{5i}{2\pi} \frac{\int_0^\infty \mathrm{d}u \, G_{0,5}^{5,0}(u) \left[ G_{0,5}^{3,0}(0,0,1,0,0|-u\varphi) \, {}_0F_4(u\varphi) - G_{0,5}^{3,0}(1|-u\varphi) \, {}_0F_4(2;u\varphi) \right]}{\int_0^\infty \mathrm{d}u \, G_{0,5}^{5,0}(u) \left[ G_{0,5}^{2,0}(0,1,0,0,0|u\varphi) \, {}_0F_4(u\varphi) + G_{0,5}^{2,0}(1|u\varphi) \, {}_0F_4(2;u\varphi) \right]} + 2 .$$

For real $\varphi$, this expression has real part equal to $-\frac{1}{2}$. Attempts to numerically integrate the above combinations of Meijer G functions are met with problems in Mathematica, so instead it is best to expand each of the hypergeometric functions $_0F_4$ as a power series in $u$ up to some large order (we used 360). After interchanging the order of summation and integration, each integral in the

resulting sum can be evaluated in Mathematica exactly as a single Meijer G function, yielding a series expression amenable to fast evaluation.

The numerical evidence strongly suggests that the $j$-invariant $j(\tau)$ is again a rational function of the complex structure parameter $\varphi$,

$$j(\tau(\varphi)) \;=\; \frac{\left(\varphi^4 - 12\varphi^3 + 14\varphi^2 + 12\varphi + 1\right)^3}{\varphi^5\left(\varphi^2 - 11\varphi - 1\right)} \;.$$

The fibre $\mathcal{E}_F$ with this $j$-invariant is again relatively complicated, so we omit its display. However, we can again find a twist (37) which gives the twisted Sen curve $\mathcal{E}_S$ which is isomorphic to $\mathcal{E}_R$ over $\mathbb{Q}$. The twist parameter $k(\varphi)$ satisfies the equation

$$\frac{1}{216}\,k(\varphi)^6\left(\varphi^6 - 18\varphi^5 + 75\varphi^4 + 75\varphi^2 + 18\varphi + 1\right) + \frac{1}{36}\,k(\varphi)^4\left(3\varphi^4 - 36\varphi^3 + 42\varphi^2 + 36\varphi + 3\right) - 4 = 0 \;,$$

and the resulting curve $\mathcal{E}_S$ is given by the Weierstrass equation

$$\begin{aligned}
y^2 \;=\; x^3 &- \frac{1}{48}\left(\varphi^4 - 12\varphi^3 + 14\varphi^2 + 12\varphi + 1\right)x \\
&+ \frac{1}{864}\left(\varphi^2 + 1\right)\left(\varphi^4 - 18\varphi^3 + 74\varphi^2 + 18\varphi + 1\right) \;.
\end{aligned}$$

There is an isogeny over $\mathbb{Q}$ corresponding to a rescaling of the lattice parameter $\operatorname{Im}\tau \mapsto \operatorname{Im}\tau/5 = \operatorname{Im}\widetilde{\tau}$. The associated $j$-function is given by

$$j(\widetilde{\tau}) \;=\; \frac{\left(\varphi^4 + 228\varphi^3 + 494\varphi^2 - 228\varphi + 1\right)^3}{\varphi\left(\varphi^2 - 11\varphi - 1\right)^5} \;.$$

For the related F-theory fibre, the twist parameter satisfies the equation

$$\begin{aligned}
\frac{1}{216}\,k(\varphi)^6&\left(\varphi^6 - 522\varphi^5 - 10005\varphi^4 - 10005\varphi^2 + 522\varphi + 1\right) \\
&+ \frac{1}{36}\,k(\varphi)^4\left(3\varphi^4 + 684\varphi^3 + 1482\varphi^2 - 684\varphi + 3\right) - 4 \;=\; 0 \;,
\end{aligned}$$

and the twisted Sen curve $\widetilde{\mathcal{E}}_S$ has the Weierstrass equation

$$\begin{aligned}
y^2 \;=\; x^3 &- \frac{1}{48}\left(\varphi^4 + 228\varphi^3 + 494\varphi^2 - 228\varphi + 1\right)x \\
&+ \frac{1}{864}\left(\varphi^2 + 1\right)\left(\varphi^4 - 522\varphi^3 - 10006\varphi^2 + 522\varphi + 1\right) \;.
\end{aligned}$$

## Acknowledgements

It is a pleasure to acknowledge fruitful conversations with Fernando Alday, Christopher Beem, Mohamed Elmi, Hans Jockers, Albrecht Klemm, Alan Lauder, Horia Magureanu, and Erik Panzer. JM is supported by EPSRC studentship #2272658, and PK thanks the Osk. Huttusen Säätiö and the Jenny and Antti Wihuri Foundation for support.

## A. Modular forms

We give a brief overview of the types of modular forms that are encountered in this paper, together with some of their useful properties.

### A.1. Elliptic modular forms

To begin, we recall the definition and some important properties of elliptic or 'ordinary' modular forms, mainly following [94].

Modular forms are holomorphic functions on the upper half plane

$$\mathbb{H} \; = \; \{z \in \mathbb{C} \,|\, \mathrm{Im}\, z > 0\}$$

which have prescribed transformation properties under the action of the Möbius transformations

$$z \mapsto \gamma z \;\stackrel{\text{def}}{=}\; \frac{az+b}{cz+d}\,, \qquad \gamma \; = \; \begin{pmatrix} a & b \\ c & d \end{pmatrix} \in \mathrm{SL}(2,\mathbb{Z})\,.$$

More generally, we can take $\gamma \in \Gamma \subset \mathrm{SL}(2,\mathbb{R})$ for some discrete subgroup $\Gamma$. Some useful subgroups we encounter are *principal congruence subgroups* $\Gamma(N)$ and *Hecke congruence subgroups* $\Gamma_0(N)$, which are defined as

$$\Gamma(N) \; = \; \left\{ \begin{pmatrix} a & b \\ c & d \end{pmatrix} \in \mathrm{SL}(2,\mathbb{Z}) \;\middle|\; b, c \equiv 0 \bmod N \right\},$$

$$\Gamma_0(N) \; = \; \left\{ \begin{pmatrix} a & b \\ c & d \end{pmatrix} \in \mathrm{SL}(2,\mathbb{Z}) \;\middle|\; c \equiv 0 \bmod N \right\}.$$

The integer $N$ is called the *level* of the subgroup, and is identified with the *conductor* of the corresponding elliptic curve.

Under a transformation $\gamma \in \Gamma$, a (*holomorphic*) *modular form* $f$ (for the group $\Gamma$) of *weight* $k$ transforms as

$$f(\gamma z) \; = \; (cz+d)^k f(z)\,, \qquad \gamma = \begin{pmatrix} a & b \\ c & d \end{pmatrix} \in \Gamma. \tag{75}$$

In addition, it is required that $f(x+iy)$ is bounded as $y \to \infty$.

For all subgroups $\Gamma \subset \mathrm{SL}(2,\mathbb{R})$, such as $\Gamma_0(N)$, that contain the element

$$T \; = \; \begin{pmatrix} 1 & 1 \\ 0 & 1 \end{pmatrix}$$

corresponding to the transformation $\tau \mapsto \tau + 1$, the transformation law (75) implies that $f$ is periodic in $\tau$, and thus has a Fourier expansion

$$f(\tau) \; = \; \sum_{n=0}^{\infty} \alpha_n \, q^n\,, \qquad q \; = \; e^{2\pi i \tau}\,.$$

We call $f$ a *cusp form* if the Fourier coefficient $\alpha_0$ vanishes. The terminology is related to the fact that such modular forms vanish at the cusps of the fundamental domain of $\Gamma$.

We denote the space of modular forms for group $\Gamma$ of weight $k$ by $M_k(\Gamma)$, and the weight-$k$ cusp forms for the same group by $S_k(\Gamma)$. Due to their transformation properties, modular forms are highly constrained, and importantly the spaces $M_k\big(\mathrm{SL}(2,\mathbb{Z})\big)$ and $M_k\big(\Gamma_0(N)\big)$, and thus also $S_k\big(\mathrm{SL}(2,\mathbb{Z})\big)$ and $S_k\big(\Gamma_0(N)\big)$, are finite-dimensional.

*Hecke operators*

If $n$ and $N$ are integers such that $\gcd(n, N) = 1$, it is possible to define a set of *Hecke operators* $T_n$ acting on the spaces $M_k(\Gamma_0(N))$ as

$$T_n f(\tau) \;=\; n^{k-1} \sum_{\gamma \in \Gamma_0(N) \backslash \mathcal{M}_n(N)} (c\tau + d)^{-k} f(\gamma\tau) \;, \qquad \text{where } \gamma \;=\; \begin{pmatrix} a & b \\ c & d \end{pmatrix}, \tag{76}$$

and $\mathcal{M}_n(N)$ is the subset of $\Gamma_0(N)$ of matrices with determinant $n$. By expanding $f$ as a Fourier series, and using a suitable set of representatives for $\mathcal{M}_n(N)$, it can be shown that these act on the Fourier coefficients as

$$\alpha_i \;\mapsto\; \sum_{r | \gcd(n,i)} r^{k-1} \alpha_{in/r^2} \;.$$

From this it also follows that any two Hecke operators commute with each other, which allows us to find a basis for $M_k(\Gamma_0(N))$ that is a simultaneous eigenbasis of all Hecke operators. A simultaneous eigenvector of all Hecke operators is a *Hecke eigenform*. We say such a form is *normalised* if $\alpha_1 = 1$.

Note that for $N = 1$, $\gcd(n, N) = 1$ for all $n$, and thus a normalised Hecke eigenform is completely determined by its Hecke eigenvalues. If $N \neq 1$ this is no longer true, but it is possible to define a space of modular forms, called *newforms*, for which this property holds.

Let $M$ be an integer such that $M \mid N$. Then for every $d \mid N/M$ and $g \in M_k(M)$, the function $f$ defined by $f(\tau) = g(d\tau)$ is a cusp form of weight $k$ and level $N$, $f \in S_k(\Gamma_0(N))$. These modular forms are said to be *oldforms*. The orthogonal complement of the space of oldforms in $S_k(\Gamma_0(N))$ under the Petersson inner product (see for example [95] for definition) is the space of *newforms*, so we have a decomposition

$$S_k(\Gamma_0(N)) \;=\; S_k^{\mathrm{old}}(\Gamma_0(N)) \oplus S_k^{\mathrm{new}}(\Gamma_0(N)) \;.$$

A modular form in $S_k^{\mathrm{new}}(\Gamma_0(N))$ is called a *newform* if it is also a normalised Hecke eigenform. Newforms form a basis of $S_k^{\mathrm{new}}(\Gamma_0(N))$.

## A.2. Number fields and field extensions

Before discussing Hilbert and Bianchi modular forms, we briefly review some aspects of field extensions and in particular number fields that will be useful in discussing these modular forms. A more complete account of this topic can be found for example in [96]. For an exposition aimed at physicists, see for example [97].

A *number field* is a field extension $\mathbb{E}$ of $\mathbb{Q}$ with finite degree. For our purposes in this paper, it is enough to concentrate on the field extensions of the form $\mathbb{Q}(\alpha)$ generated by a single element.

One of the central results in the theory of primes numbers in $\mathbb{Z}$ is unique factorisation of any integer into primes. This does not continue to hold in rings of integers $\mathcal{O}_{\mathbb{E}}$, which is defined as the subset of $\mathbb{E}$ whose elements are roots of monic integral polynomials. For example, consider the ring of integers $\mathbb{Z}[\sqrt{10}]$ in which we have

$$9 \;=\; 3 \cdot 3 \;=\; (\sqrt{10} + 1)(\sqrt{10} - 1) \;,$$

and 3 and $\sqrt{10} \pm 1$ do not divide further, nor are any of these *associates*, that is, they are not related by multiplication by a unit. To remedy this, we need to consider ideals and prime ideals of the ring of integers $\mathcal{O}_{\mathbb{E}}$ of the field $\mathbb{E}$ as 'proper' generalisations of primes in $\mathbb{Z}$. Recall that an ideal $\mathfrak{p}$ of a ring $R$ is a *prime ideal* if from $ab \in \mathfrak{p}$ it follows that $a \in \mathfrak{p}$ or $b \in \mathfrak{p}$.

Thinking of the set of ideals of the ring of integers $\mathcal{O}_{\mathbb{E}}$ as a generalisation of the set of integers of $\mathbb{Q}$, and the set of prime ideals as a generalisation of the set of primes in $\mathbb{Q}$, we can recover an analogue of the *unique factorisation theorem*.

To see this, note first that the ring of integers $\mathcal{O}_{\mathbb{E}}$ of a number field $\mathbb{E}$ is a *Dedekind domain*. That is, it is an integral domain that is integrally closed, Noetherian and whose every nonzero prime ideal is maximal.

It can be shown that in any Dedekind domain any nonzero proper ideal $\mathfrak{a}$ factorises uniquely as

$$\mathfrak{a} \;=\; \mathfrak{p}_1^{r_1}\ldots\mathfrak{p}_n^{r_n}\;,$$

where $\mathfrak{p}_i$ are distinct prime ideals of $\mathcal{O}_{\mathbb{E}}$, $r_i$ are positive integers, and both $\mathfrak{p}_i$ and $r_i$ are uniquely determined.

For instance, consider the ideal $(9) \subset \mathbb{Z}\big[\sqrt{10}\,\big]$. While $(9) = (3)(3)$, this is not a factorisation to prime ideals, as $(3)$ is not a prime ideal, which can be seen by the fact that $\mathbb{Z}\big[\sqrt{10}\,\big]/(3)$ is not an integral domain. However, $(3, \sqrt{10}+1)$ is a prime ideal as $\mathbb{Z}\big[\sqrt{10}\,\big]/(3,\sqrt{10}+1) \cong \mathbb{Z}_3$, and a similar argument shows that $(3,\sqrt{10}-1)$ is also prime. It is often useful to think of the ideal $(a,b)$ as a generalisation of the greatest common divisor of $a$ and $b$.

We have that

$$\big(3,\sqrt{10}+1\big)\big(3,\sqrt{10}-1\big) \;=\; \big(9,\,3\sqrt{10}+3,\,3\sqrt{10}-3,\,9\big) \;=\; (3)\;,$$

where the last equality follows by noting that

$$9 + \big(3\sqrt{10}-3\big) - \big(3\sqrt{10}+3\big) \;=\; 3\;.$$

Then it is clear, as multiplication of ideals is associative, that

$$\big(3,\sqrt{10}+1\big)^2\big(3,\sqrt{10}-1\big)^2 \;=\; (9)$$

is the unique decomposition of $(9)$ into prime ideals.

This example also involves another interesting property of field extensions. The ideal $(3)$ is clearly prime as an ideal of $\mathbb{Z}$. However, in the ring of integers $\mathbb{Z}\big[\sqrt{10}\,\big]$ of the field extension $\mathbb{E} = \mathbb{Q}\big(\sqrt{10}\,\big)$, the ideal $(3)$ factorises, as we have seen, into $\big(3,\sqrt{10}+1\big)\big(3,\sqrt{10}-1\big)$, and thus is no longer prime. We say that a prime $\mathfrak{p} \subset \mathcal{O}_{\mathbb{K}}$ *splits* in the field extension $\mathbb{E} \supset \mathbb{K}$ if $\mathfrak{p}\mathcal{O}_{\mathbb{E}}$ factorises in $\mathcal{O}_{\mathbb{E}}$ as

$$\mathfrak{p}\mathcal{O}_{\mathbb{E}} \;=\; \mathfrak{p}_1\ldots\mathfrak{p}_n\;,$$

where all $\mathfrak{p}_i$ are distinct, in addition to which we require that the *residue class degree*, that is, the degree of the field extension $[\mathbb{E}/\mathfrak{p}_i : \mathbb{K}/\mathfrak{p}]$, is unity for every prime ideal $\mathfrak{p}_i$ appearing in the factorisation. On the other hand, if $\mathfrak{p}$ factorises as

$$\mathfrak{p}\mathcal{O}_{\mathbb{E}} \;=\; \mathfrak{p}_1^{r_1}\ldots\mathfrak{p}_n^{r_n}\;,$$

with at least one $r_i > 1$, then $\mathfrak{p}$ is said to *ramify*. Finally, if $\mathfrak{p}\mathcal{O}_{\mathbb{E}}$ is a prime ideal in $\mathcal{O}_{\mathbb{E}}$, then $\mathfrak{p}$ is said to be *inert*.

As a simple example, consider again the number field $\mathbb{Q}\big(\sqrt{10}\,\big)$ and its ring of integers $\mathbb{Z}\big[\sqrt{10}\,\big]$. We have already seen that $(3)$ splits. One the other hand, there are two rational primes, 2 and 5, that ramify. Namely,

$$\big(5,\sqrt{10}\,\big)^2 \;=\; \big(25,10\sqrt{10},10\big) \;=\; (5)\;,$$
$$\big(2,\sqrt{10}\,\big)^2 \;=\; \big(\,4,\,4\sqrt{10},10\big) \;=\; (2)\;.$$

It is easy to see that there are no inert primes in the extension $\mathbb{Q}\big(\sqrt{10}\,\big)/\mathbb{Q}$.

Finally, a concept that is important when comparing Hecke eigenvalues of Hilbert and Bianchi modular forms to zeta function coefficients is that of the *norm of an ideal*. The norm of an ideal $\mathfrak{p}$ in a ring of integers $\mathcal{O}_{\mathbb{E}}$ is defined simply as the index of $\mathfrak{p}$ in $\mathcal{O}_{\mathbb{E}}$

$$N(\mathfrak{p}) \;=\; [\mathcal{O}_{\mathbb{E}} : \mathfrak{p}] = |\mathcal{O}_{\mathbb{E}}/\mathfrak{p}|.$$

For example, as noted, $\mathbb{Z}\big[\sqrt{10}\,\big]/\big(3, \sqrt{10}+1\big) \cong \mathbb{Z}_3$, which shows that $N\big((3, \sqrt{10}+1)\big) = 3$.

### A.3.  Hilbert modular forms

Another important class of modular forms consists of Hilbert modular forms. These are modular forms associated to totally real number fields, such as real quadratic fields, which we consider here for simplicity, although generalisation to other totally real number fields is obvious. We review some of the properties of real quadratic fields and Hilbert modular forms, following [95].

This section is mainly intended to illustrate the ideas behind generalising the notion of modular forms to more general number fields. This is, however, an extensive topic and as such, we will not discuss the Hecke theory of Hilbert or Bianchi modular forms. Instead, we refer the interested reader to [78, 98] for compehensive discussion on these topics. For the purposes of this paper, it is enough to note that in both cases the Hecke eigenvalues can be associated to prime ideals of the ring of integers, which allows us to compare the Hecke eigenvalues to the zeta function coefficients in the manner described in §4.6.

*Real quadratic fields and embeddings*

Real quadratic fields $\mathbb{F}$ are field extensions of $\mathbb{Q}$ of the form $\mathbb{Q}\big(\sqrt{n}\,\big)$, where $n$ is a square-free integer. The ring of integers of $\mathbb{F} = \mathbb{Q}\big(\sqrt{n}\,\big)$ is given by

$$\mathcal{O}_{\mathbb{F}} \;=\; \begin{cases} \mathbb{Z}\left[\frac{1+\sqrt{n}}{2}\right] \;, & \text{if } n \equiv 1 \quad \mathrm{mod}\ 4\ , \\[2mm] \mathbb{Z}\left[\sqrt{n}\,\right] \;\;, & \text{if } n \equiv 2,3\ \mathrm{mod}\ 4\ . \end{cases}$$

The analogues of the groups $\mathrm{GL}(2,\mathbb{Q})$ and $\mathrm{GL}(2,\mathbb{Z})$ are $\mathrm{GL}(2,\mathbb{F})$ and $\mathrm{GL}(2,\mathcal{O}_{\mathbb{F}})$. These can be embedded into $\mathrm{GL}(2,\mathbb{R}) \times \mathrm{GL}(2,\mathbb{R})$ using the two inequivalent embeddings of $\mathbb{F}$ into $\mathbb{R}$.[14] To be explicit, choose an embedding of $\mathbb{F}$ into $\mathbb{R}$ so that we can write every $a \in \mathbb{F}$ as $a_1 + \sqrt{n}\,a_2$, with $a_1, a_2 \in \mathbb{Q}$. This has a conjugate which is precisely the image of $a$ in the second embedding. Let us denote this by $\bar{a} = a_1 - \sqrt{n}\,a_2$. Then an element of $\gamma$ of $\mathrm{GL}(2,\mathbb{F})$ is embedded into $\mathrm{GL}(2,\mathbb{R}) \times \mathrm{GL}(2,\mathbb{R})$ by

$$\gamma \;=\; \begin{pmatrix} a & b \\ c & d \end{pmatrix} \mapsto \left[ \begin{pmatrix} a_1+\sqrt{n}\,a_2 & b_1+\sqrt{n}\,b_2 \\ c_1+\sqrt{n}\,c_2 & d_1+\sqrt{n}\,d_2 \end{pmatrix}, \begin{pmatrix} a_1-\sqrt{n}\,a_2 & b_1-\sqrt{n}\,b_2 \\ c_1-\sqrt{n}\,c_2 & d_1-\sqrt{n}\,d_2 \end{pmatrix} \right] \in \mathrm{GL}(2,\mathbb{R}) \times \mathrm{GL}(2,\mathbb{R})\ .$$

Thus $\mathrm{SL}(2,\mathbb{F})$ has a natural action on the upper half space $\mathbb{H}^2 = \mathbb{H} \times \mathbb{H}$, on which it acts by

$$(z_1, z_2) \mapsto \left( \frac{az_1 + b}{cz_1 + d}, \frac{\bar{a}z_1 + \bar{b}}{\bar{c}z_1 + \bar{d}} \right)\ .$$

---

[14] When we have to make a particular choice of the embedding, we use the conventions laid out in §1.2.

*Hilbert modular forms*

Let $\mathfrak{n}$ be a nonzero ideal of $\mathcal{O}_{\mathbb{F}}$, and let $\Gamma_0(\mathfrak{n}) \subset \mathrm{GL}(2, \mathbb{F})$ be a subgroup defined by

$$\Gamma_0(\mathfrak{n}) \overset{\mathrm{def}}{=} \left\{ \gamma = \begin{pmatrix} a & b \\ c & d \end{pmatrix} \in \mathrm{GL}(2, \mathcal{O}_F) \,\middle|\, c \in \mathfrak{n} \,, \text{ and } \det \gamma \text{ totally positive} \right\}.$$

This subgroup is *commensurable* with $\mathrm{GL}(2, \mathcal{O}_{\mathbb{F}})$, that is, $\Gamma_0(\mathfrak{n}) \cap \mathrm{SL}(2, \mathcal{O}_{\mathbb{F}})$ has a finite index in both $\Gamma_0(\mathfrak{n})$ and $\mathrm{SL}(2, \mathcal{O}_{\mathbb{F}})$.

A meromorphic function $f : \mathbb{H}^2 \to \mathbb{C}$ is a *Hilbert modular form* of weight $k = (k_1, k_2) \in \mathbb{Z}^2$ for $\Gamma$ if it satisfies the transformation law

$$f(\gamma z) \;=\; (c z_1 + d)^{k_1} (\bar{c} z_2 + \bar{d})^{k_2} f(z) \tag{77}$$

for all $\gamma \in \Gamma$. In particular

$$\begin{pmatrix} 1 & \mu \\ 0 & 1 \end{pmatrix} \in \Gamma_0(\mathfrak{n})$$

for all $\mu \in \mathcal{O}_{\mathbb{F}}$ and for all integral ideals $\mathfrak{n}$. Applying the transformation law (77), it follows that

$$f(z + \mu) \;=\; f(z)$$

for all $\mu \in \mathcal{O}_{\mathbb{F}}$. Hence, $f$ has a Fourier expansion of the form

$$f \;=\; \sum_{\nu \in \mathfrak{d}^{-1}} \alpha_\nu \exp\left( 2\pi \mathrm{i} \, \mathrm{Tr}\left( \nu z \right) \right) ,$$

where $\mathrm{Tr}$ is the field trace and $\mathfrak{d}^{-1}$ denotes the *inverse different* defined by

$$\mathfrak{d}^{-1} \overset{\mathrm{def}}{=} \left\{ x \in \mathbb{F} \,\middle|\, \mathrm{Tr}\left( xy \right) \in \mathbb{Z} \text{ for all } y \in \mathcal{O}_{\mathbb{F}} \right\}.$$

The coefficients in the Fourier expansion can be associated to integral ideals $\mathfrak{m}$ of $\mathbb{F}$:

$$\alpha_{\mathfrak{m}} \overset{\mathrm{def}}{=} \alpha_u \,,$$

where $u$ is a totally positive element satisfying $(u) = \mathfrak{m}\mathfrak{d}^{-1}$.

# B. Data for the $S_5$ Symmetric Hulek-Verrill Manifold

In this, and the following two appendices, we gather relevant arithmetic data of the three manifolds we have used as examples in the paper: the Hulek-Verrill manifold, the mirror bicubic and the mirror split quintic. Specifically, we list the numerators $R(X, T)$ of their local zeta functions outside (apparent) singularities for a few first primes, and tabulate some modular forms associated to the quadratic factor of the zeta function together with the corresponding twisted Sen curves $\mathcal{E}_S$.

## B.1. Zeta function numerators

Table 10: *The numerators of the local zeta functions of non-singular $S_5$ symmetric Hulek-Verrill manifolds without apparent singularities for primes $7 \leqslant p \leqslant 47$.*

| $\varphi$ | smooth/sing. | singularity | $R(T)$ |
|---|---|---|---|
| | | $p = 7$ | |
| 1 | singular | 1 | |
| 2 | singular | $\frac{1}{25}$ | |
| 3 | smooth | | $\left(p^3 T^2 - 2pT + 1\right)^4 \left(p^6 T^4 + 2p^3 T^3 - 54pT^2 + 2T + 1\right)$ |
| 4 | singular | $\frac{1}{9}$ | |
| 5 | smooth | | $\left(p^3 T^2 + 4pT + 1\right)^4 \left(p^3 T^2 - 34T + 1\right) \left(p^3 T^2 + 4pT + 1\right)$ |
| 6 | smooth | | $\left(p^3 T^2 - 2pT + 1\right)^4 \left(p^6 T^4 + 12p^3 T^3 - 14pT^2 + 12T + 1\right)$ |

| $\varphi$ | smooth/sing. | singularity | $R(T)$ |
|---|---|---|---|
| | | $p = 11$ | |
| 1 | singular | 1 | |
| 2 | smooth | | $\left(p^3 T^2 - 6pT + 1\right)^4 \left(p^6 T^4 + 42p^3 T^3 + 194pT^2 + 42T + 1\right)$ |
| 3 | smooth | | $\left(p^3 T^2 + 1\right)^4 \left(p^3 T^2 + 1\right) \left(p^3 T^2 + 28T + 1\right)$ |
| 4 | singular | $\frac{1}{25}$ | |
| 5 | singular | $\frac{1}{9}$ | |
| 6 | smooth | | $\left(p^3 T^2 + 1\right)^4 \left(p^6 T^4 - 2p^3 T^3 + 2pT^2 - 2T + 1\right)$ |
| 7 | smooth | | $\left(p^3 T^2 + 1\right)^4 \left(p^6 T^4 - 22p^3 T^3 + 122pT^2 - 22T + 1\right)$ |
| 8 | smooth* | | |
| 9 | smooth | | $\left(p^3 T^2 + 1\right)^4 \left(p^6 T^4 - 32p^3 T^3 + 2pT^2 - 32T + 1\right)$ |
| 10 | smooth | | $\left(p^3 T^2 + 1\right)^4 \left(p^6 T^4 - 22p^3 T^3 + 2pT^2 - 22T + 1\right)$ |

| $\varphi$ | smooth/sing. | singularity | $R(T)$ |
|---|---|---|---|
| | | $p = 13$ | |
| 1 | singular | 1 | |
| 2 | smooth | | $\left(p^3 T^2 - 2pT + 1\right)^4 \left(p^3 T^2 + 34T + 1\right) \left(p^3 T^2 - 6pT + 1\right)$ |

| $p = 13$, continued | | | |
|---|---|---|---|
| $\varphi$ | smooth/sing. | singularity | $R(T)$ |
| 3 | singular | $\frac{1}{9}$ | |
| 4 | smooth | | $\left(p^3T^2 - 2pT + 1\right)^4 \left(p^3T^2 + 42T + 1\right)\left(p^3T^2 - 2pT + 1\right)$ |
| 5 | smooth | | $\left(p^3T^2 + 4pT + 1\right)^4 \left(p^6T^4 - 36p^3T^3 + 166pT^2 - 36T + 1\right)$ |
| 6 | smooth | | $\left(p^3T^2 + 4pT + 1\right)^4 \left(p^6T^4 - 26p^3T^3 + 266pT^2 - 26T + 1\right)$ |
| 7 | smooth | | $\left(p^3T^2 + 4pT + 1\right)^4 \left(p^6T^4 + 14p^3T^3 - 54pT^2 + 14T + 1\right)$ |
| 8 | smooth* | | |
| 9 | smooth* | | |
| 10 | smooth | | $\left(p^3T^2 - 2pT + 1\right)^4 \left(p^3T^2 + 42T + 1\right)\left(p^3T^2 - 2pT + 1\right)$ |
| 11 | smooth | | $\left(p^3T^2 + 4pT + 1\right)^4 \left(p^3T^2 - 18T + 1\right)\left(p^3T^2 + 4pT + 1\right)$ |
| 12 | singular | $\frac{1}{25}$ | |

| $p = 17$ | | | |
|---|---|---|---|
| $\varphi$ | smooth/sing. | singularity | $R(T)$ |
| 1 | singular | 1 | |
| 2 | singular | $\frac{1}{9}$ | |
| 3 | smooth | | $\left(p^3T^2 + 6pT + 1\right)^4 \left(p^6T^4 + 98p^3T^3 + 674pT^2 + 98T + 1\right)$ |
| 4 | smooth | | $\left(p^3T^2 + 6pT + 1\right)^4 \left(p^3T^2 - 114T + 1\right)\left(p^3T^2 + 6pT + 1\right)$ |
| 5 | smooth* | | |
| 6 | smooth | | $\left(p^3T^2 + 1\right)^4 \left(p^6T^4 - 44p^3T^3 + 38pT^2 - 44T + 1\right)$ |
| 7 | smooth | | $\left(p^3T^2 + 6pT + 1\right)^4 \left(p^6T^4 - 2p^3T^3 + 194pT^2 - 2T + 1\right)$ |
| 8 | smooth | | $\left(p^3T^2 + 6pT + 1\right)^4 \left(p^3T^2 - 94T + 1\right)\left(p^3T^2 + 6pT + 1\right)$ |
| 9 | smooth | | $\left(p^3T^2 - 6pT + 1\right)^4 \left(p^6T^4 + 4p^3T^3 + 182pT^2 + 4T + 1\right)$ |
| 10 | smooth | | $\left(p^3T^2 - 6pT + 1\right)^4 \left(p^6T^4 + 34p^3T^3 + 2pT^2 + 34T + 1\right)$ |
| 11 | smooth | | $\left(p^3T^2 + 1\right)^4 \left(p^6T^4 - 124p^3T^3 + 518pT^2 - 124T + 1\right)$ |
| 12 | smooth | | $\left(p^3T^2 - 6pT + 1\right)^4 \left(p^3T^2 - 74T + 1\right)\left(p^3T^2 - 6pT + 1\right)$ |
| 13 | smooth | | $\left(p^3T^2 - 6pT + 1\right)^4 \left(p^3T^2 + 78T + 1\right)\left(p^3T^2 - 2pT + 1\right)$ |
| 14 | smooth | | $\left(p^3T^2 + 1\right)^4 \left(p^3T^2 + 1\right)\left(p^3T^2 - 134T + 1\right)$ |
| 15 | singular | $\frac{1}{25}$ | |
| 16 | smooth | | $\left(p^3T^2 + 6pT + 1\right)^4 \left(p^3T^2 - 2pT + 1\right)\left(p^3T^2 + 6pT + 1\right)$ |

| $p = 19$ | | | |
|---|---|---|---|
| $\varphi$ | smooth/sing. | singularity | $R(T)$ |
| 1 | singular | 1 | |
| 2 | smooth | | $\left(p^3T^2 + 4pT + 1\right)^4 \left(p^6T^4 + 76p^3T^3 + 2pT^2 + 76T + 1\right)$ |

| $\varphi$ | smooth/sing. | singularity | $R(T)$ |
|---|---|---|---|
| | $p = 19$, *continued* | | |
| 3 | smooth | | $\left(p^3T^2 - 2pT + 1\right)^4 \left(p^6T^4 - 8p^3T^3 + 242pT^2 - 8T + 1\right)$ |
| 4 | smooth | | $\left(p^3T^2 + 4pT + 1\right)^4 \left(p^3T^2 - 60T + 1\right) \left(p^3T^2 + 4pT + 1\right)$ |
| 5 | smooth | | $\left(p^3T^2 + 4pT + 1\right)^4 \left(p^3T^2 - 60T + 1\right) \left(p^3T^2 + 4pT + 1\right)$ |
| 6 | smooth | | $\left(p^3T^2 - 8pT + 1\right)^4 \left(p^6T^4 + 8p^3T^3 - 318pT^2 + 8T + 1\right)$ |
| 7 | smooth | | $\left(p^3T^2 + 4pT + 1\right)^4 \left(p^6T^4 - 44p^3T^3 - 238pT^2 - 44T + 1\right)$ |
| 8 | smooth | | $\left(p^3T^2 - 2pT + 1\right)^4 \left(p^3T^2 - 80T + 1\right) \left(p^3T^2 - 2pT + 1\right)$ |
| 9 | smooth | | $\left(p^3T^2 + 4pT + 1\right)^4 \left(p^3T^2 - 160T + 1\right) \left(p^3T^2 + 4pT + 1\right)$ |
| 10 | smooth | | $\left(p^3T^2 - 2pT + 1\right)^4 \left(p^6T^4 + 12p^3T^3 + 562pT^2 + 12T + 1\right)$ |
| 11 | smooth | | $\left(p^3T^2 + 4pT + 1\right)^4 \left(p^3T^2 - 140T + 1\right) \left(p^3T^2 + 4pT + 1\right)$ |
| 12 | smooth | | $\left(p^3T^2 - 2pT + 1\right)^4 \left(p^6T^4 + 12p^3T^3 + 82pT^2 + 12T + 1\right)$ |
| 13 | smooth | | $\left(p^3T^2 - 8pT + 1\right)^4 \left(p^6T^4 + 178p^3T^3 + 1082pT^2 + 178T + 1\right)$ |
| 14 | smooth | | $\left(p^3T^2 - 2pT + 1\right)^4 \left(p^6T^4 + 12p^3T^3 - 158pT^2 + 12T + 1\right)$ |
| 15 | smooth | | $\left(p^3T^2 - 2pT + 1\right)^4 \left(p^6T^4 + 42p^3T^3 - 38pT^2 + 42T + 1\right)$ |
| 16 | singular | $\frac{1}{25}$ | |
| 17 | singular | $\frac{1}{9}$ | |
| 18 | smooth* | | |

| $\varphi$ | smooth/sing. | singularity | $R(T)$ |
|---|---|---|---|
| | $p = 23$ | | |
| 1 | singular | $1$ | |
| 2 | smooth | | $\left(p^3T^2 + 1\right)^4 \left(p^6T^4 - 28p^3T^3 + 98pT^2 - 28T + 1\right)$ |
| 3 | smooth | | $\left(p^3T^2 + 1\right)^4 \left(p^6T^4 - 68p^3T^3 + 98pT^2 - 68T + 1\right)$ |
| 4 | smooth | | $\left(p^3T^2 + 1\right)^4 \left(p^3T^2 - 120T + 1\right) \left(p^3T^2 + 4pT + 1\right)$ |
| 5 | smooth | | $\left(p^3T^2 + 6pT + 1\right)^4 \left(p^6T^4 - 90p^3T^3 + 890pT^2 - 90T + 1\right)$ |
| 6 | smooth | | $\left(p^3T^2 + 1\right)^4 \left(p^6T^4 + 72p^3T^3 + 98pT^2 + 72T + 1\right)$ |
| 7 | smooth | | $\left(p^3T^2 + 1\right)^4 \left(p^6T^4 + 52p^3T^3 + 338pT^2 + 52T + 1\right)$ |
| 8 | smooth | | $\left(p^3T^2 + 1\right)^4 \left(p^6T^4 + 12p^3T^3 + 98pT^2 + 12T + 1\right)$ |
| 9 | smooth* | | |
| 10 | smooth | | $\left(p^3T^2 + 6pT + 1\right)^4 \left(p^3T^2 - 132T + 1\right) \left(p^3T^2 + 4pT + 1\right)$ |
| 11 | smooth | | $\left(p^3T^2 - 6pT + 1\right)^4 \left(p^6T^4 + 94p^3T^3 - 94pT^2 + 94T + 1\right)$ |
| 12 | singular | $\frac{1}{25}$ | |
| 13 | smooth | | $\left(p^3T^2 + 1\right)^4 \left(p^3T^2 + 1\right) \left(p^3T^2 + 112T + 1\right)$ |
| 14 | smooth | | $\left(p^3T^2 - 6pT + 1\right)^4 \left(p^6T^4 + 94p^3T^3 + 266pT^2 + 94T + 1\right)$ |
| 15 | smooth | | $\left(p^3T^2 + 1\right)^4 \left(p^6T^4 + 2p^3T^3 - 142pT^2 + 2T + 1\right)$ |
| 16 | smooth* | | |

| | | | |
|---|---|---|---|
| $p = 23$, *continued* | | | |
| $\varphi$ | smooth/sing. | singularity | $R(T)$ |
| 17 | smooth | | $\left(p^3T^2 - 6pT + 1\right)^4 \left(p^6T^4 + 144p^3T^3 + 626pT^2 + 144T + 1\right)$ |
| 18 | singular | $\frac{1}{9}$ | |
| 19 | smooth | | $\left(p^3T^2 + 1\right)^4 \left(p^6T^4 + 162p^3T^3 + 1178pT^2 + 162T + 1\right)$ |
| 20 | smooth | | $\left(p^3T^2 + 6pT + 1\right)^4 \left(p^6T^4 - 120p^3T^3 + 530pT^2 - 120T + 1\right)$ |
| 21 | smooth | | $\left(p^3T^2 + 6pT + 1\right)^4 \left(p^6T^4 - 120p^3T^3 + 530pT^2 - 120T + 1\right)$ |
| 22 | smooth | | $\left(p^3T^2 - 6pT + 1\right)^4 \left(p^6T^4 + 94p^3T^3 + 866pT^2 + 94T + 1\right)$ |

| | | | |
|---|---|---|---|
| $p = 29$ | | | |
| $\varphi$ | smooth/sing. | singularity | $R(T)$ |
| 1 | singular | 1 | |
| 2 | smooth | | $\left(p^3T^2 + 1\right)^4 \left(p^6T^4 - 150p^3T^3 + 362pT^2 - 150T + 1\right)$ |
| 3 | smooth | | $\left(p^3T^2 + 6pT + 1\right)^4 \left(p^6T^4 - 6p^3T^3 + 722pT^2 - 6T + 1\right)$ |
| 4 | smooth | | $\left(p^3T^2 + 6pT + 1\right)^4 \left(p^3T^2 - 190T + 1\right) \left(p^3T^2 + 6pT + 1\right)$ |
| 5 | smooth | | $\left(p^3T^2 + 6pT + 1\right)^4 \left(p^6T^4 + 44p^3T^3 + 182pT^2 + 44T + 1\right)$ |
| 6 | smooth | | $\left(p^3T^2 - 6pT + 1\right)^4 \left(p^6T^4 + 96p^3T^3 - 418pT^2 + 96T + 1\right)$ |
| 7 | singular | $\frac{1}{25}$ | |
| 8 | smooth | | $\left(p^3T^2 + 6pT + 1\right)^4 \left(p^6T^4 + 24p^3T^3 + 542pT^2 + 24T + 1\right)$ |
| 9 | smooth | | $\left(p^3T^2 + 6pT + 1\right)^4 \left(p^6T^4 - 16p^3T^3 - 418pT^2 - 16T + 1\right)$ |
| 10 | smooth | | $\left(p^3T^2 + 1\right)^4 \left(p^6T^4 - 80p^3T^3 + 542pT^2 - 80T + 1\right)$ |
| 11 | smooth | | $\left(p^3T^2 + 6pT + 1\right)^4 \left(p^6T^4 - 126p^3T^3 + 242pT^2 - 126T + 1\right)$ |
| 12 | smooth | | $\left(p^3T^2 + 1\right)^4 \left(p^6T^4 + 100p^3T^3 + 182pT^2 + 100T + 1\right)$ |
| 13 | singular | $\frac{1}{9}$ | |
| 14 | smooth | | $\left(p^3T^2 - 6pT + 1\right)^4 \left(p^6T^4 - 24p^3T^3 + 542pT^2 - 24T + 1\right)$ |
| 15 | smooth | | $\left(p^3T^2 + 6pT + 1\right)^4 \left(p^6T^4 - 156p^3T^3 + 1382pT^2 - 156T + 1\right)$ |
| 16 | smooth | | $\left(p^3T^2 + 6pT + 1\right)^4 \left(p^6T^4 - 396p^3T^3 + 2822pT^2 - 396T + 1\right)$ |
| 17 | smooth | | $\left(p^3T^2 - 6pT + 1\right)^4 \left(p^6T^4 + 6p^3T^3 - 238pT^2 + 6T + 1\right)$ |
| 18 | smooth | | $\left(p^3T^2 + 1\right)^4 \left(p^6T^4 + 10p^3T^3 - 1438pT^2 + 10T + 1\right)$ |
| 19 | smooth | | $\left(p^3T^2 + 1\right)^4 \left(p^6T^4 + 130p^3T^3 + 722pT^2 + 130T + 1\right)$ |
| 20 | smooth | | $\left(p^3T^2 - 6pT + 1\right)^4 \left(p^6T^4 + 276p^3T^3 + 1862pT^2 + 276T + 1\right)$ |
| 21 | smooth* | | |
| 22 | smooth | | $\left(p^3T^2 + 6pT + 1\right)^4 \left(p^6T^4 + 24p^3T^3 + 1262pT^2 + 24T + 1\right)$ |
| 23 | smooth | | $\left(p^3T^2 - 6pT + 1\right)^4 \left(p^3T^2 - 294T + 1\right) \left(p^3T^2 + 10pT + 1\right)$ |
| 24 | smooth | | $\left(p^3T^2 - 6pT + 1\right)^4 \left(p^6T^4 + 56p^3T^3 + 782pT^2 + 56T + 1\right)$ |
| 25 | smooth | | $\left(p^3T^2 - 6pT + 1\right)^4 \left(p^6T^4 - 44p^3T^3 - 58pT^2 - 44T + 1\right)$ |
| 26 | smooth* | | |

| $p = 29$, continued | | | |
|---|---|---|---|
| $\varphi$ | smooth/sing. | singularity | $R(T)$ |
| 27 | smooth | | $\left(p^3T^2 - 6pT + 1\right)^4 \left(p^6T^4 + 6p^3T^3 - 118pT^2 + 6T + 1\right)$ |
| 28 | smooth | | $\left(p^3T^2 - 6pT + 1\right)^4 \left(p^6T^4 + 136p^3T^3 + 782pT^2 + 136T + 1\right)$ |

| $p = 31$ | | | |
|---|---|---|---|
| $\varphi$ | smooth/sing. | singularity | $R(T)$ |
| 1 | singular | $1$ | |
| 2 | smooth | | $\left(p^3T^2 + 4pT + 1\right)^4 \left(p^6T^4 - 288p^3T^3 + 2434pT^2 - 288T + 1\right)$ |
| 3 | smooth* | | |
| 4 | smooth | | $\left(p^3T^2 + 4pT + 1\right)^4 \left(p^6T^4 + 352p^3T^3 + 2114pT^2 + 352T + 1\right)$ |
| 5 | singular | $\frac{1}{25}$ | |
| 6 | smooth | | $\left(p^3T^2 + 10pT + 1\right)^4 \left(p^6T^4 + 258p^3T^3 + 2122pT^2 + 258T + 1\right)$ |
| 7 | singular | $\frac{1}{9}$ | |
| 8 | smooth | | $\left(p^3T^2 - 8pT + 1\right)^4 \left(p^3T^2 + 268T + 1\right) \left(p^3T^2 - 8pT + 1\right)$ |
| 9 | smooth | | $\left(p^3T^2 - 8pT + 1\right)^4 \left(p^3T^2 + 168T + 1\right) \left(p^3T^2 - 8pT + 1\right)$ |
| 10 | smooth | | $\left(p^3T^2 + 4pT + 1\right)^4 \left(p^6T^4 + 52p^3T^3 - 286pT^2 + 52T + 1\right)$ |
| 11 | smooth | | $\left(p^3T^2 + 4pT + 1\right)^4 \left(p^6T^4 - 98p^3T^3 + 1514pT^2 - 98T + 1\right)$ |
| 12 | smooth | | $\left(p^3T^2 - 2pT + 1\right)^4 \left(p^6T^4 - 264p^3T^3 + 1906pT^2 - 264T + 1\right)$ |
| 13 | smooth | | $\left(p^3T^2 + 4pT + 1\right)^4 \left(p^6T^4 - 98p^3T^3 + 674pT^2 - 98T + 1\right)$ |
| 14 | smooth | | $\left(p^3T^2 + 4pT + 1\right)^4 \left(p^6T^4 - 8p^3T^3 + 1154pT^2 - 8T + 1\right)$ |
| 15 | smooth | | $\left(p^3T^2 - 8pT + 1\right)^4 \left(p^6T^4 - 200p^3T^3 + 1058pT^2 - 200T + 1\right)$ |
| 16 | smooth | | $\left(p^3T^2 + 4pT + 1\right)^4 \left(p^3T^2 - 192T + 1\right) \left(p^3T^2 + 4pT + 1\right)$ |
| 17 | smooth | | $\left(p^3T^2 - 8pT + 1\right)^4 \left(p^6T^4 - 90p^3T^3 - 422pT^2 - 90T + 1\right)$ |
| 18 | smooth | | $\left(p^3T^2 - 8pT + 1\right)^4 \left(p^3T^2 + 128T + 1\right) \left(p^3T^2 - 8pT + 1\right)$ |
| 19 | smooth* | | |
| 20 | smooth | | $\left(p^3T^2 - 8pT + 1\right)^4 \left(p^3T^2 - 8pT + 1\right) \left(p^3T^2 + 8pT + 1\right)$ |
| 21 | smooth | | $\left(p^3T^2 + 4pT + 1\right)^4 \left(p^6T^4 + 12p^3T^3 + 754pT^2 + 12T + 1\right)$ |
| 22 | smooth | | $\left(p^3T^2 + 4pT + 1\right)^4 \left(p^3T^2 - 72T + 1\right) \left(p^3T^2 + 4pT + 1\right)$ |
| 23 | smooth | | $\left(p^3T^2 - 2pT + 1\right)^4 \left(p^6T^4 - 4p^3T^3 + 1266pT^2 - 4T + 1\right)$ |
| 24 | smooth | | $\left(p^3T^2 + 4pT + 1\right)^4 \left(p^6T^4 + 82p^3T^3 + 1394pT^2 + 82T + 1\right)$ |
| 25 | smooth | | $\left(p^3T^2 + 4pT + 1\right)^4 \left(p^6T^4 + 72p^3T^3 - 446pT^2 + 72T + 1\right)$ |
| 26 | smooth | | $\left(p^3T^2 + 4pT + 1\right)^4 \left(p^3T^2 - 202T + 1\right) \left(p^3T^2 + 4pT + 1\right)$ |
| 27 | smooth | | $\left(p^3T^2 + 10pT + 1\right)^4 \left(p^6T^4 + 198p^3T^3 + 562pT^2 + 198T + 1\right)$ |
| 28 | smooth | | $\left(p^3T^2 - 8pT + 1\right)^4 \left(p^6T^4 + 1858pT^2 + 1\right)$ |
| 29 | smooth | | $\left(p^3T^2 - 2pT + 1\right)^4 \left(p^6T^4 + 196p^3T^3 + 1346pT^2 + 196T + 1\right)$ |
| 30 | smooth | | $\left(p^3T^2 + 4pT + 1\right)^4 \left(p^6T^4 - 318p^3T^3 + 1954pT^2 - 318T + 1\right)$ |

| | | | $p = 37$ |
|---|---|---|---|
| $\varphi$ | smooth/sing. | singularity | $R(T)$ |
| 1 | singular | 1 | |
| 2 | smooth | | $\left(p^3T^2 + 4pT + 1\right)^4 \left(p^6T^4 + 54p^3T^3 + 322pT^2 + 54T + 1\right)$ |
| 3 | singular | $\frac{1}{25}$ | |
| 4 | smooth | | $\left(p^3T^2 - 2pT + 1\right)^4 \left(p^6T^4 - 48p^3T^3 + 2206pT^2 - 48T + 1\right)$ |
| 5 | smooth | | $\left(p^3T^2 - 2pT + 1\right)^4 \left(p^3T^2 - 404T + 1\right) \left(p^3T^2 - 2pT + 1\right)$ |
| 6 | smooth | | $\left(p^3T^2 + 4pT + 1\right)^4 \left(p^6T^4 + 64p^3T^3 + 302pT^2 + 64T + 1\right)$ |
| 7 | smooth | | $\left(p^3T^2 + 10pT + 1\right)^4 \left(p^6T^4 - 204p^3T^3 + 838pT^2 - 204T + 1\right)$ |
| 8 | smooth | | $\left(p^3T^2 + 10pT + 1\right)^4 \left(p^6T^4 - 274p^3T^3 + 2418pT^2 - 274T + 1\right)$ |
| 9 | smooth | | $\left(p^3T^2 - 2pT + 1\right)^4 \left(p^6T^4 + 232p^3T^3 + 1166pT^2 + 232T + 1\right)$ |
| 10 | smooth | | $\left(p^3T^2 + 10pT + 1\right)^4 \left(p^6T^4 - 244p^3T^3 + 2358pT^2 - 244T + 1\right)$ |
| 11 | smooth* | | |
| 12 | smooth | | $\left(p^3T^2 + 10pT + 1\right)^4 \left(p^6T^4 + 176p^3T^3 + 1758pT^2 + 176T + 1\right)$ |
| 13 | smooth | | $\left(p^3T^2 - 8pT + 1\right)^4 \left(p^6T^4 - 10p^3T^3 + 930pT^2 - 10T + 1\right)$ |
| 14 | smooth | | $\left(p^3T^2 - 2pT + 1\right)^4 \left(p^6T^4 - 18p^3T^3 + 826pT^2 - 18T + 1\right)$ |
| 15 | smooth | | $\left(p^3T^2 - 2pT + 1\right)^4 \left(p^6T^4 + 182p^3T^3 + 186pT^2 + 182T + 1\right)$ |
| 16 | smooth | | $\left(p^3T^2 - 2pT + 1\right)^4 \left(p^6T^4 - 28p^3T^3 - 474pT^2 - 28T + 1\right)$ |
| 17 | smooth | | $\left(p^3T^2 - 2pT + 1\right)^4 \left(p^6T^4 - 18p^3T^3 - 1454pT^2 - 18T + 1\right)$ |
| 18 | smooth | | $\left(p^3T^2 + 10pT + 1\right)^4 \left(p^6T^4 + 316p^3T^3 + 3158pT^2 + 316T + 1\right)$ |
| 19 | smooth | | $\left(p^3T^2 - 2pT + 1\right)^4 \left(p^6T^4 + 162p^3T^3 + 1186pT^2 + 162T + 1\right)$ |
| 20 | smooth | | $\left(p^3T^2 - 2pT + 1\right)^4 \left(p^6T^4 - 188p^3T^3 + 1766pT^2 - 188T + 1\right)$ |
| 21 | smooth | | $\left(p^3T^2 - 2pT + 1\right)^4 \left(p^3T^2 + 346T + 1\right) \left(p^3T^2 - 2pT + 1\right)$ |
| 22 | smooth | | $\left(p^3T^2 - 2pT + 1\right)^4 \left(p^6T^4 - 258p^3T^3 + 2146pT^2 - 258T + 1\right)$ |
| 23 | smooth | | $\left(p^3T^2 - 8pT + 1\right)^4 \left(p^6T^4 + 170p^3T^3 + 1890pT^2 + 170T + 1\right)$ |
| 24 | smooth | | $\left(p^3T^2 + 4pT + 1\right)^4 \left(p^3T^2 + 358T + 1\right) \left(p^3T^2 - 2pT + 1\right)$ |
| 25 | smooth | | $\left(p^3T^2 + 10pT + 1\right)^4 \left(p^6T^4 - 24p^3T^3 + 238pT^2 - 24T + 1\right)$ |
| 26 | smooth | | $\left(p^3T^2 - 2pT + 1\right)^4 \left(p^6T^4 + 12p^3T^3 + 1846pT^2 + 12T + 1\right)$ |
| 27 | smooth | | $\left(p^3T^2 - 2pT + 1\right)^4 \left(p^6T^4 + 432p^3T^3 + 3166pT^2 + 432T + 1\right)$ |
| 28 | smooth | | $\left(p^3T^2 - 2pT + 1\right)^4 \left(p^6T^4 - 88p^3T^3 - 1074pT^2 - 88T + 1\right)$ |
| 29 | smooth | | $\left(p^3T^2 - 2pT + 1\right)^4 \left(p^6T^4 + 432p^3T^3 + 3886pT^2 + 432T + 1\right)$ |
| 30 | smooth | | $\left(p^3T^2 - 2pT + 1\right)^4 \left(p^6T^4 - 8p^3T^3 + 2126pT^2 - 8T + 1\right)$ |
| 31 | smooth | | $\left(p^3T^2 - 8pT + 1\right)^4 \left(p^6T^4 + 30p^3T^3 + 1090pT^2 + 30T + 1\right)$ |
| 32 | smooth | | $\left(p^3T^2 - 8pT + 1\right)^4 \left(p^6T^4 + 30p^3T^3 + 2410pT^2 + 30T + 1\right)$ |
| 33 | singular | $\frac{1}{9}$ | |
| 34 | smooth | | $\left(p^3T^2 - 2pT + 1\right)^4 \left(p^6T^4 - 28p^3T^3 + 1446pT^2 - 28T + 1\right)$ |
| 35 | smooth* | | |
| | | | *Continued on the following page* |

| | | | |
|---|---|---|---|
| $p = 37$, *continued* | | | |
| $\varphi$ | smooth/sing. | singularity | $R(T)$ |
| 36 | smooth | | $\left(p^3T^2 - 2pT + 1\right)^4 \left(p^6T^4 - 308p^3T^3 + 2486pT^2 - 308T + 1\right)$ |

| | | | |
|---|---|---|---|
| $p = 41$ | | | |
| $\varphi$ | smooth/sing. | singularity | $R(T)$ |
| 1 | singular | 1 | |
| 2 | smooth | | $\left(p^3T^2 - 6pT + 1\right)^4 \left(p^6T^4 - 48p^3T^3 + 1214pT^2 - 48T + 1\right)$ |
| 3 | smooth | | $\left(p^3T^2 + 12pT + 1\right)^4 \left(p^6T^4 - 230p^3T^3 + 2018pT^2 - 230T + 1\right)$ |
| 4 | smooth* | | |
| 5 | smooth | | $\left(p^3T^2 - 6pT + 1\right)^4 \left(p^6T^4 + 192p^3T^3 - 226pT^2 + 192T + 1\right)$ |
| 6 | smooth | | $\left(p^3T^2 - 12pT + 1\right)^4 \left(p^6T^4 + 396p^3T^3 + 2966pT^2 + 396T + 1\right)$ |
| 7 | smooth | | $\left(p^3T^2 + 1\right)^4 \left(p^6T^4 - 12p^3T^3 + 2342pT^2 - 12T + 1\right)$ |
| 8 | smooth | | $\left(p^3T^2 + 6pT + 1\right)^4 \left(p^6T^4 - 396p^3T^3 + 2390pT^2 - 396T + 1\right)$ |
| 9 | smooth | | $\left(p^3T^2 + 6pT + 1\right)^4 \left(p^6T^4 + 24p^3T^3 - 850pT^2 + 24T + 1\right)$ |
| 10 | smooth | | $\left(p^3T^2 + 6pT + 1\right)^4 \left(p^3T^2 - 422T + 1\right) \left(p^3T^2 + 6pT + 1\right)$ |
| 11 | smooth | | $\left(p^3T^2 + 6pT + 1\right)^4 \left(p^6T^4 - 246p^3T^3 + 1970pT^2 - 246T + 1\right)$ |
| 12 | smooth | | $\left(p^3T^2 + 1\right)^4 \left(p^6T^4 - 282p^3T^3 + 482pT^2 - 282T + 1\right)$ |
| 13 | smooth | | $\left(p^3T^2 + 6pT + 1\right)^4 \left(p^6T^4 - 106p^3T^3 + 290pT^2 - 106T + 1\right)$ |
| 14 | smooth | | $\left(p^3T^2 + 1\right)^4 \left(p^6T^4 + 8p^3T^3 - 2338pT^2 + 8T + 1\right)$ |
| 15 | smooth | | $\left(p^3T^2 - 6pT + 1\right)^4 \left(p^6T^4 + 182p^3T^3 + 1154pT^2 + 182T + 1\right)$ |
| 16 | smooth* | | |
| 17 | smooth | | $\left(p^3T^2 + 1\right)^4 \left(p^6T^4 + 148p^3T^3 + 422pT^2 + 148T + 1\right)$ |
| 18 | smooth | | $\left(p^3T^2 + 6pT + 1\right)^4 \left(p^6T^4 - 76p^3T^3 - 490pT^2 - 76T + 1\right)$ |
| 19 | smooth | | $\left(p^3T^2 + 12pT + 1\right)^4 \left(p^6T^4 - 30p^3T^3 + 578pT^2 - 30T + 1\right)$ |
| 20 | smooth | | $\left(p^3T^2 - 6pT + 1\right)^4 \left(p^6T^4 - 108p^3T^3 - 106pT^2 - 108T + 1\right)$ |
| 21 | smooth | | $\left(p^3T^2 - 6pT + 1\right)^4 \left(p^6T^4 - 128p^3T^3 - 1186pT^2 - 128T + 1\right)$ |
| 22 | smooth | | $\left(p^3T^2 + 6pT + 1\right)^4 \left(p^6T^4 - 336p^3T^3 + 2030pT^2 - 336T + 1\right)$ |
| 23 | singular | $\frac{1}{25}$ | |
| 24 | smooth | | $\left(p^3T^2 + 6pT + 1\right)^4 \left(p^6T^4 + 4p^3T^3 + 470pT^2 + 4T + 1\right)$ |
| 25 | smooth | | $\left(p^3T^2 + 6pT + 1\right)^4 \left(p^6T^4 - 36p^3T^3 - 250pT^2 - 36T + 1\right)$ |
| 26 | smooth | | $\left(p^3T^2 + 1\right)^4 \left(p^6T^4 - 312p^3T^3 + 1742pT^2 - 312T + 1\right)$ |
| 27 | smooth | | $\left(p^3T^2 + 1\right)^4 \left(p^6T^4 + 168p^3T^3 - 898pT^2 + 168T + 1\right)$ |
| 28 | smooth | | $\left(p^3T^2 + 1\right)^4 \left(p^6T^4 + 48p^3T^3 - 1138pT^2 + 48T + 1\right)$ |
| 29 | smooth | | $\left(p^3T^2 - 6pT + 1\right)^4 \left(p^6T^4 + 172p^3T^3 + 614pT^2 + 172T + 1\right)$ |
| 30 | smooth | | $\left(p^3T^2 + 1\right)^4 \left(p^6T^4 - 82p^3T^3 + 2642pT^2 - 82T + 1\right)$ |
| 31 | smooth | | $\left(p^3T^2 + 6pT + 1\right)^4 \left(p^6T^4 + 104p^3T^3 + 590pT^2 + 104T + 1\right)$ |

| | | | $p = 41$, continued |
|---|---|---|---|
| $\varphi$ | smooth/sing. | singularity | $R(T)$ |
| 32 | singular | $\frac{1}{9}$ | |
| 33 | smooth | | $\left(p^3T^2 - 6pT + 1\right)^4 \left(p^6T^4 + 292p^3T^3 + 1334pT^2 + 292T + 1\right)$ |
| 34 | smooth | | $\left(p^3T^2 - 6pT + 1\right)^4 \left(p^6T^4 + 282p^3T^3 + 3554pT^2 + 282T + 1\right)$ |
| 35 | smooth | | $\left(p^3T^2 - 6pT + 1\right)^4 \left(p^3T^2 - 162T + 1\right) \left(p^3T^2 - 6pT + 1\right)$ |
| 36 | smooth | | $\left(p^3T^2 - 6pT + 1\right)^4 \left(p^6T^4 + 392p^3T^3 + 3374pT^2 + 392T + 1\right)$ |
| 37 | smooth | | $\left(p^3T^2 + 6pT + 1\right)^4 \left(p^6T^4 + 284p^3T^3 + 1670pT^2 + 284T + 1\right)$ |
| 38 | smooth | | $\left(p^3T^2 - 12pT + 1\right)^4 \left(p^6T^4 - 44p^3T^3 - 154pT^2 - 44T + 1\right)$ |
| 39 | smooth | | $\left(p^3T^2 - 6pT + 1\right)^4 \left(p^6T^4 + 552p^3T^3 + 4334pT^2 + 552T + 1\right)$ |
| 40 | smooth | | $\left(p^3T^2 - 6pT + 1\right)^4 \left(p^6T^4 + 92p^3T^3 + 2054pT^2 + 92T + 1\right)$ |

| | | | $p = 43$ |
|---|---|---|---|
| $\varphi$ | smooth/sing. | singularity | $R(T)$ |
| 1 | singular | 1 | |
| 2 | smooth | | $\left(p^3T^2 - 8pT + 1\right)^4 \left(p^6T^4 - 212p^3T^3 + 482pT^2 - 212T + 1\right)$ |
| 3 | smooth | | $\left(p^3T^2 + 4pT + 1\right)^4 \left(p^6T^4 + 94p^3T^3 + 2186pT^2 + 94T + 1\right)$ |
| 4 | smooth | | $\left(p^3T^2 - 8pT + 1\right)^4 \left(p^6T^4 + 48p^3T^3 + 1282pT^2 + 48T + 1\right)$ |
| 5 | smooth | | $\left(p^3T^2 - 8pT + 1\right)^4 \left(p^6T^4 - 72p^3T^3 + 1762pT^2 - 72T + 1\right)$ |
| 6 | smooth | | $\left(p^3T^2 - 8pT + 1\right)^4 \left(p^3T^2 + 412T + 1\right) \left(p^3T^2 - 8pT + 1\right)$ |
| 7 | smooth | | $\left(p^3T^2 - 2pT + 1\right)^4 \left(p^6T^4 + 356p^3T^3 + 2994pT^2 + 356T + 1\right)$ |
| 8 | smooth | | $\left(p^3T^2 + 4pT + 1\right)^4 \left(p^6T^4 - 336p^3T^3 + 2626pT^2 - 336T + 1\right)$ |
| 9 | smooth | | $\left(p^3T^2 + 4pT + 1\right)^4 \left(p^6T^4 - 196p^3T^3 + 3666pT^2 - 196T + 1\right)$ |
| 10 | smooth | | $\left(p^3T^2 + 4pT + 1\right)^4 \left(p^3T^2 - 508T + 1\right) \left(p^3T^2 + 4pT + 1\right)$ |
| 11 | smooth | | $\left(p^3T^2 + 4pT + 1\right)^4 \left(p^6T^4 + 84p^3T^3 - 1454pT^2 + 84T + 1\right)$ |
| 12 | smooth* | | |
| 13 | smooth | | $\left(p^3T^2 + 4pT + 1\right)^4 \left(p^3T^2 - 508T + 1\right) \left(p^3T^2 + 4pT + 1\right)$ |
| 14 | smooth | | $\left(p^3T^2 - 8pT + 1\right)^4 \left(p^6T^4 + 368p^3T^3 + 2562pT^2 + 368T + 1\right)$ |
| 15 | smooth | | $\left(p^3T^2 - 8pT + 1\right)^4 \left(p^6T^4 + 48p^3T^3 + 1282pT^2 + 48T + 1\right)$ |
| 16 | smooth | | $\left(p^3T^2 + 4pT + 1\right)^4 \left(p^6T^4 - 196p^3T^3 - 654pT^2 - 196T + 1\right)$ |
| 17 | smooth | | $\left(p^3T^2 + 4pT + 1\right)^4 \left(p^6T^4 + 224p^3T^3 + 1986pT^2 + 224T + 1\right)$ |
| 18 | smooth | | $\left(p^3T^2 + 10pT + 1\right)^4 \left(p^6T^4 + 152p^3T^3 + 2418pT^2 + 152T + 1\right)$ |
| 19 | smooth | | $\left(p^3T^2 + 10pT + 1\right)^4 \left(p^6T^4 - 128p^3T^3 + 818pT^2 - 128T + 1\right)$ |
| 20 | smooth | | $\left(p^3T^2 - 8pT + 1\right)^4 \left(p^6T^4 - 12p^3T^3 - 1358pT^2 - 12T + 1\right)$ |
| 21 | smooth | | $\left(p^3T^2 - 8pT + 1\right)^4 \left(p^6T^4 + 168p^3T^3 - 1118pT^2 + 168T + 1\right)$ |
| 22 | smooth | | $\left(p^3T^2 - 8pT + 1\right)^4 \left(p^6T^4 + 38p^3T^3 - 2238pT^2 + 38T + 1\right)$ |
| 23 | smooth | | $\left(p^3T^2 + 4pT + 1\right)^4 \left(p^6T^4 + 44p^3T^3 + 2226pT^2 + 44T + 1\right)$ |

| $\varphi$ | smooth/sing. | singularity | $R(T)$ |
|---|---|---|---|
| $p = 43$, *continued* | | | |
| 24 | singular | $\frac{1}{9}$ | |
| 25 | smooth | | $\left(p^3T^2 + 4pT + 1\right)^4 \left(p^6T^4 + 124p^3T^3 + 1586pT^2 + 124T + 1\right)$ |
| 26 | smooth | | $\left(p^3T^2 + 10pT + 1\right)^4 \left(p^6T^4 - 178p^3T^3 + 1938pT^2 - 178T + 1\right)$ |
| 27 | smooth | | $\left(p^3T^2 - 8pT + 1\right)^4 \left(p^3T^2 + 292T + 1\right)\left(p^3T^2 - 8pT + 1\right)$ |
| 28 | smooth | | $\left(p^3T^2 - 2pT + 1\right)^4 \left(p^6T^4 + 96p^3T^3 - 1166pT^2 + 96T + 1\right)$ |
| 29 | smooth | | $\left(p^3T^2 - 8pT + 1\right)^4 \left(p^6T^4 - 162p^3T^3 + 1882pT^2 - 162T + 1\right)$ |
| 30 | smooth* | | |
| 31 | singular | $\frac{1}{25}$ | |
| 32 | smooth | | $\left(p^3T^2 - 2pT + 1\right)^4 \left(p^6T^4 - 214p^3T^3 - 366pT^2 - 214T + 1\right)$ |
| 33 | smooth | | $\left(p^3T^2 - 2pT + 1\right)^4 \left(p^6T^4 + 326p^3T^3 + 2394pT^2 + 326T + 1\right)$ |
| 34 | smooth | | $\left(p^3T^2 + 10pT + 1\right)^4 \left(p^6T^4 - 498p^3T^3 + 4618pT^2 - 498T + 1\right)$ |
| 35 | smooth | | $\left(p^3T^2 + 4pT + 1\right)^4 \left(p^6T^4 - 176p^3T^3 + 1346pT^2 - 176T + 1\right)$ |
| 36 | smooth | | $\left(p^3T^2 - 8pT + 1\right)^4 \left(p^3T^2 - 244T + 1\right)\left(p^3T^2 + 4pT + 1\right)$ |
| 37 | smooth | | $\left(p^3T^2 + 4pT + 1\right)^4 \left(p^6T^4 - 136p^3T^3 + 3426pT^2 - 136T + 1\right)$ |
| 38 | smooth | | $\left(p^3T^2 - 8pT + 1\right)^4 \left(p^6T^4 + 188p^3T^3 + 3282pT^2 + 188T + 1\right)$ |
| 39 | smooth | | $\left(p^3T^2 + 4pT + 1\right)^4 \left(p^6T^4 + 274p^3T^3 + 3506pT^2 + 274T + 1\right)$ |
| 40 | smooth | | $\left(p^3T^2 + 4pT + 1\right)^4 \left(p^6T^4 + 264p^3T^3 + 1186pT^2 + 264T + 1\right)$ |
| 41 | smooth | | $\left(p^3T^2 + 4pT + 1\right)^4 \left(p^3T^2 - 428T + 1\right)\left(p^3T^2 + 4pT + 1\right)$ |
| 42 | smooth | | $\left(p^3T^2 + 10pT + 1\right)^4 \left(p^6T^4 + 42p^3T^3 - 1982pT^2 + 42T + 1\right)$ |

| $\varphi$ | smooth/sing. | singularity | $R(T)$ |
|---|---|---|---|
| $p = 47$ | | | |
| 1 | singular | 1 | |
| 2 | smooth | | $\left(p^3T^2 + 1\right)^4 \left(p^3T^2 + 1\right)\left(p^3T^2 - 264T + 1\right)$ |
| 3 | smooth | | $\left(p^3T^2 + 1\right)^4 \left(p^3T^2 + 1\right)\left(p^3T^2 + 136T + 1\right)$ |
| 4 | smooth | | $\left(p^3T^2 + 12pT + 1\right)^4 \left(p^6T^4 - 240p^3T^3 + 1730pT^2 - 240T + 1\right)$ |
| 5 | smooth | | $\left(p^3T^2 - 6pT + 1\right)^4 \left(p^6T^4 - 216p^3T^3 + 1202pT^2 - 216T + 1\right)$ |
| 6 | smooth | | $\left(p^3T^2 + 12pT + 1\right)^4 \left(p^6T^4 - 480p^3T^3 + 3650pT^2 - 480T + 1\right)$ |
| 7 | smooth | | $\left(p^3T^2 + 1\right)^4 \left(p^6T^4 + 376p^3T^3 + 3458pT^2 + 376T + 1\right)$ |
| 8 | smooth | | $\left(p^3T^2 + 1\right)^4 \left(p^6T^4 - 144p^3T^3 + 1538pT^2 - 144T + 1\right)$ |
| 9 | smooth | | $\left(p^3T^2 + 1\right)^4 \left(p^3T^2 + 1\right)\left(p^3T^2 - 444T + 1\right)$ |
| 10 | smooth | | $\left(p^3T^2 - 12pT + 1\right)^4 \left(p^6T^4 + 72p^3T^3 + 866pT^2 + 72T + 1\right)$ |
| 11 | smooth | | $\left(p^3T^2 - 6pT + 1\right)^4 \left(p^3T^2 - 402T + 1\right)\left(p^3T^2 + 8pT + 1\right)$ |
| 12 | smooth | | $\left(p^3T^2 + 1\right)^4 \left(p^6T^4 - 124p^3T^3 + 578pT^2 - 124T + 1\right)$ |
| 13 | smooth | | $\left(p^3T^2 + 6pT + 1\right)^4 \left(p^6T^4 - 2p^3T^3 + 74pT^2 - 2T + 1\right)$ |

| $\varphi$ | smooth/sing. | singularity | $R(T)$ |
|---|---|---|---|
| | $p = 47$, continued | | |
| 14 | smooth* | | |
| 15 | smooth | | $\left(p^3T^2 - 6pT + 1\right)^4 \left(p^6T^4 - 276p^3T^3 + 3842pT^2 - 276T + 1\right)$ |
| 16 | smooth | | $\left(p^3T^2 + 1\right)^4 \left(p^3T^2 + 1\right) \left(p^3T^2 + 136T + 1\right)$ |
| 17 | smooth | | $\left(p^3T^2 - 12pT + 1\right)^4 \left(p^3T^2 + 416T + 1\right) \left(p^3T^2 - 12pT + 1\right)$ |
| 18 | smooth | | $\left(p^3T^2 + 1\right)^4 \left(p^6T^4 - 64p^3T^3 + 3458pT^2 - 64T + 1\right)$ |
| 19 | smooth | | $\left(p^3T^2 + 6pT + 1\right)^4 \left(p^6T^4 + 98p^3T^3 + 1754pT^2 + 98T + 1\right)$ |
| 20 | smooth | | $\left(p^3T^2 + 12pT + 1\right)^4 \left(p^3T^2 - 24T + 1\right) \left(p^3T^2 + 12pT + 1\right)$ |
| 21 | singular | $\frac{1}{9}$ | |
| 22 | smooth | | $\left(p^3T^2 + 1\right)^4 \left(p^6T^4 + 436p^3T^3 + 3218pT^2 + 436T + 1\right)$ |
| 23 | smooth | | $\left(p^3T^2 + 6pT + 1\right)^4 \left(p^6T^4 + 18p^3T^3 + 434pT^2 + 18T + 1\right)$ |
| 24 | smooth | | $\left(p^3T^2 + 12pT + 1\right)^4 \left(p^6T^4 + 440p^3T^3 + 3650pT^2 + 440T + 1\right)$ |
| 25 | smooth | | $\left(p^3T^2 + 1\right)^4 \left(p^6T^4 - 244p^3T^3 + 2018pT^2 - 244T + 1\right)$ |
| 26 | smooth | | $\left(p^3T^2 - 6pT + 1\right)^4 \left(p^3T^2 - 32T + 1\right) \left(p^3T^2 - 12pT + 1\right)$ |
| 27 | smooth | | $\left(p^3T^2 - 12pT + 1\right)^4 \left(p^6T^4 - 288p^3T^3 + 1346pT^2 - 288T + 1\right)$ |
| 28 | smooth | | $\left(p^3T^2 + 1\right)^4 \left(p^6T^4 - 304p^3T^3 + 3458pT^2 - 304T + 1\right)$ |
| 29 | smooth | | $\left(p^3T^2 + 1\right)^4 \left(p^6T^4 + 166p^3T^3 + 218pT^2 + 166T + 1\right)$ |
| 30 | smooth | | $\left(p^3T^2 - 6pT + 1\right)^4 \left(p^6T^4 - 106p^3T^3 + 3602pT^2 - 106T + 1\right)$ |
| 31 | smooth | | $\left(p^3T^2 + 1\right)^4 \left(p^6T^4 + 156p^3T^3 - 382pT^2 + 156T + 1\right)$ |
| 32 | singular | $\frac{1}{25}$ | |
| 33 | smooth | | $\left(p^3T^2 + 1\right)^4 \left(p^6T^4 + 536p^3T^3 + 4898pT^2 + 536T + 1\right)$ |
| 34 | smooth* | | |
| 35 | smooth | | $\left(p^3T^2 + 12pT + 1\right)^4 \left(p^6T^4 - 560p^3T^3 + 5570pT^2 - 560T + 1\right)$ |
| 36 | smooth | | $\left(p^3T^2 + 1\right)^4 \left(p^6T^4 - 304p^3T^3 + 578pT^2 - 304T + 1\right)$ |
| 37 | smooth | | $\left(p^3T^2 + 1\right)^4 \left(p^6T^4 + 16p^3T^3 - 2302pT^2 + 16T + 1\right)$ |
| 38 | smooth | | $\left(p^3T^2 + 1\right)^4 \left(p^6T^4 + 296p^3T^3 + 2978pT^2 + 296T + 1\right)$ |
| 39 | smooth | | $\left(p^3T^2 + 6pT + 1\right)^4 \left(p^6T^4 + 708p^3T^3 + 5714pT^2 + 708T + 1\right)$ |
| 40 | smooth | | $\left(p^3T^2 + 12pT + 1\right)^4 \left(p^6T^4 + 270p^3T^3 + 2930pT^2 + 270T + 1\right)$ |
| 41 | smooth | | $\left(p^3T^2 - 12pT + 1\right)^4 \left(p^6T^4 + 222p^3T^3 + 1826pT^2 + 222T + 1\right)$ |
| 42 | smooth | | $\left(p^3T^2 - 12pT + 1\right)^4 \left(p^6T^4 - 168p^3T^3 + 3266pT^2 - 168T + 1\right)$ |
| 43 | smooth | | $\left(p^3T^2 + 6pT + 1\right)^4 \left(p^6T^4 + 488p^3T^3 + 3314pT^2 + 488T + 1\right)$ |
| 44 | smooth | | $\left(p^3T^2 - 12pT + 1\right)^4 \left(p^6T^4 - 158p^3T^3 + 1826pT^2 - 158T + 1\right)$ |
| 45 | smooth | | $\left(p^3T^2 - 6pT + 1\right)^4 \left(p^6T^4 - 366p^3T^3 + 3122pT^2 - 366T + 1\right)$ |
| 46 | smooth | | $\left(p^3T^2 + 6pT + 1\right)^4 \left(p^6T^4 - 612p^3T^3 + 5714pT^2 - 612T + 1\right)$ |

## B.2. Rational points and elliptic modular forms

Table 11: *Some rational points on the $S_5$ symmetric line $\boldsymbol{\varphi} = (\varphi, \ldots, \varphi)$ in the moduli space of Hulek-Verrill manifolds together with the modular forms corresponding to the quadratic factors of the local zeta functions of these manifolds, and the related twisted Sen curves $\mathcal{E}_S$. The second to fifth columns labelled $\mathcal{E}_S(\tau/n)$ give the LMFDB label of the twisted Sen curves corresponding to the four isogenous families found in §6.2, and the penultimate column lists the label of the corresponding modular form common to all isogenous curves. The column labelled "p" lists primes at which the modular form coefficient $\alpha_p$ is not equal to the zeta function coefficient $c_p$. The last column gives the size of the isogeny class of the twisted Sen curves, confirming that there are indeed only four distinct families of twisted Sen curves.*

| $\varphi$ | $\mathcal{E}_S(\tau)$ | $\mathcal{E}_S(\tau/2)$ | $\mathcal{E}_S(\tau/3)$ | $\mathcal{E}_S(\tau/6)$ | $p$ | form label | # |
|---|---|---|---|---|---|---|---|
| $\frac{1}{385}$ | 108570.cs2 | 108570.cs4 | 108570.cs1 | 108570.cs3 | | 108570.cs1 | 4 |
| $\frac{1}{105}$ | 8190.j2 | 8190.j4 | 8190.j1 | 8190.j3 | | 8190.j1 | 4 |
| $\frac{1}{77}$ | 99484.d2 | 99484.d4 | 99484.d1 | 99484.d3 | | 99484.d1 | 4 |
| $\frac{1}{70}$ | 294630.bg4 | 294630.bg3 | 294630.bg2 | 294630.bg1 | | 294630.bg2 | 4 |
| $\frac{1}{66}$ | 244530.bh4 | 244530.bh3 | 244530.bh2 | 244530.bh1 | | 244530.bh2 | 4 |
| $\frac{1}{55}$ | 15180.m4 | 15180.m3 | 15180.m2 | 15180.m1 | | 15180.m2 | 4 |
| $\frac{1}{42}$ | 56826.h4 | 56826.h3 | 56826.h2 | 56826.h1 | | 56826.h2 | 4 |
| $\frac{2}{77}$ | 136290.cl4 | 136290.cl3 | 136290.cl2 | 136290.cl1 | | 136290.cl2 | 4 |
| $\frac{1}{35}$ | 30940.b4 | 30940.b3 | 30940.b2 | 30940.b1 | | 30940.b2 | 4 |
| $\frac{1}{33}$ | 198.d2 | 198.d4 | 198.d1 | 198.d3 | | 198.d1 | 4 |
| $\frac{1}{30}$ | 18270.h4 | 18270.h3 | 18270.h2 | 18270.h1 | | 18270.h2 | 4 |
| $\frac{2}{55}$ | 215710.l4 | 215710.l3 | 215710.l2 | 215710.l1 | | 215710.l2 | 4 |
| $\frac{3}{77}$ | 170940.t4 | 170940.t3 | 170940.t2 | 170940.t1 | | 170940.t2 | 4 |
| $\frac{1}{22}$ | 6006.n4 | 6006.n3 | 6006.n2 | 6006.n1 | | 6006.n2 | 4 |
| $\frac{1}{21}$ | 1260.j3 | 1260.j4 | 1260.j1 | 1260.j2 | | 1260.j1 | 4 |
| $\frac{4}{77}$ | 460922.d4 | 460922.d3 | 460922.d2 | 460922.d1 | | 460922.d2 | 4 |
| $\frac{3}{55}$ | 60060.bc3 | 60060.bc4 | 60060.bc1 | 60060.bc2 | | 60060.bc1 | 4 |
| $\frac{2}{35}$ | 39270.cn4 | 39270.cn3 | 39270.cn2 | 39270.cn1 | | 39270.cn2 | 4 |
| $\frac{2}{33}$ | 30690.bc4 | 30690.bc3 | 30690.bc2 | 30690.bc1 | | 30690.bc2 | 4 |
| $\frac{5}{77}$ | 2310.l5 | 2310.l7 | 2310.l3 | 2310.l4 | | 2310.l3 | 8 |
| $\frac{1}{15}$ | 1260.d4 | 1260.d3 | 1260.d2 | 1260.d1 | | 1260.d2 | 4 |
| $\frac{1}{14}$ | 910.a4 | 910.a3 | 910.a2 | 910.a1 | | 910.a2 | 4 |
| $\frac{4}{55}$ | 106590.bz4 | 106590.bz3 | 106590.bz2 | 106590.bz1 | | 106590.bz2 | 4 |
| $\frac{5}{66}$ | 422730.bw4 | 422730.bw3 | 422730.bw2 | 422730.bw1 | | 422730.bw2 | 4 |
| $\frac{13}{165}$ | 244530.c3 | 244530.c4 | 244530.c1 | 244530.c2 | | 244530.c1 | 4 |
| $\frac{3}{35}$ | 210.d5 | 210.d7 | 210.d3 | 210.d4 | | 210.d3 | 8 |
| $\frac{1}{11}$ | 220.a4 | 220.a3 | 220.a2 | 220.a1 | | 220.a2 | 4 |

| $\varphi$ | $\mathcal{E}_S(\tau)$ | $\mathcal{E}_S(\tau/2)$ | $\mathcal{E}_S(\tau/3)$ | $\mathcal{E}_S(\tau/6)$ | $p$ | form label | # |
|---|---|---|---|---|---|---|---|
| continued | | | | | | | |
| $\frac{2}{21}$ | 2394.k4 | 2394.k3 | 2394.k2 | 2394.k1 | | 2394.k2 | 4 |
| $\frac{23}{231}$ | 414414.ck3 | 414414.ck4 | 414414.ck1 | 414414.ck2 | | 414414.ck1 | 4 |
| $\frac{1}{10}$ | 30.a6 | 30.a5 | 30.a3 | 30.a1 | | 30.a3 | 8 |
| $\frac{8}{77}$ | 53130.cf4 | 53130.cf3 | 53130.cf2 | 53130.cf1 | | 53130.cf2 | 4 |
| $\frac{7}{66}$ | 81774.r4 | 81774.r3 | 81774.r2 | 81774.r1 | | 81774.r2 | 4 |
| $\frac{6}{55}$ | 2310.u7 | 2310.u5 | 2310.u3 | 2310.u1 | | 2310.u3 | 8 |
| $\frac{17}{154}$ | 358666.b4 | 358666.b3 | 358666.b2 | 358666.b1 | | 358666.b2 | 4 |
| $\frac{4}{35}$ | 2170.k4 | 2170.k3 | 2170.k2 | 2170.k1 | | 2170.k2 | 4 |
| $\frac{9}{77}$ | 15708.g4 | 15708.g3 | 15708.g2 | 15708.g1 | | 15708.g2 | 4 |
| $\frac{5}{42}$ | 23310.w4 | 23310.w3 | 23310.w2 | 23310.w1 | | 23310.w2 | 4 |
| $\frac{4}{33}$ | 5742.t4 | 5742.t3 | 5742.t2 | 5742.t1 | | 5742.t2 | 4 |
| $\frac{13}{105}$ | 376740.v4 | 376740.v3 | 376740.v2 | 376740.v1 | | 376740.v2 | 4 |
| $\frac{7}{55}$ | 2310.t4 | 2310.t3 | 2310.t2 | 2310.t1 | | 2310.t2 | 4 |
| $\frac{9}{70}$ | 140910.be4 | 140910.be3 | 140910.be2 | 140910.be1 | | 140910.be2 | 4 |
| $\frac{2}{15}$ | 1170.h4 | 1170.h3 | 1170.h2 | 1170.h1 | | 1170.h2 | 4 |
| $\frac{31}{231}$ | 214830.bt4 | 214830.bt3 | 214830.bt2 | 214830.bt1 | | 214830.bt2 | 4 |
| $\frac{3}{22}$ | 6270.k4 | 6270.k3 | 6270.k2 | 6270.k1 | | 6270.k2 | 4 |
| $\frac{1}{7}$ | 84.b4 | 84.b3 | 84.b2 | 84.b1 | | 84.b2 | 4 |
| $\frac{8}{55}$ | 87890.g4 | 87890.g3 | 87890.g2 | 87890.g1 | | 87890.g2 | 4 |
| $\frac{5}{33}$ | 13860.t4 | 13860.t3 | 13860.t2 | 13860.t1 | | 13860.t2 | 4 |
| $\frac{17}{105}$ | 117810.x4 | 117810.x3 | 117810.x2 | 117810.x1 | | 117810.x2 | 4 |
| $\frac{9}{55}$ | 197340.n4 | 197340.n3 | 197340.n2 | 197340.n1 | | 197340.n2 | 4 |
| $\frac{1}{6}$ | 90.a4 | 90.a3 | 90.a2 | 90.a1 | | 90.a2 | 4 |
| $\frac{13}{77}$ | 10010.n4 | 10010.n3 | 10010.n2 | 10010.n1 | | 10010.n2 | 4 |
| $\frac{6}{35}$ | 115710.cr4 | 115710.cr3 | 115710.cr2 | 115710.cr1 | | 115710.cr2 | 4 |
| $\frac{29}{165}$ | 488070.c4 | 488070.c3 | 488070.c2 | 488070.c1 | | 488070.c2 | 4 |
| $\frac{2}{11}$ | 462.f4 | 462.f3 | 462.f2 | 462.f1 | | 462.f2 | 4 |
| $\frac{4}{21}$ | 10710.bc4 | 10710.bc3 | 10710.bc2 | 10710.bc1 | | 10710.bc2 | 4 |
| $\frac{1}{5}$ | 20.a4 | 20.a3 | 20.a2 | 20.a1 | | 20.a2 | 4 |
| $\frac{7}{33}$ | 180180.q4 | 180180.q3 | 180180.q2 | 180180.q1 | | 180180.q2 | 4 |
| $\frac{3}{14}$ | 6006.p4 | 6006.p3 | 6006.p2 | 6006.p1 | | 6006.p2 | 4 |
| $\frac{37}{165}$ | 256410.bs4 | 256410.bs3 | 256410.bs2 | 256410.bs1 | | 256410.bs2 | 4 |
| $\frac{5}{22}$ | 43010.a4 | 43010.a3 | 43010.a2 | 43010.a1 | | 43010.a2 | 4 |
| $\frac{8}{35}$ | 7770.ba6 | 7770.ba5 | 7770.ba4 | 7770.ba3 | | 7770.ba4 | 6 |

| $\varphi$ | $\mathcal{E}_S(\tau)$ | $\mathcal{E}_S(\tau/2)$ | $\mathcal{E}_S(\tau/3)$ | $\mathcal{E}_S(\tau/6)$ | $p$ | form label | # |
|---|---|---|---|---|---|---|---|
| | | | continued | | | | |
| $\frac{7}{30}$ | 159390.s4 | 159390.s2 | 159390.s3 | 159390.s1 | | 159390.s3 | 4 |
| $\frac{5}{21}$ | 630.j4 | 630.j3 | 630.j2 | 630.j1 | | 630.j2 | 4 |
| $\frac{8}{33}$ | 12870.cc4 | 12870.cc2 | 12870.cc3 | 12870.cc1 | | 12870.cc3 | 4 |
| $\frac{9}{35}$ | 125580.s4 | 125580.s2 | 125580.s3 | 125580.s1 | | 125580.s3 | 4 |
| $\frac{4}{15}$ | 6930.ba4 | 6930.ba2 | 6930.ba3 | 6930.ba1 | | 6930.ba3 | 4 |
| $\frac{3}{11}$ | 66.a4 | 66.a3 | 66.a2 | 66.a1 | | 66.a2 | 4 |
| $\frac{2}{7}$ | 770.g4 | 770.g2 | 770.g3 | 770.g1 | | 770.g3 | 4 |
| $\frac{16}{55}$ | 381810.bn4 | 381810.bn2 | 381810.bn3 | 381810.bn1 | | 381810.bn3 | 4 |
| $\frac{3}{10}$ | 3570.j4 | 3570.j2 | 3570.j3 | 3570.j1 | | 3570.j3 | 4 |
| $\frac{10}{33}$ | 432630.dv4 | 432630.dv2 | 432630.dv3 | 432630.dv1 | | 432630.dv3 | 4 |
| $\frac{17}{55}$ | 497420.g4 | 497420.g2 | 497420.g3 | 497420.g1 | | 497420.g3 | 4 |
| $\frac{13}{42}$ | 237510.v4 | 237510.v2 | 237510.v3 | 237510.v1 | | 237510.v3 | 4 |
| $\frac{11}{35}$ | 2310.i4 | 2310.i3 | 2310.i2 | 2310.i1 | | 2310.i2 | 4 |
| $\frac{7}{22}$ | 94710.br4 | 94710.br2 | 94710.br3 | 94710.br1 | | 94710.br3 | 4 |
| $\frac{12}{35}$ | 352590.cz3 | 352590.cz2 | 352590.cz4 | 352590.cz1 | | 352590.cz4 | 4 |
| $\frac{19}{55}$ | 363660.n3 | 363660.n2 | 363660.n4 | 363660.n1 | | 363660.n4 | 4 |
| $\frac{27}{77}$ | 383460.w3 | 383460.w2 | 383460.w4 | 383460.w1 | | 383460.w4 | 4 |
| $\frac{5}{14}$ | 6510.j3 | 6510.j2 | 6510.j4 | 6510.j1 | | 6510.j4 | 4 |
| $\frac{4}{11}$ | 770.f3 | 770.f2 | 770.f4 | 770.f1 | | 770.f4 | 4 |
| $\frac{11}{30}$ | 432630.c3 | 432630.c2 | 432630.c4 | 432630.c1 | | 432630.c4 | 4 |
| $\frac{29}{77}$ | 308154.bw3 | 308154.bw2 | 308154.bw4 | 308154.bw1 | | 308154.bw4 | 4 |
| $\frac{8}{21}$ | 27846.bf3 | 27846.bf2 | 27846.bf4 | 27846.bf1 | | 27846.bf4 | 4 |
| $\frac{41}{105}$ | 284130.de3 | 284130.de2 | 284130.de4 | 284130.de1 | | 284130.de4 | 4 |
| $\frac{13}{33}$ | 180180.bx3 | 180180.bx2 | 180180.bx4 | 180180.bx1 | | 180180.bx4 | 4 |
| $\frac{2}{5}$ | 390.g3 | 390.g2 | 390.g4 | 390.g1 | | 390.g4 | 4 |
| $\frac{17}{42}$ | 396270.bu3 | 396270.bu2 | 396270.bu4 | 396270.bu1 | | 396270.bu4 | 4 |
| $\frac{9}{22}$ | 50622.j3 | 50622.j2 | 50622.j4 | 50622.j1 | | 50622.j4 | 4 |
| $\frac{32}{77}$ | 487410.bw3 | 487410.bw2 | 487410.bw4 | 487410.bw1 | | 487410.bw4 | 4 |
| $\frac{23}{55}$ | 48070.f3 | 48070.f2 | 48070.f4 | 48070.f1 | | 48070.f4 | 4 |
| $\frac{3}{7}$ | 420.c3 | 420.c2 | 420.c4 | 420.c1 | | 420.c4 | 4 |
| $\frac{5}{11}$ | 11220.k3 | 11220.k2 | 11220.k4 | 11220.k1 | | 11220.k4 | 4 |
| $\frac{16}{35}$ | 144970.p3 | 144970.p2 | 144970.p4 | 144970.p1 | | 144970.p4 | 4 |
| $\frac{7}{15}$ | 630.c3 | 630.c2 | 630.c4 | 630.c1 | | 630.c4 | 4 |
| $\frac{10}{21}$ | 159390.eo3 | 159390.eo2 | 159390.eo4 | 159390.eo1 | | 159390.eo4 | 4 |

| $\varphi$ | $\mathcal{E}_S(\tau)$ | $\mathcal{E}_S(\tau/2)$ | $\mathcal{E}_S(\tau/3)$ | $\mathcal{E}_S(\tau/6)$ | $p$ | form label | # |
|---|---|---|---|---|---|---|---|
| $\frac{37}{77}$ | 28490.b3 | 28490.b2 | 28490.b4 | 28490.b1 | | 28490.b4 | 4 |
| $\frac{16}{33}$ | 124542.bc3 | 124542.bc2 | 124542.bc4 | 124542.bc1 | | 124542.bc4 | 4 |
| $\frac{17}{35}$ | 421260.ba3 | 421260.ba2 | 421260.ba4 | 421260.ba1 | | 421260.ba4 | 4 |
| $\frac{27}{55}$ | 217140.p3 | 217140.p2 | 217140.p4 | 217140.p1 | | 217140.p4 | 4 |
| $\frac{1}{2}$ | 14.a5 | 14.a4 | 14.a6 | 14.a3 | | 14.a6 | 6 |
| $\frac{28}{55}$ | 455070.cy3 | 455070.cy2 | 455070.cy4 | 455070.cy1 | | 455070.cy4 | 4 |
| $\frac{18}{35}$ | 453390.cs3 | 453390.cs2 | 453390.cs4 | 453390.cs1 | | 453390.cs4 | 4 |
| $\frac{17}{33}$ | 16830.bl3 | 16830.bl2 | 16830.bl4 | 16830.bl1 | | 16830.bl4 | 4 |
| $\frac{11}{21}$ | 180180.cb3 | 180180.cb2 | 180180.cb4 | 180180.cb1 | | 180180.cb4 | 4 |
| $\frac{8}{15}$ | 11970.bo3 | 11970.bo2 | 11970.bo4 | 11970.bo1 | | 11970.bo4 | 4 |
| $\frac{19}{35}$ | 22610.m3 | 22610.m2 | 22610.m4 | 22610.m1 | | 22610.m4 | 4 |
| $\frac{6}{11}$ | 14190.n3 | 14190.n2 | 14190.n4 | 14190.n1 | | 14190.n4 | 4 |
| $\frac{31}{55}$ | 71610.o3 | 71610.o2 | 71610.o4 | 71610.o1 | | 71610.o4 | 4 |
| $\frac{4}{7}$ | 1218.h3 | 1218.h2 | 1218.h4 | 1218.h1 | | 1218.h4 | 4 |
| $\frac{45}{77}$ | 94710.cv3 | 94710.cv2 | 94710.cv4 | 94710.cv1 | | 94710.cv4 | 4 |
| $\frac{13}{22}$ | 81510.bi3 | 81510.bi2 | 81510.bi4 | 81510.bi1 | | 81510.bi4 | 4 |
| $\frac{3}{5}$ | 660.c3 | 660.c2 | 660.c4 | 660.c1 | | 660.c4 | 4 |
| $\frac{20}{33}$ | 90090.dn3 | 90090.dn2 | 90090.dn4 | 90090.dn1 | | 90090.dn4 | 4 |
| $\frac{13}{21}$ | 1638.e3 | 1638.e2 | 1638.e4 | 1638.e1 | | 1638.e4 | 4 |
| $\frac{7}{11}$ | 4004.a3 | 4004.a2 | 4004.a4 | 4004.a1 | | 4004.a4 | 4 |
| $\frac{9}{14}$ | 14070.e3 | 14070.e2 | 14070.e4 | 14070.e1 | | 14070.e4 | 4 |
| $\frac{23}{35}$ | 415380.m3 | 415380.m2 | 415380.m4 | 415380.m1 | | 415380.m4 | 4 |
| $\frac{2}{3}$ | 90.b3 | 90.b2 | 90.b4 | 90.b1 | | 90.b4 | 4 |
| $\frac{15}{22}$ | 261030.v3 | 261030.v2 | 261030.v4 | 261030.v1 | | 261030.v4 | 4 |
| $\frac{24}{35}$ | 418110.bh3 | 418110.bh2 | 418110.bh4 | 418110.bh1 | | 418110.bh4 | 4 |
| $\frac{53}{77}$ | 122430.ba3 | 122430.ba2 | 122430.ba4 | 122430.ba1 | | 122430.ba4 | 4 |
| $\frac{7}{10}$ | 11130.n3 | 11130.n2 | 11130.n4 | 11130.n1 | | 11130.n4 | 4 |
| $\frac{39}{55}$ | 158730.bz3 | 158730.bz2 | 158730.bz4 | 158730.bz1 | | 158730.bz4 | 4 |
| $\frac{5}{7}$ | 2660.c3 | 2660.c2 | 2660.c4 | 2660.c1 | | 2660.c4 | 4 |
| $\frac{8}{11}$ | 4026.i3 | 4026.i2 | 4026.i4 | 4026.i1 | | 4026.i4 | 4 |
| $\frac{11}{15}$ | 13860.k3 | 13860.k2 | 13860.k4 | 13860.k1 | | 13860.k4 | 4 |
| $\frac{49}{66}$ | 117810.w3 | 117810.w2 | 117810.w4 | 117810.w1 | | 117810.w4 | 4 |
| $\frac{25}{33}$ | 990.e3 | 990.e2 | 990.e4 | 990.e1 | | 990.e4 | 4 |
| $\frac{16}{21}$ | 25830.bi3 | 25830.bi2 | 25830.bi4 | 25830.bi1 | | 25830.bi4 | 4 |

| $\varphi$ | $\mathcal{E}_S(\tau)$ | $\mathcal{E}_S(\tau/2)$ | $\mathcal{E}_S(\tau/3)$ | $\mathcal{E}_S(\tau/6)$ | $p$ | form label | # |
|---|---|---|---|---|---|---|---|
| | | | | continued | | | |
| $\frac{27}{35}$ | 2730.m5 | 2730.m4 | 2730.m6 | 2730.m3 | | 2730.m6 | 6 |
| $\frac{17}{22}$ | 244970.c3 | 244970.c2 | 244970.c4 | 244970.c1 | | 244970.c4 | 4 |
| $\frac{11}{14}$ | 39270.bf3 | 39270.bf2 | 39270.bf4 | 39270.bf1 | | 39270.bf4 | 4 |
| $\frac{4}{5}$ | 310.a3 | 310.a2 | 310.a4 | 310.a1 | | 310.a4 | 4 |
| $\frac{17}{21}$ | 47124.k3 | 47124.k2 | 47124.k4 | 47124.k1 | | 47124.k4 | 4 |
| $\frac{9}{11}$ | 4620.j3 | 4620.j2 | 4620.j4 | 4620.j1 | | 4620.j4 | 4 |
| $\frac{5}{6}$ | 1170.c3 | 1170.c2 | 1170.c4 | 1170.c1 | | 1170.c4 | 4 |
| $\frac{47}{55}$ | 118910.a3 | 118910.a2 | 118910.a4 | 118910.a1 | | 118910.a4 | 4 |
| $\frac{6}{7}$ | 1974.j3 | 1974.j2 | 1974.j4 | 1974.j1 | | 1974.j4 | 4 |
| $\frac{19}{22}$ | 186846.f3 | 186846.f2 | 186846.f4 | 186846.f1 | | 186846.f4 | 4 |
| $\frac{13}{15}$ | 39780.h3 | 39780.h2 | 39780.h4 | 39780.h1 | | 39780.h4 | 4 |
| $\frac{29}{33}$ | 218196.g3 | 218196.g2 | 218196.g4 | 218196.g1 | | 218196.g4 | 4 |
| $\frac{31}{35}$ | 264740.a3 | 264740.a2 | 264740.a4 | 264740.a1 | | 264740.a4 | 4 |
| $\frac{69}{77}$ | 180642.o3 | 180642.o2 | 180642.o4 | 180642.o1 | | 180642.o4 | 4 |
| $\frac{9}{10}$ | 2130.g3 | 2130.g2 | 2130.g4 | 2130.g1 | | 2130.g4 | 4 |
| $\frac{19}{21}$ | 23940.j3 | 23940.j2 | 23940.j4 | 23940.j1 | | 23940.j4 | 4 |
| $\frac{10}{11}$ | 8690.f3 | 8690.f2 | 8690.f4 | 8690.f1 | | 8690.f4 | 4 |
| $\frac{32}{35}$ | 53130.ch3 | 53130.ch2 | 53130.ch4 | 53130.ch1 | | 53130.ch4 | 4 |
| $\frac{97}{105}$ | 61110.c3 | 61110.c2 | 61110.c4 | 61110.c1 | | 61110.c4 | 4 |
| $\frac{13}{14}$ | 18746.c3 | 18746.c2 | 18746.c4 | 18746.c1 | | 18746.c4 | 4 |
| $\frac{14}{15}$ | 23310.bl3 | 23310.bl2 | 23310.bl4 | 23310.bl1 | | 23310.bl4 | 4 |
| $\frac{20}{21}$ | 33390.ca3 | 33390.ca2 | 33390.ca4 | 33390.ca1 | | 33390.ca4 | 4 |
| $\frac{21}{22}$ | 77154.k3 | 77154.k2 | 77154.k4 | 77154.k1 | | 77154.k4 | 4 |
| $\frac{29}{30}$ | 200970.p3 | 200970.p2 | 200970.p4 | 200970.p1 | | 200970.p4 | 4 |
| $\frac{32}{33}$ | 16830.bi3 | 16830.bi2 | 16830.bi4 | 16830.bi1 | | 16830.bi4 | 4 |
| $\frac{34}{35}$ | 322490.i3 | 322490.i2 | 322490.i4 | 322490.i1 | | 322490.i4 | 4 |
| $\frac{54}{55}$ | 142230.t3 | 142230.t2 | 142230.t4 | 142230.t1 | | 142230.t4 | 4 |
| $\frac{78}{77}$ | 30030.bt6 | 30030.bt2 | 30030.bt4 | 30030.bt1 | | 30030.bt4 | 8 |
| $\frac{56}{55}$ | 345730.n4 | 345730.n2 | 345730.n3 | 345730.n1 | | 345730.n3 | 4 |
| $\frac{36}{35}$ | 3570.w6 | 3570.w3 | 3570.w4 | 3570.w2 | | 3570.w4 | 8 |
| $\frac{34}{33}$ | 306306.cv4 | 306306.cv2 | 306306.cv3 | 306306.cv1 | | 306306.cv3 | 4 |
| $\frac{31}{30}$ | 231570.a4 | 231570.a2 | 231570.a3 | 231570.a1 | | 231570.a3 | 4 |
| $\frac{23}{22}$ | 93610.a4 | 93610.a2 | 93610.a3 | 93610.a1 | | 93610.a3 | 4 |
| $\frac{22}{21}$ | 81774.bv4 | 81774.bv2 | 81774.bv3 | 81774.bv1 | | 81774.bv3 | 4 |

| $\varphi$ | $\mathcal{E}_S(\tau)$ | $\mathcal{E}_S(\tau/2)$ | $\mathcal{E}_S(\tau/3)$ | $\mathcal{E}_S(\tau/6)$ | $p$ | form label | # |
|---|---|---|---|---|---|---|---|
| | continued | | | | | | |
| $\frac{81}{77}$ | 150612.j3 | 150612.j4 | 150612.j1 | 150612.j2 | | 150612.j1 | 4 |
| $\frac{16}{15}$ | 3870.l4 | 3870.l2 | 3870.l3 | 3870.l1 | | 3870.l3 | 4 |
| $\frac{15}{14}$ | 2310.l5 | 2310.l2 | 2310.l3 | 2310.l1 | | 2310.l3 | 8 |
| $\frac{12}{11}$ | 6402.r4 | 6402.r2 | 6402.r3 | 6402.r1 | | 6402.r3 | 4 |
| $\frac{23}{21}$ | 179676.g4 | 179676.g2 | 179676.g3 | 179676.g1 | | 179676.g3 | 4 |
| $\frac{11}{10}$ | 9790.b4 | 9790.b2 | 9790.b3 | 9790.b1 | | 9790.b3 | 4 |
| $\frac{39}{35}$ | 431340.bd3 | 431340.bd4 | 431340.bd1 | 431340.bd2 | | 431340.bd1 | 4 |
| $\frac{37}{33}$ | 73260.n3 | 73260.n4 | 73260.n1 | 73260.n2 | | 73260.n1 | 4 |
| $\frac{17}{15}$ | 70380.t4 | 70380.t2 | 70380.t3 | 70380.t1 | | 70380.t3 | 4 |
| $\frac{25}{22}$ | 66990.bq4 | 66990.bq2 | 66990.bq3 | 66990.bq1 | | 66990.bq3 | 4 |
| $\frac{8}{7}$ | 910.g6 | 910.g4 | 910.g5 | 910.g3 | | 910.g5 | 6 |
| $\frac{63}{55}$ | 2310.h5 | 2310.h6 | 2310.h2 | 2310.h4 | | 2310.h2 | 8 |
| $\frac{121}{105}$ | 284130.du3 | 284130.du4 | 284130.du1 | 284130.du2 | | 284130.du1 | 4 |
| $\frac{64}{55}$ | 171930.r4 | 171930.r2 | 171930.r3 | 171930.r1 | | 171930.r3 | 4 |
| $\frac{7}{6}$ | 2394.d4 | 2394.d2 | 2394.d3 | 2394.d1 | | 2394.d3 | 4 |
| $\frac{13}{11}$ | 30316.b4 | 30316.b2 | 30316.b3 | 30316.b1 | | 30316.b3 | 4 |
| $\frac{25}{21}$ | 21420.v3 | 21420.v4 | 21420.v1 | 21420.v2 | | 21420.v1 | 4 |
| $\frac{6}{5}$ | 210.d5 | 210.d2 | 210.d3 | 210.d1 | | 210.d3 | 8 |
| $\frac{17}{14}$ | 99246.m4 | 99246.m2 | 99246.m3 | 99246.m1 | | 99246.m3 | 4 |
| $\frac{27}{22}$ | 72930.bw4 | 72930.bw2 | 72930.bw3 | 72930.bw1 | | 72930.bw3 | 4 |
| $\frac{43}{35}$ | 33110.b3 | 33110.b4 | 33110.b1 | 33110.b2 | | 33110.b1 | 4 |
| $\frac{41}{33}$ | 56826.h3 | 56826.h4 | 56826.h1 | 56826.h2 | | 56826.h1 | 4 |
| $\frac{44}{35}$ | 43890.ct6 | 43890.ct3 | 43890.ct4 | 43890.ct2 | | 43890.ct4 | 8 |
| $\frac{19}{15}$ | 44460.c3 | 44460.c4 | 44460.c1 | 44460.c2 | | 44460.c1 | 4 |
| $\frac{14}{11}$ | 53130.ci4 | 53130.ci2 | 53130.ci3 | 53130.ci1 | | 53130.ci3 | 4 |
| $\frac{9}{7}$ | 3108.g4 | 3108.g2 | 3108.g3 | 3108.g1 | | 3108.g3 | 4 |
| $\frac{13}{10}$ | 41730.k4 | 41730.k2 | 41730.k3 | 41730.k1 | | 41730.k3 | 4 |
| $\frac{4}{3}$ | 198.d4 | 198.d2 | 198.d3 | 198.d1 | | 198.d3 | 4 |
| $\frac{19}{14}$ | 208810.b4 | 208810.b2 | 208810.b3 | 208810.b1 | | 208810.b3 | 4 |
| $\frac{15}{11}$ | 20460.m3 | 20460.m4 | 20460.m1 | 20460.m2 | | 20460.m1 | 4 |
| $\frac{29}{21}$ | 18270.h3 | 18270.h4 | 18270.h1 | 18270.h2 | | 18270.h1 | 4 |
| $\frac{7}{5}$ | 4060.b4 | 4060.b2 | 4060.b3 | 4060.b1 | | 4060.b3 | 4 |
| $\frac{10}{7}$ | 17430.bp4 | 17430.bp2 | 17430.bp3 | 17430.bp1 | | 17430.bp3 | 4 |
| $\frac{16}{11}$ | 14630.o4 | 14630.o3 | 14630.o2 | 14630.o1 | | 14630.o2 | 4 |

| $\varphi$ | $\mathcal{E}_S(\tau)$ | $\mathcal{E}_S(\tau/2)$ | $\mathcal{E}_S(\tau/3)$ | $\mathcal{E}_S(\tau/6)$ | $p$ | form label | # |
|---|---|---|---|---|---|---|---|
| continued | | | | | | | |
| $\frac{51}{35}$ | 189210.cm3 | 189210.cm4 | 189210.cm1 | 189210.cm2 | | 189210.cm1 | 4 |
| $\frac{22}{15}$ | 422730.dc4 | 422730.dc3 | 422730.dc2 | 422730.dc1 | | 422730.dc2 | 4 |
| $\frac{49}{33}$ | 23562.bc3 | 23562.bc4 | 23562.bc1 | 23562.bc2 | | 23562.bc1 | 4 |
| $\frac{3}{2}$ | 30.a6 | 30.a4 | 30.a3 | 30.a2 | | 30.a3 | 8 |
| $\frac{32}{21}$ | 123354.bu4 | 123354.bu3 | 123354.bu2 | 123354.bu1 | | 123354.bu2 | 4 |
| $\frac{23}{15}$ | 2070.a3 | 2070.a4 | 2070.a1 | 2070.a2 | | 2070.a1 | 4 |
| $\frac{17}{11}$ | 159324.h4 | 159324.h3 | 159324.h2 | 159324.h1 | | 159324.h2 | 4 |
| $\frac{11}{7}$ | 7084.c3 | 7084.c4 | 7084.c1 | 7084.c2 | | 7084.c1 | 4 |
| $\frac{8}{5}$ | 2010.j4 | 2010.j3 | 2010.j2 | 2010.j1 | | 2010.j2 | 4 |
| $\frac{169}{105}$ | 483210.ci3 | 483210.ci4 | 483210.ci1 | 483210.ci2 | | 483210.ci1 | 4 |
| $\frac{125}{77}$ | 302610.ch3 | 302610.ch4 | 302610.ch1 | 302610.ch2 | | 302610.ch1 | 4 |
| $\frac{18}{11}$ | 69762.bg4 | 69762.bg3 | 69762.bg2 | 69762.bg1 | | 69762.bg2 | 4 |
| $\frac{23}{14}$ | 186438.j4 | 186438.j3 | 186438.j2 | 186438.j1 | | 186438.j2 | 4 |
| $\frac{5}{3}$ | 1260.j4 | 1260.j3 | 1260.j2 | 1260.j1 | | 1260.j2 | 4 |
| $\frac{59}{35}$ | 384090.z3 | 384090.z4 | 384090.z1 | 384090.z2 | | 384090.z1 | 4 |
| $\frac{17}{10}$ | 170170.c4 | 170170.c3 | 170170.c2 | 170170.c1 | | 170170.c2 | 4 |
| $\frac{12}{7}$ | 21210.bd4 | 21210.bd3 | 21210.bd2 | 21210.bd1 | | 21210.bd2 | 4 |
| $\frac{19}{11}$ | 2090.c3 | 2090.c4 | 2090.c1 | 2090.c2 | | 2090.c1 | 4 |
| $\frac{37}{21}$ | 60606.z3 | 60606.z4 | 60606.z1 | 60606.z2 | | 60606.z1 | 4 |
| $\frac{25}{14}$ | 162470.b4 | 162470.b3 | 162470.b2 | 162470.b1 | | 162470.b2 | 4 |
| $\frac{9}{5}$ | 1140.d3 | 1140.d4 | 1140.d1 | 1140.d2 | | 1140.d1 | 4 |
| $\frac{20}{11}$ | 4290.bb6 | 4290.bb5 | 4290.bb2 | 4290.bb1 | | 4290.bb2 | 8 |
| $\frac{11}{6}$ | 30690.r4 | 30690.r3 | 30690.r2 | 30690.r1 | | 30690.r2 | 4 |
| $\frac{13}{7}$ | 60060.bc4 | 60060.bc3 | 60060.bc2 | 60060.bc1 | | 60060.bc2 | 4 |
| $\frac{19}{10}$ | 91770.x4 | 91770.x3 | 91770.x2 | 91770.x1 | | 91770.x2 | 4 |
| $\frac{21}{11}$ | 411180.n4 | 411180.n3 | 411180.n2 | 411180.n1 | | 411180.n2 | 4 |
| $\frac{67}{35}$ | 332990.l3 | 332990.l4 | 332990.l1 | 332990.l2 | | 332990.l1 | 4 |
| $\frac{27}{14}$ | 125034.i4 | 125034.i3 | 125034.i2 | 125034.i1 | | 125034.i2 | 4 |
| $\frac{65}{33}$ | 296010.dq3 | 296010.dq4 | 296010.dq1 | 296010.dq2 | | 296010.dq1 | 4 |
| $2$ | 34.a4 | 34.a3 | 34.a2 | 34.a1 | | 34.a2 | 4 |
| $\frac{31}{15}$ | 30690.bc3 | 30690.bc4 | 30690.bc1 | 30690.bc2 | | 30690.bc1 | 4 |
| $\frac{23}{11}$ | 21252.i3 | 21252.i4 | 21252.i1 | 21252.i2 | | 21252.i1 | 4 |
| $\frac{44}{21}$ | 159390.dr4 | 159390.dr3 | 159390.dr2 | 159390.dr1 | | 159390.dr2 | 4 |
| $\frac{21}{10}$ | 413490.bg4 | 413490.bg3 | 413490.bg2 | 413490.bg1 | | 413490.bg2 | 4 |

| $\varphi$ | $\mathcal{E}_S(\tau)$ | $\mathcal{E}_S(\tau/2)$ | $\mathcal{E}_S(\tau/3)$ | $\mathcal{E}_S(\tau/6)$ | $p$ | form label | # |
|---|---|---|---|---|---|---|---|
| continued | | | | | | | |
| $\frac{32}{15}$ | **139230.cv4** | **139230.cv3** | **139230.cv2** | **139230.cv1** | | **139230.cv2** | 4 |
| $\frac{15}{7}$ | **210.b6** | **210.b7** | **210.b2** | **210.b5** | | **210.b2** | 8 |
| $\frac{13}{6}$ | **60606.k4** | **60606.k3** | **60606.k2** | **60606.k1** | | **60606.k2** | 4 |
| $\frac{24}{11}$ | **175890.bx4** | **175890.bx3** | **175890.bx2** | **175890.bx1** | | **175890.bx2** | 4 |
| $\frac{11}{5}$ | **31020.k4** | **31020.k3** | **31020.k2** | **31020.k1** | | **31020.k2** | 4 |
| $\frac{49}{22}$ | **193578.i4** | **193578.i3** | **193578.i2** | **193578.i1** | | **193578.i2** | 4 |
| $\frac{25}{11}$ | **164780.a4** | **164780.a3** | **164780.a2** | **164780.a1** | | **164780.a2** | 4 |
| $\frac{16}{7}$ | **5754.i4** | **5754.i3** | **5754.i2** | **5754.i1** | | **5754.i2** | 4 |
| $\frac{7}{3}$ | **1260.d3** | **1260.d4** | **1260.d1** | **1260.d2** | | **1260.d1** | 4 |
| $\frac{12}{5}$ | **21630.bb4** | **21630.bb3** | **21630.bb2** | **21630.bb1** | | **21630.bb2** | 4 |
| $\frac{17}{7}$ | **173740.b4** | **173740.b3** | **173740.b2** | **173740.b1** | | **173740.b2** | 4 |
| $\frac{27}{11}$ | **1914.n3** | **1914.n4** | **1914.n1** | **1914.n2** | | **1914.n1** | 4 |
| $\frac{5}{2}$ | **1290.h4** | **1290.h3** | **1290.h2** | **1290.h1** | | **1290.h2** | 4 |
| $\frac{53}{21}$ | **126882.bk3** | **126882.bk4** | **126882.bk1** | **126882.bk2** | | **126882.bk1** | 4 |
| $\frac{18}{7}$ | **71610.bs4** | **71610.bs3** | **71610.bs2** | **71610.bs1** | | **71610.bs2** | 4 |
| $\frac{13}{5}$ | **910.a3** | **910.a4** | **910.a1** | **910.a2** | | **910.a1** | 4 |
| $\frac{29}{11}$ | **19140.f4** | **19140.f3** | **19140.f2** | **19140.f1** | | **19140.f2** | 4 |
| $\frac{8}{3}$ | **2070.p4** | **2070.p3** | **2070.p2** | **2070.p1** | | **2070.p2** | 4 |
| $\frac{89}{33}$ | **123354.m3** | **123354.m4** | **123354.m1** | **123354.m2** | | **123354.m1** | 4 |
| $\frac{27}{10}$ | **118830.f4** | **118830.f3** | **118830.f2** | **118830.f1** | | **118830.f2** | 4 |
| $\frac{19}{7}$ | **65436.j3** | **65436.j4** | **65436.j1** | **65436.j2** | | **65436.j1** | 4 |
| $\frac{14}{5}$ | **2310.u7** | **2310.u4** | **2310.u3** | **2310.u2** | | **2310.u3** | 8 |
| $\frac{31}{11}$ | **456940.b3** | **456940.b4** | **456940.b1** | **456940.b2** | | **456940.b1** | 4 |
| $\frac{99}{35}$ | **247170.dj3** | **247170.dj4** | **247170.dj1** | **247170.dj2** | | **247170.dj1** | 4 |
| $\frac{17}{6}$ | **23562.h4** | **23562.h3** | **23562.h2** | **23562.h1** | | **23562.h2** | 4 |
| $\frac{20}{7}$ | **157430.n4** | **157430.n3** | **157430.n2** | **157430.n1** | | **157430.n2** | 4 |
| $\frac{61}{21}$ | **422730.bw3** | **422730.bw4** | **422730.bw1** | **422730.bw2** | | **422730.bw1** | 4 |
| $\frac{32}{11}$ | **127974.y4** | **127974.y3** | **127974.y2** | **127974.y1** | | **127974.y2** | 4 |
| 3 | **156.b4** | **156.b3** | **156.b2** | **156.b1** | | **156.b2** | 4 |
| $\frac{47}{15}$ | **71910.x3** | **71910.x4** | **71910.x1** | **71910.x2** | | **71910.x1** | 4 |
| $\frac{22}{7}$ | **441210.cn4** | **441210.cn3** | **441210.cn2** | **441210.cn1** | | **441210.cn2** | 4 |
| $\frac{19}{6}$ | **244530.c4** | **244530.c3** | **244530.c2** | **244530.c1** | | **244530.c2** | 4 |
| $\frac{35}{11}$ | **43890.bj3** | **43890.bj4** | **43890.bj1** | **43890.bj2** | | **43890.bj1** | 4 |
| $\frac{16}{5}$ | **15290.f4** | **15290.f3** | **15290.f2** | **15290.f1** | | **15290.f2** | 4 |

| $\varphi$ | $\mathcal{E}_S(\tau)$ | $\mathcal{E}_S(\tau/2)$ | $\mathcal{E}_S(\tau/3)$ | $\mathcal{E}_S(\tau/6)$ | $p$ | form label | # |
|---|---|---|---|---|---|---|---|
| continued | | | | | | | |
| $\frac{36}{11}$ | 103290.s4 | 103290.s3 | 103290.s2 | 103290.s1 | | 103290.s2 | 4 |
| $\frac{23}{7}$ | 1610.b3 | 1610.b4 | 1610.b1 | 1610.b2 | | 1610.b1 | 4 |
| $\frac{10}{3}$ | 18270.bz4 | 18270.bz3 | 18270.bz2 | 18270.bz1 | | 18270.bz2 | 4 |
| $\frac{17}{5}$ | 37740.f3 | 37740.f4 | 37740.f1 | 37740.f2 | | 37740.f1 | 4 |
| $\frac{24}{7}$ | 149226.bs4 | 149226.bs3 | 149226.bs2 | 149226.bs1 | | 149226.bs2 | 4 |
| $\frac{38}{11}$ | 415074.p4 | 415074.p3 | 415074.p2 | 415074.p1 | | 415074.p2 | 4 |
| $\frac{7}{2}$ | 4270.c4 | 4270.c3 | 4270.c2 | 4270.c1 | | 4270.c2 | 4 |
| $\frac{25}{7}$ | 45780.t4 | 45780.t3 | 45780.t2 | 45780.t1 | | 45780.t2 | 4 |
| $\frac{18}{5}$ | 61230.z4 | 61230.z3 | 61230.z2 | 61230.z1 | | 61230.z2 | 4 |
| $\frac{11}{3}$ | 198.b3 | 198.b4 | 198.b1 | 198.b2 | | 198.b1 | 4 |
| $\frac{37}{10}$ | 358530.i4 | 358530.i3 | 358530.i2 | 358530.i1 | | 358530.i2 | 4 |
| $\frac{19}{5}$ | 220780.b4 | 220780.b3 | 220780.b2 | 220780.b1 | | 220780.b2 | 4 |
| $\frac{23}{6}$ | 471546.h4 | 471546.h3 | 471546.h2 | 471546.h1 | | 471546.h2 | 4 |
| $\frac{27}{7}$ | 24780.p3 | 24780.p4 | 24780.p1 | 24780.p2 | | 24780.p1 | 4 |
| $\frac{43}{11}$ | 44462.g3 | 44462.g4 | 44462.g1 | 44462.g2 | | 44462.g1 | 4 |
| $4$ | 210.d7 | 210.d5 | 210.d4 | 210.d3 | | 210.d4 | 8 |
| $\frac{85}{21}$ | 332010.ec3 | 332010.ec4 | 332010.ec1 | 332010.ec2 | | 332010.ec1 | 4 |
| $\frac{25}{6}$ | 124830.bd4 | 124830.bd3 | 124830.bd2 | 124830.bd1 | | 124830.bd2 | 4 |
| $\frac{21}{5}$ | 4830.bf3 | 4830.bf4 | 4830.bf1 | 4830.bf2 | | 4830.bf1 | 4 |
| $\frac{64}{15}$ | 117810.cz4 | 117810.cz3 | 117810.cz2 | 117810.cz1 | | 117810.cz2 | 4 |

## B.3. Real quadratic points and Hilbert modular forms

Table 12: *Some values of the modulus $\varphi$ which lie in various real quadratic field extensions $\mathbb{Q}(\sqrt{n})$ together with the corresponding Hilbert modular forms and the twisted Sen curves $\mathcal{E}_S$. The second to fifth columns labelled $\mathcal{E}_S(\tau/n)$ give the LMFDB labels (without the number field label) of the twisted Sen curves, and the penultimate column lists the label (again without the number field label) of the corresponding modular form. The column labelled "$N(\mathfrak{p})$" lists the ideal norms of the prime ideals at which the modular form coefficient $\alpha_{\mathfrak{p}}$ is not equal to the zeta function coefficient $c_p$ in the sense discussed in §4.6.*

| $\varphi$ | $\mathcal{E}_S(\tau)$ | $\mathcal{E}_S(\tau/2)$ | $\mathcal{E}_S(\tau/3)$ | $\mathcal{E}_S(\tau/6)$ | $N(\mathfrak{p})$ | form labels |
|---|---|---|---|---|---|---|
| $-10-7\sqrt{2}$ | 2254.2-f4 | 2254.2-f1 | 2254.2-f3 | 2254.2-f2 | 23 | 2254.1-f, 2254.2-f |
| $-10+7\sqrt{2}$ | 2254.1-f3 | 2254.1-f2 | 2254.1-f4 | 2254.1-f1 | 23 | 2254.1-f, 2254.2-f |
| $-7-8\sqrt{2}$ | 1106.3-a1 | 1106.3-a3 | 1106.3-a4 | 1106.3-a2 | | 1106.2-a, 1106.3-a |
| $-7-6\sqrt{2}$ | 4991.5-e2 | 4991.5-e4 | 4991.5-e3 | 4991.5-e1 | | 4991.4-e, 4991.5-e |
| $-7-5\sqrt{2}$ | 644.1-d3 | 644.1-d1 | 644.1-d2 | 644.1-d4 | 7 | 644.1-d, 644.4-d |
| $-7-4\sqrt{2}$ | 3196.2-a3 | 3196.2-a2 | 3196.2-a4 | 3196.2-a1 | | 3196.2-a, 3196.3-a |
| $-7+4\sqrt{2}$ | 3196.3-a3 | 3196.3-a2 | 3196.3-a4 | 3196.3-a1 | | 3196.2-a, 3196.3-a |
| $-7+5\sqrt{2}$ | 644.4-d3 | 644.4-d2 | 644.4-d4 | 644.4-d1 | 7 | 644.1-d, 644.4-d |
| $-7+6\sqrt{2}$ | 4991.4-e3 | 4991.4-e2 | 4991.4-e4 | 4991.4-e1 | | 4991.4-e, 4991.5-e |
| $-7+8\sqrt{2}$ | 1106.2-a3 | 1106.2-a2 | 1106.2-a4 | 1106.2-a1 | | 1106.2-a, 1106.3-a |
| $-5-4\sqrt{2}$ | 3332.2-b1 | 3332.2-b4 | 3332.2-b2 | 3332.2-b3 | | 3332.1-b, 3332.2-b |
| $-5+4\sqrt{2}$ | 3332.1-b3 | 3332.1-b2 | 3332.1-b4 | 3332.1-b1 | | 3332.1-b, 3332.2-b |
| $-4-3\sqrt{2}$ | 1246.4-d2 | 1246.4-d3 | 1246.4-d1 | 1246.4-d4 | 7 | 1246.1-d, 1246.4-d |
| $-4+3\sqrt{2}$ | 1246.1-d3 | 1246.1-d2 | 1246.1-d4 | 1246.1-d1 | 7 | 1246.1-d, 1246.4-d |
| $-3-2\sqrt{2}$ | 17.1-a4 | 17.1-a1 | 17.1-a3 | 17.1-a2 | | 17.1-a, 17.2-a |
| $-3+2\sqrt{2}$ | 17.2-a3 | 17.2-a2 | 17.2-a4 | 17.2-a1 | | 17.1-a, 17.2-a |
| $-2-2\sqrt{2}$ | 574.2-d1 | 574.2-d4 | 574.2-d2 | 574.2-d3 | | 574.2-d, 574.3-d |
| $-2-\sqrt{2}$ | 2786.1-d3 | 2786.1-d2 | 2786.1-d4 | 2786.1-d1 | 7 | 2786.1-d, 2786.4-d |
| $-2+\sqrt{2}$ | 2786.4-d3 | 2786.4-d2 | 2786.4-d4 | 2786.4-d1 | 7 | 2786.1-d, 2786.4-d |
| $-2+2\sqrt{2}$ | 574.3-d3 | 574.3-d2 | 574.3-d4 | 574.3-d1 | | 574.2-d, 574.3-d |
| $-1-2\sqrt{2}$ | 3836.3-d2 | 3836.3-d4 | 3836.3-d3 | 3836.3-d1 | | 3836.2-d, 3836.3-d |
| $-1-\sqrt{2}$ | 124.2-b1 | 124.2-b4 | 124.2-b2 | 124.2-b3 | | 124.1-b, 124.2-b |
| $-1+\sqrt{2}$ | 124.1-b3 | 124.1-b2 | 124.1-b4 | 124.1-b1 | | 124.1-b, 124.2-b |
| $-1+2\sqrt{2}$ | 3836.2-d3 | 3836.2-d2 | 3836.2-d4 | 3836.2-d1 | | 3836.2-d, 3836.3-d |
| $-\sqrt{2}$ | 322.4-c2 | 322.4-c4 | 322.4-c3 | 322.4-c1 | | 322.1-c, 322.4-c |
| $\sqrt{2}$ | 322.1-c3 | 322.1-c2 | 322.1-c4 | 322.1-c1 | | 322.1-c, 322.4-c |
| $1-2\sqrt{2}$ | 511.3-e1 | 511.3-e3 | 511.3-e4 | 511.3-e2 | | 511.2-e, 511.3-e |
| $1-\sqrt{2}$ | 28.2-a3 | 28.2-a6 | 28.2-a7 | 28.2-a4 | | 28.1-a, 28.2-a |
| $1+\sqrt{2}$ | 28.1-a5 | 28.1-a2 | 28.1-a6 | 28.1-a1 | | 28.1-a, 28.2-a |
| $1+2\sqrt{2}$ | 511.2-e3 | 511.2-e2 | 511.2-e4 | 511.2-e1 | | 511.2-e, 511.3-e |
| $2-\sqrt{2}$ | 254.1-a1 | 254.1-a3 | 254.1-a4 | 254.1-a2 | | 254.1-a, 254.2-a |

| $\varphi$ | $\mathcal{E}_S(\tau)$ | $\mathcal{E}_S(\tau/2)$ | $\mathcal{E}_S(\tau/3)$ | $\mathcal{E}_S(\tau/6)$ | $N(\mathfrak{p})$ | form labels |
|---|---|---|---|---|---|---|
| $2+\sqrt{2}$ | 254.2-a1 | 254.2-a4 | 254.2-a2 | 254.2-a3 | | 254.1-a, 254.2-a |
| $3-2\sqrt{2}$ | 28.1-a4 | 28.1-a5 | 28.1-a8 | 28.1-a6 | | 28.1-a, 28.2-a |
| $3+2\sqrt{2}$ | 28.2-a2 | 28.2-a3 | 28.2-a1 | 28.2-a7 | | 28.1-a, 28.2-a |
| $4-3\sqrt{2}$ | 4194.1-b1 | 4194.1-b2 | 4194.1-b4 | 4194.1-b3 | | 4194.1-b, 4194.2-b |
| $4-2\sqrt{2}$ | 1154.1-a1 | 1154.1-a2 | 1154.1-a4 | 1154.1-a3 | | 1154.1-a, 1154.2-a |
| $4+2\sqrt{2}$ | 1154.2-a3 | 1154.2-a4 | 1154.2-a2 | 1154.2-a1 | | 1154.1-a, 1154.2-a |
| $4+3\sqrt{2}$ | 4194.2-b4 | 4194.2-b1 | 4194.2-b3 | 4194.2-b2 | | 4194.1-b, 4194.2-b |
| $5-4\sqrt{2}$ | 1148.2-b2 | 1148.2-b1 | 1148.2-b4 | 1148.2-b3 | | 1148.2-b, 1148.3-b |
| $5-2\sqrt{2}$ | 2737.4-e1 | 2737.4-e2 | 2737.4-e4 | 2737.4-e3 | | 2737.4-e, 2737.5-e |
| $5+2\sqrt{2}$ | 2737.5-e1 | 2737.5-e3 | 2737.5-e4 | 2737.5-e2 | | 2737.4-e, 2737.5-e |
| $5+4\sqrt{2}$ | 1148.3-b3 | 1148.3-b2 | 1148.3-b4 | 1148.3-b1 | | 1148.2-b, 1148.3-b |
| $6-4\sqrt{2}$ | 3038.2-d5 | 3038.2-d4 | 3038.2-d8 | 3038.2-d6 | 7 | 3038.1-d, 3038.2-d |
| $6+4\sqrt{2}$ | 3038.1-d1 | 3038.1-d6 | 3038.1-d2 | 3038.1-d5 | 7 | 3038.1-d, 3038.2-d |
| $7-5\sqrt{2}$ | 2884.2-a2 | 2884.2-a1 | 2884.2-a4 | 2884.2-a3 | 7 | 2884.2-a, 2884.3-a |
| $7+5\sqrt{2}$ | 2884.3-a4 | 2884.3-a1 | 2884.3-a3 | 2884.3-a2 | 7 | 2884.2-a, 2884.3-a |
| $9-8\sqrt{2}$ | 2914.3-f2 | 2914.3-f1 | 2914.3-f4 | 2914.3-f3 | | 2914.2-f, 2914.3-f |
| $9-6\sqrt{2}$ | 639.1-b2 | 639.1-b1 | 639.1-b4 | 639.1-b3 | | 639.1-b, 639.2-b |
| $9-4\sqrt{2}$ | 3332.1-d5 | 3332.1-d4 | 3332.1-d8 | 3332.1-d6 | | 3332.1-d, 3332.2-d |
| $9+4\sqrt{2}$ | 3332.2-d3 | 3332.2-d6 | 3332.2-d7 | 3332.2-d5 | | 3332.1-d, 3332.2-d |
| $9+6\sqrt{2}$ | 639.2-b1 | 639.2-b4 | 639.2-b2 | 639.2-b3 | | 639.1-b, 639.2-b |
| $9+8\sqrt{2}$ | 2914.2-f3 | 2914.2-f2 | 2914.2-f4 | 2914.2-f1 | | 2914.2-f, 2914.3-f |
| $10-7\sqrt{2}$ | 578.1-b2 | 578.1-b1 | 578.1-b4 | 578.1-b3 | 17 | 578.1-b, 578.1-f |
| $10+7\sqrt{2}$ | 578.1-f2 | 578.1-f1 | 578.1-f3 | 578.1-f4 | 17 | 578.1-b, 578.1-f |
| $-9-6\sqrt{3}$ | 759.4-e2 | 759.4-e4 | 759.4-e3 | 759.4-e1 | | 759.1-e, 759.4-e |
| $-9+6\sqrt{3}$ | 759.1-e3 | 759.1-e2 | 759.1-e4 | 759.1-e1 | | 759.1-e, 759.4-e |
| $-7-4\sqrt{3}$ | 52.1-b4 | 52.1-b1 | 52.1-b3 | 52.1-b2 | | 52.1-b, 52.2-b |
| $-7+4\sqrt{3}$ | 52.2-b3 | 52.2-b2 | 52.2-b4 | 52.2-b1 | | 52.1-b, 52.2-b |
| $-5-3\sqrt{3}$ | 426.2-a2 | 426.2-a3 | 426.2-a1 | 426.2-a4 | | 426.1-a, 426.2-a |
| $-5+3\sqrt{3}$ | 426.1-a3 | 426.1-a2 | 426.1-a4 | 426.1-a1 | | 426.1-a, 426.2-a |
| $-3-2\sqrt{3}$ | 564.2-a1 | 564.2-a4 | 564.2-a2 | 564.2-a3 | | 564.1-a, 564.2-a |
| $-3+2\sqrt{3}$ | 564.1-a3 | 564.1-a2 | 564.1-a4 | 564.1-a1 | | 564.1-a, 564.2-a |
| $-2-2\sqrt{3}$ | 3666.3-m2 | 3666.3-m4 | 3666.3-m3 | 3666.3-m1 | | 3666.2-m, 3666.3-m |
| $-2-\sqrt{3}$ | 708.2-d4 | 708.2-d1 | 708.2-d3 | 708.2-d2 | | 708.1-d, 708.2-d |
| $-2+\sqrt{3}$ | 708.1-d3 | 708.1-d2 | 708.1-d4 | 708.1-d1 | | 708.1-d, 708.2-d |
| $-2+2\sqrt{3}$ | 3666.2-m3 | 3666.2-m2 | 3666.2-m4 | 3666.2-m1 | | 3666.2-m, 3666.3-m |
| $-1-2\sqrt{3}$ | 1199.2-e1 | 1199.2-e3 | 1199.2-e4 | 1199.2-e2 | | 1199.2-e, 1199.3-e |

| $\varphi$ | $\mathcal{E}_S(\tau)$ | $\mathcal{E}_S(\tau/2)$ | $\mathcal{E}_S(\tau/3)$ | $\mathcal{E}_S(\tau/6)$ | $N(\mathfrak{p})$ | form labels |
|---|---|---|---|---|---|---|
| $-1-\sqrt{3}$ | 286.4-b1 | 286.4-b4 | 286.4-b2 | 286.4-b3 | | 286.1-b, 286.4-b |
| $-1+\sqrt{3}$ | 286.1-b3 | 286.1-b2 | 286.1-b4 | 286.1-b1 | | 286.1-b, 286.4-b |
| $-1+2\sqrt{3}$ | 1199.3-e3 | 1199.3-e2 | 1199.3-e4 | 1199.3-e1 | | 1199.2-e, 1199.3-e |
| $-\sqrt{3}$ | 132.1-b4 | 132.1-b7 | 132.1-b6 | 132.1-b3 | | 132.1-b, 132.2-b |
| $\sqrt{3}$ | 132.2-b3 | 132.2-b2 | 132.2-b7 | 132.2-b1 | | 132.1-b, 132.2-b |
| $1-\sqrt{3}$ | 1074.1-c1 | 1074.1-c3 | 1074.1-c4 | 1074.1-c2 | | 1074.1-c, 1074.2-c |
| $1+\sqrt{3}$ | 1074.2-c3 | 1074.2-c2 | 1074.2-c4 | 1074.2-c1 | | 1074.1-c, 1074.2-c |
| $2-\sqrt{3}$ | 92.1-b1 | 92.1-b3 | 92.1-b4 | 92.1-b2 | | 92.1-b, 92.2-b |
| $2+\sqrt{3}$ | 92.2-b2 | 92.2-b3 | 92.2-b1 | 92.2-b4 | | 92.1-b, 92.2-b |
| $3-2\sqrt{3}$ | 111.2-a1 | 111.2-a2 | 111.2-a4 | 111.2-a3 | | 111.1-a, 111.2-a |
| $3-\sqrt{3}$ | 2598.1-d1 | 2598.1-d2 | 2598.1-d4 | 2598.1-d3 | | 2598.1-d, 2598.2-d |
| $3+\sqrt{3}$ | 2598.2-d2 | 2598.2-d4 | 2598.2-d3 | 2598.2-d1 | | 2598.1-d, 2598.2-d |
| $3+2\sqrt{3}$ | 111.1-a3 | 111.1-a2 | 111.1-a4 | 111.1-a1 | | 111.1-a, 111.2-a |
| $4-2\sqrt{3}$ | 1518.3-e4 | 1518.3-e5 | 1518.3-e8 | 1518.3-e6 | | 1518.2-e, 1518.3-e |
| $4+2\sqrt{3}$ | 1518.2-e1 | 1518.2-e6 | 1518.2-e2 | 1518.2-e4 | | 1518.2-e, 1518.3-e |
| $6-3\sqrt{3}$ | 3732.1-f2 | 3732.1-f1 | 3732.1-f4 | 3732.1-f3 | | 3732.1-f, 3732.2-f |
| $6+3\sqrt{3}$ | 3732.2-f1 | 3732.2-f4 | 3732.2-f2 | 3732.2-f3 | | 3732.1-f, 3732.2-f |
| $7-4\sqrt{3}$ | 132.1-b5 | 132.1-b4 | 132.1-b8 | 132.1-b6 | | 132.1-b, 132.2-b |
| $7+4\sqrt{3}$ | 132.2-b6 | 132.2-b3 | 132.2-b4 | 132.2-b7 | | 132.1-b, 132.2-b |
| $8-4\sqrt{3}$ | 2306.2-c2 | 2306.2-c1 | 2306.2-c4 | 2306.2-c3 | | 2306.1-c, 2306.2-c |
| $8+4\sqrt{3}$ | 2306.1-c3 | 2306.1-c4 | 2306.1-c1 | 2306.1-c2 | | 2306.1-c, 2306.2-c |
| $10-6\sqrt{3}$ | 4962.1-a2 | 4962.1-a1 | 4962.1-a4 | 4962.1-a3 | | 4962.1-a, 4962.2-a |
| $10+6\sqrt{3}$ | 4962.2-a3 | 4962.2-a2 | 4962.2-a4 | 4962.2-a1 | | 4962.1-a, 4962.2-a |
| $-9-4\sqrt{5}$ | 4880.1-c4 | 4880.1-c1 | 4880.1-c3 | 4880.1-c2 | | 4880.1-c, 4880.2-c |
| $-9+4\sqrt{5}$ | 4880.2-c3 | 4880.2-c2 | 4880.2-c4 | 4880.2-c1 | | 4880.1-c, 4880.2-c |
| $-8-4\sqrt{5}$ | 4604.1-a1 | 4604.1-a4 | 4604.1-a2 | 4604.1-a3 | | 4604.1-a, 4604.2-a |
| $-8+4\sqrt{5}$ | 4604.2-a3 | 4604.2-a2 | 4604.2-a4 | 4604.2-a1 | | 4604.1-a, 4604.2-a |
| $-5-2\sqrt{5}$ | 2480.1-a3 | 2480.1-a2 | 2480.1-a4 | 2480.1-a1 | | 2480.1-a, 2480.2-a |
| $-5+2\sqrt{5}$ | 2480.2-a3 | 2480.2-a2 | 2480.2-a4 | 2480.2-a1 | | 2480.1-a, 2480.2-a |
| $-2-\sqrt{5}$ | 176.2-a2 | 176.2-a3 | 176.2-a1 | 176.2-a4 | | 176.1-a, 176.2-a |
| $-2+\sqrt{5}$ | 176.1-a3 | 176.1-a2 | 176.1-a4 | 176.1-a1 | | 176.1-a, 176.2-a |
| $-1-\sqrt{5}$ | 1220.2-f3 | 1220.2-f4 | 1220.2-f2 | 1220.2-f1 | | 1220.1-f, 1220.2-f |
| $-1+\sqrt{5}$ | 1220.1-f3 | 1220.1-f2 | 1220.1-f4 | 1220.1-f1 | | 1220.1-f, 1220.2-f |
| $2-2\sqrt{5}$ | 836.3-c1 | 836.3-c2 | 836.3-c5 | 836.3-c3 | 19 | 836.2-c, 836.3-c |
| $2-\sqrt{5}$ | 464.2-b1 | 464.2-b3 | 464.2-b4 | 464.2-b2 | | 464.1-b, 464.2-b |
| $2+\sqrt{5}$ | 464.1-b4 | 464.1-b1 | 464.1-b3 | 464.1-b2 | | 464.1-b, 464.2-b |

| *Continued* | | | | | | |
|---|---|---|---|---|---|---|
| $\varphi$ | $\mathcal{E}_S(\tau)$ | $\mathcal{E}_S(\tau/2)$ | $\mathcal{E}_S(\tau/3)$ | $\mathcal{E}_S(\tau/6)$ | $N(\mathfrak{p})$ | form labels |
| $2+2\sqrt{5}$ | **836.2-c4** | **836.2-c3** | **836.2-c6** | **836.2-c1** | 19 | **836.2-c, 836.3-c** |
| $3-\sqrt{5}$ | **1084.1-c1** | **1084.1-c2** | **1084.1-c4** | **1084.1-c3** | | **1084.1-c, 1084.2-c** |
| $3+\sqrt{5}$ | **1084.2-c1** | **1084.2-c4** | **1084.2-c2** | **1084.2-c3** | | **1084.1-c, 1084.2-c** |
| $5-4\sqrt{5}$ | **3905.2-b2** | **3905.2-b1** | **3905.2-b4** | **3905.2-b3** | | **3905.2-b, 3905.3-b** |
| $5+4\sqrt{5}$ | **3905.3-b3** | **3905.3-b2** | **3905.3-b4** | **3905.3-b1** | | **3905.2-b, 3905.3-b** |
| $9-4\sqrt{5}$ | **80.1-a5** | **80.1-a4** | **80.1-a8** | **80.1-a6** | | **80.1-a** |
| $9+4\sqrt{5}$ | **80.1-a7** | **80.1-a4** | **80.1-a3** | **80.1-a6** | | **80.1-a** |
| $-5-2\sqrt{7}$ | **57.1-d2** | **57.1-d3** | **57.1-d1** | **57.1-d4** | | **57.1-d, 57.4-d** |
| $-5+2\sqrt{7}$ | **57.4-d3** | **57.4-d2** | **57.4-d4** | **57.4-d1** | | **57.1-d, 57.4-d** |
| $3-\sqrt{7}$ | **654.2-b1** | **654.2-b2** | **654.2-b4** | **654.2-b3** | | **654.2-b, 654.3-b** |
| $3+\sqrt{7}$ | **654.3-b1** | **654.3-b4** | **654.3-b2** | **654.3-b3** | | **654.2-b, 654.3-b** |
| $5-\sqrt{7}$ | **666.1-h3** | **666.1-h4** | **666.1-h7** | **666.1-h5** | | **666.1-h, 666.2-h** |
| $5+\sqrt{7}$ | **666.2-h3** | **666.2-h6** | **666.2-h7** | **666.2-h4** | | **666.1-h, 666.2-h** |
| $-19-6\sqrt{10}$ | **265.1-d4** | **265.1-d1** | **265.1-d3** | **265.1-d2** | | **265.1-d, 265.2-d** |
| $-19+6\sqrt{10}$ | **265.2-d3** | **265.2-d2** | **265.2-d4** | **265.2-d1** | | **265.1-d, 265.2-d** |
| $-3-\sqrt{10}$ | **156.3-d2** | **156.3-d3** | **156.3-d1** | **156.3-d4** | | **156.2-d, 156.3-d** |
| $-3+\sqrt{10}$ | **156.2-d3** | **156.2-d2** | **156.2-d4** | **156.2-d1** | | **156.2-d, 156.3-d** |
| $13-4\sqrt{10}$ | **372.4-b2** | **372.4-b1** | **372.4-b4** | **372.4-b3** | | **372.1-b, 372.4-b** |
| $13+4\sqrt{10}$ | **372.1-b1** | **372.1-b4** | **372.1-b2** | **372.1-b3** | | **372.1-b, 372.4-b** |

## B.4. Quadratic imaginary points and Bianchi modular forms

Table 13: *Some values of the modulus $\varphi$ which lie in various imaginary quadratic field extensions $\mathbb{Q}(\sqrt{n})$ together with the corresponding Hilbert modular forms and the twisted Sen curves $\mathcal{E}_S$. The second to fifth columns labelled $\mathcal{E}_S(\tau/n)$ give the LMFDB labels (without the number field label) of the twisted Sen curves, and the penultimate column lists the label (again without the number field label) of the corresponding modular form. The column labelled "$N(\mathfrak{p})$" lists the ideal norms of the prime ideals at which the modular form coefficient $\alpha_{\mathfrak{p}}$ is not equal to the zeta function coefficient $c_p$ in the sense discussed in §4.6.*

| $\varphi$ | $\mathcal{E}_S(\tau)$ | $\mathcal{E}_S(\tau/2)$ | $\mathcal{E}_S(\tau/3)$ | $\mathcal{E}_S(\tau/6)$ | $N(\mathfrak{p})$ | form labels |
|---|---|---|---|---|---|---|
| $-7-4i\sqrt{2}$ | **7524.8-c6** | **7524.8-c3** | **7524.8-c1** | **7524.8-c8** | $19, 73$ | **7524.5-c, 7524.8-c** |
| $-7+4i\sqrt{2}$ | **7524.5-c6** | **7524.5-c3** | **7524.5-c1** | **7524.5-c8** | $19, 73$ | **7524.5-c, 7524.8-c** |
| $-4-i\sqrt{2}$ | **27558.3-d4** | **27558.3-d3** | **27558.3-d1** | **27558.3-d2** | $19, 73$ | **27558.3-d, 27558.4-d** |
| $-4+i\sqrt{2}$ | **27558.4-d4** | **27558.4-d3** | **27558.4-d1** | **27558.4-d2** | $19, 73$ | **27558.3-d, 27558.4-d** |
| $-3-2i\sqrt{2}$ | **9129.1-a2** | **9129.1-a3** | **9129.1-a1** | **9129.1-a4** | $19, 73$ | **9129.1-a, 9129.8-a** |
| $-3+2i\sqrt{2}$ | **9129.8-a2** | **9129.8-a3** | **9129.8-a1** | **9129.8-a4** | $19, 73$ | **9129.1-a, 9129.8-a** |

| $\varphi$ | $\mathcal{E}_S(\tau)$ | $\mathcal{E}_S(\tau/2)$ | $\mathcal{E}_S(\tau/3)$ | $\mathcal{E}_S(\tau/6)$ | $N(\mathfrak{p})$ | form labels |
|---|---|---|---|---|---|---|
| | | | *Continued* | | | |
| $-1-2i\sqrt{2}$ | 6732.7-a3 | 6732.7-a5 | 6732.7-a8 | 6732.7-a7 | $19,73$ | 6732.6-a, 6732.7-a |
| $-1-i\sqrt{2}$ | 4716.3-b1 | 4716.3-b2 | 4716.3-b4 | 4716.3-b3 | $19,73$ | 4716.3-b, 4716.4-b |
| $-1+i\sqrt{2}$ | 4716.4-b1 | 4716.4-b2 | 4716.4-b4 | 4716.4-b3 | $19,73$ | 4716.3-b, 4716.4-b |
| $-1+2i\sqrt{2}$ | 6732.6-a3 | 6732.6-a5 | 6732.6-a8 | 6732.6-a7 | $19,73$ | 6732.6-a, 6732.7-a |
| $-2i\sqrt{2}$ | 3894.2-a2 | 3894.2-a3 | 3894.2-a1 | 3894.2-a4 | $19,73$ | 3894.2-a, 3894.7-a |
| $-i\sqrt{2}$ | 978.3-a1 | 978.3-a3 | 978.3-a4 | 978.3-a2 | $19,73$ | 978.2-a, 978.3-a |
| $i\sqrt{2}$ | 978.2-a1 | 978.2-a3 | 978.2-a4 | 978.2-a2 | $19,73$ | 978.2-a, 978.3-a |
| $2i\sqrt{2}$ | 3894.7-a2 | 3894.7-a3 | 3894.7-a1 | 3894.7-a4 | $19,73$ | 3894.2-a, 3894.7-a |
| $1-8i\sqrt{2}$ | 42054.8-b3 | 42054.8-b4 | 42054.8-b1 | 42054.8-b2 | $19,73$ | 42054.1-b, 42054.8-b |
| $1-4i\sqrt{2}$ | 10956.2-a4 | 10956.2-a3 | 10956.2-a1 | 10956.2-a2 | $19,73$ | 10956.2-a, 10956.7-a |
| $1-2i\sqrt{2}$ | 267.3-a4 | 267.3-a2 | 267.3-a1 | 267.3-a3 | $19,73$ | 267.2-a, 267.3-a |
| $1-i\sqrt{2}$ | 1356.1-a2 | 1356.1-a3 | 1356.1-a4 | 1356.1-a1 | $19,73$ | 1356.1-a, 1356.4-a |
| $1+i\sqrt{2}$ | 1356.4-a2 | 1356.4-a3 | 1356.4-a4 | 1356.4-a1 | $19,73$ | 1356.1-a, 1356.4-a |
| $1+2i\sqrt{2}$ | 267.2-a4 | 267.2-a2 | 267.2-a1 | 267.2-a3 | $19,73$ | 267.2-a, 267.3-a |
| $1+4i\sqrt{2}$ | 10956.7-a4 | 10956.7-a3 | 10956.7-a1 | 10956.7-a2 | $19,73$ | 10956.2-a, 10956.7-a |
| $1+8i\sqrt{2}$ | 42054.1-b3 | 42054.1-b4 | 42054.1-b1 | 42054.1-b2 | $19,73$ | 42054.1-b, 42054.8-b |
| $2-2i\sqrt{2}$ | 16866.3-d3 | 16866.3-d1 | 16866.3-d4 | 16866.3-d2 | $19,73$ | 16866.3-d, 16866.4-d |
| $2-i\sqrt{2}$ | 8118.7-g4 | 8118.7-g2 | 8118.7-g1 | 8118.7-g3 | $19,73$ | 8118.6-g, 8118.7-g |
| $2+i\sqrt{2}$ | 8118.6-g4 | 8118.6-g2 | 8118.6-g1 | 8118.6-g3 | $19,73$ | 8118.6-g, 8118.7-g |
| $2+2i\sqrt{2}$ | 16866.4-d3 | 16866.4-d1 | 16866.4-d4 | 16866.4-d2 | $19,73$ | 16866.3-d, 16866.4-d |
| $5-2i\sqrt{2}$ | 31977.16-a3 | 31977.16-a4 | 31977.16-a1 | 31977.16-a2 | $19,73$ | 31977.9-a, 31977.16-a |
| $5-i\sqrt{2}$ | 37764.3-a2 | 37764.3-a3 | 37764.3-a1 | 37764.3-a4 | $19,73$ | 37764.3-a, 37764.4-a |
| $5+i\sqrt{2}$ | 37764.4-a2 | 37764.4-a3 | 37764.4-a1 | 37764.4-a4 | $19,73$ | 37764.3-a, 37764.4-a |
| $5+2i\sqrt{2}$ | 31977.9-a3 | 31977.9-a4 | 31977.9-a1 | 31977.9-a2 | $19,73$ | 31977.9-a, 31977.16-a |
| $-3-5i\sqrt{3}$ | 145236.8-h3 | 145236.8-h4 | 145236.8-h10 | 145236.8-h8 | $7,13$ | 145236.5-h, 145236.8-h |
| $-3-4i\sqrt{3}$ | 4161.1-a1 | 4161.1-a2 | 4161.1-a4 | 4161.1-a3 | | 4161.1-a, 4161.4-a |
| $-3+4i\sqrt{3}$ | 4161.4-a1 | 4161.4-a2 | 4161.4-a4 | 4161.4-a3 | | 4161.1-a, 4161.4-a |
| $-3+5i\sqrt{3}$ | 145236.5-h4 | 145236.5-h5 | 145236.5-h10 | 145236.5-h9 | $7,13$ | 145236.5-h, 145236.8-h |
| $-2-3i\sqrt{3}$ | 135408.7-a1 | 135408.7-a2 | 135408.7-a4 | 135408.7-a3 | | 135408.2-a, 135408.7-a |
| $-2-2i\sqrt{3}$ | 111972.3-d3 | 111972.3-d6 | 111972.3-d8 | 111972.3-d7 | $7$ | 111972.3-d, 111972.6-d |
| $-2-i\sqrt{3}$ | 50736.4-a2 | 50736.4-a3 | 50736.4-a1 | 50736.4-a4 | | 50736.1-a, 50736.4-a |
| $-2+i\sqrt{3}$ | 50736.1-a3 | 50736.1-a4 | 50736.1-a1 | 50736.1-a2 | | 50736.1-a, 50736.4-a |
| $-2+2i\sqrt{3}$ | 111972.6-d4 | 111972.6-d2 | 111972.6-d8 | 111972.6-d7 | $7$ | 111972.3-d, 111972.6-d |
| $-2+3i\sqrt{3}$ | 135408.2-a2 | 135408.2-a1 | 135408.2-a4 | 135408.2-a3 | | 135408.2-a, 135408.7-a |
| $-1-2i\sqrt{3}$ | 13936.2-a1 | 13936.2-a2 | 13936.2-a4 | 13936.2-a3 | | 13936.2-a, 13936.3-a |
| $-1+2i\sqrt{3}$ | 13936.3-a2 | 13936.3-a1 | 13936.3-a4 | 13936.3-a3 | | 13936.2-a, 13936.3-a |

*Continued on the following page*

| $\varphi$ | $\mathcal{E}_S(\tau)$ | $\mathcal{E}_S(\tau/2)$ | $\mathcal{E}_S(\tau/3)$ | $\mathcal{E}_S(\tau/6)$ | $N(\mathfrak{p})$ | form labels |
|---|---|---|---|---|---|---|
| | Continued | | | | | |
| $-i\sqrt{3}$ | 2928.2-a2 | 2928.2-a4 | 2928.2-a1 | 2928.2-a3 | | 2928.1-a, 2928.2-a |
| $i\sqrt{3}$ | 2928.1-a2 | 2928.1-a4 | 2928.1-a1 | 2928.1-a3 | | 2928.1-a, 2928.2-a |
| $1-8i\sqrt{3}$ | 141276.2-b3 | 141276.2-b4 | 141276.2-b1 | 141276.2-b2 | | 141276.2-b, 141276.3-b |
| $1-4i\sqrt{3}$ | 82992.2-b4 | 82992.2-b3 | 82992.2-b1 | 82992.2-b2 | | 82992.2-b, 82992.7-b |
| $1-i\sqrt{3}$ | 3684.2-b2 | 3684.2-b3 | 3684.2-b4 | 3684.2-b1 | | 3684.1-b, 3684.2-b |
| $1+i\sqrt{3}$ | 3684.1-b2 | 3684.1-b3 | 3684.1-b4 | 3684.1-b1 | | 3684.1-b, 3684.2-b |
| $1+4i\sqrt{3}$ | 82992.7-b3 | 82992.7-b2 | 82992.7-b1 | 82992.7-b7 | | 82992.2-b, 82992.7-b |
| $1+8i\sqrt{3}$ | 141276.3-b2 | 141276.3-b3 | 141276.3-b1 | 141276.3-b4 | | 141276.2-b, 141276.3-b |
| $2-2i\sqrt{3}$ | 65572.4-a3 | 65572.4-a2 | 65572.4-a4 | 65572.4-a1 | 13 | 65572.3-a, 65572.4-a |
| $2-i\sqrt{3}$ | 14896.3-a4 | 14896.3-a2 | 14896.3-a1 | 14896.3-a3 | | 14896.3-a, 14896.4-a |
| $2+i\sqrt{3}$ | 14896.4-a3 | 14896.4-a1 | 14896.4-a4 | 14896.4-a2 | | 14896.3-a, 14896.4-a |
| $2+2i\sqrt{3}$ | 65572.3-a3 | 65572.3-a2 | 65572.3-a4 | 65572.3-a1 | 13 | 65572.3-a, 65572.4-a |
| $3-2i\sqrt{3}$ | 34608.4-b3 | 34608.4-b2 | 34608.4-b4 | 34608.4-b1 | | 34608.1-b, 34608.4-b |
| $3-i\sqrt{3}$ | 77196.2-d3 | 77196.2-d2 | 77196.2-d1 | 77196.2-d4 | 7 | 77196.2-d, 77196.3-d |
| $3+i\sqrt{3}$ | 77196.3-d3 | 77196.3-d4 | 77196.3-d1 | 77196.3-d2 | 7 | 77196.2-d, 77196.3-d |
| $3+2i\sqrt{3}$ | 34608.1-b3 | 34608.1-b2 | 34608.1-b4 | 34608.1-b1 | | 34608.1-b, 34608.4-b |
| $4-2i\sqrt{3}$ | 7644.4-c6 | 7644.4-c5 | 7644.4-c1 | 7644.4-c3 | 7 | 7644.3-c, 7644.4-c |
| $4+2i\sqrt{3}$ | 7644.3-c6 | 7644.3-c5 | 7644.3-c1 | 7644.3-c3 | 7 | 7644.3-c, 7644.4-c |
| $5-4i\sqrt{3}$ | 6643.3-a3 | 6643.3-a2 | 6643.3-a4 | 6643.3-a1 | | 6643.3-a, 6643.6-a |
| $5+4i\sqrt{3}$ | 6643.6-a3 | 6643.6-a2 | 6643.6-a4 | 6643.6-a1 | | 6643.3-a, 6643.6-a |
| $9-8i\sqrt{3}$ | 7644.3-c5 | 7644.3-c6 | 7644.3-c10 | 7644.3-c2 | | 7644.3-c, 7644.4-c |
| $9+8i\sqrt{3}$ | 7644.4-c5 | 7644.4-c6 | 7644.4-c10 | 7644.4-c2 | | 7644.3-c, 7644.4-c |
| $-3-4i\sqrt{7}$ | 3388.5-c1 | 3388.5-c3 | 3388.5-c8 | 3388.5-c6 | 67 | 3388.5-c, 3388.5-d |
| $-3+4i\sqrt{7}$ | 3388.5-d1 | 3388.5-d3 | 3388.5-d8 | 3388.5-d6 | 67 | 3388.5-c, 3388.5-d |
| $-2-i\sqrt{7}$ | 2552.10-a1 | 2552.10-a2 | 2552.10-a4 | 2552.10-a3 | 29, 37 | 2552.7-a, 2552.10-a |
| $-2+i\sqrt{7}$ | 2552.7-a1 | 2552.7-a2 | 2552.7-a4 | 2552.7-a3 | 29, 37 | 2552.7-a, 2552.10-a |
| $5-4i\sqrt{7}$ | 23564.8-b2 | 23564.8-b3 | 23564.8-b4 | 23564.8-b1 | 11, 79 | 23564.5-b, 23564.8-b |
| $5+4i\sqrt{7}$ | 23564.5-b2 | 23564.5-b3 | 23564.5-b4 | 23564.5-b1 | 11, 79 | 23564.5-b, 23564.8-b |

## C. Data for the $\mathbb{Z}_2$ Symmetric Mirror Bicubic

### C.1. Zeta function numerators

Table 14: *The numerators of the local zeta functions of non-singular $\mathbb{Z}_2$ symmetric mirror bicubics without apparent singularities for primes $7 \leq p \leq 47$.*

| $\varphi$ | smooth/sing. | singularity | $R(T)$ |
|---|---|---|---|
| | | $p = 7$ | |
| 1 | singular | $-\frac{1}{27}$ | |
| 2 | smooth* | | |
| 3 | smooth* | | |
| 4 | smooth | | $\left(p^3T^2 + 4pT + 1\right)\left(p^6T^4 + 20p^3T^3 + 12pT^2 + 20T + 1\right)$ |
| 5 | smooth | | $\left(p^3T^2 - 2pT + 1\right)\left(p^6T^4 + 5p^4T^3 + 90pT^2 + 5pT + 1\right)$ |
| 6 | singular | $\frac{1}{216}$ | |

| $\varphi$ | smooth/sing. | singularity | $R(T)$ |
|---|---|---|---|
| | | $p = 11$ | |
| 1 | smooth | | $\left(p^3T^2 + 1\right)\left(p^6T^4 + 36p^3T^3 + 170pT^2 + 36T + 1\right)$ |
| 2 | singular | $-\frac{1}{27}$ | |
| 3 | smooth | | $\left(p^3T^2 + 6pT + 1\right)\left(p^6T^4 - 15p^3T^3 + 152pT^2 - 15T + 1\right)$ |
| 4 | smooth | | $\left(p^3T^2 - 3pT + 1\right)\left(p^6T^4 + 21p^3T^3 + 170pT^2 + 21T + 1\right)$ |
| 5 | smooth | | $\left(p^3T^2 + 1\right)\left(p^6T^4 + 45p^3T^3 + 224pT^2 + 45T + 1\right)$ |
| 6 | smooth* | | |
| 7 | smooth | | $\left(p^3T^2 + 3pT + 1\right)\left(p^6T^4 - 3p^3T^3 - 28pT^2 - 3T + 1\right)$ |
| 8 | singular | $\frac{1}{216}$ | |
| 9 | smooth | | $\left(p^3T^2 + 3pT + 1\right)\left(p^6T^4 + 51p^3T^3 + 170pT^2 + 51T + 1\right)$ |
| 10 | smooth* | | |

| $\varphi$ | smooth/sing. | singularity | $R(T)$ |
|---|---|---|---|
| | | $p = 13$ | |
| 1 | smooth | | $\left(p^3T^2 + 4pT + 1\right)\left(p^6T^4 + 23p^3T^3 + 60pT^2 + 23T + 1\right)$ |
| 2 | smooth | | $\left(p^3T^2 - 2pT + 1\right)\left(p^6T^4 + 20p^3T^3 + 192pT^2 + 20T + 1\right)$ |
| 3 | smooth* | | |
| 4 | smooth | | $\left(p^3T^2 + 4pT + 1\right)\left(p^6T^4 + 23p^3T^3 + 330pT^2 + 23T + 1\right)$ |
| 5 | singular | $\frac{1}{216}$ | |
| 6 | smooth | | $\left(p^3T^2 - 2pT + 1\right)\left(p^6T^4 - 4p^4T^3 + 174pT^2 - 4pT + 1\right)$ |
| 7 | smooth* | | |
| | | |  |

| | | | |
|---|---|---|---|
| $p = 13$, *continued* | | | |
| $\varphi$ | smooth/sing. | singularity | $R(T)$ |
| 8 | smooth | | $\left(p^3T^2 + 4pT + 1\right)\left(p^6T^4 + 14p^3T^3 - 138pT^2 + 14T + 1\right)$ |
| 9 | smooth | | $\left(p^3T^2 + pT + 1\right)\left(p^6T^4 + 17p^3T^3 + 144pT^2 + 17T + 1\right)$ |
| 10 | smooth | | $\left(p^3T^2 - 2pT + 1\right)\left(p^6T^4 + 56p^3T^3 + 228pT^2 + 56T + 1\right)$ |
| 11 | smooth | | $\left(p^3T^2 + 4pT + 1\right)\left(p^6T^4 - p^4T^3 + 168pT^2 - pT + 1\right)$ |
| 12 | singular | $-\frac{1}{27}$ | |

| | | | |
|---|---|---|---|
| $p = 17$ | | | |
| $\varphi$ | smooth/sing. | singularity | $R(T)$ |
| 1 | smooth | | $\left(p^3T^2 - 6pT + 1\right)\left(p^6T^4 - 12p^3T^3 + 128pT^2 - 12T + 1\right)$ |
| 2 | smooth | | $\left(p^3T^2 + 3pT + 1\right)\left(p^6T^4 + 105p^3T^3 + 488pT^2 + 105T + 1\right)$ |
| 3 | smooth | | $\left(p^3T^2 - 3pT + 1\right)\left(p^6T^4 + 93p^3T^3 + 254pT^2 + 93T + 1\right)$ |
| 4 | smooth | | $\left(p^3T^2 + 6pT + 1\right)\left(p^6T^4 + 21p^3T^3 - 250pT^2 + 21T + 1\right)$ |
| 5 | singular | $-\frac{1}{27}$ | |
| 6 | smooth* | | |
| 7 | smooth | | $\left(p^3T^2 + 3pT + 1\right)\left(p^6T^4 - 39p^3T^3 + 38pT^2 - 39T + 1\right)$ |
| 8 | smooth | | $\left(p^3T^2 + 6pT + 1\right)\left(p^6T^4 + 6p^4T^3 + 542pT^2 + 6pT + 1\right)$ |
| 9 | smooth | | $\left(p^3T^2 + 6pT + 1\right)\left(p^6T^4 - 132p^3T^3 + 668pT^2 - 132T + 1\right)$ |
| 10 | singular | $\frac{1}{216}$ | |
| 11 | smooth | | $\left(p^3T^2 + 1\right)\left(p^6T^4 + 108p^3T^3 + 362pT^2 + 108T + 1\right)$ |
| 12 | smooth | | $\left(p^3T^2 + 1\right)\left(p^6T^4 + 117p^3T^3 + 524pT^2 + 117T + 1\right)$ |
| 13 | smooth | | $\left(p^3T^2 - 6pT + 1\right)\left(p^6T^4 - 39p^3T^3 - 142pT^2 - 39T + 1\right)$ |
| 14 | smooth* | | |
| 15 | smooth | | $\left(p^3T^2 - 3pT + 1\right)\left(p^6T^4 + 6p^4T^3 + 25p^2T^2 + 6pT + 1\right)$ |
| 16 | smooth | | $\left(p^3T^2 + 3pT + 1\right)\left(p^6T^4 + 15p^3T^3 - 196pT^2 + 15T + 1\right)$ |

| | | | |
|---|---|---|---|
| $p = 19$ | | | |
| $\varphi$ | smooth/sing. | singularity | $R(T)$ |
| 1 | smooth | | $\left(p^3T^2 - 2pT + 1\right)\left(p^6T^4 + 59p^3T^3 + 366pT^2 + 59T + 1\right)$ |
| 2 | smooth | | $\left(p^3T^2 + pT + 1\right)\left(p^6T^4 - 70p^3T^3 + 579pT^2 - 70T + 1\right)$ |
| 3 | smooth | | $\left(p^3T^2 - 5pT + 1\right)\left(p^6T^4 + 89p^3T^3 + 432pT^2 + 89T + 1\right)$ |
| 4 | smooth | | $\left(p^3T^2 - 5pT + 1\right)\left(p^6T^4 - 109p^3T^3 + 612pT^2 - 109T + 1\right)$ |
| 5 | smooth | | $\left(p^3T^2 + pT + 1\right)\left(p^6T^4 - 43p^3T^3 + 714pT^2 - 43T + 1\right)$ |
| 6 | smooth* | | |
| 7 | singular | $-\frac{1}{27}$ | |

| $p = 19$, continued | | | |
|---|---|---|---|
| $\varphi$ | smooth/sing. | singularity | $R(T)$ |
| 8 | smooth | | $\left(p^3T^2 - 2pT + 1\right)\left(p^6T^4 + 158p^3T^3 + 816pT^2 + 158T + 1\right)$ |
| 9 | smooth | | $\left(p^3T^2 + 4pT + 1\right)\left(p^6T^4 - p^4T^3 - 468pT^2 - pT + 1\right)$ |
| 10 | smooth | | $\left(p^3T^2 + 4pT + 1\right)\left(p^6T^4 + 89p^3T^3 + 270pT^2 + 89T + 1\right)$ |
| 11 | singular | $\frac{1}{216}$ | |
| 12 | smooth | | $\left(p^3T^2 - 2pT + 1\right)\left(p^6T^4 - 4p^3T^3 + 330pT^2 - 4T + 1\right)$ |
| 13 | smooth | | $\left(p^3T^2 - 8pT + 1\right)\left(p^6T^4 + 74p^3T^3 + 210pT^2 + 74T + 1\right)$ |
| 14 | smooth | | $\left(p^3T^2 - 2pT + 1\right)\left(p^6T^4 + 41p^3T^3 + 420pT^2 + 41T + 1\right)$ |
| 15 | smooth | | $\left(p^3T^2 + 7pT + 1\right)\left(p^6T^4 + 131p^3T^3 + 510pT^2 + 131T + 1\right)$ |
| 16 | smooth* | | |
| 17 | smooth | | $\left(p^3T^2 + 4pT + 1\right)\left(p^6T^4 - 28p^3T^3 + 198pT^2 - 28T + 1\right)$ |
| 18 | smooth | | $\left(p^3T^2 - 2pT + 1\right)\left(p^6T^4 - 4p^3T^3 - 156pT^2 - 4T + 1\right)$ |

| $p = 23$ | | | |
|---|---|---|---|
| $\varphi$ | smooth/sing. | singularity | $R(T)$ |
| 1 | smooth | | $\left(p^3T^2 + 1\right)\left(p^6T^4 + 144p^3T^3 + 662pT^2 + 144T + 1\right)$ |
| 2 | smooth | | $\left(p^3T^2 - 6pT + 1\right)\left(p^6T^4 - 57p^3T^3 - 382pT^2 - 57T + 1\right)$ |
| 3 | smooth* | | |
| 4 | smooth | | $\left(p^3T^2 - 3pT + 1\right)\left(p^6T^4 + 39p^3T^3 - 220pT^2 + 39T + 1\right)$ |
| 5 | smooth | | $\left(p^3T^2 - 6pT + 1\right)\left(p^6T^4 - 102p^3T^3 + 428pT^2 - 102T + 1\right)$ |
| 6 | smooth | | $\left(p^3T^2 + 6pT + 1\right)\left(p^6T^4 + 3p^3T^3 + 122pT^2 + 3T + 1\right)$ |
| 7 | smooth | | $\left(p^3T^2 - 3pT + 1\right)\left(p^6T^4 - 42p^3T^3 - 409pT^2 - 42T + 1\right)$ |
| 8 | smooth | | $\left(p^3T^2 + 3pT + 1\right)\left(p^6T^4 + 141p^3T^3 + 680pT^2 + 141T + 1\right)$ |
| 9 | smooth | | $\left(p^3T^2 - 3pT + 1\right)\left(p^6T^4 + 111p^3T^3 + 986pT^2 + 111T + 1\right)$ |
| 10 | smooth* | | |
| 11 | smooth | | $\left(p^3T^2 + 102T + 1\right)\left(p^3T^2 - 6pT + 1\right)\left(p^3T^2 - 3pT + 1\right)$ |
| 12 | smooth | | $\left(p^3T^2 + 6pT + 1\right)\left(p^6T^4 - 51p^3T^3 + 716pT^2 - 51T + 1\right)$ |
| 13 | smooth | | $\left(p^3T^2 + 3pT + 1\right)\left(p^6T^4 + 15p^3T^3 + 464pT^2 + 15T + 1\right)$ |
| 14 | smooth | | $\left(p^3T^2 - 6pT + 1\right)\left(p^6T^4 + 96p^3T^3 + 752pT^2 + 96T + 1\right)$ |
| 15 | smooth | | $\left(p^3T^2 + 6pT + 1\right)\left(p^6T^4 - 6p^3T^3 + 716pT^2 - 6T + 1\right)$ |
| 16 | smooth | | $\left(p^3T^2 + 1\right)\left(p^6T^4 + 135p^3T^3 + 1094pT^2 + 135T + 1\right)$ |
| 17 | singular | $-\frac{1}{27}$ | |
| 18 | singular | $\frac{1}{216}$ | |
| 19 | smooth | | $\left(p^3T^2 - 9pT + 1\right)\left(p^6T^4 + 36p^3T^3 + 311pT^2 + 36T + 1\right)$ |
| 20 | smooth | | $\left(p^3T^2 + 1\right)\left(p^6T^4 - 72p^3T^3 + 878pT^2 - 72T + 1\right)$ |
| 21 | smooth | | $\left(p^3T^2 + 9pT + 1\right)\left(p^6T^4 + 81p^3T^3 + 554pT^2 + 81T + 1\right)$ |

| | $p = 23$, continued | | |
|---|---|---|---|
| $\varphi$ | smooth/sing. | singularity | $R(T)$ |
| 22 | smooth | | $\left(p^3T^2 + 1\right)\left(p^6T^4 + 27p^3T^3 - 364pT^2 + 27T + 1\right)$ |

| | $p = 29$ | | |
|---|---|---|---|
| $\varphi$ | smooth/sing. | singularity | $R(T)$ |
| 1 | smooth | | $\left(p^3T^2 + 6pT + 1\right)\left(p^6T^4 - 15p^3T^3 - 244pT^2 - 15T + 1\right)$ |
| 2 | smooth | | $\left(p^3T^2 + 9pT + 1\right)\left(p^6T^4 - 90p^3T^3 + 665pT^2 - 90T + 1\right)$ |
| 3 | smooth | | $\left(p^3T^2 + 1\right)\left(p^6T^4 + 117p^3T^3 + 152pT^2 + 117T + 1\right)$ |
| 4 | smooth | | $\left(p^3T^2 - 3pT + 1\right)\left(p^3T^2 + 3pT + 1\right)\left(p^3T^2 + 6pT + 1\right)$ |
| 5 | smooth | | $\left(p^3T^2 + 9pT + 1\right)\left(p^6T^4 - 27p^3T^3 - 604pT^2 - 27T + 1\right)$ |
| 6 | smooth | | $\left(p^3T^2 + 1\right)\left(p^6T^4 + 9p^3T^3 - 550pT^2 + 9T + 1\right)$ |
| 7 | smooth* | | |
| 8 | smooth | | $\left(p^3T^2 - 3pT + 1\right)\left(p^6T^4 + 66p^3T^3 + 1115pT^2 + 66T + 1\right)$ |
| 9 | singular | $\frac{1}{216}$ | |
| 10 | smooth | | $\left(p^3T^2 - 9pT + 1\right)\left(p^6T^4 + 36p^3T^3 + 1475pT^2 + 36T + 1\right)$ |
| 11 | smooth* | | |
| 12 | smooth | | $\left(p^3T^2 + 1\right)\left(p^6T^4 + 216p^3T^3 + 1448pT^2 + 216T + 1\right)$ |
| 13 | smooth | | $\left(p^3T^2 - 9pT + 1\right)\left(p^6T^4 - 9p^3T^3 + 854pT^2 - 9T + 1\right)$ |
| 14 | smooth | | $\left(p^3T^2 - 6pT + 1\right)\left(p^6T^4 + 195p^3T^3 + 1520pT^2 + 195T + 1\right)$ |
| 15 | singular | $-\frac{1}{27}$ | |
| 16 | smooth | | $\left(p^3T^2 + 6pT + 1\right)\left(p^6T^4 - 60p^3T^3 + 332pT^2 - 60T + 1\right)$ |
| 17 | smooth | | $\left(p^3T^2 + 6pT + 1\right)\left(p^6T^4 + 156p^3T^3 + 692pT^2 + 156T + 1\right)$ |
| 18 | smooth | | $\left(p^3T^2 - 6pT + 1\right)\left(p^6T^4 - 12p^3T^3 - 622pT^2 - 12T + 1\right)$ |
| 19 | smooth | | $\left(p^3T^2 - 6pT + 1\right)\left(p^6T^4 + 24p^3T^3 - 118pT^2 + 24T + 1\right)$ |
| 20 | smooth | | $\left(p^3T^2 - 6pT + 1\right)\left(p^6T^4 - 120p^3T^3 + 1466pT^2 - 120T + 1\right)$ |
| 21 | smooth | | $\left(p^3T^2 + 3pT + 1\right)\left(p^6T^4 - 93p^3T^3 - 442pT^2 - 93T + 1\right)$ |
| 22 | smooth | | $\left(p^3T^2 + 1\right)\left(p^6T^4 + 360p^3T^3 + 2582pT^2 + 360T + 1\right)$ |
| 23 | smooth | | $\left(p^3T^2 - 3pT + 1\right)\left(p^6T^4 - 177p^3T^3 + 404pT^2 - 177T + 1\right)$ |
| 24 | smooth | | $\left(p^3T^2 + 6pT + 1\right)\left(p^6T^4 + 237p^3T^3 + 1952pT^2 + 237T + 1\right)$ |
| 25 | smooth | | $\left(p^3T^2 + 6pT + 1\right)\left(p^6T^4 - 276p^3T^3 + 1934pT^2 - 276T + 1\right)$ |
| 26 | smooth | | $\left(p^3T^2 + 6pT + 1\right)\left(p^6T^4 + 156p^3T^3 + 4p^2T^2 + 156T + 1\right)$ |
| 27 | smooth | | $\left(p^3T^2 - 6pT + 1\right)\left(p^6T^4 - 264p^3T^3 + 1484pT^2 - 264T + 1\right)$ |
| 28 | smooth | | $\left(p^3T^2 - 6pT + 1\right)\left(p^6T^4 + 195p^3T^3 + 1646pT^2 + 195T + 1\right)$ |

| | | | $p = 31$ |
|---|---|---|---|
| $\varphi$ | smooth/sing. | singularity | $R(T)$ |
| 1 | smooth | | $\left(p^3T^2 + 232T + 1\right)\left(p^3T^2 - 5pT + 1\right)\left(p^3T^2 + 4pT + 1\right)$ |
| 2 | smooth* | | |
| 3 | smooth | | $\left(p^3T^2 + pT + 1\right)\left(p^6T^4 - p^3T^3 - 18p^2T^2 - T + 1\right)$ |
| 4 | smooth | | $\left(p^3T^2 + 4pT + 1\right)\left(p^6T^4 - 4p^3T^3 + 438pT^2 - 4T + 1\right)$ |
| 5 | smooth | | $\left(p^3T^2 + pT + 1\right)\left(p^6T^4 + 179p^3T^3 + 1206pT^2 + 179T + 1\right)$ |
| 6 | smooth | | $\left(p^3T^2 - 233T + 1\right)\left(p^3T^2 - 5pT + 1\right)\left(p^3T^2 + 10pT + 1\right)$ |
| 7 | smooth | | $\left(p^3T^2 + 4pT + 1\right)\left(p^6T^4 - 139p^3T^3 + 1518pT^2 - 139T + 1\right)$ |
| 8 | singular | $-\frac{1}{27}$ | |
| 9 | smooth | | $\left(p^3T^2 + 4pT + 1\right)\left(p^6T^4 - 175p^3T^3 + 1554pT^2 - 175T + 1\right)$ |
| 10 | smooth | | $\left(p^3T^2 - 5pT + 1\right)\left(p^6T^4 - 67p^3T^3 - 12pT^2 - 67T + 1\right)$ |
| 11 | smooth | | $\left(p^3T^2 - 8pT + 1\right)\left(p^6T^4 - p^3T^3 + 162pT^2 - T + 1\right)$ |
| 12 | smooth | | $\left(p^3T^2 + 4pT + 1\right)\left(p^6T^4 + 32p^3T^3 - 462pT^2 + 32T + 1\right)$ |
| 13 | smooth | | $\left(p^3T^2 - 2pT + 1\right)\left(p^6T^4 + 56p^3T^3 - 1014pT^2 + 56T + 1\right)$ |
| 14 | smooth | | $\left(p^3T^2 - 2pT + 1\right)\left(p^6T^4 - 322p^3T^3 + 2226pT^2 - 322T + 1\right)$ |
| 15 | smooth | | $\left(p^3T^2 + 4pT + 1\right)\left(p^6T^4 - 76p^3T^3 + 312pT^2 - 76T + 1\right)$ |
| 16 | smooth | | $\left(p^3T^2 - 5pT + 1\right)\left(p^6T^4 + 275p^3T^3 + 1878pT^2 + 275T + 1\right)$ |
| 17 | smooth | | $\left(p^3T^2 - 8pT + 1\right)\left(p^6T^4 + 188p^3T^3 + 1782pT^2 + 188T + 1\right)$ |
| 18 | smooth | | $\left(p^3T^2 + 7pT + 1\right)\left(p^6T^4 + 389p^3T^3 + 3018pT^2 + 389T + 1\right)$ |
| 19 | smooth | | $\left(p^3T^2 + 10pT + 1\right)\left(p^6T^4 - 163p^3T^3 + 1404pT^2 - 163T + 1\right)$ |
| 20 | smooth | | $\left(p^3T^2 + 10pT + 1\right)\left(p^6T^4 + 80p^3T^3 + 234pT^2 + 80T + 1\right)$ |
| 21 | smooth | | $\left(p^3T^2 + 7pT + 1\right)\left(p^6T^4 + 92p^3T^3 + 21pT^2 + 92T + 1\right)$ |
| 22 | smooth | | $\left(p^3T^2 - 8pT + 1\right)\left(p^6T^4 - 19p^3T^3 - 252pT^2 - 19T + 1\right)$ |
| 23 | smooth | | $\left(p^3T^2 - 5pT + 1\right)\left(p^6T^4 + 104p^3T^3 + 141pT^2 + 104T + 1\right)$ |
| 24 | smooth | | $\left(p^3T^2 + 4pT + 1\right)\left(p^6T^4 + 338p^3T^3 + 2580pT^2 + 338T + 1\right)$ |
| 25 | smooth | | $\left(p^3T^2 - 2pT + 1\right)\left(p^6T^4 + 38p^3T^3 + 1452pT^2 + 38T + 1\right)$ |
| 26 | smooth | | $\left(p^3T^2 - 2pT + 1\right)\left(p^6T^4 - 52p^3T^3 - 690pT^2 - 52T + 1\right)$ |
| 27 | smooth* | | |
| 28 | smooth | | $\left(p^3T^2 - 8pT + 1\right)\left(p^6T^4 + 170p^3T^3 + 1638pT^2 + 170T + 1\right)$ |
| 29 | smooth | | $\left(p^3T^2 - 5pT + 1\right)\left(p^6T^4 + 32p^3T^3 + 1473pT^2 + 32T + 1\right)$ |
| 30 | singular | $\frac{1}{216}$ | |

| | | | $p = 37$ |
|---|---|---|---|
| $\varphi$ | smooth/sing. | singularity | $R(T)$ |
| 1 | smooth | | $\left(p^3T^2 - 2pT + 1\right)\left(p^6T^4 - 22p^3T^3 + 2310pT^2 - 22T + 1\right)$ |

| $\varphi$ | smooth/sing. | singularity | $R(T)$ |
|---|---|---|---|
| | | | $p = 37,\ continued$ |
| 2 | smooth | | $\left(p^3T^2 - 5pT + 1\right)\left(p^6T^4 - 415p^3T^3 + 3078pT^2 - 415T + 1\right)$ |
| 3 | smooth | | $\left(p^3T^2 - 2pT + 1\right)\left(p^6T^4 + 122p^3T^3 + 1662pT^2 + 122T + 1\right)$ |
| 4 | smooth | | $\left(p^3T^2 + 4pT + 1\right)\left(p^6T^4 - 10p^3T^3 + 756pT^2 - 10T + 1\right)$ |
| 5 | smooth | | $\left(p^3T^2 + 4pT + 1\right)\left(p^6T^4 + 98p^3T^3 + 1548pT^2 + 98T + 1\right)$ |
| 6 | singular | $\frac{1}{216}$ | |
| 7 | smooth | | $\left(p^3T^2 - 8pT + 1\right)\left(p^6T^4 + 92p^3T^3 + 930pT^2 + 92T + 1\right)$ |
| 8 | smooth | | $\left(p^3T^2 - 2pT + 1\right)\left(p^6T^4 + 77p^3T^3 - 804pT^2 + 77T + 1\right)$ |
| 9 | smooth | | $\left(p^3T^2 - 8pT + 1\right)\left(p^6T^4 + 389p^3T^3 + 2388pT^2 + 389T + 1\right)$ |
| 10 | smooth | | $\left(p^3T^2 + 7pT + 1\right)\left(p^6T^4 - 94p^3T^3 + 2211pT^2 - 94T + 1\right)$ |
| 11 | smooth | | $\left(p^3T^2 - 2pT + 1\right)\left(p^6T^4 - 202p^3T^3 + 2346pT^2 - 202T + 1\right)$ |
| 12 | smooth | | $\left(p^3T^2 - 2pT + 1\right)\left(p^6T^4 + 248p^3T^3 + 1932pT^2 + 248T + 1\right)$ |
| 13 | smooth | | $\left(p^3T^2 + 10pT + 1\right)\left(p^6T^4 + 128p^3T^3 + 642pT^2 + 128T + 1\right)$ |
| 14 | smooth | | $\left(p^3T^2 - 2pT + 1\right)\left(p^6T^4 - 265p^3T^3 + 2310pT^2 - 265T + 1\right)$ |
| 15 | smooth | | $\left(p^3T^2 - 2pT + 1\right)\left(p^6T^4 + 68p^3T^3 + 1014pT^2 + 68T + 1\right)$ |
| 16 | smooth | | $\left(p^3T^2 + 7pT + 1\right)\left(p^6T^4 - 166p^3T^3 + 33pT^2 - 166T + 1\right)$ |
| 17 | smooth | | $\left(p^3T^2 - 8pT + 1\right)\left(p^6T^4 - 178p^3T^3 + 1362pT^2 - 178T + 1\right)$ |
| 18 | smooth | | $\left(p^3T^2 - 2pT + 1\right)\left(p^6T^4 - 4p^3T^3 + 2094pT^2 - 4T + 1\right)$ |
| 19 | smooth | | $\left(p^3T^2 - 8pT + 1\right)\left(p^6T^4 - 115p^3T^3 + 1938pT^2 - 115T + 1\right)$ |
| 20 | smooth | | $\left(p^3T^2 - 11pT + 1\right)\left(p^6T^4 - 31p^3T^3 - 1560pT^2 - 31T + 1\right)$ |
| 21 | smooth | | $\left(p^3T^2 + 4pT + 1\right)\left(p^6T^4 + 8p^4T^3 + 2376pT^2 + 8pT + 1\right)$ |
| 22 | smooth | | $\left(p^3T^2 + 4pT + 1\right)\left(p^6T^4 + 26p^3T^3 + 26T + 1\right)$ |
| 23 | smooth | | $\left(p^3T^2 - 2pT + 1\right)\left(p^6T^4 + 104p^3T^3 - 210pT^2 + 104T + 1\right)$ |
| 24 | smooth* | | |
| 25 | smooth* | | |
| 26 | singular | $-\frac{1}{27}$ | |
| 27 | smooth | | $\left(p^3T^2 + 7pT + 1\right)\left(p^6T^4 - 274p^3T^3 + 3057pT^2 - 274T + 1\right)$ |
| 28 | smooth | | $\left(p^3T^2 + 10pT + 1\right)\left(p^6T^4 - 124p^3T^3 - 96pT^2 - 124T + 1\right)$ |
| 29 | smooth | | $\left(p^3T^2 - 2pT + 1\right)\left(p^6T^4 + 203p^3T^3 + 564pT^2 + 203T + 1\right)$ |
| 30 | smooth | | $\left(p^3T^2 + pT + 1\right)\left(p^6T^4 - 97p^3T^3 + 66p^2T^2 - 97T + 1\right)$ |
| 31 | smooth | | $\left(p^3T^2 + 7pT + 1\right)\left(p^6T^4 + 311p^3T^3 + 1644pT^2 + 311T + 1\right)$ |
| 32 | smooth | | $\left(p^3T^2 + 7pT + 1\right)\left(p^6T^4 + 41p^3T^3 + 24p^2T^2 + 41T + 1\right)$ |
| 33 | smooth | | $\left(p^3T^2 + pT + 1\right)\left(p^6T^4 + 146p^3T^3 + 2469pT^2 + 146T + 1\right)$ |
| 34 | smooth | | $\left(p^3T^2 - 2pT + 1\right)\left(p^6T^4 - 103p^3T^3 + 1482pT^2 - 103T + 1\right)$ |
| 35 | smooth | | $\left(p^3T^2 + 10pT + 1\right)\left(p^6T^4 + 335p^3T^3 + 3198pT^2 + 335T + 1\right)$ |
| 36 | smooth | | $\left(p^3T^2 + 7pT + 1\right)\left(p^6T^4 - 31p^3T^3 - 588pT^2 - 31T + 1\right)$ |

| | | | $p = 41$ |
|---|---|---|---|
| $\varphi$ | smooth/sing. | singularity | $R(T)$ |
| 1 | smooth | | $\left(p^3T^2 - 6pT + 1\right)\left(p^6T^4 - 111p^3T^3 + 392pT^2 - 111T + 1\right)$ |
| 2 | smooth | | $\left(p^3T^2 + 6pT + 1\right)\left(p^6T^4 - 294p^3T^3 + 770pT^2 - 294T + 1\right)$ |
| 3 | singular | $-\frac{1}{27}$ | |
| 4 | smooth | | $\left(p^3T^2 + 1\right)\left(p^6T^4 + 423p^3T^3 + 2804pT^2 + 423T + 1\right)$ |
| 5 | smooth | | $\left(p^3T^2 + 6pT + 1\right)\left(p^6T^4 + 354p^3T^3 + 2732pT^2 + 354T + 1\right)$ |
| 6 | smooth | | $\left(p^3T^2 - 6pT + 1\right)\left(p^6T^4 + 105p^3T^3 + 500pT^2 + 105T + 1\right)$ |
| 7 | smooth | | $\left(p^3T^2 + 3pT + 1\right)\left(p^6T^4 - 138p^3T^3 + 977pT^2 - 138T + 1\right)$ |
| 8 | smooth | | $\left(p^3T^2 - 12pT + 1\right)\left(p^6T^4 - 591p^3T^3 + 4676pT^2 - 591T + 1\right)$ |
| 9 | smooth | | $\left(p^3T^2 + 9pT + 1\right)\left(p^6T^4 - 108p^3T^3 + 887pT^2 - 108T + 1\right)$ |
| 10 | smooth | | $\left(p^3T^2 - 3pT + 1\right)\left(p^6T^4 - 105p^3T^3 + 2228pT^2 - 105T + 1\right)$ |
| 11 | smooth* | | |
| 12 | smooth | | $\left(p^3T^2 + 6pT + 1\right)\left(p^6T^4 - 150p^3T^3 + 1310pT^2 - 150T + 1\right)$ |
| 13 | smooth | | $\left(p^3T^2 - 183T + 1\right)\left(p^3T^2 - 9pT + 1\right)\left(p^3T^2 + 6pT + 1\right)$ |
| 14 | smooth | | $\left(p^3T^2 - 12pT + 1\right)\left(p^6T^4 - 249p^3T^3 + 2336pT^2 - 249T + 1\right)$ |
| 15 | singular | $\frac{1}{216}$ | |
| 16 | smooth | | $\left(p^3T^2 + 9pT + 1\right)\left(p^6T^4 - 387p^3T^3 + 3614pT^2 - 387T + 1\right)$ |
| 17 | smooth | | $\left(p^3T^2 - 6pT + 1\right)\left(p^6T^4 - 57p^3T^3 - 2218pT^2 - 57T + 1\right)$ |
| 18 | smooth | | $\left(p^3T^2 + 3pT + 1\right)\left(p^6T^4 - 237p^3T^3 - 40pT^2 - 237T + 1\right)$ |
| 19 | smooth* | | |
| 20 | smooth | | $\left(p^3T^2 - 6pT + 1\right)\left(p^6T^4 + 276p^3T^3 + 2858pT^2 + 276T + 1\right)$ |
| 21 | smooth | | $\left(p^3T^2 + 1\right)\left(p^6T^4 - 99p^3T^3 + 158pT^2 - 99T + 1\right)$ |
| 22 | smooth | | $\left(p^3T^2 + 1\right)\left(p^6T^4 - 36p^3T^3 + 212pT^2 - 36T + 1\right)$ |
| 23 | smooth | | $\left(p^3T^2 + 1\right)\left(p^6T^4 + 914pT^2 + 1\right)$ |
| 24 | smooth | | $\left(p^3T^2 + 6pT + 1\right)\left(p^6T^4 + 273p^3T^3 + 3128pT^2 + 273T + 1\right)$ |
| 25 | smooth | | $\left(p^3T^2 + 6pT + 1\right)\left(p^6T^4 + 30p^3T^3 - 1930pT^2 + 30T + 1\right)$ |
| 26 | smooth | | $\left(p^3T^2 - 3pT + 1\right)\left(p^6T^4 + 687p^3T^3 + 5684pT^2 + 687T + 1\right)$ |
| 27 | smooth | | $\left(p^3T^2 + 6pT + 1\right)\left(p^6T^4 + 30p^3T^3 + 140pT^2 + 30T + 1\right)$ |
| 28 | smooth | | $\left(p^3T^2 - 9pT + 1\right)\left(p^6T^4 + 405p^3T^3 + 2210pT^2 + 405T + 1\right)$ |
| 29 | smooth | | $\left(p^3T^2 + 1\right)\left(p^6T^4 - 8p^2T^2 + 1\right)$ |
| 30 | smooth | | $\left(p^3T^2 + 12pT + 1\right)\left(p^6T^4 + 24p^3T^3 + 2606pT^2 + 24T + 1\right)$ |
| 31 | smooth | | $\left(p^3T^2 + 9pT + 1\right)\left(p^6T^4 + 225p^3T^3 + 2264pT^2 + 225T + 1\right)$ |
| 32 | smooth | | $\left(p^3T^2 - 6pT + 1\right)\left(p^6T^4 + 69p^3T^3 + 3128pT^2 + 69T + 1\right)$ |
| 33 | smooth | | $\left(p^3T^2 + 3pT + 1\right)\left(p^6T^4 + 141p^3T^3 + 374pT^2 + 141T + 1\right)$ |
| 34 | smooth | | $\left(p^3T^2 + 3pT + 1\right)\left(p^6T^4 + 555p^3T^3 + 4370pT^2 + 555T + 1\right)$ |
| 35 | smooth | | $\left(p^3T^2 - 9pT + 1\right)\left(p^6T^4 + 1967pT^2 + 1\right)$ |

| $p = 41$, *continued* | | | |
|---|---|---|---|
| $\varphi$ | smooth/sing. | singularity | $R(T)$ |
| 36 | smooth | | $\left(p^3T^2 + 12pT + 1\right)\left(p^6T^4 - 138p^3T^3 + 1400pT^2 - 138T + 1\right)$ |
| 37 | smooth | | $\left(p^3T^2 - 3pT + 1\right)\left(p^6T^4 - 69p^3T^3 + 1436pT^2 - 69T + 1\right)$ |
| 38 | smooth | | $\left(p^3T^2 + 6pT + 1\right)\left(p^6T^4 + 336p^3T^3 + 4010pT^2 + 336T + 1\right)$ |
| 39 | smooth | | $\left(p^3T^2 - 6pT + 1\right)\left(p^6T^4 + 339p^3T^3 + 2912pT^2 + 339T + 1\right)$ |
| 40 | smooth | | $\left(p^3T^2 + 1\right)\left(p^6T^4 + 180p^3T^3 - 706pT^2 + 180T + 1\right)$ |

| $p = 43$ | | | |
|---|---|---|---|
| $\varphi$ | smooth/sing. | singularity | $R(T)$ |
| 1 | singular | $\frac{1}{216}$ | |
| 2 | smooth | | $\left(p^3T^2 - 8pT + 1\right)\left(p^6T^4 + 104p^3T^3 + 1896pT^2 + 104T + 1\right)$ |
| 3 | smooth | | $\left(p^3T^2 - 8pT + 1\right)\left(p^6T^4 + 59p^3T^3 - 1254pT^2 + 59T + 1\right)$ |
| 4 | smooth* | | |
| 5 | smooth | | $\left(p^3T^2 - 2pT + 1\right)\left(p^6T^4 + 332p^3T^3 + 2250pT^2 + 332T + 1\right)$ |
| 6 | smooth | | $\left(p^3T^2 + 412T + 1\right)\left(p^3T^2 - 5pT + 1\right)\left(p^3T^2 + 7pT + 1\right)$ |
| 7 | smooth | | $\left(p^3T^2 + 4pT + 1\right)\left(p^6T^4 - 340p^3T^3 + 4134pT^2 - 340T + 1\right)$ |
| 8 | smooth | | $\left(p^3T^2 + 10pT + 1\right)\left(p^6T^4 + 131p^3T^3 + 24p^2T^2 + 131T + 1\right)$ |
| 9 | smooth | | $\left(p^3T^2 + 4pT + 1\right)\left(p^6T^4 + 272p^3T^3 + 1398pT^2 + 272T + 1\right)$ |
| 10 | smooth | | $\left(p^3T^2 - 2pT + 1\right)\left(p^6T^4 - 415p^3T^3 + 3708pT^2 - 415T + 1\right)$ |
| 11 | smooth | | $\left(p^3T^2 - 8pT + 1\right)\left(p^6T^4 - 103p^3T^3 + 2256pT^2 - 103T + 1\right)$ |
| 12 | smooth | | $\left(p^3T^2 + 4pT + 1\right)\left(p^6T^4 + 146p^3T^3 - 510pT^2 + 146T + 1\right)$ |
| 13 | smooth | | $\left(p^3T^2 + pT + 1\right)\left(p^6T^4 + 113p^3T^3 + 1860pT^2 + 113T + 1\right)$ |
| 14 | smooth | | $\left(p^3T^2 + 10pT + 1\right)\left(p^6T^4 + 5p^3T^3 - 1776pT^2 + 5T + 1\right)$ |
| 15 | smooth | | $\left(p^3T^2 - 11pT + 1\right)\left(p^6T^4 + 557p^3T^3 + 4374pT^2 + 557T + 1\right)$ |
| 16 | smooth | | $\left(p^3T^2 + pT + 1\right)\left(p^6T^4 - 247p^3T^3 + 1626pT^2 - 247T + 1\right)$ |
| 17 | smooth | | $\left(p^3T^2 - 5pT + 1\right)\left(p^6T^4 - 304p^3T^3 + 2847pT^2 - 304T + 1\right)$ |
| 18 | smooth | | $\left(p^3T^2 - 5pT + 1\right)\left(p^6T^4 + 353p^3T^3 + 2262pT^2 + 353T + 1\right)$ |
| 19 | smooth | | $\left(p^3T^2 + pT + 1\right)\left(p^6T^4 - 76p^3T^3 + 1509pT^2 - 76T + 1\right)$ |
| 20 | smooth | | $\left(p^3T^2 - 11pT + 1\right)\left(p^6T^4 + 395p^3T^3 + 3294pT^2 + 395T + 1\right)$ |
| 21 | smooth | | $\left(p^3T^2 + 10pT + 1\right)\left(p^6T^4 + 626p^3T^3 + 5172pT^2 + 626T + 1\right)$ |
| 22 | smooth | | $\left(p^3T^2 + pT + 1\right)\left(p^6T^4 - 121p^3T^3 - 30pT^2 - 121T + 1\right)$ |
| 23 | smooth | | $\left(p^3T^2 - 332T + 1\right)\left(p^3T^2 + 4pT + 1\right)^2$ |
| 24 | smooth | | $\left(p^3T^2 - 8pT + 1\right)\left(p^6T^4 + 50p^3T^3 - 1974pT^2 + 50T + 1\right)$ |
| 25 | smooth | | $\left(p^3T^2 - 2pT + 1\right)\left(p^6T^4 + 143p^3T^3 + 3510pT^2 + 143T + 1\right)$ |
| 26 | smooth | | $\left(p^3T^2 + 4pT + 1\right)\left(p^6T^4 - 34p^3T^3 + 66pT^2 - 34T + 1\right)$ |
| 27 | smooth | | $\left(p^3T^2 + 10pT + 1\right)\left(p^6T^4 - 103p^3T^3 + 1122pT^2 - 103T + 1\right)$ |

| | $p = 43$, *continued* | | |
|---|---|---|---|
| $\varphi$ | smooth/sing. | singularity | $R(T)$ |
| 28 | smooth | | $\left(p^3T^2 + 4pT + 1\right)\left(p^6T^4 - 124p^3T^3 + 1542pT^2 - 124T + 1\right)$ |
| 29 | smooth | | $\left(p^3T^2 + 7pT + 1\right)\left(p^6T^4 - 487p^3T^3 + 3654pT^2 - 487T + 1\right)$ |
| 30 | smooth | | $\left(p^3T^2 - 2pT + 1\right)\left(p^6T^4 + 287p^3T^3 + 1836pT^2 + 287T + 1\right)$ |
| 31 | smooth | | $\left(p^3T^2 + 10pT + 1\right)\left(p^6T^4 + 176p^3T^3 - 156pT^2 + 176T + 1\right)$ |
| 32 | smooth | | $\left(p^3T^2 - 8pT + 1\right)\left(p^6T^4 + 428p^3T^3 + 3192pT^2 + 428T + 1\right)$ |
| 33 | smooth | | $\left(p^3T^2 - 8pT + 1\right)\left(p^6T^4 - 67p^3T^3 + 2400pT^2 - 67T + 1\right)$ |
| 34 | smooth | | $\left(p^3T^2 - 8pT + 1\right)\left(p^6T^4 + 113p^3T^3 + 726pT^2 + 113T + 1\right)$ |
| 35 | singular | $-\frac{1}{27}$ | |
| 36 | smooth | | $\left(p^3T^2 + pT + 1\right)\left(p^6T^4 - 4p^3T^3 - 1911pT^2 - 4T + 1\right)$ |
| 37 | smooth | | $\left(p^3T^2 + pT + 1\right)\left(p^6T^4 + 59p^3T^3 + 1644pT^2 + 59T + 1\right)$ |
| 38 | smooth | | $\left(p^3T^2 + 4pT + 1\right)\left(p^6T^4 + 92p^3T^3 + 1704pT^2 + 92T + 1\right)$ |
| 39 | smooth | | $\left(p^3T^2 - 8pT + 1\right)\left(p^6T^4 - 310p^3T^3 + 4074pT^2 - 310T + 1\right)$ |
| 40 | smooth | | $\left(p^3T^2 + 4pT + 1\right)\left(p^6T^4 + 2p^3T^3 + 30p^2T^2 + 2T + 1\right)$ |
| 41 | smooth* | | |
| 42 | smooth | | $\left(p^3T^2 + pT + 1\right)\left(p^6T^4 - 121p^3T^3 + 132pT^2 - 121T + 1\right)$ |

| | $p = 47$ | | |
|---|---|---|---|
| $\varphi$ | smooth/sing. | singularity | $R(T)$ |
| 1 | smooth | | $\left(p^3T^2 + 12pT + 1\right)\left(p^6T^4 + 96p^3T^3 + 2924pT^2 + 96T + 1\right)$ |
| 2 | smooth | | $\left(p^3T^2 - 6pT + 1\right)\left(p^6T^4 + 366p^3T^3 + 1016pT^2 + 366T + 1\right)$ |
| 3 | smooth | | $\left(p^3T^2 + 12pT + 1\right)\left(p^6T^4 - 84p^3T^3 - 1018pT^2 - 84T + 1\right)$ |
| 4 | smooth | | $\left(p^3T^2 + 3pT + 1\right)\left(p^6T^4 - 453p^3T^3 + 4166pT^2 - 453T + 1\right)$ |
| 5 | smooth | | $\left(p^3T^2 + 9pT + 1\right)\left(p^6T^4 + 63p^3T^3 - 784pT^2 + 63T + 1\right)$ |
| 6 | smooth | | $\left(p^3T^2 - 9pT + 1\right)\left(p^6T^4 + 162p^3T^3 + 3563pT^2 + 162T + 1\right)$ |
| 7 | smooth | | $\left(p^3T^2 - 6pT + 1\right)\left(p^6T^4 - 228p^3T^3 + 2600pT^2 - 228T + 1\right)$ |
| 8 | smooth | | $\left(p^3T^2 - 3pT + 1\right)\left(p^6T^4 - 204p^3T^3 + 2159pT^2 - 204T + 1\right)$ |
| 9 | smooth | | $\left(p^3T^2 + 6pT + 1\right)\left(p^6T^4 - 6p^3T^3 + 1610pT^2 - 6T + 1\right)$ |
| 10 | smooth* | | |
| 11 | smooth | | $\left(p^3T^2 - 3pT + 1\right)\left(p^6T^4 - 186p^3T^3 + 1079pT^2 - 186T + 1\right)$ |
| 12 | smooth | | $\left(p^3T^2 + 1\right)\left(p^6T^4 - 270p^3T^3 + 2942pT^2 - 270T + 1\right)$ |
| 13 | smooth | | $\left(p^3T^2 + 1\right)\left(p^6T^4 + 18p^3T^3 + 3482pT^2 + 18T + 1\right)$ |
| 14 | smooth | | $\left(p^3T^2 + 6pT + 1\right)\left(p^6T^4 - 105p^3T^3 + 2276pT^2 - 105T + 1\right)$ |
| 15 | smooth | | $\left(p^3T^2 - 12pT + 1\right)\left(p^6T^4 + 318p^3T^3 + 1736pT^2 + 318T + 1\right)$ |
| 16 | smooth | | $\left(p^3T^2 - 3pT + 1\right)\left(p^6T^4 + 237p^3T^3 + 1646pT^2 + 237T + 1\right)$ |
| 17 | smooth | | $\left(p^3T^2 - 6pT + 1\right)\left(p^6T^4 + 645p^3T^3 + 6290pT^2 + 645T + 1\right)$ |

| $\varphi$ | smooth/sing. | singularity | $R(T)$ |
|---|---|---|---|
| | | | $p = 47,$ *continued* |
| 18 | smooth | | $\left(p^3T^2 - 6pT + 1\right)\left(p^6T^4 - 12p^3T^3 - 946pT^2 - 12T + 1\right)$ |
| 19 | smooth | | $\left(p^3T^2 - 12pT + 1\right)\left(p^6T^4 - 321p^3T^3 + 3446pT^2 - 321T + 1\right)$ |
| 20 | smooth | | $\left(p^3T^2 + 1\right)\left(p^6T^4 - 36p^3T^3 + 1538pT^2 - 36T + 1\right)$ |
| 21 | smooth | | $\left(p^3T^2 + 12pT + 1\right)\left(p^6T^4 + 42p^3T^3 + 3698pT^2 + 42T + 1\right)$ |
| 22 | smooth | | $\left(p^3T^2 + 6pT + 1\right)\left(p^6T^4 - 78p^3T^3 + 3590pT^2 - 78T + 1\right)$ |
| 23 | smooth | | $\left(p^3T^2 + 6pT + 1\right)\left(p^6T^4 + 453p^3T^3 + 5408pT^2 + 453T + 1\right)$ |
| 24 | smooth | | $\left(p^3T^2 + 3pT + 1\right)\left(p^6T^4 + 3p^4T^3 - 1702pT^2 + 3pT + 1\right)$ |
| 25 | smooth | | $\left(p^3T^2 + 12pT + 1\right)\left(p^6T^4 + 627p^3T^3 + 5156pT^2 + 627T + 1\right)$ |
| 26 | smooth | | $\left(p^3T^2 - 6pT + 1\right)\left(p^6T^4 + 204p^3T^3 + 4p^2T^2 + 204T + 1\right)$ |
| 27 | smooth* | | |
| 28 | smooth | | $\left(p^3T^2 + 1\right)\left(p^6T^4 - 99p^3T^3 + 1322pT^2 - 99T + 1\right)$ |
| 29 | smooth | | $\left(p^3T^2 + 3pT + 1\right)\left(p^6T^4 + 465p^3T^3 + 4958pT^2 + 465T + 1\right)$ |
| 30 | smooth | | $\left(p^3T^2 - 9pT + 1\right)\left(p^6T^4 + 585p^3T^3 + 6182pT^2 + 585T + 1\right)$ |
| 31 | smooth | | $\left(p^3T^2 - 6pT + 1\right)\left(p^6T^4 - 57p^3T^3 - 2620pT^2 - 57T + 1\right)$ |
| 32 | smooth | | $\left(p^3T^2 + 3pT + 1\right)\left(p^6T^4 - 165p^3T^3 + 1124pT^2 - 165T + 1\right)$ |
| 33 | smooth | | $\left(p^3T^2 + 6pT + 1\right)\left(p^6T^4 + 390p^3T^3 + 4760pT^2 + 390T + 1\right)$ |
| 34 | smooth | | $\left(p^3T^2 + 9pT + 1\right)\left(p^6T^4 - 288p^3T^3 + 3185pT^2 - 288T + 1\right)$ |
| 35 | smooth | | $\left(p^3T^2 - 9pT + 1\right)\left(p^6T^4 + 153p^3T^3 + 2240pT^2 + 153T + 1\right)$ |
| 36 | smooth | | $\left(p^3T^2 - 3pT + 1\right)\left(p^6T^4 - 60p^3T^3 - 505pT^2 - 60T + 1\right)$ |
| 37 | smooth | | $\left(p^3T^2 - 12pT + 1\right)\left(p^6T^4 + 156p^3T^3 + 1178pT^2 + 156T + 1\right)$ |
| 38 | smooth | | $\left(p^3T^2 + 6pT + 1\right)\left(p^6T^4 + 120p^3T^3 - 1774pT^2 + 120T + 1\right)$ |
| 39 | smooth | | $\left(p^3T^2 - 12pT + 1\right)\left(p^6T^4 - 708p^3T^3 + 5570pT^2 - 708T + 1\right)$ |
| 40 | singular | $-\frac{1}{27}$ | |
| 41 | smooth | | $\left(p^3T^2 + 3pT + 1\right)\left(p^6T^4 - 543p^3T^3 + 4850pT^2 - 543T + 1\right)$ |
| 42 | singular | $\frac{1}{216}$ | |
| 43 | smooth | | $\left(p^3T^2 + 1\right)\left(p^6T^4 - 369p^3T^3 + 1862pT^2 - 369T + 1\right)$ |
| 44 | smooth | | $\left(p^3T^2 + 1\right)\left(p^6T^4 + 837p^3T^3 + 7586pT^2 + 837T + 1\right)$ |
| 45 | smooth | | $\left(p^3T^2 + 1\right)\left(p^6T^4 + 189p^3T^3 + 1700pT^2 + 189T + 1\right)$ |
| 46 | smooth | | $\left(p^3T^2 - 3pT + 1\right)\left(p^6T^4 + 12p^3T^3 - 1171pT^2 + 12T + 1\right)$ |

## C.2. Rational points and elliptic modular forms

Table 15: *Select rational points on the $\mathbb{Z}_2$ symmetric line $\boldsymbol{\varphi} = (\varphi, \varphi)$ in the moduli space of mirror bicubic manifolds together with the modular forms corresponding to the quadratic factors of the local zeta functions of these manifolds, and the related twisted Sen curves $\mathcal{E}_S$. The second and third columns labelled $\mathcal{E}_S(\tau/n)$ give the LMFDB label of the twisted Sen curves corresponding to the two isogenous families, and the penultimate column lists the label of the corresponding modular form common to all isogenous curves. The column labelled "p" lists primes at which the modular form coefficient $\alpha_p$ is not equal to the zeta function coefficient $c_p$. The last column gives the size of the isogeny class of the twisted Sen curves, confirming that there are indeed only two distinct families of twisted Sen curves.*

| $\varphi$ | $\mathcal{E}_S(\tau)$ | $\mathcal{E}_S(\tau/3)$ | $p$ | form label | # |
|---|---|---|---|---|---|
| $\frac{1}{165}$ | 163350.bo1 | 163350.bo2 | 43 | 163350.2.a.bo | 2 |
| $\frac{1}{77}$ | 154154.bg1 | 154154.bg2 | | 154154.2.a.bg | 2 |
| $\frac{1}{70}$ | 475300.c1 | 475300.c2 | | 475300.2.a.c | 2 |
| $\frac{1}{66}$ | 405108.o1 | 405108.o2 | | 405108.2.a.o | 2 |
| $\frac{1}{55}$ | 248050.c1 | 248050.c2 | | 248050.2.a.c | 2 |
| $\frac{1}{42}$ | 121716.bi1 | 121716.bi2 | | 121716.2.a.bi | 2 |
| $\frac{1}{35}$ | 75950.bi1 | 75950.bi2 | | 75950.2.a.bi | 2 |
| $\frac{1}{33}$ | 32670.q1 | 32670.q2 | | 32670.2.a.q | 2 |
| $\frac{1}{30}$ | 51300.c1 | 51300.c2 | | 51300.2.a.c | 2 |
| $\frac{1}{21}$ | 2646.h1 | 2646.h2 | | 2646.2.a.h | 2 |
| $\frac{3}{55}$ | 308550.cz1 | 308550.cz2 | | 308550.2.a.cz | 2 |
| $\frac{2}{35}$ | 218050.cg1 | 218050.cg2 | | 218050.2.a.cg | 2 |
| $\frac{2}{33}$ | 189486.s1 | 189486.s2 | 31 | 189486.2.a.s | 2 |
| $\frac{1}{15}$ | 9450.ba1 | 9450.ba2 | 43 | 9450.2.a.ba | 2 |
| $\frac{1}{14}$ | 8036.d1 | 8036.d2 | | 8036.2.a.d | 2 |
| $\frac{3}{35}$ | 213150.dh1 | 213150.dh2 | | 213150.2.a.dh | 2 |
| $\frac{1}{11}$ | 4598.h1 | 4598.h2 | | 4598.2.a.h | 2 |
| $\frac{2}{21}$ | 13230.cj1 | 13230.cj2 | 31 | 13230.2.a.cj | 2 |
| $\frac{1}{10}$ | 3700.b1 | 3700.b2 | | 3700.2.a.b | 2 |
| $\frac{4}{35}$ | 350350.fw1 | 350350.fw2 | 31 | 350350.2.a.fw | 2 |
| $\frac{9}{77}$ | 177870.ck1 | 177870.ck2 | | 177870.2.a.ck | 2 |
| $\frac{4}{33}$ | 307098.da1 | 307098.da2 | | 307098.2.a.da | 2 |

| $\varphi$ | $\mathcal{E}_S(\tau)$ | $\mathcal{E}_S(\tau/3)$ | $p$ | form label | # |
|---|---|---|---|---|---|
| | continued | | | | |
| $\frac{3}{22}$ | 149556.j1 | 149556.j2 | 43 | 149556.2.a.j | 2 |
| $\frac{1}{7}$ | 1666.c1 | 1666.c2 | | 1666.2.a.c | 2 |
| $\frac{5}{33}$ | 228690.df1 | 228690.df2 | | 228690.2.a.df | 2 |
| $\frac{1}{6}$ | 1188.b1 | 1188.b2 | | 1188.2.a.b | 2 |
| $\frac{2}{11}$ | 15730.r1 | 15730.r2 | | 15730.2.a.r | 2 |
| $\frac{4}{21}$ | 113778.y1 | 113778.y2 | | 113778.2.a.y | 2 |
| $\frac{1}{5}$ | 50.a2 | 50.a4 | | 50.2.a.a | 4 |
| $\frac{3}{14}$ | 55860.u1 | 55860.u2 | | 55860.2.a.u | 2 |
| $\frac{5}{22}$ | 379940.l1 | 379940.l2 | | 379940.2.a.l | 2 |
| $\frac{5}{21}$ | 171990.cp1 | 171990.cp2 | | 171990.2.a.cp | 2 |
| $\frac{4}{15}$ | 55350.v1 | 55350.v2 | 43 | 55350.2.a.v | 2 |
| $\frac{3}{11}$ | 16698.t1 | 16698.t2 | | 16698.2.a.t | 2 |
| $\frac{2}{7}$ | 5978.k1 | 5978.k2 | | 5978.2.a.k | 2 |
| $\frac{3}{10}$ | 27300.v1 | 27300.v2 | | 27300.2.a.v | 2 |
| $\frac{1}{3}$ | 270.a1 | 270.a2 | | 270.2.a.a | 2 |
| $\frac{5}{14}$ | 146020.b1 | 146020.b2 | | 146020.2.a.b | 2 |
| $\frac{4}{11}$ | 28798.x1 | 28798.x2 | 31 | 28798.2.a.x | 2 |
| $\frac{8}{21}$ | 209034.cw1 | 209034.cw2 | | 209034.2.a.cw | 2 |
| $\frac{13}{33}$ | 84942.t1 | 84942.t2 | | 84942.2.a.t | 2 |
| $\frac{2}{5}$ | 2950.k1 | 2950.k2 | | 2950.2.a.k | 2 |
| $\frac{9}{22}$ | 384780.v1 | 384780.v2 | | 384780.2.a.v | 2 |
| $\frac{3}{7}$ | 3234.h1 | 3234.h2 | | 3234.2.a.h | 2 |
| $\frac{5}{11}$ | 88330.g1 | 88330.g2 | | 88330.2.a.g | 2 |
| $\frac{7}{15}$ | 160650.df1 | 160650.df2 | | 160650.2.a.df | 2 |
| $\frac{27}{55}$ | 127050.dx1 | 127050.dx2 | | 127050.2.a.dx | 2 |
| $\frac{1}{2}$ | 116.b1 | 116.b2 | | 116.2.a.b | 2 |
| $\frac{8}{15}$ | 103950.gh1 | 103950.gh2 | | 103950.2.a.gh | 2 |
| $\frac{6}{11}$ | 125598.bl1 | 125598.bl2 | | 125598.2.a.bl | 2 |

| continued | | | | | |
|---|---|---|---|---|---|
| $\varphi$ | $\mathcal{E}_S(\tau)$ | $\mathcal{E}_S(\tau/3)$ | $p$ | form label | # |
| $\frac{4}{7}$ | **11270.j1** | **11270.j2** | | **11270.2.a.j** | 2 |
| $\frac{3}{5}$ | **6450.n1** | **6450.n2** | | **6450.2.a.n** | 2 |
| $\frac{7}{11}$ | **8470.m1** | **8470.m2** | | **8470.2.a.m** | 2 |
| $\frac{9}{14}$ | **151116.k1** | **151116.k2** | | **151116.2.a.k** | 2 |
| $\frac{2}{3}$ | **1026.j1** | **1026.j2** | | **1026.2.a.j** | 2 |
| $\frac{7}{10}$ | **139300.c1** | **139300.c2** | | **139300.2.a.c** | 2 |
| $\frac{5}{7}$ | **34790.i1** | **34790.i2** | | **34790.2.a.i** | 2 |
| $\frac{8}{11}$ | **54934.e1** | **54934.e2** | | **54934.2.a.e** | 2 |
| $\frac{11}{15}$ | **193050.g1** | **193050.g2** | | **193050.2.a.g** | 2 |
| $\frac{16}{21}$ | **399546.bn1** | **399546.bn2** | | **399546.2.a.bn** | 2 |
| $\frac{4}{5}$ | **5650.j1** | **5650.j2** | | **5650.2.a.j** | 2 |
| $\frac{17}{21}$ | **224910.v1** | **224910.v2** | | **224910.2.a.v** | 2 |
| $\frac{9}{11}$ | **92202.h1** | **92202.h2** | | **92202.2.a.h** | 2 |
| $\frac{5}{6}$ | **25380.n1** | **25380.n2** | 31 | **25380.2.a.n** | 2 |
| $\frac{6}{7}$ | **3822.bh1** | **3822.bh2** | | **3822.2.a.bh** | 2 |
| $\frac{9}{10}$ | **75900.be1** | **75900.be2** | | **75900.2.a.be** | 2 |
| $\frac{10}{11}$ | **340010.r1** | **340010.r2** | | **340010.2.a.r** | 2 |
| 1 | **14.a5** | **14.a6** | 31 | **14.2.a.a** | 6 |
| $\frac{16}{15}$ | **201150.bk1** | **201150.bk2** | | **201150.2.a.bk** | 2 |
| $\frac{12}{11}$ | **243210.br1** | **243210.br2** | | **243210.2.a.br** | 2 |
| $\frac{11}{10}$ | **337700.i1** | **337700.i2** | 29 | **337700.2.a.i** | 2 |
| $\frac{8}{7}$ | **21854.j1** | **21854.j2** | | **21854.2.a.j** | 2 |
| $\frac{7}{6}$ | **49140.f1** | **49140.f2** | | **49140.2.a.f** | 2 |
| $\frac{25}{21}$ | **383670.cf1** | **383670.cf2** | | **383670.2.a.cf** | 2 |
| $\frac{6}{5}$ | **25050.w1** | **25050.w2** | | **25050.2.a.w** | 2 |
| $\frac{19}{15}$ | **282150.bv1** | **282150.bv2** | | **282150.2.a.bv** | 2 |
| $\frac{9}{7}$ | **1470.f1** | **1470.f2** | | **1470.2.a.f** | 2 |
| $\frac{13}{10}$ | **24700.d1** | **24700.d2** | | **24700.2.a.d** | 2 |
| | | | | *Continued on the following page* | |

| $\varphi$ | $\mathcal{E}_S(\tau)$ | $\mathcal{E}_S(\tau/3)$ | $p$ | form label | # |
|---|---|---|---|---|---|
| | continued | | | | |
| $\frac{4}{3}$ | **1998.k1** | **1998.k2** | | **1998.2.a.k** | 2 |
| $\frac{48}{35}$ | **80850.fv1** | **80850.fv2** | | **80850.2.a.fv** | 2 |
| $\frac{7}{5}$ | **33950.n1** | **33950.n2** | | **33950.2.a.n** | 2 |
| $\frac{10}{7}$ | **135730.i1** | **135730.i2** | | **135730.2.a.i** | 2 |
| $\frac{16}{11}$ | **107206.j1** | **107206.j2** | | **107206.2.a.j** | 2 |
| $\frac{3}{2}$ | **996.b1** | **996.b2** | 43 | **996.2.a.b** | 2 |
| $\frac{11}{7}$ | **20482.f1** | **20482.f2** | 31 | **20482.2.a.f** | 2 |
| $\frac{8}{5}$ | **11050.o1** | **11050.o2** | | **11050.2.a.o** | 2 |
| $\frac{18}{11}$ | **360822.bk1** | **360822.bk2** | | **360822.2.a.bk** | 2 |
| $\frac{5}{3}$ | **6210.g1** | **6210.g2** | | **6210.2.a.g** | 2 |
| $\frac{12}{7}$ | **97314.s1** | **97314.s2** | | **97314.2.a.s** | 2 |
| $\frac{9}{5}$ | **4650.q1** | **4650.q2** | | **4650.2.a.q** | 2 |
| $\frac{11}{6}$ | **119988.c1** | **119988.c2** | | **119988.2.a.c** | 2 |
| $\frac{13}{7}$ | **228046.a1** | **228046.a2** | | **228046.2.a.a** | 2 |
| $\frac{21}{11}$ | **86394.bh1** | **86394.bh2** | | **86394.2.a.bh** | 2 |
| $\frac{27}{14}$ | **436884.p1** | **436884.p2** | 29 | **436884.2.a.p** | 2 |
| $2$ | **110.c1** | **110.c2** | | **110.2.a.c** | 2 |
| $\frac{23}{11}$ | **439714.b1** | **439714.b2** | 43 | **439714.2.a.b** | 2 |
| $\frac{32}{15}$ | **395550.bh1** | **395550.bh2** | | **395550.2.a.bh** | 2 |
| $\frac{15}{7}$ | **151410.bo1** | **151410.bo2** | | **151410.2.a.bo** | 2 |
| $\frac{13}{6}$ | **167076.o1** | **167076.o2** | | **167076.2.a.o** | 2 |
| $\frac{24}{11}$ | **478434.ba1** | **478434.ba2** | | **478434.2.a.ba** | 2 |
| $\frac{11}{5}$ | **83050.d1** | **83050.d2** | | **83050.2.a.d** | 2 |
| $\frac{25}{11}$ | **8470.o1** | **8470.o2** | | **8470.2.a.o** | 2 |
| $\frac{16}{7}$ | **43022.d1** | **43022.d2** | | **43022.2.a.d** | 2 |
| $\frac{7}{3}$ | **378.b2** | **378.b3** | | **378.2.a.b** | 3 |
| $\frac{12}{5}$ | **49350.ck1** | **49350.ck2** | | **49350.2.a.ck** | 2 |
| $\frac{17}{7}$ | **388178.g1** | **388178.g2** | | **388178.2.a.g** | 2 |

| $\varphi$ | $\mathcal{E}_S(\tau)$ | $\mathcal{E}_S(\tau/3)$ | $p$ | form label | # |
|---|---|---|---|---|---|
| continued | | | | | |
| $\frac{27}{11}$ | **134310.ba1** | **134310.ba2** | | **134310.2.a.ba** | 2 |
| $\frac{5}{2}$ | **2740.b1** | **2740.b2** | | **2740.2.a.b** | 2 |
| $\frac{18}{7}$ | **144942.cb1** | **144942.cb2** | | **144942.2.a.cb** | 2 |
| $\frac{13}{5}$ | **57850.j1** | **57850.j2** | | **57850.2.a.j** | 2 |
| $\frac{8}{3}$ | **3942.f1** | **3942.f2** | | **3942.2.a.f** | 2 |
| $\frac{27}{10}$ | **221700.j1** | **221700.j2** | | **221700.2.a.j** | 2 |
| $\frac{19}{7}$ | **121030.a1** | **121030.a2** | | **121030.2.a.a** | 2 |
| $\frac{14}{5}$ | **134050.m1** | **134050.m2** | | **134050.2.a.m** | 2 |
| $\frac{31}{11}$ | **397606.f1** | **397606.f2** | | **397606.2.a.f** | 2 |
| $\frac{17}{6}$ | **284580.a1** | **284580.a2** | | **284580.2.a.a** | 2 |
| $\frac{20}{7}$ | **268030.l1** | **268030.l2** | | **268030.2.a.l** | 2 |
| $\frac{43}{15}$ | **406350.cn1** | **406350.cn2** | | **406350.2.a.cn** | 2 |
| $\frac{32}{11}$ | **8470.t1** | **8470.t2** | | **8470.2.a.t** | 2 |
| $3$ | **246.d1** | **246.d2** | | **246.2.a.d** | 2 |
| $\frac{31}{10}$ | **238700.f1** | **238700.f2** | | **238700.2.a.f** | 2 |
| $\frac{19}{6}$ | **354996.a1** | **354996.a2** | | **354996.2.a.a** | 2 |
| $\frac{16}{5}$ | **21850.f1** | **21850.f2** | | **21850.2.a.f** | 2 |
| $\frac{23}{7}$ | **353878.d1** | **353878.d2** | | **353878.2.a.d** | 2 |
| $\frac{10}{3}$ | **24570.s1** | **24570.s2** | | **24570.2.a.s** | 2 |
| $\frac{17}{5}$ | **24650.c1** | **24650.c2** | | **24650.2.a.c** | 2 |
| $\frac{24}{7}$ | **192570.ce1** | **192570.ce2** | | **192570.2.a.ce** | 2 |
| $\frac{7}{2}$ | **5348.a1** | **5348.a2** | | **5348.2.a.a** | 2 |
| $\frac{25}{7}$ | **167090.i1** | **167090.i2** | | **167090.2.a.i** | 2 |
| $\frac{18}{5}$ | **73650.bn1** | **73650.bn2** | | **73650.2.a.bn** | 2 |
| $\frac{11}{3}$ | **2970.f1** | **2970.f2** | | **2970.2.a.f** | 2 |
| $\frac{81}{22}$ | **68244.k1** | **68244.k2** | | **68244.2.a.k** | 2 |
| $\frac{19}{5}$ | **246050.y1** | **246050.y2** | | **246050.2.a.y** | 2 |
| $\frac{27}{7}$ | **6762.o1** | **6762.o2** | | **6762.2.a.o** | 2 |

| continued | | | | | |
|---|---|---|---|---|---|
| $\varphi$ | $\mathcal{E}_S(\tau)$ | $\mathcal{E}_S(\tau/3)$ | $p$ | form label | # |
| 4 | **218.a1** | **218.a2** | 29 | **218.2.a.a** | 2 |
| $\frac{25}{6}$ | **122580.k1** | **122580.k2** | | **122580.2.a.k** | 2 |
| $\frac{21}{5}$ | **150150.cv1** | **150150.cv2** | | **150150.2.a.cv** | 2 |
| $\frac{47}{11}$ | **56870.c1** | **56870.c2** | | **56870.2.a.c** | 2 |

## C.3.    Real quadratic points and Hilbert modular forms

Table 16: *Some values of the modulus $\varphi$ which lie in various real quadratic field extensions $\mathbb{Q}(\sqrt{n})$ together with the corresponding Hilbert modular forms and the twisted Sen curves $\mathcal{E}_S$. The second and third columns labelled $\mathcal{E}_S(\tau/n)$ give the LMFDB labels (without the number field label) of the twisted Sen curves, and the penultimate column lists the label (again without the number field label) of the corresponding modular form. The column labelled "$N(\mathfrak{p})$" lists the ideal norms of the prime ideals at which the modular form coefficient $\alpha_{\mathfrak{p}}$ is not equal to the zeta function coefficient $c_p$ in the sense discussed in §4.6.*

| $\varphi$ | $\mathcal{E}_S(\tau)$ | $\mathcal{E}_S(\tau/3)$ | $N(\mathfrak{p})$ | form labels |
|---|---|---|---|---|
| $-10 - 7\sqrt{2}$ | **1838.2-c2** | **1838.2-c1** | | **1838.1-c, 1838.2-c** |
| $-10 + 7\sqrt{2}$ | **1838.1-c2** | **1838.1-c1** | | **1838.1-c, 1838.2-c** |
| $-9 - 6\sqrt{2}$ | **3906.2-l2** | **3906.2-l1** | | **3906.2-l, 3906.3-l** |
| $-9 + 6\sqrt{2}$ | **3906.3-l2** | **3906.3-l1** | | **3906.2-l, 3906.3-l** |
| $-7 - 5\sqrt{2}$ | **1106.2-c1** | **1106.2-c3** | | **1106.2-c, 1106.3-c** |
| $-7 + 5\sqrt{2}$ | **1106.3-c3** | **1106.3-c2** | | **1106.2-c, 1106.3-c** |
| $-4 - 3\sqrt{2}$ | **3346.3-c1** | **3346.3-c2** | | **3346.2-c, 3346.3-c** |
| $-4 + 3\sqrt{2}$ | **3346.2-c2** | **3346.2-c1** | | **3346.2-c, 3346.3-c** |
| $-3 - 4\sqrt{2}$ | **1058.1-e1** | **1058.1-e2** | | **1058.1-d, 1058.1-e** |
| $-3 - 2\sqrt{2}$ | **142.2-a2** | **142.2-a1** | | **142.1-a, 142.2-a** |
| $-3 + 2\sqrt{2}$ | **142.1-a2** | **142.1-a1** | | **142.1-a, 142.2-a** |
| $-3 + 4\sqrt{2}$ | **1058.1-d2** | **1058.1-d1** | | **1058.1-d, 1058.1-e** |
| $-2 - \sqrt{2}$ | **2702.4-b2** | **2702.4-b1** | 31 | **2702.1-b, 2702.4-b** |
| $-2 + \sqrt{2}$ | **2702.1-b2** | **2702.1-b1** | 31 | **2702.1-b, 2702.4-b** |
| $-1 - \sqrt{2}$ | **782.1-f1** | **782.1-f2** | | **782.1-f, 782.4-f** |
| | | | | *Continued on the following page* |

| continued | | | | |
|---|---|---|---|---|
| $\varphi$ | $\mathcal{E}_S(\tau)$ | $\mathcal{E}_S(\tau/3)$ | $N(\mathfrak{p})$ | form labels |
| $-1+\sqrt{2}$ | **782.4-f2** | **782.4-f1** | | **782.1-f, 782.4-f** |
| $-2\sqrt{2}$ | **238.2-c2** | **238.2-c3** | | **238.2-c, 238.3-c** |
| $-\sqrt{2}$ | **2914.1-b1** | **2914.1-b2** | | **2914.1-b, 2914.4-b** |
| $\sqrt{2}$ | **2914.4-b2** | **2914.4-b1** | | **2914.1-b, 2914.4-b** |
| $2\sqrt{2}$ | **238.3-c3** | **238.3-c2** | | **238.2-c, 238.3-c** |
| $1-\sqrt{2}$ | **674.1-d1** | **674.1-d2** | | **674.1-d, 674.2-d** |
| $1+\sqrt{2}$ | **674.2-d2** | **674.2-d1** | | **674.1-d, 674.2-d** |
| $2-\sqrt{2}$ | **3134.2-b1** | **3134.2-b2** | | **3134.1-b, 3134.2-b** |
| $2+\sqrt{2}$ | **3134.1-b1** | **3134.1-b2** | | **3134.1-b, 3134.2-b** |
| $3-2\sqrt{2}$ | **446.1-a1** | **446.1-a2** | | **446.1-a, 446.2-a** |
| $3+2\sqrt{2}$ | **446.2-a1** | **446.2-a2** | | **446.1-a, 446.2-a** |
| $4-3\sqrt{2}$ | **2482.2-c1** | **2482.2-c2** | | **2482.2-c, 2482.3-c** |
| $4+3\sqrt{2}$ | **2482.3-c2** | **2482.3-c1** | | **2482.2-c, 2482.3-c** |
| $5-4\sqrt{2}$ | **2114.3-f1** | **2114.3-f2** | 31 | **2114.2-f, 2114.3-f** |
| $5+4\sqrt{2}$ | **2114.2-f2** | **2114.2-f1** | 31 | **2114.2-f, 2114.3-f** |
| $7-5\sqrt{2}$ | **350.1-a2** | **350.1-a3** | | **350.1-a, 350.2-a** |
| $7+5\sqrt{2}$ | **350.2-a3** | **350.2-a2** | | **350.1-a, 350.2-a** |
| $10-7\sqrt{2}$ | **3998.1-c1** | **3998.1-c2** | | **3998.1-c, 3998.2-c** |
| $10+7\sqrt{2}$ | **3998.2-c1** | **3998.2-c2** | | **3998.1-c, 3998.2-c** |
| $-7-4\sqrt{3}$ | **22.2-a4** | **22.2-a2** | | **22.1-a, 22.2-a** |
| $-7+4\sqrt{3}$ | **22.1-a4** | **22.1-a3** | | **22.1-a, 22.2-a** |
| $-5-3\sqrt{3}$ | **3454.1-a1** | **3454.1-a2** | | **3454.1-a, 3454.4-a** |
| $-5+3\sqrt{3}$ | **3454.4-a2** | **3454.4-a1** | | **3454.1-a, 3454.4-a** |
| $-3-2\sqrt{3}$ | **3522.2-g1** | **3522.2-g2** | | **3522.1-g, 3522.2-g** |
| $-3+2\sqrt{3}$ | **3522.1-g2** | **3522.1-g1** | | **3522.1-g, 3522.2-g** |
| $-2-\sqrt{3}$ | **622.2-d2** | **622.2-d1** | | **622.1-d, 622.2-d** |
| $-2+\sqrt{3}$ | **622.1-d2** | **622.1-d1** | | **622.1-d, 622.2-d** |
| $-1-\sqrt{3}$ | **3022.1-a1** | **3022.1-a2** | | **3022.1-a, 3022.2-a** |
| | | | | *Continued on the following page* |

| $\varphi$ | $\mathcal{E}_S(\tau)$ | $\mathcal{E}_S(\tau/3)$ | $N(\mathfrak{p})$ | form labels |
|---|---|---|---|---|
| | | continued | | |
| $-1+\sqrt{3}$ | **3022.2-a2** | **3022.2-a1** | | **3022.1-a, 3022.2-a** |
| $1-\sqrt{3}$ | **2806.1-d1** | **2806.1-d2** | | **2806.1-d, 2806.4-d** |
| $1+\sqrt{3}$ | **2806.4-d2** | **2806.4-d1** | | **2806.1-d, 2806.4-d** |
| $2-\sqrt{3}$ | **838.2-b1** | **838.2-b2** | | **838.1-b, 838.2-b** |
| $2+\sqrt{3}$ | **838.1-b1** | **838.1-b2** | | **838.1-b, 838.2-b** |
| $3-2\sqrt{3}$ | **1518.1-f1** | **1518.1-f2** | | **1518.1-f, 1518.4-f** |
| $3+2\sqrt{3}$ | **1518.4-f2** | **1518.4-f1** | | **1518.1-f, 1518.4-f** |
| $4-4\sqrt{3}$ | **4202.1-d1** | **4202.1-d2** | | **4202.1-d, 4202.4-d** |
| $4+4\sqrt{3}$ | **4202.4-d2** | **4202.4-d1** | | **4202.1-d, 4202.4-d** |
| $5-3\sqrt{3}$ | **2374.2-c1** | **2374.2-c2** | | **2374.1-c, 2374.2-c** |
| $5+3\sqrt{3}$ | **2374.1-c2** | **2374.1-c1** | | **2374.1-c, 2374.2-c** |
| $7-4\sqrt{3}$ | **554.2-f1** | **554.2-f2** | | **554.1-f, 554.2-f** |
| $7+4\sqrt{3}$ | **554.1-f1** | **554.1-f2** | | **554.1-f, 554.2-f** |
| $-9-4\sqrt{5}$ | **244.2-b3** | **244.2-b2** | | **244.1-b, 244.2-b** |
| $-9+4\sqrt{5}$ | **244.1-b3** | **244.1-b2** | | **244.1-b, 244.2-b** |
| $-5-2\sqrt{5}$ | **4220.2-d2** | **4220.2-d1** | | **4220.1-d, 4220.2-d** |
| $-5+2\sqrt{5}$ | **4220.1-d2** | **4220.1-d1** | | **4220.1-d, 4220.2-d** |
| $-2-\sqrt{5}$ | **836.1-b2** | **836.1-b3** | | **836.1-b, 836.4-b** |
| $-2+\sqrt{5}$ | **836.4-b3** | **836.4-b2** | | **836.1-b, 836.4-b** |
| $2-\sqrt{5}$ | **620.1-c2** | **620.1-c3** | | **620.1-c, 620.2-c** |
| $2+\sqrt{5}$ | **620.2-c3** | **620.2-c2** | | **620.1-c, 620.2-c** |
| $9-4\sqrt{5}$ | **76.2-b3** | **76.2-b4** | | **76.1-b, 76.2-b** |
| $9+4\sqrt{5}$ | **76.1-b2** | **76.1-b4** | | **76.1-b, 76.2-b** |
| $-8-3\sqrt{7}$ | **298.1-g2** | **298.1-g1** | | **298.1-g, 298.2-g** |
| $-8+3\sqrt{7}$ | **298.2-g2** | **298.2-g1** | | **298.1-g, 298.2-g** |
| $-19-6\sqrt{10}$ | **74.2-a1** | **74.2-a2** | | **74.1-a, 74.2-a** |
| $-19+6\sqrt{10}$ | **74.1-a2** | **74.1-a1** | | **74.1-a, 74.2-a** |

## C.4. Quadratic imaginary points and Bianchi Modular Forms

Table 17: *Some values of the modulus $\varphi$ which lie in various imaginary quadratic field extensions $\mathbb{Q}(\sqrt{n})$ together with the corresponding Hilbert modular forms and the twisted Sen curves $\mathcal{E}_S$. The second and third columns labelled $\mathcal{E}_S(\tau/n)$ give the LMFDB labels (without the number field label) of the twisted Sen curves, and the penultimate column lists the label (again without the number field label) of the corresponding modular form. The column labelled "$N(\mathfrak{p})$" lists the ideal norms of the prime ideals at which the modular form coefficient $\alpha_{\mathfrak{p}}$ is not equal to the zeta function coefficient $c_p$ in the sense discussed in §4.6.*

| $\varphi$ | $\mathcal{E}_S(\tau)$ | $\mathcal{E}_S(\tau/3)$ | $N(\mathfrak{p})$ | form labels |
|---|---|---|---|---|
| $-7 - 4\mathrm{i}\sqrt{2}$ | 22002.5-a2 | 22002.5-a1 | 19 | 22002.4-a, 22002.5-a |
| $-7 + 4\mathrm{i}\sqrt{2}$ | 22002.4-a2 | 22002.4-a1 | 19 | 22002.4-a, 22002.5-a |
| $-2 - \mathrm{i}\sqrt{2}$ | 25602.3-a2 | 25602.3-a1 | 19 | 25602.3-a, 25602.6-a |
| $-2 + \mathrm{i}\sqrt{2}$ | 25602.6-a2 | 25602.6-a1 | 19 | 25602.3-a, 25602.6-a |
| $-1 - 2\mathrm{i}\sqrt{2}$ | 9762.2-a2 | 9762.2-a1 | | 9762.2-a, 9762.3-a |
| $-1 - \mathrm{i}\sqrt{2}$ | 6402.7-a2 | 6402.7-a1 | 19 | 6402.2-b, 6402.7-a |
| $-1 + \mathrm{i}\sqrt{2}$ | 6402.2-b2 | 6402.2-b1 | 19 | 6402.2-b, 6402.7-a |
| $-1 + 2\mathrm{i}\sqrt{2}$ | 9762.3-a2 | 9762.3-a1 | | 9762.2-a, 9762.3-a |
| $-4\mathrm{i}\sqrt{2}$ | 46658.4-a1 | 46658.4-a2 | 19 | 46658.1-a, 46658.4-a |
| $-2\mathrm{i}\sqrt{2}$ | 11666.2-a1 | 11666.2-a2 | 19 | 11666.2-a, 11666.3-a |
| $-\mathrm{i}\sqrt{2}$ | 2918.2-b1 | 2918.2-b2 | 19 | 2918.1-a, 2918.2-b |
| $\mathrm{i}\sqrt{2}$ | 2918.1-a1 | 2918.1-a2 | 19 | 2918.1-a, 2918.2-b |
| $2\mathrm{i}\sqrt{2}$ | 11666.3-a1 | 11666.3-a2 | 19 | 11666.2-a, 11666.3-a |
| $4\mathrm{i}\sqrt{2}$ | 46658.1-a1 | 46658.1-a2 | 19 | 46658.1-a, 46658.4-a |
| $1 - 2\mathrm{i}\sqrt{2}$ | 4962.4-b1 | 4962.4-b2 | 19 | 4962.1-b, 4962.4-b |
| $1 - \mathrm{i}\sqrt{2}$ | 6726.4-a1 | 6726.4-a2 | 19 | 6726.4-a, 6726.5-a |
| $1 + \mathrm{i}\sqrt{2}$ | 6726.5-a1 | 6726.5-a2 | 19 | 6726.4-a, 6726.5-a |
| $1 + 2\mathrm{i}\sqrt{2}$ | 4962.1-b1 | 4962.1-b2 | 19 | 4962.1-b, 4962.4-b |
| $2 - 4\mathrm{i}\sqrt{2}$ | 8322.7-a1 | 8322.7-a2 | 43 | 8322.2-b, 8322.7-a |
| $2 - \mathrm{i}\sqrt{2}$ | 26898.4-a1 | 26898.4-a2 | 19 | 26898.1-a, 26898.4-a |
| $2 + \mathrm{i}\sqrt{2}$ | 26898.1-a1 | 26898.1-a2 | 19 | 26898.1-a, 26898.4-a |
| $2 + 4\mathrm{i}\sqrt{2}$ | 8322.2-b1 | 8322.2-b2 | 43 | 8322.2-b, 8322.7-a |
| $-9 - 3\mathrm{i}\sqrt{3}$ | 72228.1-a2 | 72228.1-a1 | | 72228.1-a, 72228.4-a |
| $-9 + 3\mathrm{i}\sqrt{3}$ | 72228.4-a2 | 72228.4-a1 | | 72228.1-a, 72228.4-a |
| $-3 - \mathrm{i}\sqrt{3}$ | 103044.2-a2 | 103044.2-a1 | | 103044.2-a, 103044.3-a |
| $-3 + \mathrm{i}\sqrt{3}$ | 103044.3-a2 | 103044.3-a1 | | 103044.2-a, 103044.3-a |

| $\varphi$ | $\mathcal{E}_S(\tau)$ | $\mathcal{E}_S(\tau/3)$ | $N(\mathfrak{p})$ | form labels |
|---|---|---|---|---|
| continued | | | | |
| $-2 - 2\mathrm{i}\sqrt{3}$ | **46228.2-a2** | **46228.2-a1** | | **46228.2-a, 46228.7-a** |
| $-2 - \mathrm{i}\sqrt{3}$ | **34972.3-a2** | **34972.3-a1** | | **34972.2-a, 34972.3-a** |
| $-2 + \mathrm{i}\sqrt{3}$ | **34972.2-a2** | **34972.2-a1** | | **34972.2-a, 34972.3-a** |
| $-2 + 2\mathrm{i}\sqrt{3}$ | **46228.7-a1** | **46228.7-a2** | | **46228.2-a, 46228.7-a** |
| $-1 - 2\mathrm{i}\sqrt{3}$ | **30628.4-b2** | **30628.4-b1** | | **30628.4-b, 30628.5-b** |
| $-1 - \mathrm{i}\sqrt{3}$ | **11452.3-a2** | **11452.3-a1** | | **11452.2-a, 11452.3-a** |
| $-1 + \mathrm{i}\sqrt{3}$ | **11452.2-a1** | **11452.2-a2** | | **11452.2-a, 11452.3-a** |
| $-1 + 2\mathrm{i}\sqrt{3}$ | **30628.5-b1** | **30628.5-b2** | | **30628.4-b, 30628.5-b** |
| $-3\mathrm{i}\sqrt{3}$ | **59052.1-d4** | **59052.1-d3** | | **59052.1-d, 59052.8-d** |
| $-2\mathrm{i}\sqrt{3}$ | **104988.4-a2** | **104988.4-a1** | | **104988.1-a, 104988.4-a** |
| $-\mathrm{i}\sqrt{3}$ | **6564.1-a1** | **6564.1-a2** | | **6564.1-a, 6564.2-a** |
| $\mathrm{i}\sqrt{3}$ | **6564.2-a1** | **6564.2-a2** | | **6564.1-a, 6564.2-a** |
| $2\mathrm{i}\sqrt{3}$ | **104988.1-a1** | **104988.1-a2** | | **104988.1-a, 104988.4-a** |
| $3\mathrm{i}\sqrt{3}$ | **59052.8-d1** | **59052.8-d2** | | **59052.1-d, 59052.8-d** |
| $1 - 4\mathrm{i}\sqrt{3}$ | **15652.4-a2** | **15652.4-a1** | | **15652.4-a, 15652.5-a** |
| $1 - 2\mathrm{i}\sqrt{3}$ | **123916.4-a2** | **123916.4-a1** | | **123916.1-a, 123916.4-a** |
| $1 - \mathrm{i}\sqrt{3}$ | **11884.1-a1** | **11884.1-a2** | | **11884.1-a, 11884.2-a** |
| $1 + \mathrm{i}\sqrt{3}$ | **11884.2-a1** | **11884.2-a2** | | **11884.1-a, 11884.2-a** |
| $1 + 2\mathrm{i}\sqrt{3}$ | **123916.1-a1** | **123916.1-a2** | | **123916.1-a, 123916.4-a** |
| $1 + 4\mathrm{i}\sqrt{3}$ | **15652.5-a1** | **15652.5-a2** | | **15652.4-a, 15652.5-a** |
| $2 - 2\mathrm{i}\sqrt{3}$ | **47092.4-b1** | **47092.4-b2** | | **47092.1-b, 47092.4-b** |
| $2 - \mathrm{i}\sqrt{3}$ | **36484.1-a1** | **36484.1-a2** | | **36484.1-a, 36484.4-a** |
| $2 + \mathrm{i}\sqrt{3}$ | **36484.4-a1** | **36484.4-a2** | | **36484.1-a, 36484.4-a** |
| $2 + 2\mathrm{i}\sqrt{3}$ | **47092.1-b1** | **47092.1-b2** | | **47092.1-b, 47092.4-b** |
| $3 - 6\mathrm{i}\sqrt{3}$ | **119028.7-d3** | **119028.7-d2** | | **119028.2-d, 119028.7-d** |
| $3 - 2\mathrm{i}\sqrt{3}$ | **81228.3-a1** | **81228.3-a2** | | **81228.2-a, 81228.3-a** |
| $3 - \mathrm{i}\sqrt{3}$ | **106932.3-b1** | **106932.3-b2** | | **106932.3-b, 106932.6-b** |
| $3 + \mathrm{i}\sqrt{3}$ | **106932.6-b1** | **106932.6-b2** | | **106932.3-b, 106932.6-b** |
| $3 + 2\mathrm{i}\sqrt{3}$ | **81228.2-a1** | **81228.2-a2** | | **81228.2-a, 81228.3-a** |
| $3 + 6\mathrm{i}\sqrt{3}$ | **119028.2-d2** | **119028.2-d3** | | **119028.2-d, 119028.7-d** |

| $\varphi$ | $\mathcal{E}_S(\tau)$ | $\mathcal{E}_S(\tau/3)$ | $N(\mathfrak{p})$ | form labels |
|---|---|---|---|---|
| continued | | | | |
| $4 - 2\mathrm{i}\sqrt{3}$ | **82516.4-b1** | **82516.4-b2** | | **82516.3-b, 82516.4-b** |
| $4 + 2\mathrm{i}\sqrt{3}$ | **82516.3-b1** | **82516.3-b2** | | **82516.3-b, 82516.4-b** |
| $5 - 2\mathrm{i}\sqrt{3}$ | **144004.3-a1** | **144004.3-a2** | | **144004.3-a, 144004.6-a** |
| $5 + 2\mathrm{i}\sqrt{3}$ | **144004.6-a1** | **144004.6-a2** | | **144004.3-a, 144004.6-a** |
| $6 - 6\mathrm{i}\sqrt{3}$ | **25788.1-a1** | **25788.1-a2** | | **25788.1-a, 25788.4-a** |
| $6 + 6\mathrm{i}\sqrt{3}$ | **25788.4-a1** | **25788.4-a2** | | **25788.1-a, 25788.4-a** |
| $-3 - \mathrm{i}\sqrt{7}$ | **46012.4-a2** | **46012.4-a1** | 37 | **46012.4-a** |
| $-2 - 6\mathrm{i}\sqrt{7}$ | **10508.8-a1** | **10508.8-a2** | 37 | **10508.5-a, 10508.8-a** |
| $-2 + \mathrm{i}\sqrt{7}$ | **43516.13-b2** | **43516.13-b1** | 29 | **43516.13-b** |
| $-2 + 6\mathrm{i}\sqrt{7}$ | **10508.5-a2** | **10508.5-a1** | 37 | **10508.5-a, 10508.8-a** |
| $1 - \mathrm{i}\sqrt{7}$ | **812.3-b1** | **812.3-b2** | 37 | **812.3-b, 812.4-b** |
| $1 + \mathrm{i}\sqrt{7}$ | **812.4-b1** | **812.4-b2** | 37 | **812.3-b, 812.4-b** |

## D. Data for the $\mathbb{Z}_2$ Symmetric Mirror Split Quintic

### D.1. Zeta function numerators

Table 18: *The numerators of the local zeta functions of non-singular $\mathbb{Z}_2$ symmetric mirror split quintics without apparent singularities for primes $7 \leq p \leq 47$.*

| | | | $p = 7$ |
|---|---|---|---|
| $\varphi$ | smooth/sing. | singularity | $R(T)$ |
| 1 | smooth* | | |
| 2 | singular | $-\frac{1}{32}$ | |
| 3 | smooth* | | |
| 4 | smooth | | $\left(p^3T^2 + 2pT + 1\right)\left(p^6T^4 - 20p^3T^3 + 40pT^2 - 20T + 1\right)$ |
| 5 | smooth | | $\left(p^3T^2 + 2pT + 1\right)\left(p^6T^4 + 10p^3T^3 + 10p^2T^2 + 10T + 1\right)$ |
| 6 | smooth | | $\left(p^3T^2 + 26T + 1\right)\left(p^3T^2 + 2pT + 1\right)^2$ |

| | | | $p = 11$ |
|---|---|---|---|
| $\varphi$ | smooth/sing. | singularity | $R(T)$ |
| 1 | singular | $\frac{1}{2}\left(11 \pm 5\sqrt{5}\right)$ | |
| 2 | smooth | | $\left(p^3T^2 - 2pT + 1\right)\left(p^6T^4 + 6p^3T^3 + 6p^2T^2 + 6T + 1\right)$ |
| 3 | smooth | | $\left(p^3T^2 - 2pT + 1\right)\left(p^6T^4 + 46p^3T^3 + 226pT^2 + 46T + 1\right)$ |
| 4 | smooth | | $\left(p^3T^2 + 3pT + 1\right)\left(p^6T^4 + 51p^3T^3 + 226pT^2 + 51T + 1\right)$ |
| 5 | smooth | | $\left(p^3T^2 - 2pT + 1\right)\left(p^6T^4 - 4p^4T^3 + 166pT^2 - 4pT + 1\right)$ |
| 6 | smooth | | $\left(p^3T^2 + 3pT + 1\right)\left(p^6T^4 + 41p^3T^3 + 86pT^2 + 41T + 1\right)$ |
| 7 | smooth | | $\left(p^3T^2 - 2pT + 1\right)\left(p^6T^4 - 4p^3T^3 - 124pT^2 - 4T + 1\right)$ |
| 8 | smooth | | $\left(p^3T^2 + 3pT + 1\right)\left(p^6T^4 + p^3T^3 + 126pT^2 + T + 1\right)$ |
| 9 | smooth | | $\left(p^3T^2 + 3pT + 1\right)\left(p^6T^4 - 9p^3T^3 - 64pT^2 - 9T + 1\right)$ |
| 10 | singular | $-\frac{1}{32}$ | |

| | | | $p = 13$ |
|---|---|---|---|
| $\varphi$ | smooth/sing. | singularity | $R(T)$ |
| 1 | smooth | | $\left(p^3T^2 - 4pT + 1\right)\left(p^6T^4 + 50p^3T^3 + 320pT^2 + 50T + 1\right)$ |
| 2 | smooth | | $\left(p^3T^2 + pT + 1\right)\left(p^6T^4 + 75p^3T^3 + 380pT^2 + 75T + 1\right)$ |
| 3 | smooth | | $\left(p^3T^2 + pT + 1\right)\left(p^6T^4 - 15p^3T^3 - 140pT^2 - 15T + 1\right)$ |
| 4 | smooth | | $\left(p^3T^2 + pT + 1\right)\left(p^6T^4 - 45p^3T^3 + 220pT^2 - 45T + 1\right)$ |
| 5 | smooth | | $\left(p^3T^2 + 6pT + 1\right)\left(p^6T^4 - 110pT^2 + 1\right)$ |
| 6 | smooth* | | |
| 7 | smooth | | $\left(p^3T^2 - 4pT + 1\right)\left(p^6T^4 + 20p^3T^3 + 80pT^2 + 20T + 1\right)$ |
| | | | *Continued on the following page* |

| $p = 13$, continued | | | |
|---|---|---|---|
| $\varphi$ | smooth/sing. | singularity | $R(T)$ |
| 8 | smooth | | $\left(p^3T^2 - 4pT + 1\right)\left(p^6T^4 + 10p^3T^3 - 150pT^2 + 10T + 1\right)$ |
| 9 | smooth* | | |
| 10 | smooth | | $\left(p^3T^2 + 6pT + 1\right)\left(p^6T^4 - 20p^3T^3 + 30pT^2 - 20T + 1\right)$ |
| 11 | singular | $-\frac{1}{32}$ | |
| 12 | smooth | | $\left(p^3T^2 + 32T + 1\right)\left(p^3T^2 - 4pT + 1\right)^2$ |

| $p = 17$ | | | |
|---|---|---|---|
| $\varphi$ | smooth/sing. | singularity | $R(T)$ |
| 1 | smooth | | $\left(p^3T^2 + 2pT + 1\right)\left(p^6T^4 + 30p^3T^3 + 500pT^2 + 30T + 1\right)$ |
| 2 | smooth | | $\left(p^3T^2 - 3pT + 1\right)\left(p^6T^4 - 15p^3T^3 + 100pT^2 - 15T + 1\right)$ |
| 3 | smooth | | $\left(p^3T^2 + 2pT + 1\right)\left(p^6T^4 + 100p^3T^3 + 620pT^2 + 100T + 1\right)$ |
| 4 | smooth* | | |
| 5 | smooth | | $\left(p^3T^2 + 7pT + 1\right)\left(p^6T^4 + 35p^3T^3 + 110pT^2 + 35T + 1\right)$ |
| 6 | smooth | | $\left(p^3T^2 - 3pT + 1\right)\left(p^6T^4 + 95p^3T^3 + 360pT^2 + 95T + 1\right)$ |
| 7 | smooth | | $\left(p^3T^2 + 2pT + 1\right)\left(p^6T^4 + 70pT^2 + 1\right)$ |
| 8 | singular | $-\frac{1}{32}$ | |
| 9 | smooth | | $\left(p^3T^2 - 3pT + 1\right)\left(p^6T^4 - 35p^3T^3 + 180pT^2 - 35T + 1\right)$ |
| 10 | smooth | | $\left(p^3T^2 + 7pT + 1\right)\left(p^6T^4 - 35p^3T^3 - 60pT^2 - 35T + 1\right)$ |
| 11 | smooth | | $\left(p^3T^2 + 2pT + 1\right)\left(p^6T^4 + 120p^3T^3 + 740pT^2 + 120T + 1\right)$ |
| 12 | smooth | | $\left(p^3T^2 + 2pT + 1\right)\left(p^6T^4 - 30pT^2 + 1\right)$ |
| 13 | smooth | | $\left(p^3T^2 + 2pT + 1\right)\left(p^6T^4 - 30p^3T^3 + 90pT^2 - 30T + 1\right)$ |
| 14 | smooth* | | |
| 15 | smooth | | $\left(p^3T^2 - 3pT + 1\right)\left(p^6T^4 + 35p^3T^3 - 100pT^2 + 35T + 1\right)$ |
| 16 | smooth | | $\left(p^3T^2 - 74T + 1\right)\left(p^3T^2 + 2pT + 1\right)^2$ |

| $p = 19$ | | | |
|---|---|---|---|
| $\varphi$ | smooth/sing. | singularity | $R(T)$ |
| 1 | smooth | | $\left(p^3T^2 + 1\right)\left(p^6T^4 - 60p^3T^3 + 522pT^2 - 60T + 1\right)$ |
| 2 | singular | $\frac{1}{2}\left(11 \pm 5\sqrt{5}\right)$ | |
| 3 | singular | $-\frac{1}{32}$ | |
| 4 | smooth | | $\left(p^3T^2 + 1\right)\left(p^6T^4 + 80p^3T^3 + 622pT^2 + 80T + 1\right)$ |
| 5 | smooth | | $\left(p^3T^2 + 5pT + 1\right)\left(p^6T^4 - 105p^3T^3 + 672pT^2 - 105T + 1\right)$ |
| 6 | smooth | | $\left(p^3T^2 + 5pT + 1\right)\left(p^6T^4 + 75p^3T^3 + 172pT^2 + 75T + 1\right)$ |
| 7 | smooth | | $\left(p^3T^2 - 5pT + 1\right)\left(p^6T^4 - 25p^3T^3 - 178pT^2 - 25T + 1\right)$ |

| $p = 19$, continued | | | |
|---|---|---|---|
| $\varphi$ | smooth/sing. | singularity | $R(T)$ |
| 8 | smooth | | $\left(p^3T^2 - 5pT + 1\right)\left(p^6T^4 - 45p^3T^3 + 622pT^2 - 45T + 1\right)$ |
| 9 | singular | $\frac{1}{2}\left(11 \pm 5\sqrt{5}\right)$ | |
| 10 | smooth | | $\left(p^3T^2 - 5pT + 1\right)\left(p^6T^4 + 5p^3T^3 + 422pT^2 + 5T + 1\right)$ |
| 11 | smooth | | $\left(p^3T^2 + 1\right)\left(p^6T^4 + 140p^3T^3 + 922pT^2 + 140T + 1\right)$ |
| 12 | smooth | | $\left(p^3T^2 + 1\right)\left(p^6T^4 + 40p^3T^3 + 122pT^2 + 40T + 1\right)$ |
| 13 | smooth | | $\left(p^3T^2 + 1\right)\left(p^6T^4 + 170p^3T^3 + 972pT^2 + 170T + 1\right)$ |
| 14 | smooth* | | |
| 15 | smooth* | | |
| 16 | smooth | | $\left(p^3T^2 + 1\right)\left(p^6T^4 - 80p^3T^3 + 322pT^2 - 80T + 1\right)$ |
| 17 | smooth | | $\left(p^3T^2 - 5pT + 1\right)\left(p^6T^4 - 5p^3T^3 - 12p^2T^2 - 5T + 1\right)$ |
| 18 | smooth | | $\left(p^3T^2 + 1\right)^2\left(p^3T^2 + 60T + 1\right)$ |

| $p = 23$ | | | |
|---|---|---|---|
| $\varphi$ | smooth/sing. | singularity | $R(T)$ |
| 1 | smooth | | $\left(p^3T^2 + pT + 1\right)\left(p^6T^4 - 25p^3T^3 - 50pT^2 - 25T + 1\right)$ |
| 2 | smooth | | $\left(p^3T^2 + pT + 1\right)\left(p^6T^4 - 15p^3T^3 - 470pT^2 - 15T + 1\right)$ |
| 3 | smooth | | $\left(p^3T^2 + 6pT + 1\right)\left(p^6T^4 + 80p^3T^3 + 500pT^2 + 80T + 1\right)$ |
| 4 | smooth | | $\left(p^3T^2 - 4pT + 1\right)\left(p^6T^4 - 50p^3T^3 + 340pT^2 - 50T + 1\right)$ |
| 5 | smooth | | $\left(p^3T^2 - 4pT + 1\right)\left(p^6T^4 + 40p^3T^3 + 410pT^2 + 40T + 1\right)$ |
| 6 | smooth | | $\left(p^3T^2 - 4pT + 1\right)\left(p^6T^4 + 180p^3T^3 + 930pT^2 + 180T + 1\right)$ |
| 7 | smooth | | $\left(p^3T^2 - 9pT + 1\right)\left(p^6T^4 + 65p^3T^3 + 200pT^2 + 65T + 1\right)$ |
| 8 | smooth | | $\left(p^3T^2 + 6pT + 1\right)\left(p^6T^4 - 340pT^2 + 1\right)$ |
| 9 | smooth | | $\left(p^3T^2 - 4pT + 1\right)\left(p^6T^4 + 80p^3T^3 + 730pT^2 + 80T + 1\right)$ |
| 10 | smooth | | $\left(p^3T^2 + 6pT + 1\right)\left(p^6T^4 + 120p^3T^3 + 1070pT^2 + 120T + 1\right)$ |
| 11 | smooth | | $\left(p^3T^2 + pT + 1\right)\left(p^6T^4 - 45p^3T^3 - 260pT^2 - 45T + 1\right)$ |
| 12 | smooth | | $\left(p^3T^2 + 6pT + 1\right)\left(p^6T^4 + 140p^3T^3 + 930pT^2 + 140T + 1\right)$ |
| 13 | smooth* | | |
| 14 | smooth | | $\left(p^3T^2 - 4pT + 1\right)\left(p^6T^4 - 160p^3T^3 + 710pT^2 - 160T + 1\right)$ |
| 15 | smooth | | $\left(p^3T^2 + 68T + 1\right)\left(p^3T^2 - 6pT + 1\right)\left(p^3T^2 + 6pT + 1\right)$ |
| 16 | smooth | | $\left(p^3T^2 + 6pT + 1\right)\left(p^6T^4 - 60p^3T^3 + 730pT^2 - 60T + 1\right)$ |
| 17 | smooth | | $\left(p^3T^2 - 4pT + 1\right)\left(p^6T^4 + 80p^3T^3 + 930pT^2 + 80T + 1\right)$ |
| 18 | singular | $-\frac{1}{32}$ | |
| 19 | smooth | | $\left(p^3T^2 - 4pT + 1\right)\left(p^6T^4 - 60p^3T^3 + 1060pT^2 - 60T + 1\right)$ |
| 20 | smooth | | $\left(p^3T^2 + 18T + 1\right)\left(p^3T^2 - 6pT + 1\right)\left(p^3T^2 + 6pT + 1\right)$ |
| 21 | smooth | | $\left(p^3T^2 + 6pT + 1\right)\left(p^6T^4 + 10p^3T^3 + 740pT^2 + 10T + 1\right)$ |

| $p = 23$, continued | | | |
|---|---|---|---|
| $\varphi$ | smooth/sing. | singularity | $R(T)$ |
| 22 | smooth | | $\left(p^3T^2 + 182T + 1\right)\left(p^3T^2 + pT + 1\right)^2$ |

| $p = 29$ | | | |
|---|---|---|---|
| $\varphi$ | smooth/sing. | singularity | $R(T)$ |
| 1 | smooth | | $\left(p^3T^2 + 1\right)\left(p^6T^4 + 80p^3T^3 + 1082pT^2 + 80T + 1\right)$ |
| 2 | smooth | | $\left(p^3T^2 + 5pT + 1\right)\left(p^6T^4 - 65p^3T^3 + 732pT^2 - 65T + 1\right)$ |
| 3 | smooth | | $\left(p^3T^2 - 10pT + 1\right)\left(p^6T^4 - 70p^3T^3 + 932pT^2 - 70T + 1\right)$ |
| 4 | singular | $\frac{1}{2}\left(11 \pm 5\sqrt{5}\right)$ | |
| 5 | smooth* | | |
| 6 | smooth | | $\left(p^3T^2 + 1\right)\left(p^6T^4 + 270p^3T^3 + 1432pT^2 + 270T + 1\right)$ |
| 7 | singular | $\frac{1}{2}\left(11 \pm 5\sqrt{5}\right)$ | |
| 8 | smooth | | $\left(p^3T^2 + 1\right)\left(p^6T^4 + 100p^3T^3 + 1582pT^2 + 100T + 1\right)$ |
| 9 | smooth | | $\left(p^3T^2 + 10pT + 1\right)\left(p^6T^4 + 160p^3T^3 + 1532pT^2 + 160T + 1\right)$ |
| 10 | singular | $-\frac{1}{32}$ | |
| 11 | smooth | | $\left(p^3T^2 + 1\right)\left(p^6T^4 - 120p^3T^3 + 982pT^2 - 120T + 1\right)$ |
| 12 | smooth | | $\left(p^3T^2 + 1\right)\left(p^3T^2 + 200T + 1\right)\left(p^3T^2 - 10pT + 1\right)$ |
| 13 | smooth | | $\left(p^3T^2 - 5pT + 1\right)\left(p^6T^4 + 75p^3T^3 - 568pT^2 + 75T + 1\right)$ |
| 14 | smooth | | $\left(p^3T^2 + 5pT + 1\right)\left(p^6T^4 + 95p^3T^3 + 1232pT^2 + 95T + 1\right)$ |
| 15 | smooth | | $\left(p^3T^2 + 1\right)\left(p^6T^4 + 240p^3T^3 + 1832pT^2 + 240T + 1\right)$ |
| 16 | smooth | | $\left(p^3T^2 + 10pT + 1\right)\left(p^6T^4 + 190p^3T^3 + 882pT^2 + 190T + 1\right)$ |
| 17 | smooth | | $\left(p^3T^2 + 10pT + 1\right)\left(p^6T^4 - 170p^3T^3 + 1582pT^2 - 170T + 1\right)$ |
| 18 | smooth | | $\left(p^3T^2 + 1\right)\left(p^6T^4 + 50p^3T^3 + 82pT^2 + 50T + 1\right)$ |
| 19 | smooth | | $\left(p^3T^2 - 10pT + 1\right)\left(p^6T^4 - 10p^4T^3 + 2332pT^2 - 10pT + 1\right)$ |
| 20 | smooth | | $\left(p^3T^2 - 5pT + 1\right)\left(p^6T^4 + 295p^3T^3 + 2132pT^2 + 295T + 1\right)$ |
| 21 | smooth | | $\left(p^3T^2 + 1\right)\left(p^6T^4 + 40p^3T^3 + 182pT^2 + 40T + 1\right)$ |
| 22 | smooth | | $\left(p^3T^2 + 5pT + 1\right)\left(p^6T^4 + 65p^3T^3 + 1032pT^2 + 65T + 1\right)$ |
| 23 | smooth | | $\left(p^3T^2 + 1\right)\left(p^6T^4 + 130p^3T^3 - 268pT^2 + 130T + 1\right)$ |
| 24 | smooth | | $\left(p^3T^2 + 1\right)\left(p^6T^4 - 320p^3T^3 + 2282pT^2 - 320T + 1\right)$ |
| 25 | smooth | | $\left(p^3T^2 + 5pT + 1\right)\left(p^6T^4 - 125p^3T^3 + 432pT^2 - 125T + 1\right)$ |
| 26 | smooth* | | |
| 27 | smooth | | $\left(p^3T^2 + 1\right)\left(p^6T^4 + 40p^3T^3 + 782pT^2 + 40T + 1\right)$ |
| 28 | smooth | | $\left(p^3T^2 + 1\right)^2\left(p^3T^2 + 90T + 1\right)$ |

| | | | $p = 31$ |
|---|---|---|---|
| $\varphi$ | smooth/sing. | singularity | $R(T)$ |
| 1 | singular | $-\frac{1}{32}$ | |
| 2 | smooth | | $\left(p^3T^2 + 8pT + 1\right)\left(p^6T^4 - 34p^3T^3 + 1156pT^2 - 34T + 1\right)$ |
| 3 | smooth | | $\left(p^3T^2 + 3pT + 1\right)\left(p^6T^4 + 211p^3T^3 + 556pT^2 + 211T + 1\right)$ |
| 4 | smooth* | | |
| 5 | singular | $\frac{1}{2}\left(11 \pm 5\sqrt{5}\right)$ | |
| 6 | singular | $\frac{1}{2}\left(11 \pm 5\sqrt{5}\right)$ | |
| 7 | smooth | | $\left(p^3T^2 + 8pT + 1\right)\left(p^6T^4 + 96p^3T^3 - 474pT^2 + 96T + 1\right)$ |
| 8 | smooth | | $\left(p^3T^2 - 2pT + 1\right)\left(p^6T^4 + 216p^3T^3 + 596pT^2 + 216T + 1\right)$ |
| 9 | smooth | | $\left(p^3T^2 + 48T + 1\right)\left(p^3T^2 - 2pT + 1\right)^2$ |
| 10 | smooth | | $\left(p^3T^2 + 3pT + 1\right)\left(p^6T^4 + 71p^3T^3 + 46pT^2 + 71T + 1\right)$ |
| 11 | smooth | | $\left(p^3T^2 - 7pT + 1\right)\left(p^6T^4 + 151p^3T^3 + 1756pT^2 + 151T + 1\right)$ |
| 12 | smooth | | $\left(p^3T^2 + 8pT + 1\right)\left(p^6T^4 - 4p^3T^3 - 474pT^2 - 4T + 1\right)$ |
| 13 | smooth | | $\left(p^3T^2 - 2pT + 1\right)\left(p^6T^4 - 334p^3T^3 + 2746pT^2 - 334T + 1\right)$ |
| 14 | smooth* | | |
| 15 | smooth | | $\left(p^3T^2 + 8pT + 1\right)\left(p^6T^4 + 106p^3T^3 + 1066pT^2 + 106T + 1\right)$ |
| 16 | smooth | | $\left(p^3T^2 - 2pT + 1\right)\left(p^6T^4 + 36p^3T^3 - 174pT^2 + 36T + 1\right)$ |
| 17 | smooth | | $\left(p^3T^2 + 8pT + 1\right)\left(p^6T^4 + 276p^3T^3 + 1546pT^2 + 276T + 1\right)$ |
| 18 | smooth | | $\left(p^3T^2 + 8pT + 1\right)\left(p^6T^4 - 184p^3T^3 + 406pT^2 - 184T + 1\right)$ |
| 19 | smooth | | $\left(p^3T^2 - 2pT + 1\right)\left(p^6T^4 + 96p^3T^3 + 36p^2T^2 + 96T + 1\right)$ |
| 20 | smooth | | $\left(p^3T^2 + 8pT + 1\right)\left(p^6T^4 + 426p^3T^3 + 2896pT^2 + 426T + 1\right)$ |
| 21 | smooth | | $\left(p^3T^2 - 7pT + 1\right)\left(p^6T^4 + 51p^3T^3 - 144pT^2 + 51T + 1\right)$ |
| 22 | smooth | | $\left(p^3T^2 + 8pT + 1\right)\left(p^6T^4 + 176p^3T^3 + 546pT^2 + 176T + 1\right)$ |
| 23 | smooth | | $\left(p^3T^2 + 3pT + 1\right)\left(p^6T^4 - 389p^3T^3 + 3006pT^2 - 389T + 1\right)$ |
| 24 | smooth | | $\left(p^3T^2 - 2pT + 1\right)\left(p^6T^4 - 34p^3T^3 + 1146pT^2 - 34T + 1\right)$ |
| 25 | smooth | | $\left(p^3T^2 - 7pT + 1\right)\left(p^6T^4 + 141p^3T^3 + 666pT^2 + 141T + 1\right)$ |
| 26 | smooth | | $\left(p^3T^2 - 7pT + 1\right)\left(p^6T^4 - 109p^3T^3 - 84pT^2 - 109T + 1\right)$ |
| 27 | smooth | | $\left(p^3T^2 - 2pT + 1\right)\left(p^6T^4 + 66p^3T^3 + 346pT^2 + 66T + 1\right)$ |
| 28 | smooth | | $\left(p^3T^2 - 7pT + 1\right)\left(p^6T^4 + 61p^3T^3 + 546pT^2 + 61T + 1\right)$ |
| 29 | smooth | | $\left(p^3T^2 - 2pT + 1\right)\left(p^6T^4 - 34p^3T^3 + 196pT^2 - 34T + 1\right)$ |
| 30 | smooth | | $\left(p^3T^2 + 8T + 1\right)\left(p^3T^2 - 7pT + 1\right)^2$ |

| | | | $p = 37$ |
|---|---|---|---|
| $\varphi$ | smooth/sing. | singularity | $R(T)$ |
| 1 | smooth | | $\left(p^3T^2 - 3pT + 1\right)\left(p^6T^4 - 205p^3T^3 + 230pT^2 - 205T + 1\right)$ |

| $\varphi$ | smooth/sing. | singularity | $R(T)$ |
|---|---|---|---|
| | | | $p=37$, *continued* |
| 2 | smooth | | $\left(p^3T^2 + 2pT + 1\right)\left(p^6T^4 - 20p^3T^3 + 1270pT^2 - 20T + 1\right)$ |
| 3 | smooth | | $\left(p^3T^2 + 2pT + 1\right)\left(p^6T^4 + 20p^3T^3 + 410pT^2 + 20T + 1\right)$ |
| 4 | smooth | | $\left(p^3T^2 - 8pT + 1\right)\left(p^6T^4 + 300p^3T^3 + 1930pT^2 + 300T + 1\right)$ |
| 5 | smooth | | $\left(p^3T^2 + 7pT + 1\right)\left(p^6T^4 + 315p^3T^3 + 3060pT^2 + 315T + 1\right)$ |
| 6 | smooth | | $\left(p^3T^2 + 2pT + 1\right)\left(p^6T^4 - 210p^3T^3 + 1530pT^2 - 210T + 1\right)$ |
| 7 | smooth | | $\left(p^3T^2 - 8pT + 1\right)\left(p^6T^4 - 270p^3T^3 + 1910pT^2 - 270T + 1\right)$ |
| 8 | smooth* | | |
| 9 | smooth | | $\left(p^3T^2 - 8pT + 1\right)\left(p^6T^4 + 510p^3T^3 + 3890pT^2 + 510T + 1\right)$ |
| 10 | smooth | | $\left(p^3T^2 - 3pT + 1\right)\left(p^6T^4 + 585p^3T^3 + 4770pT^2 + 585T + 1\right)$ |
| 11 | smooth | | $\left(p^3T^2 - 3pT + 1\right)\left(p^6T^4 - 255p^3T^3 + 2130pT^2 - 255T + 1\right)$ |
| 12 | smooth | | $\left(p^3T^2 + 2pT + 1\right)\left(p^6T^4 + 140p^3T^3 + 1930pT^2 + 140T + 1\right)$ |
| 13 | smooth | | $\left(p^3T^2 - 8pT + 1\right)\left(p^6T^4 - 120p^3T^3 + 2110pT^2 - 120T + 1\right)$ |
| 14 | smooth | | $\left(p^3T^2 + 2pT + 1\right)\left(p^6T^4 + 90p^3T^3 + 1330pT^2 + 90T + 1\right)$ |
| 15 | smooth | | $\left(p^3T^2 - 3pT + 1\right)\left(p^6T^4 + 75p^3T^3 - 1440pT^2 + 75T + 1\right)$ |
| 16 | smooth | | $\left(p^3T^2 + 2pT + 1\right)\left(p^6T^4 + 30p^3T^3 + 470pT^2 + 30T + 1\right)$ |
| 17 | smooth | | $\left(p^3T^2 - 8pT + 1\right)\left(p^6T^4 - 70p^3T^3 + 2410pT^2 - 70T + 1\right)$ |
| 18 | smooth | | $\left(p^3T^2 + 2pT + 1\right)\left(p^6T^4 + 110p^3T^3 + 1550pT^2 + 110T + 1\right)$ |
| 19 | smooth | | $\left(p^3T^2 + 2pT + 1\right)\left(p^6T^4 + 210p^3T^3 + 1700pT^2 + 210T + 1\right)$ |
| 20 | smooth | | $\left(p^3T^2 + 7pT + 1\right)\left(p^6T^4 + 65p^3T^3 + 660pT^2 + 65T + 1\right)$ |
| 21 | smooth | | $\left(p^3T^2 - 8pT + 1\right)\left(p^6T^4 + 180p^3T^3 + 810pT^2 + 180T + 1\right)$ |
| 22 | singular | $-\frac{1}{32}$ | |
| 23 | smooth | | $\left(p^3T^2 + 2pT + 1\right)\left(p^6T^4 + 60p^3T^3 - 250pT^2 + 60T + 1\right)$ |
| 24 | smooth | | $\left(p^3T^2 + 7pT + 1\right)\left(p^6T^4 - 175p^3T^3 + 620pT^2 - 175T + 1\right)$ |
| 25 | smooth | | $\left(p^3T^2 + 7pT + 1\right)\left(p^6T^4 + 155p^3T^3 + 1250pT^2 + 155T + 1\right)$ |
| 26 | smooth | | $\left(p^3T^2 + 2pT + 1\right)\left(p^6T^4 + 340p^3T^3 + 2830pT^2 + 340T + 1\right)$ |
| 27 | smooth | | $\left(p^3T^2 + 2pT + 1\right)\left(p^6T^4 - 20p^3T^3 - 330pT^2 - 20T + 1\right)$ |
| 28 | smooth | | $\left(p^3T^2 - 8pT + 1\right)\left(p^6T^4 + 50p^3T^3 + 930pT^2 + 50T + 1\right)$ |
| 29 | smooth* | | |
| 30 | smooth | | $\left(p^3T^2 + 2pT + 1\right)\left(p^6T^4 - 230p^3T^3 + 310pT^2 - 230T + 1\right)$ |
| 31 | smooth | | $\left(p^3T^2 + 12pT + 1\right)\left(p^6T^4 + 120p^3T^3 + 20pT^2 + 120T + 1\right)$ |
| 32 | smooth | | $\left(p^3T^2 - 3pT + 1\right)\left(p^6T^4 - 135p^3T^3 + 200pT^2 - 135T + 1\right)$ |
| 33 | smooth | | $\left(p^3T^2 - 8pT + 1\right)\left(p^6T^4 - 310p^3T^3 + 2320pT^2 - 310T + 1\right)$ |
| 34 | smooth | | $\left(p^3T^2 + 7pT + 1\right)\left(p^6T^4 + 195p^3T^3 + 890pT^2 + 195T + 1\right)$ |
| 35 | smooth | | $\left(p^3T^2 + 2pT + 1\right)\left(p^6T^4 - 160p^3T^3 + 330pT^2 - 160T + 1\right)$ |
| 36 | smooth | | $\left(p^3T^2 + 66T + 1\right)\left(p^3T^2 - 3pT + 1\right)^2$ |

| | | $p = 41$ | |
|---|---|---|---|
| $\varphi$ | smooth/sing. | singularity | $R(T)$ |
| 1 | smooth | | $\left(p^3T^2 + 8pT + 1\right)\left(p^6T^4 + 6p^3T^3 - 1714pT^2 + 6T + 1\right)$ |
| 2 | smooth | | $\left(p^3T^2 + 8pT + 1\right)\left(p^6T^4 - 104p^3T^3 + 1046pT^2 - 104T + 1\right)$ |
| 3 | smooth | | $\left(p^3T^2 + 8pT + 1\right)\left(p^6T^4 - 194p^3T^3 + 1686pT^2 - 194T + 1\right)$ |
| 4 | smooth | | $\left(p^3T^2 - 2pT + 1\right)\left(p^6T^4 - 24p^3T^3 + 506pT^2 - 24T + 1\right)$ |
| 5 | smooth | | $\left(p^3T^2 + 3pT + 1\right)\left(p^6T^4 + 341p^3T^3 + 1696pT^2 + 341T + 1\right)$ |
| 6 | smooth | | $\left(p^3T^2 - 2pT + 1\right)\left(p^6T^4 + 86p^3T^3 + 946pT^2 + 86T + 1\right)$ |
| 7 | smooth | | $\left(p^3T^2 + 3pT + 1\right)\left(p^6T^4 + 61p^3T^3 + 3276pT^2 + 61T + 1\right)$ |
| 8 | smooth | | $\left(p^3T^2 + 3pT + 1\right)\left(p^6T^4 - 619p^3T^3 + 4656pT^2 - 619T + 1\right)$ |
| 9 | singular | $-\frac{1}{32}$ | |
| 10 | smooth | | $\left(p^3T^2 - 2pT + 1\right)\left(p^6T^4 - 244p^3T^3 + 1026pT^2 - 244T + 1\right)$ |
| 11 | smooth | | $\left(p^3T^2 + 8pT + 1\right)\left(p^6T^4 - 54p^3T^3 - 654pT^2 - 54T + 1\right)$ |
| 12 | smooth | | $\left(p^3T^2 - 2pT + 1\right)\left(p^6T^4 - 24p^3T^3 - 1894pT^2 - 24T + 1\right)$ |
| 13 | smooth | | $\left(p^3T^2 - 12pT + 1\right)\left(p^6T^4 + 126p^3T^3 + 1946pT^2 + 126T + 1\right)$ |
| 14 | singular | $\frac{1}{2}\left(11 \pm 5\sqrt{5}\right)$ | |
| 15 | smooth | | $\left(p^3T^2 - 2pT + 1\right)\left(p^6T^4 + 396p^3T^3 + 46p^2T^2 + 396T + 1\right)$ |
| 16 | smooth* | | |
| 17 | smooth | | $\left(p^3T^2 - 2pT + 1\right)\left(p^6T^4 + 56p^3T^3 - 2274pT^2 + 56T + 1\right)$ |
| 18 | smooth | | $\left(p^3T^2 - 2pT + 1\right)\left(p^6T^4 + 456p^3T^3 + 2826pT^2 + 456T + 1\right)$ |
| 19 | smooth | | $\left(p^3T^2 - 7pT + 1\right)\left(p^6T^4 + 521p^3T^3 + 3556pT^2 + 521T + 1\right)$ |
| 20 | smooth | | $\left(p^3T^2 + 8pT + 1\right)\left(p^6T^4 + 316p^3T^3 + 2176pT^2 + 316T + 1\right)$ |
| 21 | smooth | | $\left(p^3T^2 + 3pT + 1\right)\left(p^6T^4 + 491p^3T^3 + 3896pT^2 + 491T + 1\right)$ |
| 22 | smooth | | $\left(p^3T^2 - 12pT + 1\right)\left(p^6T^4 + 26p^3T^3 - 2904pT^2 + 26T + 1\right)$ |
| 23 | smooth* | | |
| 24 | smooth | | $\left(p^3T^2 + 3pT + 1\right)\left(p^6T^4 - 269p^3T^3 + 2306pT^2 - 269T + 1\right)$ |
| 25 | smooth | | $\left(p^3T^2 - 2pT + 1\right)\left(p^6T^4 + 76p^3T^3 - 694pT^2 + 76T + 1\right)$ |
| 26 | smooth | | $\left(p^3T^2 + 8pT + 1\right)\left(p^6T^4 + 66p^3T^3 + 2276pT^2 + 66T + 1\right)$ |
| 27 | smooth | | $\left(p^3T^2 + 8pT + 1\right)\left(p^6T^4 - 394p^3T^3 + 2586pT^2 - 394T + 1\right)$ |
| 28 | smooth | | $\left(p^3T^2 - 7pT + 1\right)\left(p^6T^4 + 271p^3T^3 + 1906pT^2 + 271T + 1\right)$ |
| 29 | smooth | | $\left(p^3T^2 + 3pT + 1\right)\left(p^6T^4 + p^4T^3 + 1196pT^2 + pT + 1\right)$ |
| 30 | smooth | | $\left(p^3T^2 - 2pT + 1\right)\left(p^6T^4 + 36p^3T^3 + 2246pT^2 + 36T + 1\right)$ |
| 31 | smooth | | $\left(p^3T^2 - 7pT + 1\right)\left(p^6T^4 - 409p^3T^3 + 3336pT^2 - 409T + 1\right)$ |
| 32 | smooth | | $\left(p^3T^2 - 222T + 1\right)\left(p^3T^2 + 8pT + 1\right)^2$ |
| 33 | smooth | | $\left(p^3T^2 - 12pT + 1\right)\left(p^6T^4 - 124p^3T^3 + 2496pT^2 - 124T + 1\right)$ |
| 34 | smooth | | $\left(p^3T^2 - 2pT + 1\right)\left(p^6T^4 + 396p^3T^3 + 3286pT^2 + 396T + 1\right)$ |
| 35 | smooth | | $\left(p^3T^2 + 3pT + 1\right)\left(p^6T^4 - 189p^3T^3 + 2326pT^2 - 189T + 1\right)$ |

| $\varphi$ | smooth/sing. | singularity | $R(T)$ |
|---|---|---|---|
| | | | $p = 41$, *continued* |
| 36 | smooth | | $\left(p^3 T^2 - 12pT + 1\right)\left(p^6 T^4 + 126p^3 T^3 - 854pT^2 + 126T + 1\right)$ |
| 37 | smooth | | $\left(p^3 T^2 - 7pT + 1\right)\left(p^6 T^4 - 49p^3 T^3 - 924pT^2 - 49T + 1\right)$ |
| 38 | singular | $\frac{1}{2}\left(11 \pm 5\sqrt{5}\right)$ | |
| 39 | smooth | | $\left(p^3 T^2 + 3pT + 1\right)\left(p^6 T^4 + 231p^3 T^3 + 2456pT^2 + 231T + 1\right)$ |
| 40 | smooth | | $\left(p^3 T^2 - 422T + 1\right)\left(p^3 T^2 + 8pT + 1\right)^2$ |

| $\varphi$ | smooth/sing. | singularity | $R(T)$ |
|---|---|---|---|
| | | | $p = 43$ |
| 1 | smooth | | $\left(p^3 T^2 + 6pT + 1\right)\left(p^6 T^4 - 10p^3 T^3 - 2930pT^2 - 10T + 1\right)$ |
| 2 | smooth | | $\left(p^3 T^2 - 4pT + 1\right)\left(p^6 T^4 + 1730pT^2 + 1\right)$ |
| 3 | smooth | | $\left(p^3 T^2 + pT + 1\right)\left(p^6 T^4 - 25p^3 T^3 - 210pT^2 - 25T + 1\right)$ |
| 4 | smooth | | $\left(p^3 T^2 + 11pT + 1\right)\left(p^6 T^4 - 35p^3 T^3 + 2680pT^2 - 35T + 1\right)$ |
| 5 | smooth* | | |
| 6 | smooth | | $\left(p^3 T^2 + 6pT + 1\right)\left(p^6 T^4 + 70p^3 T^3 + 1410pT^2 + 70T + 1\right)$ |
| 7 | smooth | | $\left(p^3 T^2 + 6pT + 1\right)\left(p^6 T^4 + 300p^3 T^3 + 1050pT^2 + 300T + 1\right)$ |
| 8 | smooth | | $\left(p^3 T^2 - 4pT + 1\right)\left(p^6 T^4 - 70p^3 T^3 - 930pT^2 - 70T + 1\right)$ |
| 9 | smooth | | $\left(p^3 T^2 + pT + 1\right)\left(p^6 T^4 + 45p^3 T^3 - 150pT^2 + 45T + 1\right)$ |
| 10 | smooth | | $\left(p^3 T^2 - 4pT + 1\right)\left(p^6 T^4 - 200p^3 T^3 + 930pT^2 - 200T + 1\right)$ |
| 11 | smooth | | $\left(p^3 T^2 + 6pT + 1\right)\left(p^6 T^4 + 410p^3 T^3 + 3330pT^2 + 410T + 1\right)$ |
| 12 | smooth | | $\left(p^3 T^2 - 4pT + 1\right)\left(p^6 T^4 + 60p^3 T^3 + 410pT^2 + 60T + 1\right)$ |
| 13 | smooth | | $\left(p^3 T^2 + pT + 1\right)\left(p^6 T^4 + 355p^3 T^3 + 1430pT^2 + 355T + 1\right)$ |
| 14 | smooth | | $\left(p^3 T^2 + pT + 1\right)\left(p^6 T^4 - 285p^3 T^3 + 3460pT^2 - 285T + 1\right)$ |
| 15 | smooth | | $\left(p^3 T^2 + 11pT + 1\right)\left(p^6 T^4 - 485p^3 T^3 + 4680pT^2 - 485T + 1\right)$ |
| 16 | smooth | | $\left(p^3 T^2 - 4pT + 1\right)\left(p^6 T^4 - 310p^3 T^3 + 1450pT^2 - 310T + 1\right)$ |
| 17 | smooth | | $\left(p^3 T^2 - 9pT + 1\right)\left(p^6 T^4 + 65p^3 T^3 - 210pT^2 + 65T + 1\right)$ |
| 18 | smooth | | $\left(p^3 T^2 - 4pT + 1\right)\left(p^6 T^4 + 10p^4 T^3 + 3470pT^2 + 10pT + 1\right)$ |
| 19 | smooth | | $\left(p^3 T^2 + pT + 1\right)\left(p^6 T^4 + 535p^3 T^3 + 5270pT^2 + 535T + 1\right)$ |
| 20 | smooth | | $\left(p^3 T^2 + 11pT + 1\right)\left(p^6 T^4 + 5p^4 T^3 + 3230pT^2 + 5pT + 1\right)$ |
| 21 | smooth | | $\left(p^3 T^2 - 4pT + 1\right)\left(p^6 T^4 - 120p^3 T^3 + 2670pT^2 - 120T + 1\right)$ |
| 22 | smooth | | $\left(p^3 T^2 - 4pT + 1\right)\left(p^6 T^4 - 50p^3 T^3 + 1930pT^2 - 50T + 1\right)$ |
| 23 | smooth | | $\left(p^3 T^2 + pT + 1\right)\left(p^6 T^4 + 185p^3 T^3 + 1220pT^2 + 185T + 1\right)$ |
| 24 | smooth | | $\left(p^3 T^2 - 9pT + 1\right)\left(p^6 T^4 - 375p^3 T^3 + 3320pT^2 - 375T + 1\right)$ |
| 25 | smooth | | $\left(p^3 T^2 - 4pT + 1\right)\left(p^6 T^4 + 140p^3 T^3 + 1450pT^2 + 140T + 1\right)$ |
| 26 | smooth | | $\left(p^3 T^2 - 4pT + 1\right)\left(p^6 T^4 + 90p^3 T^3 - 350pT^2 + 90T + 1\right)$ |
| 27 | smooth | | $\left(p^3 T^2 + 6pT + 1\right)\left(p^6 T^4 + 650p^3 T^3 + 5450pT^2 + 650T + 1\right)$ |

| $\varphi$ | smooth/sing. | singularity | $R(T)$ |
|---|---|---|---|
| | | | $p = 43,$ *continued* |
| 28 | smooth* | | |
| 29 | smooth | | $\left(p^3T^2 + pT + 1\right)\left(p^6T^4 - 325p^3T^3 + 2190pT^2 - 325T + 1\right)$ |
| 30 | smooth | | $\left(p^3T^2 - 4pT + 1\right)\left(p^6T^4 + 220p^3T^3 + 3290pT^2 + 220T + 1\right)$ |
| 31 | smooth | | $\left(p^3T^2 - 4pT + 1\right)\left(p^6T^4 + 560p^3T^3 + 4910pT^2 + 560T + 1\right)$ |
| 32 | smooth | | $\left(p^3T^2 + 11pT + 1\right)\left(p^6T^4 + 205p^3T^3 + 3550pT^2 + 205T + 1\right)$ |
| 33 | smooth | | $\left(p^3T^2 + pT + 1\right)\left(p^6T^4 - 15p^3T^3 - 1130pT^2 - 15T + 1\right)$ |
| 34 | smooth | | $\left(p^3T^2 - 9pT + 1\right)\left(p^6T^4 + 25p^3T^3 + 1070pT^2 + 25T + 1\right)$ |
| 35 | smooth | | $\left(p^3T^2 + 6pT + 1\right)\left(p^6T^4 - 210p^3T^3 - 180pT^2 - 210T + 1\right)$ |
| 36 | smooth | | $\left(p^3T^2 - 4pT + 1\right)\left(p^6T^4 + 80p^3T^3 + 1870pT^2 + 80T + 1\right)$ |
| 37 | smooth | | $\left(p^3T^2 - 4pT + 1\right)\left(p^6T^4 - 90p^3T^3 + 1210pT^2 - 90T + 1\right)$ |
| 38 | smooth | | $\left(p^3T^2 - 4pT + 1\right)\left(p^6T^4 - 120p^3T^3 + 3170pT^2 - 120T + 1\right)$ |
| 39 | singular | $-\frac{1}{32}$ | |
| 40 | smooth | | $\left(p^3T^2 + pT + 1\right)\left(p^6T^4 - 295p^3T^3 + 2230pT^2 - 295T + 1\right)$ |
| 41 | smooth | | $\left(p^3T^2 - 4pT + 1\right)\left(p^6T^4 + 540p^3T^3 + 5150pT^2 + 540T + 1\right)$ |
| 42 | smooth | | $\left(p^3T^2 - 408T + 1\right)\left(p^3T^2 + 6pT + 1\right)^2$ |

## D.2.   Rational points and elliptic modular forms

Table 19: *Select rational points on the $\mathbb{Z}_2$ symmetric line $\boldsymbol{\varphi} = (\varphi, \varphi)$ in the moduli space of mirror split quintic manifolds together with the modular forms corresponding to the quadratic factors of the local zeta functions of these manifolds, and the related twisted Sen curves $\mathcal{E}_S$. The second and third columns labelled $\mathcal{E}_S(\tau/n)$ give the LMFDB label of the twisted Sen curves corresponding to the two isogenous families, and the penultimate column lists the label of the corresponding modular form common to all isogenous curves. The column labelled "p" lists primes at which the modular form coefficient $\alpha_p$ is not equal to the zeta function coefficient $c_p$. The last column gives the size of the isogeny class of the twisted Sen curves, confirming that there are indeed only two distinct families of twisted Sen curves.*

| $\varphi$ | $\mathcal{E}_S(\tau)$ | $\mathcal{E}_S(\tau/5)$ | $p$ | form label | # |
|---|---|---|---|---|---|
| $\frac{1}{70}$ | **396830.e1** | **396830.e2** | | **396830.2.a.e** | 2 |
| $\frac{1}{66}$ | **335346.f1** | **335346.f2** | 23 | **335346.2.a.f** | 2 |
| $\frac{1}{55}$ | **199595.b1** | **199595.b2** | | **199595.2.a.b** | 2 |
| $\frac{1}{42}$ | **93450.cz1** | **93450.cz2** | 43 | **93450.2.a.cz** | 2 |
| $\frac{1}{35}$ | **56315.a1** | **56315.a2** | | **56315.2.a.a** | 2 |
| $\frac{1}{33}$ | **47883.a1** | **47883.a2** | $5, 17$ | **47883.2.a.a** | 2 |

| $\varphi$ | $\mathcal{E}_S(\tau)$ | $\mathcal{E}_S(\tau/5)$ | $p$ | form label | # |
|---|---|---|---|---|---|
| continued | | | | | |
| $\frac{1}{30}$ | **36870.e1** | **36870.e2** | | **36870.2.a.e** | 2 |
| $\frac{2}{55}$ | **465410.e1** | **465410.e2** | | **465410.2.a.e** | 2 |
| $\frac{1}{22}$ | **15950.l1** | **15950.l2** | | **15950.2.a.l** | 2 |
| $\frac{1}{21}$ | **14091.b1** | **14091.b2** | | **14091.2.a.b** | 2 |
| $\frac{2}{35}$ | **139370.p1** | **139370.p2** | | **139370.2.a.p** | 2 |
| $\frac{2}{33}$ | **119526.g1** | **119526.g2** | 5 | **119526.2.a.g** | 2 |
| $\frac{1}{15}$ | **5835.c1** | **5835.c2** | 23 | **5835.2.a.c** | 2 |
| $\frac{1}{14}$ | **4886.f1** | **4886.f2** | 5 | **4886.2.a.f** | 2 |
| $\frac{3}{35}$ | **248955.b1** | **248955.b2** | | **248955.2.a.b** | 2 |
| $\frac{1}{11}$ | **2651.a1** | **2651.a2** | | **2651.2.a.a** | 2 |
| $\frac{2}{21}$ | **37758.p1** | **37758.p2** | 5 | **37758.2.a.p** | 2 |
| $\frac{1}{10}$ | **2090.l1** | **2090.l2** | | **2090.2.a.l** | 2 |
| $\frac{4}{35}$ | **192430.h1** | **192430.h2** | | **192430.2.a.h** | 2 |
| $\frac{4}{33}$ | **166650.cd1** | **166650.cd2** | | **166650.2.a.cd** | 2 |
| $\frac{2}{15}$ | **16530.bb1** | **16530.bb2** | 17 | **16530.2.a.bb** | 2 |
| $\frac{1}{7}$ | **175.a1** | **175.a2** | | **175.2.a.a** | 2 |
| $\frac{5}{33}$ | **475035.c1** | **475035.c2** | | **475035.2.a.c** | 2 |
| $\frac{1}{6}$ | **606.f1** | **606.f2** | | **606.2.a.f** | 2 |
| $\frac{2}{11}$ | **7898.f1** | **7898.f2** | 5 | **7898.2.a.f** | 2 |
| $\frac{4}{21}$ | **56658.s1** | **56658.s2** | 5 | **56658.2.a.s** | 2 |
| $\frac{1}{5}$ | **395.a1** | **395.a2** | | **395.2.a.a** | 2 |
| $\frac{3}{14}$ | **81774.bg1** | **81774.bg2** | $5, 17$ | **81774.2.a.bg** | 2 |
| $\frac{5}{22}$ | **183590.b1** | **183590.b2** | | **183590.2.a.b** | 2 |
| $\frac{8}{35}$ | **296870.f1** | **296870.f2** | 43 | **296870.2.a.f** | 2 |
| $\frac{5}{21}$ | **164955.a1** | **164955.a2** | | **164955.2.a.a** | 2 |
| $\frac{8}{33}$ | **259314.i1** | **259314.i2** | 41 | **259314.2.a.i** | 2 |
| $\frac{9}{35}$ | **483945.c1** | **483945.c2** | | **483945.2.a.c** | 2 |
| $\frac{4}{15}$ | **26070.ba1** | **26070.ba2** | | **26070.2.a.ba** | 2 |

| continued | | | | | |
|---|---|---|---|---|---|
| $\varphi$ | $\mathcal{E}_S(\tau)$ | $\mathcal{E}_S(\tau/5)$ | $p$ | form label | # |
| $\frac{3}{11}$ | **15675.c1** | **15675.c2** | | **15675.2.a.c** | 2 |
| $\frac{2}{7}$ | **2786.b1** | **2786.b2** | 23 | **2786.2.a.b** | 2 |
| $\frac{3}{10}$ | **12630.k1** | **12630.k2** | | **12630.2.a.k** | 2 |
| $\frac{7}{22}$ | **327866.b1** | **327866.b2** | 41 | **327866.2.a.b** | 2 |
| $\frac{1}{3}$ | **123.a1** | **123.a2** | 5 | **123.2.a.a** | 2 |
| $\frac{5}{14}$ | **65870.c1** | **65870.c2** | | **65870.2.a.c** | 2 |
| $\frac{8}{21}$ | **93450.db1** | **93450.db2** | | **93450.2.a.db** | 2 |
| $\frac{2}{5}$ | **1310.c1** | **1310.c2** | 29 | **1310.2.a.c** | 2 |
| $\frac{9}{22}$ | **170346.u1** | **170346.u2** | 5 | **170346.2.a.u** | 2 |
| $\frac{3}{7}$ | **5691.a2** | **5691.a1** | 5 | **5691.2.a.a** | 2 |
| $\frac{5}{11}$ | **38555.a2** | **38555.a1** | | **38555.2.a.a** | 2 |
| $\frac{16}{35}$ | **499030.d2** | **499030.d1** | 17 | **499030.2.a.d** | 2 |
| $\frac{7}{15}$ | **1155.c2** | **1155.c1** | | **1155.2.a.c** | 2 |
| $\frac{16}{33}$ | **438306.d2** | **438306.d1** | $5, 23$ | **438306.2.a.d** | 2 |
| $\frac{1}{2}$ | **50.b3** | **50.b2** | | **50.2.a.b** | 4 |
| $\frac{8}{15}$ | **44430.e2** | **44430.e1** | | **44430.2.a.e** | 2 |
| $\frac{6}{11}$ | **53526.d2** | **53526.d1** | 17 | **53526.2.a.d** | 2 |
| $\frac{4}{7}$ | **4774.h2** | **4774.h1** | 5 | **4774.2.a.h** | 2 |
| $\frac{3}{5}$ | **2715.a2** | **2715.a1** | | **2715.2.a.a** | 2 |
| $\frac{7}{11}$ | **70763.a2** | **70763.a1** | 5 | **70763.2.a.a** | 2 |
| $\frac{9}{14}$ | **63042.w2** | **63042.w1** | | **63042.2.a.w** | 2 |
| $\frac{2}{3}$ | **426.c2** | **426.c1** | 5 | **426.2.a.c** | 2 |
| $\frac{7}{10}$ | **57470.l2** | **57470.l1** | 17 | **57470.2.a.l** | 2 |
| $\frac{5}{7}$ | **14315.a2** | **14315.a1** | | **14315.2.a.a** | 2 |
| $\frac{8}{11}$ | **22550.t2** | **22550.t1** | | **22550.2.a.t** | 2 |
| $\frac{11}{15}$ | **316635.b2** | **316635.b1** | | **316635.2.a.b** | 2 |
| $\frac{31}{42}$ | **358050.ff1** | **358050.ff2** | | **358050.2.a.ff** | 2 |
| $\frac{16}{21}$ | **163002.d2** | **163002.d1** | $23, 41$ | **163002.2.a.d** | 2 |

*Continued on the following page*

| continued | | | | | |
|---|---|---|---|---|---|
| $\varphi$ | $\mathcal{E}_S(\tau)$ | $\mathcal{E}_S(\tau/5)$ | $p$ | form label | # |
| $\frac{11}{14}$ | **272426.s2** | **272426.s1** | 5 | **272426.2.a.s** | 2 |
| $\frac{4}{5}$ | **2290.c2** | **2290.c1** | | **2290.2.a.c** | 2 |
| $\frac{9}{11}$ | **37257.a2** | **37257.a1** | 5 | **37257.2.a.a** | 2 |
| $\frac{28}{33}$ | **254562.bw2** | **254562.bw1** | | **254562.2.a.bw** | 2 |
| $\frac{6}{7}$ | **19950.da2** | **19950.da1** | | **19950.2.a.da** | 2 |
| $\frac{13}{15}$ | **429195.c2** | **429195.c1** | | **429195.2.a.c** | 2 |
| $\frac{9}{10}$ | **30270.v2** | **30270.v1** | | **30270.2.a.v** | 2 |
| $\frac{10}{11}$ | **135410.m2** | **135410.m1** | | **135410.2.a.m** | 2 |
| $\frac{13}{14}$ | **369278.a2** | **369278.a1** | 5 | **369278.2.a.a** | 2 |
| $\frac{14}{15}$ | **491190.x2** | **491190.x1** | | **491190.2.a.x** | 2 |
| $\frac{21}{22}$ | **473550.gs1** | **473550.gs2** | 43 | **473550.2.a.gs** | 2 |
| 1 | **11.a3** | **11.a2** | | **11.2.a.a** | 3 |
| $\frac{16}{15}$ | **78270.k2** | **78270.k1** | | **78270.2.a.k** | 2 |
| $\frac{15}{14}$ | **479010.o2** | **479010.o1** | | **479010.2.a.o** | 2 |
| $\frac{12}{11}$ | **94314.c2** | **94314.c1** | 5 | **94314.2.a.c** | 2 |
| $\frac{11}{10}$ | **130790.k2** | **130790.k1** | | **130790.2.a.k** | 2 |
| $\frac{8}{7}$ | **8414.e2** | **8414.e1** | 5 | **8414.2.a.e** | 2 |
| $\frac{7}{6}$ | **18858.i2** | **18858.i1** | 5 | **18858.2.a.i** | 2 |
| $\frac{13}{11}$ | **218075.b2** | **218075.b1** | 23 | **218075.2.a.b** | 2 |
| $\frac{27}{22}$ | **415074.q2** | **415074.q1** | | **415074.2.a.q** | 2 |
| $\frac{14}{11}$ | **249326.e2** | **249326.e1** | 5 | **249326.2.a.e** | 2 |
| $\frac{9}{7}$ | **13881.a2** | **13881.a1** | 5 | **13881.2.a.a** | 2 |
| $\frac{13}{10}$ | **176930.d2** | **176930.d1** | | **176930.2.a.d** | 2 |
| $\frac{4}{3}$ | **150.c3** | **150.c4** | | **150.2.a.c** | 4 |
| $\frac{15}{11}$ | **282315.c2** | **282315.c1** | | **282315.2.a.c** | 2 |
| $\frac{7}{5}$ | **665.a2** | **665.a1** | | **665.2.a.a** | 2 |
| $\frac{10}{7}$ | **50330.e2** | **50330.e1** | 17 | **50330.2.a.e** | 2 |
| $\frac{16}{11}$ | **39622.d2** | **39622.d1** | 29 | **39622.2.a.d** | 2 |
| | | | | *Continued on the following page* | |

| continued | | | | | |
|---|---|---|---|---|---|
| $\varphi$ | $\mathcal{E}_S(\tau)$ | $\mathcal{E}_S(\tau/5)$ | $p$ | form label | # |
| $\frac{3}{2}$ | **366.f2** | **366.f1** | 5 | **366.2.a.f** | 2 |
| $\frac{32}{21}$ | **285978.u2** | **285978.u1** | 5 | **285978.2.a.u** | 2 |
| $\frac{17}{11}$ | **353243.a2** | **353243.a1** | 5 | **353243.2.a.a** | 2 |
| $\frac{11}{7}$ | **59675.b2** | **59675.b1** | | **59675.2.a.b** | 2 |
| $\frac{8}{5}$ | **4010.e2** | **4010.e1** | 23 | **4010.2.a.e** | 2 |
| $\frac{34}{21}$ | **463386.bs2** | **463386.bs1** | 5 | **463386.2.a.bs** | 2 |
| $\frac{18}{11}$ | **130350.cv2** | **130350.cv1** | | **130350.2.a.cv** | 2 |
| $\frac{5}{3}$ | **2235.a2** | **2235.a1** | | **2235.2.a.a** | 2 |
| $\frac{17}{10}$ | **6970.f2** | **6970.f1** | | **6970.2.a.f** | 2 |
| $\frac{12}{7}$ | **34818.f2** | **34818.f1** | | **34818.2.a.f** | 2 |
| $\frac{19}{11}$ | **430331.a2** | **430331.a1** | 5 | **430331.2.a.a** | 2 |
| $\frac{25}{14}$ | **239470.n2** | **239470.n1** | | **239470.2.a.n** | 2 |
| $\frac{9}{5}$ | **6585.a2** | **6585.a1** | | **6585.2.a.a** | 2 |
| $\frac{20}{11}$ | **235510.k2** | **235510.k1** | | **235510.2.a.k** | 2 |
| $\frac{11}{6}$ | **42306.v2** | **42306.v1** | 17 | **42306.2.a.v** | 2 |
| $\frac{13}{7}$ | **80171.a2** | **80171.a1** | 5, 23 | **80171.2.a.a** | 2 |
| $\frac{19}{10}$ | **347510.j2** | **347510.j1** | | **347510.2.a.j** | 2 |
| 2 | **38.b2** | **38.b1** | 5 | **38.2.a.b** | 2 |
| $\frac{23}{11}$ | **120175.b2** | **120175.b1** | 17 | **120175.2.a.b** | 2 |
| $\frac{21}{10}$ | **413490.dj2** | **413490.dj1** | | **413490.2.a.dj** | 2 |
| $\frac{32}{15}$ | **134430.e2** | **134430.e1** | | **134430.2.a.e** | 2 |
| $\frac{15}{7}$ | **102795.a2** | **102795.a1** | | **102795.2.a.a** | 2 |
| $\frac{13}{6}$ | **56550.bv2** | **56550.bv1** | | **56550.2.a.bv** | 2 |
| $\frac{24}{11}$ | **161634.u2** | **161634.u1** | 5, 41 | **161634.2.a.u** | 2 |
| $\frac{11}{5}$ | **27995.a2** | **27995.a1** | | **27995.2.a.a** | 2 |
| $\frac{25}{11}$ | **138655.a2** | **138655.a1** | | **138655.2.a.a** | 2 |
| $\frac{16}{7}$ | **14350.o2** | **14350.o1** | | **14350.2.a.o** | 2 |
| $\frac{23}{10}$ | **483230.f2** | **483230.f1** | | **483230.2.a.f** | 2 |
| | | | | *Continued on the following page* | |

| continued | | | | | |
|---|---|---|---|---|---|
| $\varphi$ | $\mathcal{E}_S(\tau)$ | $\mathcal{E}_S(\tau/5)$ | $p$ | form label | # |
| $\frac{7}{3}$ | **4011.a2** | **4011.a1** | $5, 29$ | **4011.2.a.a** | 2 |
| $\frac{12}{5}$ | **16230.h2** | **16230.h1** | $17$ | **16230.2.a.h** | 2 |
| $\frac{17}{7}$ | **127211.a2** | **127211.a1** | | **127211.2.a.a** | 2 |
| $\frac{27}{11}$ | **87747.b2** | **87747.b1** | $5, 23$ | **87747.2.a.b** | 2 |
| $\frac{37}{15}$ | **250305.b2** | **250305.b1** | | **250305.2.a.b** | 2 |
| $\frac{5}{2}$ | **890.g2** | **890.g1** | | **890.2.a.g** | 2 |
| $\frac{28}{11}$ | **419650.y2** | **419650.y1** | | **419650.2.a.y** | 2 |
| $\frac{18}{7}$ | **46662.bb2** | **46662.bb1** | $5$ | **46662.2.a.bb** | 2 |
| $\frac{13}{5}$ | **37115.a2** | **37115.a1** | | **37115.2.a.a** | 2 |
| $\frac{8}{3}$ | **1254.j2** | **1254.j1** | | **1254.2.a.j** | 2 |
| $\frac{27}{10}$ | **70230.h2** | **70230.h1** | | **70230.2.a.h** | 2 |
| $\frac{19}{7}$ | **153083.b2** | **153083.b1** | $5, 41$ | **153083.2.a.b** | 2 |
| $\frac{14}{5}$ | **41930.d2** | **41930.d1** | | **41930.2.a.d** | 2 |
| $\frac{17}{6}$ | **88638.l2** | **88638.l1** | $5$ | **88638.2.a.l** | 2 |
| $\frac{20}{7}$ | **83230.s2** | **83230.s1** | | **83230.2.a.s** | 2 |
| $\frac{32}{11}$ | **65318.f2** | **65318.f1** | $5, 43$ | **65318.2.a.f** | 2 |
| $3$ | **75.a2** | **75.a1** | | **75.2.a.a** | 2 |
| $\frac{64}{21}$ | **467418.w2** | **467418.w1** | $5, 17$ | **467418.2.a.w** | 2 |
| $\frac{22}{7}$ | **193886.f2** | **193886.f1** | | **193886.2.a.f** | 2 |
| $\frac{19}{6}$ | **105906.n2** | **105906.n1** | $5$ | **105906.2.a.n** | 2 |
| $\frac{16}{5}$ | **6490.f2** | **6490.f1** | | **6490.2.a.f** | 2 |
| $\frac{36}{11}$ | **209946.y2** | **209946.y1** | $23$ | **209946.2.a.y** | 2 |
| $\frac{23}{7}$ | **207851.a2** | **207851.a1** | $5$ | **207851.2.a.a** | 2 |
| $\frac{10}{3}$ | **7170.k2** | **7170.k1** | | **7170.2.a.k** | 2 |
| $\frac{17}{5}$ | **57035.a2** | **57035.a1** | | **57035.2.a.a** | 2 |
| $\frac{24}{7}$ | **55482.f2** | **55482.f1** | $5$ | **55482.2.a.f** | 2 |
| $\frac{7}{2}$ | **1526.f2** | **1526.f1** | $23$ | **1526.2.a.f** | 2 |
| $\frac{25}{7}$ | **47215.a2** | **47215.a1** | $23$ | **47215.2.a.a** | 2 |
| | | | | *Continued on the following page* | |

| $\varphi$ | $\mathcal{E}_S(\tau)$ | $\mathcal{E}_S(\tau/5)$ | $p$ | form label | # |
|---|---|---|---|---|---|
| | continued | | | | |
| $\frac{18}{5}$ | **20730.o2** | **20730.o1** | | **20730.2.a.o** | 2 |
| $\frac{40}{11}$ | **369710.d2** | **369710.d1** | 17 | **369710.2.a.d** | 2 |
| $\frac{11}{3}$ | **8283.a2** | **8283.a1** | 5 | **8283.2.a.a** | 2 |
| $\frac{26}{7}$ | **50050.bs2** | **50050.bs1** | 29 | **50050.2.a.bs** | 2 |
| $\frac{19}{5}$ | **67355.a2** | **67355.a1** | | **67355.2.a.a** | 2 |
| $\frac{23}{6}$ | **141450.ce2** | **141450.ce1** | | **141450.2.a.ce** | 2 |
| $\frac{27}{7}$ | **29379.a2** | **29379.a1** | 17 | **29379.2.a.a** | 2 |
| $4$ | **58.b2** | **58.b1** | 5 | **58.2.a.b** | 2 |
| $\frac{29}{7}$ | **292523.a2** | **292523.a1** | 5 | **292523.2.a.a** | 2 |
| $\frac{25}{6}$ | **31830.q2** | **31830.q1** | | **31830.2.a.q** | 2 |
| $\frac{21}{5}$ | **77595.d2** | **77595.d1** | | **77595.2.a.d** | 2 |
| $\frac{64}{15}$ | **200670.g2** | **200670.g1** | $23, 29$ | **200670.2.a.g** | 2 |
| $\frac{30}{7}$ | **306390.l2** | **306390.l1** | | **306390.2.a.l** | 2 |

## D.3.  Real quadratic points and Hilbert modular forms

Table 20: *Some values of the modulus $\varphi$ which lie in various real quadratic field extensions $\mathbb{Q}(\sqrt{n})$ together with the corresponding Hilbert modular forms and the twisted Sen curves $\mathcal{E}_S$. The second and third columns labelled $\mathcal{E}_S(\tau/n)$ give the LMFDB labels (without the number field label) of the twisted Sen curves, and the penultimate column lists the label (again without the number field label) of the corresponding modular form. The column labelled "$N(\mathfrak{p})$" lists the ideal norms of the prime ideals at which the modular form coefficient $\alpha_{\mathfrak{p}}$ is not equal to the zeta function coefficient $c_p$ in the sense discussed in §4.6.*

| $\varphi$ | $\mathcal{E}_S(\tau)$ | $\mathcal{E}_S(\tau/5)$ | $N(\mathfrak{p})$ | form labels |
|---|---|---|---|---|
| $-10 - 7\sqrt{2}$ | **142.1-c2** | **142.1-c1** | 17 | **142.1-c, 142.2-c** |
| $-10 + 7\sqrt{2}$ | **142.2-c1** | **142.2-c2** | 17 | **142.1-c, 142.2-c** |
| $-8 - 8\sqrt{2}$ | **3438.1-c2** | **3438.1-c1** | | **3438.1-c, 3438.2-c** |
| $-8 + 8\sqrt{2}$ | **3438.2-c1** | **3438.2-c2** | | **3438.1-c, 3438.2-c** |
| $-6 - 4\sqrt{2}$ | **1522.2-a2** | **1522.2-a1** | 23 | **1522.1-a, 1522.2-a** |
| $-6 + 4\sqrt{2}$ | **1522.1-a1** | **1522.1-a2** | 23 | **1522.1-a, 1522.2-a** |

*Continued on the following page*

| $\varphi$ | $\mathcal{E}_S(\tau)$ | $\mathcal{E}_S(\tau/5)$ | $N(\mathfrak{p})$ | form labels |
|---|---|---|---|---|
| | | | | continued |
| $-4-3\sqrt{2}$ | **1138.1-d1** | **1138.1-d2** | | **1138.1-d, 1138.2-d** |
| $-4-2\sqrt{2}$ | **3202.2-b1** | **3202.2-b2** | | **3202.1-b, 3202.2-b** |
| $-4+2\sqrt{2}$ | **3202.1-b1** | **3202.1-b2** | | **3202.1-b, 3202.2-b** |
| $-4+3\sqrt{2}$ | **1138.2-d1** | **1138.2-d2** | | **1138.1-d, 1138.2-d** |
| $-3-2\sqrt{2}$ | **89.2-a2** | **89.2-a1** | | **89.1-a, 89.2-a** |
| $-3+2\sqrt{2}$ | **89.1-a1** | **89.1-a2** | | **89.1-a, 89.2-a** |
| $-2-2\sqrt{2}$ | **1422.1-c1** | **1422.1-c2** | 41 | **1422.1-c, 1422.2-c** |
| $-2-\sqrt{2}$ | **558.2-a2** | **558.2-a1** | | **558.1-a, 558.2-a** |
| $-2+\sqrt{2}$ | **558.1-a1** | **558.1-a2** | | **558.1-a, 558.2-a** |
| $-2+2\sqrt{2}$ | **1422.2-c1** | **1422.2-c2** | 41 | **1422.1-c, 1422.2-c** |
| $-8\sqrt{2}$ | **1282.1-a1** | **1282.1-a2** | | **1282.1-a, 1282.2-a** |
| $-2\sqrt{2}$ | **1838.2-f2** | **1838.2-f1** | | **1838.1-f, 1838.2-f** |
| $-\sqrt{2}$ | **482.2-c2** | **482.2-c1** | | **482.1-c, 482.2-c** |
| $\sqrt{2}$ | **482.1-c1** | **482.1-c2** | | **482.1-c, 482.2-c** |
| $2\sqrt{2}$ | **1838.1-f1** | **1838.1-f2** | | **1838.1-f, 1838.2-f** |
| $8\sqrt{2}$ | **1282.2-a1** | **1282.2-a2** | | **1282.1-a, 1282.2-a** |
| $1-2\sqrt{2}$ | **4473.4-f2** | **4473.4-f1** | | **4473.1-f, 4473.4-f** |
| $1+2\sqrt{2}$ | **4473.1-f1** | **4473.1-f2** | | **4473.1-f, 4473.4-f** |
| $2-2\sqrt{2}$ | **542.2-a2** | **542.2-a1** | 23 | **542.1-a, 542.2-a** |
| $2-\sqrt{2}$ | **382.2-c2** | **382.2-c1** | 23 | **382.1-c, 382.2-c** |
| $2+\sqrt{2}$ | **382.1-c1** | **382.1-c2** | 23 | **382.1-c, 382.2-c** |
| $2+2\sqrt{2}$ | **542.1-a2** | **542.1-a1** | 23 | **542.1-a, 542.2-a** |
| $3-3\sqrt{2}$ | **3609.2-b2** | **3609.2-b1** | | **3609.1-b, 3609.2-b** |
| $3-2\sqrt{2}$ | **89.2-a2** | **89.2-a1** | | **89.1-a, 89.2-a** |
| $3-\sqrt{2}$ | **3353.4-c2** | **3353.4-c1** | | **3353.1-c, 3353.4-c** |
| $3+\sqrt{2}$ | **3353.1-c2** | **3353.1-c1** | | **3353.1-c, 3353.4-c** |
| $3+2\sqrt{2}$ | **89.1-a1** | **89.1-a2** | | **89.1-a, 89.2-a** |
| $3+3\sqrt{2}$ | **3609.1-b1** | **3609.1-b2** | | **3609.1-b, 3609.2-b** |

| continued | | | | |
|---|---|---|---|---|
| $\varphi$ | $\mathcal{E}_S(\tau)$ | $\mathcal{E}_S(\tau/5)$ | $N(\mathfrak{p})$ | form labels |
| $4 - 8\sqrt{2}$ | **3906.1-i2** | **3906.1-i3** | 41 | **3906.1-i, 3906.4-i** |
| $4 - 5\sqrt{2}$ | **306.2-d2** | **306.2-d1** | | **306.1-d, 306.2-d** |
| $4 - 4\sqrt{2}$ | **558.2-f2** | **558.2-f1** | | **558.1-f, 558.2-f** |
| $4 - 3\sqrt{2}$ | **82.1-a2** | **82.1-a1** | | **82.1-a, 82.2-a** |
| $4 - 2\sqrt{2}$ | **738.1-b2** | **738.1-b1** | 17 | **738.1-b, 738.2-b** |
| $4 + 2\sqrt{2}$ | **738.2-b2** | **738.2-b1** | 17 | **738.1-b, 738.2-b** |
| $4 + 3\sqrt{2}$ | **82.2-a2** | **82.2-a1** | | **82.1-a, 82.2-a** |
| $4 + 4\sqrt{2}$ | **558.1-f1** | **558.1-f2** | | **558.1-f, 558.2-f** |
| $4 + 5\sqrt{2}$ | **306.1-d1** | **306.1-d2** | | **306.1-d, 306.2-d** |
| $4 + 8\sqrt{2}$ | **3906.4-i1** | **3906.4-i2** | 41 | **3906.1-i, 3906.4-i** |
| $5 - 4\sqrt{2}$ | **217.4-a2** | **217.4-a1** | 41 | **217.1-a, 217.4-a** |
| $5 - 3\sqrt{2}$ | **1057.2-e2** | **1057.2-e1** | 23 | **1057.2-e, 1057.3-e** |
| $5 + 3\sqrt{2}$ | **1057.3-e1** | **1057.3-e2** | 23 | **1057.2-e, 1057.3-e** |
| $5 + 4\sqrt{2}$ | **217.1-a2** | **217.1-a1** | 41 | **217.1-a, 217.4-a** |
| $6 - 5\sqrt{2}$ | **4354.3-d1** | **4354.3-d2** | | **4354.2-d, 4354.3-d** |
| $6 - 4\sqrt{2}$ | **62.1-a1** | **62.1-a4** | 17 | **62.1-a, 62.2-a** |
| $6 - 3\sqrt{2}$ | **2718.1-c2** | **2718.1-c1** | | **2718.1-c, 2718.2-c** |
| $6 + 3\sqrt{2}$ | **2718.2-c2** | **2718.2-c1** | | **2718.1-c, 2718.2-c** |
| $6 + 4\sqrt{2}$ | **62.2-a1** | **62.2-a2** | 17 | **62.1-a, 62.2-a** |
| $6 + 5\sqrt{2}$ | **4354.2-d2** | **4354.2-d1** | | **4354.2-d, 4354.3-d** |
| $7 - 4\sqrt{2}$ | **4743.3-c1** | **4743.3-c2** | | **4743.2-c, 4743.3-c** |
| $7 - 3\sqrt{2}$ | **1271.2-d1** | **1271.2-d2** | 23 | **1271.2-d, 1271.3-d** |
| $7 + 3\sqrt{2}$ | **1271.3-d2** | **1271.3-d1** | 23 | **1271.2-d, 1271.3-d** |
| $7 + 4\sqrt{2}$ | **4743.2-c2** | **4743.2-c1** | | **4743.2-c, 4743.3-c** |
| $8 - 6\sqrt{2}$ | **818.2-b1** | **818.2-b2** | 23 | **818.1-b, 818.2-b** |
| $8 - 4\sqrt{2}$ | **1502.2-a1** | **1502.2-a2** | | **1502.1-a, 1502.2-a** |
| $8 - 2\sqrt{2}$ | **1246.3-a2** | **1246.3-a1** | | **1246.2-a, 1246.3-a** |
| $8 + 2\sqrt{2}$ | **1246.2-a2** | **1246.2-a1** | | **1246.2-a, 1246.3-a** |
| | | | | *Continued on the following page* |

| $\varphi$ | $\mathcal{E}_S(\tau)$ | $\mathcal{E}_S(\tau/5)$ | $N(\mathfrak{p})$ | form labels |
|---|---|---|---|---|
| continued | | | | |
| $8 + 4\sqrt{2}$ | **1502.1-a2** | **1502.1-a1** | | **1502.1-a, 1502.2-a** |
| $8 + 6\sqrt{2}$ | **818.1-b2** | **818.1-b1** | 23 | **818.1-b, 818.2-b** |
| $10 - 7\sqrt{2}$ | **738.2-a1** | **738.2-a2** | | **738.1-a, 738.2-a** |
| $10 + 7\sqrt{2}$ | **738.1-a1** | **738.1-a2** | | **738.1-a, 738.2-a** |
| $-7 - 4\sqrt{3}$ | **71.2-d1** | **71.2-d2** | | **71.1-d, 71.2-d** |
| $-7 + 4\sqrt{3}$ | **71.1-d1** | **71.1-d2** | | **71.1-d, 71.2-d** |
| $-5 - 3\sqrt{3}$ | **1342.2-l1** | **1342.2-l2** | 23 | **1342.2-l, 1342.3-l** |
| $-5 + 3\sqrt{3}$ | **1342.3-l1** | **1342.3-l2** | 23 | **1342.2-l, 1342.3-l** |
| $-4 - 2\sqrt{3}$ | **1418.2-b2** | **1418.2-b1** | | **1418.1-b, 1418.2-b** |
| $-4 + 2\sqrt{3}$ | **1418.1-b1** | **1418.1-b2** | | **1418.1-b, 1418.2-b** |
| $-3 - 2\sqrt{3}$ | **1977.1-b1** | **1977.1-b2** | | **1977.1-b, 1977.2-b** |
| $-3 + 2\sqrt{3}$ | **1977.2-b1** | **1977.2-b2** | | **1977.1-b, 1977.2-b** |
| $-2 - 2\sqrt{3}$ | **22.1-b3** | **22.1-b2** | | **22.1-b, 22.2-b** |
| $-2 - \sqrt{3}$ | **109.1-c2** | **109.1-c1** | | **109.1-c, 109.2-c** |
| $-2 + \sqrt{3}$ | **109.2-c1** | **109.2-c2** | | **109.1-c, 109.2-c** |
| $-2 + 2\sqrt{3}$ | **22.2-b2** | **22.2-b3** | | **22.1-b, 22.2-b** |
| $-1 - 7\sqrt{3}$ | **1606.4-k1** | **1606.4-k2** | | **1606.1-k, 1606.4-k** |
| $-1 - \sqrt{3}$ | **622.2-c1** | **622.2-c2** | 23 | **622.1-c, 622.2-c** |
| $-1 + \sqrt{3}$ | **622.1-c1** | **622.1-c2** | 23 | **622.1-c, 622.2-c** |
| $-1 + 7\sqrt{3}$ | **1606.1-k1** | **1606.1-k2** | | **1606.1-k, 1606.4-k** |
| $-2\sqrt{3}$ | **726.1-p2** | **726.1-p1** | | **726.1-b, 726.1-p** |
| $-\sqrt{3}$ | **1077.1-b2** | **1077.1-b1** | | **1077.1-b, 1077.2-b** |
| $\sqrt{3}$ | **1077.2-b1** | **1077.2-b2** | | **1077.1-b, 1077.2-b** |
| $2\sqrt{3}$ | **726.1-b1** | **726.1-b2** | | **726.1-b, 726.1-p** |
| $1 - \sqrt{3}$ | **358.2-b2** | **358.2-b1** | | **358.1-b, 358.2-b** |
| $1 + \sqrt{3}$ | **358.1-b2** | **358.1-b1** | | **358.1-b, 358.2-b** |
| $2 - 2\sqrt{3}$ | **1078.2-b2** | **1078.2-b1** | | **1078.1-b, 1078.2-b** |
| $2 - \sqrt{3}$ | **109.1-c2** | **109.1-c1** | | **109.1-c, 109.2-c** |
| | | | | *Continued on the following page* |

| $\varphi$ | $\mathcal{E}_S(\tau)$ | $\mathcal{E}_S(\tau/5)$ | $N(\mathfrak{p})$ | form labels |
|---|---|---|---|---|
| continued | | | | |
| $2+\sqrt{3}$ | **109.2-c1** | **109.2-c2** | | **109.1-c, 109.2-c** |
| $2+2\sqrt{3}$ | **1078.1-b1** | **1078.1-b2** | | **1078.1-b, 1078.2-b** |
| $3-3\sqrt{3}$ | **4026.3-m2** | **4026.3-m1** | | **4026.2-m, 4026.3-m** |
| $3-2\sqrt{3}$ | **393.1-d2** | **393.1-d1** | | **393.1-d, 393.2-d** |
| $3-\sqrt{3}$ | **2454.2-l2** | **2454.2-l1** | | **2454.1-l, 2454.2-l** |
| $3+\sqrt{3}$ | **2454.1-l2** | **2454.1-l1** | | **2454.1-l, 2454.2-l** |
| $3+2\sqrt{3}$ | **393.2-d2** | **393.2-d1** | | **393.1-d, 393.2-d** |
| $3+3\sqrt{3}$ | **4026.2-m1** | **4026.2-m2** | | **4026.2-m, 4026.3-m** |
| $4-4\sqrt{3}$ | **142.2-a2** | **142.2-a1** | | **142.1-a, 142.2-a** |
| $4-3\sqrt{3}$ | **2629.3-d2** | **2629.3-d1** | | **2629.2-d, 2629.3-d** |
| $4-2\sqrt{3}$ | **362.1-a2** | **362.1-a1** | | **362.1-a, 362.2-a** |
| $4+2\sqrt{3}$ | **362.2-a1** | **362.2-a2** | | **362.1-a, 362.2-a** |
| $4+3\sqrt{3}$ | **2629.2-d1** | **2629.2-d2** | | **2629.2-d, 2629.3-d** |
| $4+4\sqrt{3}$ | **142.1-a1** | **142.1-a2** | | **142.1-a, 142.2-a** |
| $5-3\sqrt{3}$ | **22.1-b4** | **22.1-b1** | | **22.1-b, 22.2-b** |
| $5-2\sqrt{3}$ | **4537.2-d2** | **4537.2-d1** | | **4537.2-d, 4537.3-d** |
| $5+2\sqrt{3}$ | **4537.3-d2** | **4537.3-d1** | | **4537.2-d, 4537.3-d** |
| $5+3\sqrt{3}$ | **22.2-b4** | **22.2-b1** | | **22.1-b, 22.2-b** |
| $6-4\sqrt{3}$ | **1446.2-d1** | **1446.2-d2** | | **1446.1-d, 1446.2-d** |
| $6-3\sqrt{3}$ | **33.1-d3** | **33.1-d4** | | **33.1-d, 33.2-d** |
| $6+3\sqrt{3}$ | **33.2-d1** | **33.2-d2** | | **33.1-d, 33.2-d** |
| $6+4\sqrt{3}$ | **1446.1-d2** | **1446.1-d1** | | **1446.1-d, 1446.2-d** |
| $7-4\sqrt{3}$ | **71.2-d1** | **71.2-d2** | | **71.1-d, 71.2-d** |
| $7+4\sqrt{3}$ | **71.1-d1** | **71.1-d2** | | **71.1-d, 71.2-d** |
| $8-4\sqrt{3}$ | **1342.3-d1** | **1342.3-d2** | | **1342.2-d, 1342.3-d** |
| $8-2\sqrt{3}$ | **3406.2-d1** | **3406.2-d2** | | **3406.2-d, 3406.3-d** |
| $8+2\sqrt{3}$ | **3406.3-d2** | **3406.3-d1** | | **3406.2-d, 3406.3-d** |
| $8+4\sqrt{3}$ | **1342.2-d2** | **1342.2-d1** | | **1342.2-d, 1342.3-d** |

| $\varphi$ | $\mathcal{E}_S(\tau)$ | $\mathcal{E}_S(\tau/5)$ | $N(\mathfrak{p})$ | form labels |
|---|---|---|---|---|
| continued | | | | |
| $9 - 5\sqrt{3}$ | **3234.2-n1** | **3234.2-n2** | 23 | **3234.1-n, 3234.2-n** |
| $9 + 5\sqrt{3}$ | **3234.1-n1** | **3234.1-n2** | 23 | **3234.1-n, 3234.2-n** |
| $10 - 6\sqrt{3}$ | **1322.1-b1** | **1322.1-b2** | | **1322.1-b, 1322.2-b** |
| $10 + 6\sqrt{3}$ | **1322.2-b2** | **1322.2-b1** | | **1322.1-b, 1322.2-b** |
| $-9 - 4\sqrt{5}$ | **199.1-a1** | **199.1-a2** | | **199.1-a, 199.2-a** |
| $-9 + 4\sqrt{5}$ | **199.2-a1** | **199.2-a2** | | **199.1-a, 199.2-a** |
| $-7 - 5\sqrt{5}$ | **1900.2-i2** | **1900.2-i1** | | **1900.1-i, 1900.2-i** |
| $-7 + 5\sqrt{5}$ | **1900.1-i1** | **1900.1-i2** | | **1900.1-i, 1900.2-i** |
| $-5 - 2\sqrt{5}$ | **4905.2-d2** | **4905.2-d1** | | **4905.1-d, 4905.2-d** |
| $-5 + 2\sqrt{5}$ | **4905.1-d1** | **4905.1-d2** | | **4905.1-d, 4905.2-d** |
| $-4 - 4\sqrt{5}$ | **3476.4-b2** | **3476.4-b1** | | **3476.1-b, 3476.4-b** |
| $-4 - 2\sqrt{5}$ | **3916.4-a1** | **3916.4-a2** | | **3916.1-a, 3916.4-a** |
| $-4 + 2\sqrt{5}$ | **3916.1-a1** | **3916.1-a2** | | **3916.1-a, 3916.4-a** |
| $-4 + 4\sqrt{5}$ | **3476.1-b1** | **3476.1-b2** | | **3476.1-b, 3476.4-b** |
| $-3 - \sqrt{5}$ | **2684.4-e2** | **2684.4-e1** | | **2684.1-e, 2684.4-e** |
| $-3 + \sqrt{5}$ | **2684.1-e1** | **2684.1-e2** | | **2684.1-e, 2684.4-e** |
| $-1 - \sqrt{5}$ | **2356.3-a2** | **2356.3-a1** | | **2356.2-a, 2356.3-a** |
| $-1 + \sqrt{5}$ | **2356.2-a1** | **2356.2-a2** | | **2356.2-a, 2356.3-a** |
| $-5\sqrt{5}$ | **1255.2-d1** | **1255.2-d2** | | **1255.1-d, 1255.2-d** |
| $-\sqrt{5}$ | **2945.1-c2** | **2945.1-c1** | | **2945.1-c, 2945.4-c** |
| $\sqrt{5}$ | **2945.4-c1** | **2945.4-c2** | | **2945.1-c, 2945.4-c** |
| $5\sqrt{5}$ | **1255.1-d1** | **1255.1-d2** | | **1255.1-d, 1255.2-d** |
| $1 - 2\sqrt{5}$ | **3249.1-e2** | **3249.1-e1** | | **3249.1-e, 3249.1-j** |
| $1 - \sqrt{5}$ | **1476.2-a2** | **1476.2-a1** | 29 | **1476.1-a, 1476.2-a** |
| $1 + \sqrt{5}$ | **1476.1-a1** | **1476.1-a2** | 29 | **1476.1-a, 1476.2-a** |
| $1 + 2\sqrt{5}$ | **3249.1-j1** | **3249.1-j2** | | **3249.1-e, 3249.1-j** |
| $2 - 2\sqrt{5}$ | **3916.2-a2** | **3916.2-a1** | | **3916.2-a, 3916.3-a** |
| $2 + 2\sqrt{5}$ | **3916.3-a1** | **3916.3-a2** | | **3916.2-a, 3916.3-a** |

| continued | | | | |
|---|---|---|---|---|
| $\varphi$ | $\mathcal{E}_S(\tau)$ | $\mathcal{E}_S(\tau/5)$ | $N(\mathfrak{p})$ | form labels |
| $3 - \sqrt{5}$ | **1100.2-b2** | **1100.2-b1** | | **1100.1-b, 1100.2-b** |
| $3 + \sqrt{5}$ | **1100.1-b1** | **1100.1-b2** | | **1100.1-b, 1100.2-b** |
| $4 - 3\sqrt{5}$ | **4321.2-c2** | **4321.2-c1** | | **4321.2-c, 4321.3-c** |
| $4 - 2\sqrt{5}$ | **396.1-d2** | **396.1-d1** | | **396.1-d, 396.2-d** |
| $4 + 2\sqrt{5}$ | **396.2-d2** | **396.2-d1** | | **396.1-d, 396.2-d** |
| $4 + 3\sqrt{5}$ | **4321.3-c1** | **4321.3-c2** | | **4321.2-c, 4321.3-c** |
| $5 - 3\sqrt{5}$ | **3020.2-a1** | **3020.2-a2** | | **3020.1-a, 3020.2-a** |
| $5 - 2\sqrt{5}$ | **505.2-a2** | **505.2-a1** | | **505.1-a, 505.2-a** |
| $5 + 2\sqrt{5}$ | **505.1-a1** | **505.1-a2** | | **505.1-a, 505.2-a** |
| $5 + 3\sqrt{5}$ | **3020.1-a1** | **3020.1-a2** | | **3020.1-a, 3020.2-a** |
| $6 - 3\sqrt{5}$ | **1359.1-a1** | **1359.1-a2** | | **1359.1-a, 1359.2-a** |
| $6 - 2\sqrt{5}$ | **404.1-b2** | **404.1-b1** | 41 | **404.1-b, 404.2-b** |
| $6 + 2\sqrt{5}$ | **404.2-b2** | **404.2-b1** | 41 | **404.1-b, 404.2-b** |
| $6 + 3\sqrt{5}$ | **1359.2-a2** | **1359.2-a1** | | **1359.1-a, 1359.2-a** |
| $7 - 3\sqrt{5}$ | **596.2-b1** | **596.2-b2** | | **596.1-b, 596.2-b** |
| $7 - 2\sqrt{5}$ | **2871.1-g1** | **2871.1-g2** | | **2871.1-g, 2871.4-g** |
| $7 + 2\sqrt{5}$ | **2871.4-g2** | **2871.4-g1** | | **2871.1-g, 2871.4-g** |
| $7 + 3\sqrt{5}$ | **596.1-b1** | **596.1-b2** | | **596.1-b, 596.2-b** |
| $8 - 4\sqrt{5}$ | **4100.1-k1** | **4100.1-k2** | | **4100.1-k, 4100.2-k** |
| $8 + 4\sqrt{5}$ | **4100.2-k2** | **4100.2-k1** | | **4100.1-k, 4100.2-k** |
| $9 - 4\sqrt{5}$ | **199.1-a1** | **199.1-a2** | | **199.1-a, 199.2-a** |
| $9 - \sqrt{5}$ | **3724.1-c1** | **3724.1-c2** | | **3724.1-c, 3724.2-c** |
| $9 + \sqrt{5}$ | **3724.2-c2** | **3724.2-c1** | | **3724.1-c, 3724.2-c** |
| $9 + 4\sqrt{5}$ | **199.2-a1** | **199.2-a2** | | **199.1-a, 199.2-a** |
| $10 - 2\sqrt{5}$ | **3420.2-g2** | **3420.2-g3** | | **3420.1-g, 3420.2-g** |
| $10 + 2\sqrt{5}$ | **3420.1-g2** | **3420.1-g1** | | **3420.1-g, 3420.2-g** |
| $-8 - 3\sqrt{7}$ | **131.1-c1** | **131.1-c2** | | **131.1-c, 131.2-c** |
| $-8 + 3\sqrt{7}$ | **131.2-c1** | **131.2-c2** | | **131.1-c, 131.2-c** |
| | | | | *Continued on the following page* |

| $\varphi$ | $\mathcal{E}_S(\tau)$ | $\mathcal{E}_S(\tau/5)$ | $N(\mathfrak{p})$ | form labels |
|---|---|---|---|---|
| continued | | | | |
| $-3-\sqrt{7}$ | **562.1-d2** | **562.1-d1** | | **562.1-d, 562.2-d** |
| $-3+\sqrt{7}$ | **562.2-d1** | **562.2-d2** | | **562.1-d, 562.2-d** |
| $2-\sqrt{7}$ | **597.4-c2** | **597.4-c1** | | **597.1-c, 597.4-c** |
| $2+\sqrt{7}$ | **597.1-c2** | **597.1-c1** | | **597.1-c, 597.4-c** |
| $3-\sqrt{7}$ | **298.1-h2** | **298.1-h1** | | **298.1-h, 298.2-h** |
| $3+\sqrt{7}$ | **298.2-h1** | **298.2-h2** | | **298.1-h, 298.2-h** |
| $5-2\sqrt{7}$ | **57.4-c2** | **57.4-c1** | | **57.1-c, 57.4-c** |
| $5+2\sqrt{7}$ | **57.1-c2** | **57.1-c1** | | **57.1-c, 57.4-c** |
| $6-3\sqrt{7}$ | **279.2-g2** | **279.2-g3** | | **279.1-g, 279.2-g** |
| $6-2\sqrt{7}$ | **38.2-c1** | **38.2-c4** | | **38.1-c, 38.2-c** |
| $6+2\sqrt{7}$ | **38.1-c2** | **38.1-c3** | | **38.1-c, 38.2-c** |
| $6+3\sqrt{7}$ | **279.1-g1** | **279.1-g2** | | **279.1-g, 279.2-g** |
| $8-3\sqrt{7}$ | **131.1-c1** | **131.1-c2** | | **131.1-c, 131.2-c** |
| $8+3\sqrt{7}$ | **131.2-c1** | **131.2-c2** | | **131.1-c, 131.2-c** |
| $6-2\sqrt{10}$ | **82.2-b1** | **82.2-b2** | | **82.1-b, 82.2-b** |
| $6+2\sqrt{10}$ | **82.1-b2** | **82.1-b1** | | **82.1-b, 82.2-b** |
| $8-\sqrt{10}$ | **150.1-i1** | **150.1-i4** | | **150.1-i, 150.2-i** |
| $8+\sqrt{10}$ | **150.2-i4** | **150.2-i2** | | **150.1-i, 150.2-i** |

## D.4. Quadratic imaginary points and Bianchi modular forms

Table 21: *Some values of the modulus $\varphi$ which lie in various imaginary quadratic field extensions $\mathbb{Q}(\sqrt{n})$ together with the corresponding Hilbert modular forms and the twisted Sen curves $\mathcal{E}_S$. The second and third columns labelled $\mathcal{E}_S(\tau/n)$ give the LMFDB labels (without the number field label) of the twisted Sen curves, and the penultimate column lists the label (again without the number field label) of the corresponding modular form. The column labelled "$N(\mathfrak{p})$" lists the ideal norms of the prime ideals at which the modular form coefficient $\alpha_{\mathfrak{p}}$ is not equal to the zeta function coefficient $c_p$ in the sense discussed in §4.6.*

| $\varphi$ | $\mathcal{E}_S(\tau)$ | $\mathcal{E}_S(\tau/5)$ | $N(\mathfrak{p})$ | form labels |
|---|---|---|---|---|
| $-6 - \mathrm{i}\sqrt{3}$ | **22971.5-d1** | **22971.5-d2** | | **22971.4-e, 22971.5-d** |
| $-6 + \mathrm{i}\sqrt{3}$ | **22971.4-e1** | **22971.4-e2** | | **22971.4-e, 22971.5-d** |
| $-4 - 4\mathrm{i}\sqrt{3}$ | **69796.2-b2** | **69796.2-b1** | | **69796.1-b, 69796.2-b** |
| $-4 - \mathrm{i}\sqrt{3}$ | **80161.3-a2** | **80161.3-a1** | | **80161.2-a, 80161.3-a** |
| $-4 + \mathrm{i}\sqrt{3}$ | **80161.2-a1** | **80161.2-a2** | | **80161.2-a, 80161.3-a** |
| $-4 + 4\mathrm{i}\sqrt{3}$ | **69796.1-b2** | **69796.1-b1** | | **69796.1-b, 69796.2-b** |
| $-3 - 3\mathrm{i}\sqrt{3}$ | **95988.4-b2** | **95988.4-b1** | | **95988.1-c, 95988.4-b** |
| $-3 - 2\mathrm{i}\sqrt{3}$ | **90489.4-a2** | **90489.4-a1** | | **90489.4-a, 90489.5-a** |
| $-3 - \mathrm{i}\sqrt{3}$ | **27732.2-a2** | **27732.2-a1** | | **27732.1-a, 27732.2-a** |
| $-3 + \mathrm{i}\sqrt{3}$ | **27732.1-a1** | **27732.1-a2** | | **27732.1-a, 27732.2-a** |
| $-3 + 2\mathrm{i}\sqrt{3}$ | **90489.5-a2** | **90489.5-a1** | | **90489.4-a, 90489.5-a** |
| $-3 + 3\mathrm{i}\sqrt{3}$ | **95988.1-c2** | **95988.1-c1** | | **95988.1-c, 95988.4-b** |
| $-2 - 2\mathrm{i}\sqrt{3}$ | **11476.2-b2** | **11476.2-b1** | | **11476.2-b, 11476.3-b** |
| $-2 - \mathrm{i}\sqrt{3}$ | **8113.8-a2** | **8113.8-a1** | | **8113.1-a, 8113.8-a** |
| $-2 + \mathrm{i}\sqrt{3}$ | **8113.1-a2** | **8113.1-a1** | | **8113.1-a, 8113.8-a** |
| $-2 + 2\mathrm{i}\sqrt{3}$ | **11476.3-b2** | **11476.3-b1** | | **11476.2-b, 11476.3-b** |
| $-1 - 4\mathrm{i}\sqrt{3}$ | **66367.6-a1** | **66367.6-a2** | | **66367.3-a, 66367.6-a** |
| $-1 - 3\mathrm{i}\sqrt{3}$ | **134932.4-a2** | **134932.4-a1** | | **134932.4-a, 134932.5-a** |
| $-1 - 2\mathrm{i}\sqrt{3}$ | **26377.2-b2** | **26377.2-b1** | | **26377.2-b, 26377.3-b** |
| $-1 - \mathrm{i}\sqrt{3}$ | **2284.2-a2** | **2284.2-a1** | | **2284.1-a, 2284.2-a** |
| $-1 + \mathrm{i}\sqrt{3}$ | **2284.1-a2** | **2284.1-a1** | | **2284.1-a, 2284.2-a** |
| $-1 + 2\mathrm{i}\sqrt{3}$ | **26377.3-b2** | **26377.3-b1** | | **26377.2-b, 26377.3-b** |
| $-1 + 3\mathrm{i}\sqrt{3}$ | **134932.5-a2** | **134932.5-a1** | | **134932.4-a, 134932.5-a** |
| | | | | *Continued on the following page* |

| continued | | | | |
|---|---|---|---|---|
| $\varphi$ | $\mathcal{E}_S(\tau)$ | $\mathcal{E}_S(\tau/5)$ | $N(\mathfrak{p})$ | form labels |
| $-1 + 4i\sqrt{3}$ | **66367.3-a2** | **66367.3-a1** | | **66367.3-a, 66367.6-a** |
| $-4i\sqrt{3}$ | **98508.2-a1** | **98508.2-a2** | | **98508.1-b, 98508.2-a** |
| $-3i\sqrt{3}$ | **12153.1-b1** | **12153.1-b2** | | **12153.1-b, 12153.2-b** |
| $-2i\sqrt{3}$ | **19452.2-a1** | **19452.2-a2** | | **19452.1-a, 19452.2-a** |
| $-i\sqrt{3}$ | **1137.1-a2** | **1137.1-a1** | | **1137.1-a, 1137.2-a** |
| $i\sqrt{3}$ | **1137.2-a2** | **1137.2-a1** | | **1137.1-a, 1137.2-a** |
| $2i\sqrt{3}$ | **19452.1-a2** | **19452.1-a1** | | **19452.1-a, 19452.2-a** |
| $3i\sqrt{3}$ | **12153.2-b2** | **12153.2-b1** | | **12153.1-b, 12153.2-b** |
| $4i\sqrt{3}$ | **98508.1-b2** | **98508.1-b1** | | **98508.1-b, 98508.2-a** |
| $1 - 4i\sqrt{3}$ | **51583.2-b1** | **51583.2-b2** | | **51583.2-b, 51583.3-b** |
| $1 - 3i\sqrt{3}$ | **101668.3-b1** | **101668.3-b2** | | **101668.2-b, 101668.3-b** |
| $1 - 2i\sqrt{3}$ | **19513.7-a1** | **19513.7-a2** | | **19513.2-a, 19513.7-a** |
| $1 - i\sqrt{3}$ | **1756.2-b1** | **1756.2-b2** | | **1756.1-b, 1756.2-b** |
| $1 + i\sqrt{3}$ | **1756.1-b1** | **1756.1-b2** | | **1756.1-b, 1756.2-b** |
| $1 + 2i\sqrt{3}$ | **19513.2-a2** | **19513.2-a1** | | **19513.2-a, 19513.7-a** |
| $1 + 3i\sqrt{3}$ | **101668.2-b2** | **101668.2-b1** | | **101668.2-b, 101668.3-b** |
| $1 + 4i\sqrt{3}$ | **51583.3-b1** | **51583.3-b2** | | **51583.2-b, 51583.3-b** |
| $2 - 3i\sqrt{3}$ | **106609.4-a1** | **106609.4-a2** | | **106609.4-a, 106609.5-a** |
| $2 - 2i\sqrt{3}$ | **6196.1-b1** | **6196.1-b2** | | **6196.1-b, 6196.2-b** |
| $2 - i\sqrt{3}$ | **4417.1-b1** | **4417.1-b2** | | **4417.1-b, 4417.4-a** |
| $2 + i\sqrt{3}$ | **4417.4-a1** | **4417.4-a2** | | **4417.1-b, 4417.4-a** |
| $2 + 2i\sqrt{3}$ | **6196.2-b1** | **6196.2-b2** | | **6196.1-b, 6196.2-b** |
| $2 + 3i\sqrt{3}$ | **106609.5-a2** | **106609.5-a1** | | **106609.4-a, 106609.5-a** |
| $3 - 3i\sqrt{3}$ | **40548.1-c2** | **40548.1-c1** | 43 | **40548.1-c, 40548.4-d** |
| $3 - 2i\sqrt{3}$ | **35049.3-b1** | **35049.3-b2** | | **35049.2-b, 35049.3-b** |
| $3 - i\sqrt{3}$ | **10308.2-b1** | **10308.2-b2** | | **10308.1-b, 10308.2-b** |
| $3 + i\sqrt{3}$ | **10308.1-b1** | **10308.1-b2** | | **10308.1-b, 10308.2-b** |
| $3 + 2i\sqrt{3}$ | **35049.2-b1** | **35049.2-b2** | | **35049.2-b, 35049.3-b** |
| | | | | *Continued on the following page* |

| $\varphi$ | $\mathcal{E}_S(\tau)$ | $\mathcal{E}_S(\tau/5)$ | $N(\mathfrak{p})$ | form labels |
|---|---|---|---|---|
| continued | | | | |
| $3+3\mathrm{i}\sqrt{3}$ | **40548.4-d2** | **40548.4-d1** | 43 | **40548.1-c, 40548.4-d** |
| $4-4\mathrm{i}\sqrt{3}$ | **25444.2-b2** | **25444.2-b1** | | **25444.1-b, 25444.2-b** |
| $4-3\mathrm{i}\sqrt{3}$ | **145297.2-a2** | **145297.2-a1** | | **145297.2-a, 145297.7-a** |
| $4-2\mathrm{i}\sqrt{3}$ | **50092.2-a1** | **50092.2-a2** | | **50092.2-a, 50092.3-a** |
| $4-\mathrm{i}\sqrt{3}$ | **19969.2-a1** | **19969.2-a2** | | **19969.2-a, 19969.3-a** |
| $4+\mathrm{i}\sqrt{3}$ | **19969.3-a1** | **19969.3-a2** | | **19969.2-a, 19969.3-a** |
| $4+2\mathrm{i}\sqrt{3}$ | **50092.3-a1** | **50092.3-a2** | | **50092.2-a, 50092.3-a** |
| $4+3\mathrm{i}\sqrt{3}$ | **145297.7-a2** | **145297.7-a1** | | **145297.2-a, 145297.7-a** |
| $4+4\mathrm{i}\sqrt{3}$ | **25444.1-b2** | **25444.1-b1** | | **25444.1-b, 25444.2-b** |
| $5-2\mathrm{i}\sqrt{3}$ | **68857.3-a1** | **68857.3-a2** | | **68857.2-a, 68857.3-a** |
| $5-\mathrm{i}\sqrt{3}$ | **32452.7-a1** | **32452.7-a2** | | **32452.2-b, 32452.7-a** |
| $5+\mathrm{i}\sqrt{3}$ | **32452.2-b1** | **32452.2-b2** | | **32452.2-b, 32452.7-a** |
| $5+2\mathrm{i}\sqrt{3}$ | **68857.2-a1** | **68857.2-a2** | | **68857.2-a, 68857.3-a** |
| $6-3\mathrm{i}\sqrt{3}$ | **71211.2-a2** | **71211.2-a1** | | **71211.2-a, 71211.3-a** |
| $6-2\mathrm{i}\sqrt{3}$ | **22332.2-b1** | **22332.2-b2** | | **22332.1-b, 22332.2-b** |
| $6-\mathrm{i}\sqrt{3}$ | **45201.2-d1** | **45201.2-d2** | | **45201.2-d, 45201.7-d** |
| $6+\mathrm{i}\sqrt{3}$ | **45201.7-d1** | **45201.7-d2** | | **45201.2-d, 45201.7-d** |
| $6+2\mathrm{i}\sqrt{3}$ | **22332.1-b2** | **22332.1-b1** | | **22332.1-b, 22332.2-b** |
| $6+3\mathrm{i}\sqrt{3}$ | **71211.3-a2** | **71211.3-a1** | | **71211.2-a, 71211.3-a** |
| $7-2\mathrm{i}\sqrt{3}$ | **109129.1-b1** | **109129.1-b2** | | **109129.1-b, 109129.4-b** |
| $7-\mathrm{i}\sqrt{3}$ | **54652.3-a1** | **54652.3-a2** | | **54652.2-a, 54652.3-a** |
| $7+\mathrm{i}\sqrt{3}$ | **54652.2-a1** | **54652.2-a2** | | **54652.2-a, 54652.3-a** |
| $7+2\mathrm{i}\sqrt{3}$ | **109129.4-b2** | **109129.4-b1** | | **109129.1-b, 109129.4-b** |
| $8-2\mathrm{i}\sqrt{3}$ | **126844.2-a2** | **126844.2-a1** | | **126844.2-a, 126844.3-a** |
| $8-\mathrm{i}\sqrt{3}$ | **57553.1-b1** | **57553.1-b2** | | **57553.1-b, 57553.4-b** |
| $8+\mathrm{i}\sqrt{3}$ | **57553.4-b1** | **57553.4-b2** | | **57553.1-b, 57553.4-b** |
| $8+2\mathrm{i}\sqrt{3}$ | **126844.3-a2** | **126844.3-a1** | | **126844.2-a, 126844.3-a** |
| $9-3\mathrm{i}\sqrt{3}$ | **41268.3-b2** | **41268.3-b1** | | **41268.2-b, 41268.3-b** |

| $\varphi$ | $\mathcal{E}_S(\tau)$ | $\mathcal{E}_S(\tau/5)$ | $N(\mathfrak{p})$ | form labels |
|---|---|---|---|---|
| continued | | | | |
| $9 - 2i\sqrt{3}$ | **144057.2-a2** | **144057.2-a1** | | **144057.2-a, 144057.3-a** |
| $9 - i\sqrt{3}$ | **53004.2-b1** | **53004.2-b2** | | **53004.2-b, 53004.3-b** |
| $9 + i\sqrt{3}$ | **53004.3-b1** | **53004.3-b2** | | **53004.2-b, 53004.3-b** |
| $9 + 2i\sqrt{3}$ | **144057.3-a2** | **144057.3-a1** | | **144057.2-a, 144057.3-a** |
| $9 + 3i\sqrt{3}$ | **41268.2-b1** | **41268.2-b2** | | **41268.2-b, 41268.3-b** |
| $10 - 2i\sqrt{3}$ | **42028.6-b2** | **42028.6-b1** | | **42028.3-b, 42028.6-b** |
| $10 - i\sqrt{3}$ | **45217.1-a2** | **45217.1-a1** | | **45217.1-a, 45217.4-a** |
| $10 + i\sqrt{3}$ | **45217.4-a2** | **45217.4-a1** | | **45217.1-a, 45217.4-a** |
| $10 + 2i\sqrt{3}$ | **42028.3-b2** | **42028.3-b1** | | **42028.3-b, 42028.6-b** |
| $-8 - 8i\sqrt{7}$ | **20484.3-d1** | **20484.3-d2** | 29 | **20484.3-d, 20484.4-d** |
| $-8 + 8i\sqrt{7}$ | **20484.4-d1** | **20484.4-d2** | 29 | **20484.3-d, 20484.4-d** |
| $-2 - 2i\sqrt{7}$ | **25236.3-d2** | **25236.3-d1** | | **25236.3-d, 25236.4-d** |
| $-2 + 2i\sqrt{7}$ | **25236.4-d2** | **25236.4-d1** | | **25236.3-d, 25236.4-d** |
| $2 - i\sqrt{7}$ | **11209.4-a1** | **11209.4-a2** | 11 | **11209.1-a, 11209.4-a** |
| $2 + i\sqrt{7}$ | **11209.1-a1** | **11209.1-a2** | 11 | **11209.1-a, 11209.4-a** |
| $6 + 2i\sqrt{7}$ | **40832.7-d4** | **40832.7-d1** | 37 | **40832.7-d** |
| $9 - i\sqrt{7}$ | **44836.5-a2** | **44836.5-a1** | 11 | **44836.5-a** |
| $9 + i\sqrt{7}$ | **44836.8-a2** | **44836.8-a1** | | **44836.8-a** |

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
