# Peer review of "Flux Vacua and Modularity for Z2 Symmetric Calabi-Yau Manifolds"

_SciPost Physics_

## Round 1 · Referee Report · Anonymous · 2023-6-15

Strengths

1-
In the article at hand, the authors provide a large class of families of Calabi-Yau threefolds that are conjecturally modular in the sense that the Hodge structure on their midde cohomology has a direct summand of type $(2,1) + (1,2)$ and rank $2$. This class is characterized by the property that their complex structure parameter space $\mathcal{M}$ admits an action by a $\mathbb{Z}_2$ which is realized as a permutation of coordinates.

2-
This action has two effects: One the one hand, it induces the splitting of the Hodge structure and on the other hand the fixed point loci $\mathcal{F}\subset \mathcal{M}$ are solutions to the vacuum equations of supersymmetric flux compactifications in string theory. Therefore, the authors use this class of families of Calabi-Yau threefolds to give an extensive verification of the flux modularity conjecture of Kachru et al. which says that Calabi-Yau threefolds corresponding to such vacua are modular in the above sense.

3-
Since this class requires $\dim \mathcal{M} > 1$, the authors extend the method of Dwork on Calabi-Yau crystals to compute the zeta function of $X_\varphi, \varphi\in \mathcal{F}$, from the case of one-parameter families done in an earlier work by Candelas, de la Ossa, and van Straten, to the multiparameter case. The details are deferred to a future publication. The authors use this to test whether the numerator of the zeta function of $X_\varphi$ has a quadratic factor, corresponding to a two-dimensional Galois representation and hence to a direct summand of rank 2 in the Hodge structure.

4-
Using this method, the authors test the flux modularity conjecture for various explicit families of Calabi-Yau threefolds and - for each family - a very large number of moduli for which $X_\varphi$ is defined over a number field $K$ with $[K:\mathbb{Q}] \leq 2$.

5-
Furthermore, the authors point out a remarkable relation between an abstract elliptic curve over $\mathbb{Q}$ (and its isogeny class) obtained from the two-dimensional irreducible Galois representation corresponding to the quadratic factor in the numerator of the zeta function of $X_\varphi$ and the elliptic curve over $\mathbb{C}$ that appears as the fiber of the Calabi-Yau fourfold in the F-theory description of the type IIB flux compactification governed by that representation.

6-
All this - the numerators of the zeta functions, the modular forms corresponding to the various isogeneous elliptic curves - is supported by a large amount of tables.

Weaknesses

None

Report

The paper is generally very well and clearly written. It provides very valuable foundations for future investigations of modularity of Calabi–Yau threefolds and the geometry of the Calabi–Yau fourfolds for such flux compactifications. In summary, I strongly recommend it for publication after the following minor points have been taken into account.

Requested changes

1-
On p. 13, in the definition of $\mathfrak{A}_{i,j}$ after (25), $\varphi_s$ should have an upper index $s$.

2-
There is an issue with terminology and notation for algebraic varieties. The standard in mathematics is that if an algebraic variety $X$ is defined over a field $K$, i.e. the coefficients of the equations for $X$ take values in $K$, then one writes $X/K$. If the solutions to these equations take values in a field $K' \supseteq K$, then one writes for the set of $K'$-points $X(K')$, see e.g. the beginning of Chapter V.2 in Silverman, ref. [65].

This issue appears at several places in the paper:

On p. 14, last paragraph, it should read $X/\mathbb{K}$ instead of $X(\mathbb{K})$.

On p. 15, first paragraph in Subsection 4.1, it should read "defining a variety $E/\mathbb{F}_{p^n}$ over" instead of $E(\mathbb{F}_{p^n})$. The following sentence, explaining the symbol $E(\mathbb{F}_{p^n})$ is, however, correct, as this describes the solutions to the equations, not the equations themselves.

On p.15 last paragraph, however, it should read twice $E/\mathbb{F}_{p^n}$ as the singularities of $E$ are determined by the equations for $E$, not their solutions (the vanishing of the discriminant is determined by the coefficients of the equations defining $E$).

On p.16, second paragraph, it should read $X_{\mathbf{\varphi}}/\mathbb{K}$ instead of $X_{\mathbf{\varphi}}(\mathbb{K})$ if this really is referring to the field of definition of $X_{\mathbf{\varphi}}$. If, however, the set of points is referred to, then one should write that "the $\mathbb{K}$--points of the variety $X_{\varphi}$ will be denoted by $X_{\varphi}(\mathbb{K})$". I rather suspect that the authors mean the latter alternative.

On p.25, in the third paragraph of Subsection 4.6 it should read: $E/\mathbb{Q}(\sqrt{n})$ defined over $\mathbb{Q}(\sqrt{n})$.

3-
On p. 24, second line, it should read $\Lambda$ and $\Lambda'$.

4-
On p. 24 two lines before eq. (40) it should read $\wp(z,\Lambda)$ instead of $\wp(z,L)$.

5-
On p. 33, in the first line of Subsection 5.2, there is a typo in "Frobenius".

6-
On p. 34, in eq. (53) the matrices $\mathbb{Y}_i$ do not seem to be defined. Then, below, in the formula for $\theta_iE$, the (1,3) entry of the matrix should read $\ell^T \mathbf{Y}_{ij}$, the (4,4) entry should be $0$. In this formula, the vectors $\mathbf{Y}_{ij}$ are not defined. Of course, from the context it is clear that $(\mathbb{Y}_i)_{jk}=Y_{ijk}$ and $(\mathbf{Y}_{ij})_k = Y_{ijk}$.

Furthermore, while $\mathbf{\epsilon}$ was introduced in Section 3.2 as a vector of formal parameters in a nilpotent ring, here it is interpreted as a vector of (nilpotent) matrices. This implicit identification should be stated explicitly.

---

## Round 1 · Referee Report · Anonymous · 2023-6-25

Report on the paper scipost 202304−00018:

*Flux Vacua and Modularity for $Z_2$ symmetric CY manifolds*

by P. Candelas, X. de la Ossa, P. Kuusela and J. McGovern

The authors aim to provide evidence in support of a conjecture of Kachru, Nally and Yang concerning the modularity of flux vacua. This is an interesting question and the paper contains some extensive work in this direction. I find particularly interesting their foray in the Deligne conjecture, which has made an appearance in physics quite some time ago before the work on Yang, but which otherwise is rarely considered in physics. Also interesting is the consideration of the KNY modularity conjecture in the context of algebraic extensions of the rationals. I recommend the paper for publication but would like to make a few comments for the authors to consider.

**Remarks:**

1. I believe that there is a typo in the definition of $j$ and that as written the Eisenstein series $E_4$ should be replaced by $G_4$. It seems that the authors use the Koblitz definition of the weight 12 form $\Delta_K$, given by

$$\Delta_K \;=\; g_2^3 \;-\; 27 g_3^2.$$

   While classical, dating back to the 19th century, this definition does not have rational coefficients and is not normalized in the standard way via the Dedekind eta function as $\Delta = \eta^{24}$. Using this Koblitz definition of $\Delta_K$, and adjusting the definition of the $j$-function given in the paper by replacing $E_4$ by $G_4$, i.e. using

$$j \;=\; 1728 \cdot 60^3 \frac{G_4^3}{\Delta_K},$$

   leads to a more pristine form of the $j$-function as

$$j \;=\; \frac{E_4^3}{\Delta}$$

   that now does involve the integrally normalized $E_4$, as well as the integrally normalized discriminant form $\Delta \;=\; \eta^{24}$. This latter form of $j$ avoids the various factors of $(2\pi)$ that

arise from a lattice starting point. Irrespective of form in which the authors end up defining the $j$-function, and in view of the fact that different versions of the discriminant form exist in the literature, it would be useful to have the definitions of $\Delta$ and $E_4$, or $G_4$, handy, perhaps in the modular form appendix of the paper.

2. It would be useful to give the Sturm bounds for the modular forms that appear in the paper.

3. In the present paper and in previous work the authors proceed by computing the zeta function of a variety and then identify "motivic pieces" of the cohomology by factorizing the zeta function. It would be interesting, and more direct, to proceed in an alternative way by first attempting to identify the motives first and then computing the motivic $L$-functions. Work in this direction has appeared in the physics literature some time ago for certain families of CYs, although in a different physical context (Kadir et al, 1012.5807 [hep-th]).

4. Finally, I've noticed a few trivial typos on pages 14, 27 and 134.

---

## Editorial Decision

resubmitted